# A Unified Analysis of Stochastic Gradient Descent with Arbitrary Data Permutations and Beyond

**Yipeng Li**[1]*  **Xinchen Lyu**[2,3]†  **Zhenyu Liu**[1]†

[1]Tsinghua Shenzhen International Graduate School, Tsinghua University
[2]National Engineering Research Center for Mobile Network Technologies,
Beijing University of Posts and Telecommunications
[3]Department of Broadband Communication, Pengcheng Laboratory

`liyp25@mails.tsinghua.edu.cn, lvxinchen@bupt.edu.cn`
`zhenyuliu@sz.tsinghua.edu.cn`

## Abstract

We aim to provide a unified convergence analysis for permutation-based Stochastic Gradient Descent (SGD), where data examples are permuted before each epoch. By examining the relations among permutations, we classify existing permutation-based SGD algorithms into three categories: Arbitrary Permutations, Independent Permutations (including Random Reshuffling and FlipFlop [Rajput et al., 2022]), Dependent Permutations (including GraBs [Lu et al., 2022a; Cooper et al., 2023]). Existing unified analyses failed to encompass the Dependent Permutations category due to the inter-epoch permutation dependency. In this work, we propose a generalized assumption that explicitly characterizes the dependence of permutations across epochs. Building upon this assumption, we develop a unified framework for permutation-based SGD with arbitrary permutations of examples, incorporating all the existing permutation-based SGD algorithms. Furthermore, we adapt our framework for Federated Learning (FL), developing a unified framework for regularized client participation FL with arbitrary permutations of clients.

## 1 Introduction

We study the finite-sum minimization problem

$$\min_{\mathbf{x}\in\mathbb{R}^d}\left[f(\mathbf{x}) := \frac{1}{N}\sum_{n=0}^{N-1} f_n(\mathbf{x})\right], \tag{1}$$

where $d$ denotes the dimension of the model parameter vector, $N$ denotes the number of the local objective functions $\{f_n\}$ and each $f_n : \mathbb{R}^d \to \mathbb{R}$ is assumed to be differentiable.

### 1.1 Initial Motivation: Example Ordering in Permutation-based SGD

**Permutation-based SGD.** One popular way to solve problem (1) is Stochastic Gradient Descent (SGD). It updates the parameter vector iteratively according to the rule

$$\mathbf{x}^{n+1} = \mathbf{x}^n - \gamma \nabla f_{\pi(n)}(\mathbf{x}^n),$$

where $\gamma$ denotes the step size and $\pi(n)$ denotes the index of the local objective function at iteration $n$. For classic SGD, $\pi(n)$ is chosen uniformly with replacement from $\{0, 1, \dots, N-1\}$; for permutation-based SGD, $\pi(n)$ is the $(n+1)$-th element of a permutation $\pi$ of $\{0, 1, \dots, N-1\}$. Due to its simple implementation and empirical superiority [Bottou, 2012], permutation-based SGD

---

*Part of the work was done when the author was with Beijing University of Posts and Telecommunications.
†Corresponding authors.

has garnered more attention recently. The first topic of this paper is the convergence analysis of permutation-based SGD (Algorithm 1).

**A measure of the quality of the permutation of examples.** The convergence rate of permutation-based SGD is determined by the permutations of examples. Thus, to study it, we need a measure of the quality of the permutation of examples. Note that we say that a permutation is good if it leads to a high convergence rate of permutation-based SGD, and vice versa. For a small finite step size $\gamma$, the cumulative updates in any epoch $q$ are

$$
\mathbf{x}_{q+1} - \mathbf{x}_q \approx \underbrace{-\gamma N \nabla f(\mathbf{x}_q) + \gamma^2 \sum_{n=0}^{N-1} \sum_{i<n} \nabla^2 f_{\pi(n)}(\mathbf{x}_q) \nabla f(\mathbf{x}_q)}_{\text{optimization vector}}
$$

$$
+ \underbrace{\gamma^2 \sum_{n=0}^{N-1} \sum_{i<n} \nabla^2 f_{\pi(n)}(\mathbf{x}_q) \left( \nabla f_{\pi(i)}(\mathbf{x}_q) - \nabla f(\mathbf{x}_q) \right)}_{\text{error vector}},
$$

where this equation is from Smith et al. [2021, Equation (13)] (we replace $=$ with $\approx$ as we omit $\mathcal{O}(\gamma^3 N^3)$), and it can be proved by Taylor expansion. Here, we additionally assume that each $f_n$ is twice differentiable. The optimization vector is beneficial; the error vector is detrimental and depends on the order of examples. Thus, the goal is to suppress the error vector. For instance, we use Lebesgue 2-norm for both vectors and matrices:

$$
\|\text{Error vector}\| \leq \gamma^2 \sum_{n=0}^{N-1} \left\| \nabla^2 f_{\pi(n)}(\mathbf{x}_q) \right\| \left\| \sum_{i<n} \left( \nabla f_{\pi(i)}(\mathbf{x}_q) - \nabla f(\mathbf{x}_q) \right) \right\| \leq \gamma^2 L N \bar{\phi}_q,
$$

where the last inequality is due to $L$-smoothness (see Definition 3) and Definition 1 (here, $p = 2$).

**Definition 1** (Order Error, Lu et al. [2022b,a]). *The order error $\bar{\phi}_q$ in any epoch $q$ is defined as*

$$
\bar{\phi}_q := \max_{n \in [N]} \left\{ \phi_q^n := \left\| \sum_{i=0}^{n-1} \left( \nabla f_{\pi_q(i)}(\mathbf{x}_q) - \nabla f(\mathbf{x}_q) \right) \right\|_p \right\}.
$$

This implies *the order error $\bar{\phi}_q$ can be used as a measure of the quality of the permutation of examples*: a smaller $\bar{\phi}_q$ means a faster convergence rate, and a better permutation, and vice versa. Even though the order error was proposed in Lu et al. [2022b], where the authors justified its validity on synthetic experiments empirically, the rationale behind it (that is, the above analysis) has not been well understood until this work.

**Existing permutation-based SGD algorithms.** Based on *the relations among permutations*, we classify the existing permutation-based SGD algorithms into three categories:

- Arbitrary Permutations (AP): Permutations are generated without any specific structure, allowing for completely arbitrary permutations for all epochs.

- Independent Permutations (IP): Permutations are independent across epochs.

  - Random Reshuffling (RR): The permutation in each epoch is generated randomly.

  - FlipFlop [Rajput et al., 2022]: See Appendix F.3 for details.

  - Greedy Ordering [Lu et al., 2022b; Mohtashami et al., 2022]: The permutation in each epoch is generated by a greedy algorithm.

- Dependent Permutations (DP): Permutations are dependent across epochs, with the permutation in one epoch affected by the permutations in previous epochs (explicitly).

  - One Permutation (OP): The initial (first-epoch) permutation is used repeatedly for all the subsequent epochs. When the initial permutation is arbitrary, it is called Incremental Gradient (IG); when the initial permutation is random, it is called Shuffle Once (SO).

– GraBs: It includes GraB [Lu et al., 2022a] and PairGraB [Lu et al., 2022a; Cooper et al., 2023]. In particular, GraB has been proven to outperform RR, and even be a theoretically optimal permutation-based SGD algorithm [Cha et al., 2023]. See Appendix C for details.

For Greedy Ordering and OP, as done in prior works (see Lu et al. [2022b] for Greedy Ordering and Mishchenko et al. [2020] for OP), we use the bound of AP as their bounds (*The bound of AP applies to all the other algorithms.*) and will not discuss them in the remainder of this paper.

**A more general assumption.** For AP or IP, the relation among permutations is arbitrary or independent, and thus we can bound the order error for any epoch and then apply this bound for all the epochs. To deal with these categories, Lu et al. [2022b] proposed Assumption 1 (Lu et al. [2022b] considered an interval of arbitrary length, not necessarily an epoch.).

**Assumption 1** (Lu et al. [2022b,a])**.** *There exist nonnegative constants $B$ and $D$ such that for all* $\mathbf{x}_q$ *(the outputs of Algorithm 1),*

$$\left(\bar{\phi}_q\right)^2 \le B \left\|\nabla f(\mathbf{x}_q)\right\|^2 + D.$$

By proving that Assumption 1 holds for AP and IP with specific values of $B$ and $D$ (under some standard assumptions in SGD), prior works [Lu et al., 2022b; Mohtashami et al., 2022; Koloskova et al., 2024] successfully incorporate them into one framework. However, none of the unified frameworks of permutation-based SGD has successfully incorporated DP. The main reason for the failure can be that, existing works implicitly deal with the order error $\bar{\phi}_q$ separately across epochs (as in Assumption 1), while in DP (in particular, GraBs), the example orders across consecutive epochs are dependent. This limitation sparked our initial motivation for this work—*to develop a unified convergence analysis framework of permutation-based SGD that includes DP.*

To achieve this, we propose a more general assumption (Assumption 2) than Assumption 1.

**Assumption 2.** *There exist nonnegative constants* $\{A_i\}_{i=1}^q$, $\{B_i\}_{i=0}^q$ *and $D$ such that for all* $\mathbf{x}_q$ *(the outputs of Algorithm 1),*

$$\left(\bar{\phi}_q\right)^2 \le \sum_{i=1}^q A_i \left(\bar{\phi}_{q-i}\right)^2 + \sum_{i=0}^q B_i \left\|\nabla f(\mathbf{x}_{q-i})\right\|^2 + D.$$

This assumption explicitly characterizes the *dependence* between permutations across different epochs. In particular, when $A_i = 0$ and $B_i = 0$ for all $i \in \{1, 2, \ldots q\}$, it reduces to Assumption 1. Our analytical framework transforms the task of obtaining the algorithm's bound into proving that Assumption 2 holds. Especially for DP, the task is to identify the relation between order errors. For instance, for GraBs, the main task is to establish the relation between $\bar{\phi}_q$ and $\bar{\phi}_{q-1}$ for $q \ge 1$.

### 1.2 New Challenges: Client Ordering in FL with Regularized Client Participation

**FL with regularized client participation.** Federated Learning (FL) [McMahan et al., 2017] is one of the most popular distributed machine learning paradigms, which aims to learn from data distributed across multiple clients while ensuring data security and privacy. In cross-device FL [Kairouz et al., 2021], only a small fraction of clients can participate in the training process simultaneously. FL with regularized client participation (regularized-participation FL) is one realistic participation pattern, where each client participates once before any client is reused, which can be caused by diurnal variation [Eichner et al., 2019]. Wang and Ji [2022] and Cho et al. [2023] showed that the regularized participation pattern is better than the vanilla participation pattern [Yang et al., 2021]. The second topic of this paper is the convergence analysis of FL with regularized participation (Algorithm 2).

**New challenges.** Equation (1) is also the problem of FL if the local objective functions $f_n$ represent the clients (In contrast, in SGD, the local objective functions represent the examples.). This correlation raises one important question: Is it possible to apply the example ordering algorithms in SGD to client ordering in FL? The answer is affirmative; in fact, prior works [Cho et al., 2023; Malinovsky et al., 2023a] have taken a first step toward this. In this paper, we aim to advance this line of work by *developing a unified framework of regularized-participation FL with arbitrary permutations of clients*. Compared to SGD, the main challenges of FL lie in the following two aspects:

1. Partially parallel updates. In a round of federated training, the selected clients are in parallel.

2. Local updates. It performs multiple local updates on each local objective function.

(i) The first challenge causes that $\bar{\phi}_q$ (for SGD) cannot be applied in FL. To address it, we introduce a new order error $\bar{\varphi}_q$ for FL in Definition 2. (ii) The second challenge causes that we can only access the pseudo-gradients (that is, $\mathbf{p}_q^n$ in Algorithm 2) of the local objective functions, instead of the true gradients, which complicates our analysis largely.

**Definition 2.** *The order error $\bar{\varphi}_q$ in any epoch $q$ in FL is defined as*

$$\bar{\varphi}_q := \max_{n \in [N]} \left\{ \varphi_q^{v(n)} := \left\| \sum_{i=0}^{\color{red}{v(n)}-1} \left( \nabla f_{\pi_q(i)}(\mathbf{x}_q) - \nabla f(\mathbf{x}_q) \right) \right\|_p \right\},$$

*where $v(n) := \lfloor \frac{n}{S} \rfloor \cdot S$ and $S \geq 2$ denotes the number of selected clients in each round (see Algorithm 2). The main difference compared to $\bar{\phi}_q$ is highlighted in red.*

### 1.3 Main Contributions

The main contributions are as follows:

- Example ordering in SGD (Section 3). We propose a more general assumption (Assumption 2) to bound the order error, which explicitly characterizes the dependence between permutations across different epochs. Based on it, we develop a unified framework for permutation-based SGD with arbitrary permutations of examples, which is *the first unified framework that includes DP*. We prove that all the existing permutation-based SGD algorithms can be incorporated into our framework.

- Client ordering in FL with regularized participation (Section 4). We develop a unified framework for regularized-participation FL with arbitrary permutations of clients, which is *the first unified framework that focuses on client ordering*. In particular, we propose FL-GraB to accelerate the training of FL.

## 2 Related Works

Detailed related works are deferred to Appendices A and C.

### 2.1 Permutation-based SGD

Table 1: Existing unified analyses of permutation-based SGD and their upper bounds (of convergence rates). Numerical constants and polylogarithmic factors are hided. We translate all the bounds with our notations, assumptions, and step size choice. We let $\alpha = 0$ in Assumption 3 for comparison.

| | **Alg.** | Lu et al. [2022b] | Koloskova et al. [2024][1] | This work |
|---|---|---|---|---|
| AP[2] | AP | $\frac{LF_0}{Q} + \left(\frac{LF_0 N\varsigma}{NQ}\right)^{\frac{2}{3}}$ | $\frac{LF_0}{NQ} + \left(\frac{LF_0 N\varsigma}{NQ}\right)^{\frac{2}{3}} \wedge \left(\frac{LF_0 N\varsigma^2}{NQ}\right)^{\frac{1}{2}}$ [3] | $\frac{LF_0}{Q} + \left(\frac{LF_0 N\varsigma}{NQ}\right)^{\frac{2}{3}}$ |
| IP | RR | $\frac{LF_0}{Q} + \left(\frac{LF_0 \sqrt{N}\varsigma}{NQ}\right)^{\frac{2}{3}}$ | Same as AP | $\frac{LF_0}{Q} + \left(\frac{LF_0 \sqrt{N}\varsigma}{NQ}\right)^{\frac{2}{3}}$ |
| | FlipFlop | – | – | $\frac{LF_0}{Q} + \left(\frac{LF_0 \sqrt{N}\varsigma}{NQ}\right)^{\frac{2}{3}}$ |
| DP | GraBs | – | – | $\frac{\tilde{L}F_0 + (L_{2,\infty} F_0\varsigma)^{\frac{2}{3}}}{Q} + \left(\frac{L_{2,\infty} F_0\varsigma}{NQ}\right)^{\frac{2}{3}}$ [4] |

---

[1] It uses a stronger assumption than Lu et al. [2022b] and ours, equivalent to Assumption 3 with $\alpha = 0$.

[2] For a clearer comparison with other bounds, we keep "$N$" in both numerators and denominators in the bounds of AP.

[3] Here, $a \wedge b$ represents $\min\{a, b\}$.

[4] For GraBs (that is, GraB-proto, GraB, PairGraB), $\tilde{L} = L + L_{2,\infty} + L_\infty$. See Definition 3 for $L$, $L_{2,\infty}$ and $L_\infty$.

---

**Convergence analyses of permutation-based SGD.** The most relevant works are the unified analyses [Lu et al., 2022b; Koloskova et al., 2024], which are summarized in Table 1.

- **AP.** For AP, the best bound is from Koloskova et al. [2024]. (i) Their advantage lies in the first term, which is not dominant when the number of epochs $Q$ is large. (ii) They bound the order error over a period of $\Theta(\frac{1}{\gamma L})$, rather than an epoch, which complicates the analysis of other algorithms largely. For instance, for RR, their bound is worse than the other works.

- **IP.** (i) For RR, Lu et al. [2022b] and this work achieve the same convergence rate, which matches the baseline [Mishchenko et al., 2020]. (ii) In addition, this work includes FlipFlop for the first time, showing that FlipFlop can achieve the same rate as RR.

- **DP.** For GraBs, this work includes GraBs for the first time. In addition, our bounds (for GraBs) match the original bounds of GraBs [Lu et al., 2022a; Cooper et al., 2023] under a weaker assumption (Assumption 3).

## 2.2 FL with Regularized Client Participation

Table 2: Existing upper bounds (of $f(x_Q) - f_*$) for regularized-participation FL algorithms. Numerical constants and polylogarithmic factors are hided. We set $\eta = 1$ (in Algorithm 2) and $\alpha = 0$ (in Assumption 3) for comparison. See Appendix G.3.

| Alg. | Prior works | This work (unified framework) |
|---|---|---|
| FL-AP | $\frac{F_0}{NQ^2} + \frac{L^2 S^2 \varsigma^2}{\mu^3 N^2 Q^2} + \frac{L^2 \varsigma^2}{\mu^3 Q^2}$ [1] | $F_0 \exp\left(-\frac{\mu Q}{L}\right) + \frac{L^2 S^2 \varsigma^2}{\mu^3 N^2 Q^2} + \frac{L^2 \varsigma^2}{\mu^3 Q^2}$ |
| FL-RR | $\frac{L}{\mu} F_0 \exp\left(-\frac{\mu K N^{\frac{1}{S}} Q}{L}\right) + \frac{L^2 S^2 \varsigma^2}{\mu^3 N^2 Q^2} + \frac{L^2 \varsigma^2}{\mu^3 N Q^2}$ [2] | $F_0 \exp\left(-\frac{\mu Q}{L}\right) + \frac{L^2 S^2 \varsigma^2}{\mu^3 N^2 Q^2} + \frac{L^2 \varsigma^2}{\mu^3 N Q^2}$ |
| FL-GraB | – | $\left(F_0 + \frac{\varsigma^2}{L}\right) \exp\left(-\frac{\mu Q}{L}\right) + \frac{L_{2,\infty}^2 S^2 \varsigma^2}{\mu^3 N^2 Q^2} + \frac{L_{2,\infty}^2 \varsigma^2}{\mu^3 N^2 Q^2}$ |

[1] It's from Cho et al. [2023]'s Theorem 2. The difference in the first term can be due to the step size choice.
[2] It's from Malinovsky et al. [2023a]'s Theorem 6.1. The difference in the optimization term is because they consider $\mu$-strongly convex objectives (in contrast, we consider PL condition), and use the advanced technique of shuffling variance in Mishchenko et al. [2020].

**Convergence analyses of FL with regularized client participation.** The convergence analyses of regularized-participation FL have been studied in Wang and Ji [2022], Cho et al. [2023] and Malinovsky et al. [2023a], where Wang and Ji [2022] and Cho et al. [2023] considered regularized-participation FL with AP (FL-AP) and Malinovsky et al. [2023a] considered regularized-participation FL with RR (FL-RR). This work aims to develop a unified framework that includes these cases. Since Cho et al. [2023] considered the $\mu$-PL condition and Malinovsky et al. [2023a] considered the $\mu$-strongly convex objective functions, we next translate our bounds (non-convex) with PL condition (see Appendix G.3). Table 2 shows that *our bounds match the existing bounds* when the optimization term (the first term) is omitted (when $Q$ is large).

## 2.3 Preliminaries of GraBs

The goal of GraB (including other variants) is to find a permutation to minimize the order error $\bar{\phi}_q :=$ $\max_{n \in [N]} \left\| \sum_{i=0}^{n-1} \left( \nabla f_{\pi_q(i)}(\mathbf{x}_q) - \nabla f(\mathbf{x}_q) \right) \right\|_\infty$ (Notably, in GraBs, $\bar{\phi}_q$ is defined by $\|\cdot\|_\infty$), which is aligned with the goal of herding [Welling, 2009]. With this insight, Lu et al. [2022a] proposed GraB (to produce good permutations online) based on the theory of herding and balancing [Harvey and Samadi, 2014; Alweiss et al., 2021]. Consider $N$ vectors $\{\mathbf{z}_n\}_{n=0}^{N-1}$ such that $\sum_{n=0}^{N-1} \mathbf{z}_n = 0$ and $\|\mathbf{z}_n\| \leq 1$. Then, GraB applies the process of balancing and then reordering.

- **Balancing.** For any permutation $\pi$, assign the signs $\{\epsilon_n\}_{n=0}^{N-1}$ ($\epsilon_n \in \{-1, +1\}$) to the permuted vectors $\{\mathbf{z}_{\pi(n)}\}_{n=0}^{N-1}$ using the *balancing* algorithms (such as Algorithm 3 in Appendix C).

- **Reordering.** With the assigned signs and the old permutation $\pi$, produce a new permutation $\pi'$ by the *reordering* algorithm (that is, Algorithm 4 in Appendix C).

Then, Lemma 3 (in Appendix D) proves that the following equation holds:

$$\underbrace{\max_{n \in [N]} \left\| \sum_{i=0}^{n-1} \mathbf{z}_{\pi'(i)} \right\|_\infty}_{\text{the new herding error}} \leq \frac{1}{2} \underbrace{\max_{n \in [N]} \left\| \sum_{i=0}^{n-1} \mathbf{z}_{\pi(i)} \right\|_\infty}_{\text{the old herding error}} + \frac{1}{2} \underbrace{\max_{n \in [N]} \left\| \sum_{i=0}^{n-1} \epsilon_i \mathbf{z}_{\pi(i)} \right\|_\infty}_{=\tilde{\mathcal{O}}(1), \text{ if } \{\epsilon_i\} \text{ are assigned by Alg. 3}}, \tag{2}$$

where we call the three terms, the herding error under $\pi'$ (new), the herding error under $\pi$ (old), and the signed herding error under $\pi$, respectively. Equation (2) ensures that the herding error will be reduced (from $\pi$ to $\pi'$) as long as the signed herding error is small. That is, the herding error can be progressively reduced by balancing and reordering the vectors. By iteratively applying this process (balancing and then reordering), the herding error will approach the signed herding error, which

is proved to be $\tilde{\mathcal{O}}(1)$, if the signs are assigned by Algorithm 3 [Alweiss et al., 2021]. The above content introduces the key idea of GraB. We show how to use Equation (2) in GraB in Section 3.2.

(i) Importantly, GraB has been proven to outperform RR, and even be a theoretically optimal permutation-based SGD algorithm [Cha et al., 2023]. (ii) Furthermore, the GraB algorithms have also been applied in distributed optimization (not FL) [Cooper et al., 2023] and multi-objective optimization [Yang and Kwok, 2025]. Both give us strong motivations to incorporate this DP algorithm into one unified framework.

## 3 Permutation-based SGD

**Notations.** We use $\|\cdot\|_p$ to denote the Lebesgue $p$-norm; For simplicity, we use $\|\cdot\|$ to denote the Lebesgue 2-norm. We use $\lesssim$ to denote "less than" up to some numerical constants and polylogarithmic factors. See more notations in Appendix B.

**Setup.** For SGD, we consider the problem in Equation (1). In the context of SGD, the local objective functions represent the examples. See Algorithm 1. Here, $\pi$ denotes a permutation of $\{0, 1, \ldots, N-1\}$ (at the same time, it serves as the training order of examples). At the end of each epoch, it produces the

---

**Algorithm 1:** Permutation-based SGD

**Input:** $\pi_0, \mathbf{x}_0$; **Output:** $\{\mathbf{x}_q\}$
1 **for** $q = 0, 1, \ldots, Q - 1$ **do**
2 $\quad \mathbf{x}_q^0 \leftarrow \mathbf{x}_q$
3 $\quad$ **for** $n = 0, 1, \ldots, N - 1$ **do**
4 $\quad\quad \mathbf{x}_q^{n+1} \leftarrow \mathbf{x}_q^n - \gamma \nabla f_{\pi_q(n)}(\mathbf{x}_q^n)$
5 $\quad \mathbf{x}_{q+1} \leftarrow \mathbf{x}_q^N$
6 $\quad \pi_{q+1} \leftarrow \texttt{Permute}(\cdots)$

---

next-epoch permutation by some permuting algorithm (Algorithm 1). The details of the function `Permute` in Algorithm 1 for GraBs are specified in Table 6.

### 3.1 Main Theorem

Theorem 1 gives our main framework for permutation-based SGD. See Appendix J for the additional extension for Theorem 1.

Definition 3 will help us deal with the multiple smoothness constants in GraBs. We assume that the global objective function $f$ is lower bounded by $f_*$ and let $F_0 = f(\mathbf{x}_0) - f_*$.

**Definition 3** ($L_{p,p'}$-smoothness). *We say $f$ is $L_{p,p'}$-smooth, if it is differentiable and for $\mathbf{x}, \mathbf{y} \in \mathbb{R}^d$,*

$$\|\nabla f(\mathbf{x}) - \nabla f(\mathbf{y})\|_p \leq L_{p,p'} \|\mathbf{x} - \mathbf{y}\|_{p'} .$$

*If $p = p'$, we write $L_{p,p'}$ as $L_p$; if $p = p' = 2$, we write $L_{p,p'}$ as $L$ for convenience.*

**Theorem 1.** *Let the global objective function $f$ be $L$-smooth and each local objective functions $f_n$ be $L_{2,p}$-smooth and $L_p$-smooth ($p \geq 2$). Let $\nu \geq 0$ be a numerical constant. Suppose that there exist $\tilde{B}$ and $\tilde{D}$ such that for $0 \leq q \leq \nu - 1$,*

$$(\bar{\phi}_q)^2 \leq \tilde{B} \|\nabla f(\mathbf{x}_q)\|^2 + \tilde{D},$$

*and there exist $\{A_i\}$, $\{B_i\}$ and $D$ such that for $q \geq \nu$,*

$$(\bar{\phi}_q)^2 \leq \sum_{i=1}^{\nu} A_i (\bar{\phi}_{q-i})^2 + \sum_{i=0}^{\nu} B_i \|\nabla f(\mathbf{x}_{q-i})\|^2 + D.$$

*If $\gamma \leq \min\left\{ \frac{1}{LN}, \frac{1}{32 L_{2,p} N}, \frac{\sqrt{1 - \sum_{i=1}^{\nu} A_i}}{4 L_{2,p} \sqrt{\sum_{i=0}^{\nu} B_i}}, \frac{\sqrt{1 - \sum_{i=1}^{\nu} A_i}}{4 L_{2,p} \sqrt{\tilde{B}}}, \frac{1}{32 L_p N} \right\}$, then*

$$\frac{1}{Q} \sum_{q=0}^{Q-1} \|\nabla f(\mathbf{x}_q)\|^2 \leq \frac{5 F_0}{\gamma N Q} + c \gamma^2 L_{2,p}^2 \frac{1}{Q} \nu \tilde{D} + c \gamma^2 L_{2,p}^2 D,$$

*where $c = {}^{10}\!/_{(1 - \sum_{i=1}^{\nu} A_i)}$ is a numerical constant.*

### 3.2 Case Studies

In this section, we prove that the existing algorithms can be incorporated into our framework (Theorem 1) under some given assumptions (e.g., Assumption 3). The key results are summarized in Table 3. The specific choices of $A_i$, $B_i$ and $D$ are summarized in Table 7. Due to space limitations, we focus on GraBs, and deferred the details of other algorithms to Appendix F.

Table 3: Upper bounds of permutation-based SGD. Numerical constants and polylogarithmic factors are hided. To maintain consistency with GraBs, we use high-probability bounds for RR, FlipFlop and GraBs, instead of in-expectation bounds.

| | Alg. | Upper bound of $(\bar{\phi}_q)^2$ | Upper bound (of $\frac{1}{Q}\sum_{q=0}^{Q-1}\|\nabla f(\mathbf{x}_q)\|^2$) |
|---|---|---|---|
| AP | AP (Prop. 2) | $(\bar{\phi}_q)^2 \leq N^2\alpha^2\|\nabla f(\mathbf{x}_q)\|^2 + N^2\varsigma^2$ | $\frac{LF_0(1+\alpha)}{Q} + \left(\frac{LF_0 N\varsigma}{NQ}\right)^{\frac{2}{3}}$ [1] |
| IP | RR (Prop. 3) | $(\bar{\phi}_q)^2 \lesssim N\alpha^2\|\nabla f(\mathbf{x}_q)\|^2 + N\varsigma^2$ | $\frac{LF_0\left(1+\frac{\alpha}{\sqrt{N}}\right)}{Q} + \left(\frac{LF_0\sqrt{N}\varsigma}{NQ}\right)^{\frac{2}{3}}$ |
| | FlipFlop[2] (Prop. 4) | $(\bar{\phi}_q)^2 \lesssim N\alpha^2\|\nabla f(\mathbf{x}_q)\|^2 + N\varsigma^2$ | $\frac{LF_0\left(1+\frac{\alpha}{\sqrt{N}}\right)}{Q} + \left(\frac{LF_0\sqrt{N}\varsigma}{NQ}\right)^{\frac{2}{3}}$ |
| DP | GraB-proto (Prop. 1) | $(\bar{\phi}_q)^2 \lesssim (\bar{\phi}_{q-1})^2 + (N^2+\alpha^2)\|\nabla f(\mathbf{x}_{q-1})\|^2 + \varsigma^2$ | $\frac{\tilde{L}F_0+(L_{2,\infty}F_0\varsigma)^{\frac{2}{3}}}{Q} + \left(\frac{L_{2,\infty}F_0\varsigma}{NQ}\right)^{\frac{2}{3}}$ [3] |
| | GraB (Prop. 6) | $(\bar{\phi}_q)^2 \lesssim (\bar{\phi}_{q-1})^2 + (N^2+\alpha^2)\|\nabla f(\mathbf{x}_{q-1})\|^2 + (\bar{\phi}_{q-2})^2 + N^2\|\nabla f(\mathbf{x}_{q-2})\|^2 + \varsigma^2$ | $\frac{\tilde{L}F_0+(L_{2,\infty}F_0\varsigma)^{\frac{2}{3}}}{Q} + \left(\frac{L_{2,\infty}F_0\varsigma}{NQ}\right)^{\frac{2}{3}}$ [3] |
| | PairGraB (Prop. 7) | $(\bar{\phi}_q)^2 \lesssim (\bar{\phi}_{q-1})^2 + (N^2+\alpha^2)\|\nabla f(\mathbf{x}_{q-1})\|^2 + \varsigma^2$ | $\frac{\tilde{L}F_0+(L_{2,\infty}F_0\varsigma)^{\frac{2}{3}}}{Q} + \left(\frac{L_{2,\infty}F_0\varsigma}{NQ}\right)^{\frac{2}{3}}$ [3] |

[1] For a clearer comparison with other bounds, we keep "$N$" in both numerators and denominators in the bounds of AP.
[2] For FlipFlop, we bound the order error over a period of $2N$ instead of $N$ (Appendix F.3). Rajput et al. [2022] gave the upper bound of FlipFlop for quadratic functions (in contrast, this work gives the bound for non-convex objectives).
[3] For GraBs (that is, GraB-proto, GraB, PairGraB), $\tilde{L} = L + L_{2,\infty}(1+\alpha) + L_\infty$.

**Assumption 3.** *There exist nonnegative constants $\varsigma$ and $\alpha$ such that for any $n \in \{0, 1, \ldots, N-1\}$,*

$$\|\nabla f_n(\mathbf{x}) - \nabla f(\mathbf{x})\|^2 \leq \alpha^2\|\nabla f(\mathbf{x})\|^2 + \varsigma^2, \ \forall \mathbf{x} \in \mathbb{R}^d.$$

**Analysis of GraB-proto.** To clearly present our theory and analyze GraBs, we focus on GraB-proto, the simplified version of the original GraB. The key characteristic of GraB-proto (and other variants) is that the example order depends on the example order of previous epochs. Thus, the goal is *to find the relation between $\bar{\phi}_q$ and $\bar{\phi}_{q-1}$*. Specifically, for all $q \geq 1$ and $n \in [N]$,

$$\phi_q^n \leq 2L_\infty N\|\mathbf{x}_q - \mathbf{x}_{q-1}\|_\infty + \max_{n\in[N]}\left\|\sum_{i=0}^{n-1}\left(\nabla f_{\pi_q(i)}(\mathbf{x}_{q-1}) - \nabla f(\mathbf{x}_{q-1})\right)\right\|_\infty. \tag{3}$$

(i) First, note that the first term on the right hand side in Equation (3) is the well-studied "parameter deviation" [Mishchenko et al., 2020], whose upper bound is provided in Lemma 6. (ii) Second, since in GraB-proto, $\{\mathbf{z}_{\pi(i)}\}$ correspond to $\{\nabla f_{\pi_{q-1}(i)}(x_{q-1}) - \nabla f(x_{q-1})\}$ and $\{\mathbf{z}_{\pi'(i)}\}$ correspond to $\{\nabla f_{\pi_q(i)}(x_{q-1}) - \nabla f(x_{q-1})\}$, we can apply Equation (2) to the second term in Equation (3) (we denote this term as $T_2$):

$$T_2 \text{ in (3)} \leq \frac{1}{2}\max_{n\in[N]}\left\|\sum_{i=0}^{n-1}\left(\nabla f_{\pi_{q-1}(i)}(\mathbf{x}_{q-1}) - \nabla f(\mathbf{x}_{q-1})\right)\right\|_\infty + \frac{1}{2}CG_{q-1} = \frac{1}{2}\bar{\phi}_{q-1} + \frac{1}{2}CG_{q-1},$$

where $C = \mathcal{O}\left(\log\left(\frac{dN}{\delta}\right)\right) = \tilde{\mathcal{O}}(1)$ is from Alweiss et al. [2021, Theorem 1.1] and $G_{q-1} := \sqrt{\alpha^2\|\nabla f(\mathbf{x}_{q-1})\|^2 + \varsigma^2}$. Here, we use Assumption 3 to scale the vector length to be no greater than 1. Now, combining (i) and (ii) gives the relation in Proposition 1.

**Proposition 1** (GraB-proto). *Suppose that Assumption 3 holds. If each $f_n$ is $L_\infty$-smooth and $\gamma \leq \frac{1}{32L_\infty N}$, we obtain that, for $q = 0$, $(\bar{\phi}_0)^2 \leq N^2\alpha^2\|\nabla f(\mathbf{x}_0)\|^2 + N^2\varsigma^2$, and for $q \geq 1$, with probability at least $1 - \delta$,*

$$(\bar{\phi}_q)^2 \leq \frac{3}{4}(\bar{\phi}_{q-1})^2 + \left(\frac{1}{50}N^2 + C^2\alpha^2\right)\|\nabla f(\mathbf{x}_{q-1})\|^2 + C^2\varsigma^2,$$

where $C = \mathcal{O}\left(\log\left(\frac{dN}{\delta}\right)\right) = \tilde{\mathcal{O}}(1)$. *Applying Theorem 1 and tuning the step size, we obtain that, with probability at least $1 - Q\delta$,*[3]

$$\frac{1}{Q}\sum_{q=0}^{Q-1}\|\nabla f(\mathbf{x}_q)\|^2 = \mathcal{O}\left(\frac{(L + L_{2,\infty}(1+\alpha) + L_\infty)F_0 + (L_{2,\infty}F_0\varsigma)^{\frac{2}{3}}}{Q} + \left(\frac{L_{2,\infty}F_0C\varsigma}{NQ}\right)^{\frac{2}{3}}\right).$$

**Analyses of GraB and PairGraB.** See Table 3. (i) First, the upper bounds of GraB and PairGraB are almost identical to that of GraB-proto. (ii) Second, the $\bar\phi_q$ of GraB is affected by the factors from the previous two epochs (such as $\bar\phi_{q-1}$ and $\bar\phi_{q-2}$). This is because GraB uses the average of the stale gradients for centering, while PairGraB is free of centering (see Appendix C). Formal statements of GraB and PairGraB are in Propositions 6 and 7 (in Appendix F).

## 4 Federated Learning

**Setup.** In this section, we adapt our theory on example ordering in SGD for client ordering in FL. For FL, we consider the same problem as that in SGD (that is, Equation 1). Notably, in FL, the local objective functions represent the clients in FL. We focus on FL with regularized client participation (regularized-participation FL), where each client participate once before any client is reused [Wang and Ji, 2022]. More concretely, see Algorithm 2. During each epoch, it selects $S$ clients at a time from the permuted clients (under the permutation $\pi$) to complete a round of federated training, until all the clients have participated. Pay attention that one "epoch" may include multiple "rounds". At the end of each epoch, it produces the next-epoch permutation by some permuting

---

**Algorithm 2:** Regularized-participation FL

**Input:** $\pi_0, \mathbf{x}_0$; **Output:** $\{\mathbf{x}_q\}$
1 **for** $q = 0, 1, \ldots, Q-1$ **do**
2     $\mathbf{w} \leftarrow \mathbf{x}_q$
3     **for** $n = 0, 1, \ldots, N-1$ **do**
4        Initialize $\mathbf{x}_{q,0}^n \leftarrow \mathbf{w}$
5        **for** $k = 0, 1, \ldots, K-1$ **do**
6           $\mathbf{x}_{q,k+1}^n \leftarrow \mathbf{x}_{q,k}^n - \gamma\nabla f_{\pi_q(n)}(\mathbf{x}_{q,k}^n)$
7        $\mathbf{p}_q^n \leftarrow \mathbf{x}_{q,0}^n - \mathbf{x}_{q,K}^n$
8        **if** $(n+1) \mod S = 0$ **then**
9           $\mathbf{w} \leftarrow \mathbf{w} - \frac{1}{S}\sum_{s=0}^{S-1}\mathbf{p}_q^{n-s}$
10    $\mathbf{x}_{q+1} \leftarrow \mathbf{x}_q - \eta(\mathbf{x}_q - \mathbf{w})$
11    $\pi_{q+1} \leftarrow \texttt{Permute}(\cdots)$

---

algorithm. Here, we also consider the global amplification [Wang and Ji, 2022] (see Line 10). Considering that we mainly study the client ordering of FL in this paper, we use Gradient Descent (GD) as the local solver of FL (see Lines 5 and 6) for simplicity. We assume $N \mod S = 0$.

### 4.1 Main Theorem

Theorem 2 gives our main framework for regularized-participation FL. See Appendix K for the additional extension for Theorem 2.

**Theorem 2.** *Let the global objective function $f$ be $L$-smooth and each local objective functions $f_n$ be $L_{2,p}$-smooth and $L_p$-smooth ($p \geq 2$). Suppose that Assumption 3 holds. Suppose $N \mod S = 0$. Let $\nu \geq 0$ be a numerical constant. Suppose that there exist $\tilde B$ and $\tilde D$ such that for $0 \leq q \leq \nu - 1$,*

$$(\bar\varphi_q)^2 \leq \tilde B\|\nabla f(\mathbf{x}_q)\|^2 + \tilde D,$$

*and there exist $\{A_i\}$, $\{B_i\}$ and $D$ such that for $q \geq \nu$,*

$$(\bar\varphi_q)^2 \leq \sum_{i=1}^{\nu}A_i(\bar\varphi_{q-i})^2 + \sum_{i=0}^{\nu}B_i\|\nabla f(\mathbf{x}_{q-i})\|^2 + D.$$

*If $\gamma \leq \min\left\{\frac{1}{\eta LKN\frac{1}{S}}, \frac{1}{32L_{2,p}KN\frac{1}{S}(1+\alpha)}, \frac{\sqrt{1-\sum_{i=1}^{\nu}A_i}}{4L_{2,p}K\frac{1}{S}\sqrt{\sum_{i=0}^{\nu}B_i}}, \frac{\sqrt{1-\sum_{i=1}^{\nu}A_i}}{4L_{2,p}K\frac{1}{S}\sqrt{\tilde B}}, \frac{1}{32L_pKN\frac{1}{S}}\right\}$, then*

$$\frac{1}{Q}\sum_{q=0}^{Q-1}\|\nabla f(\mathbf{x}_q)\|^2 \leq \frac{5F_0}{\gamma\eta KN\frac{1}{S}Q} + c\gamma^2 L_{2,p}^2 K^2\frac{1}{S^2}\frac{1}{Q}\nu\tilde D + 2\gamma^2 L_{2,p}^2 K^2\varsigma^2 + c\gamma^2 L_{2,p}^2 K^2\frac{1}{S^2}D \quad (4)$$

*where $c = {10}/{(1-\sum_{i=1}^{\nu}A_i)}$ is a numerical constant.*

---

[3]To main consistency with Lu et al. [2022a] and Cooper et al. [2023], we use a failure probability of $Q\delta$ rather than $\delta$. Appendix F.9 shows that the framework can provide bounds that hold with probability $1 - \delta$.

We note that the third term (containing $\varsigma$) on the right hand side in Equation (4) is not subsumed into the assumptions of the order error. This is because this term comes from the local updates, which is affected by the example order within each client, rather than by the client order in FL.

## 4.2 Case Studies

Table 4: Upper bounds of FL with regularized client participation. Numerical constants and poly-logarithmic factors are hided.

| Alg. | Upper bound of $(\bar{\varphi}_q)^2$ | Upper bound (of $\frac{1}{Q}\sum_{q=0}^{Q-1}\|\nabla f(\mathbf{x}_q)\|^2$) |
|---|---|---|
| FL-AP[1] | – | $\frac{LF_0}{Q} + \left(\frac{LF_0 S\varsigma}{NQ}\right)^{\frac{2}{3}} + \left(\frac{LF_0 N\varsigma}{NQ}\right)^{\frac{2}{3}}$[2] |
| FL-AP (Prop. 8) | $(\bar{\varphi}_q)^2 \leq N^2\alpha^2 \|\nabla f(\mathbf{x}_q)\|^2 + N^2\varsigma^2$ | $\frac{LF_0(1+\alpha)}{Q} + \left(\frac{LF_0 S\varsigma}{NQ}\right)^{\frac{2}{3}} + \left(\frac{LF_0 N\varsigma}{NQ}\right)^{\frac{2}{3}}$[2] |
| FL-RR (Prop. 9) | $(\bar{\varphi}_q)^2 \lesssim N\alpha^2 \|\nabla f(\mathbf{x}_q)\|^2 + N\varsigma^2$ | $\frac{LF_0(1+\alpha)}{Q} + \left(\frac{LF_0 S\varsigma}{NQ}\right)^{\frac{2}{3}} + \left(\frac{LF_0\sqrt{N}\varsigma}{NQ}\right)^{\frac{2}{3}}$ |
| FL-GraB (Prop. 10) | $(\bar{\varphi}_q)^2 \lesssim (\bar{\varphi}_{q-1})^2 + \left(N^2+\alpha^2\right)\|\nabla f(\mathbf{x}_{q-1})\|^2$ $+S^2\varsigma^2 + \varsigma^2$ | $\frac{\tilde{L}F_0 + (L_{2,\infty}F_0\varsigma)^{\frac{2}{3}}}{Q} + \left(\frac{L_{2,\infty}F_0 S\varsigma}{NQ}\right)^{\frac{2}{3}} + \left(\frac{L_{2,\infty}F_0 S\varsigma}{NQ}\right)^{\frac{2}{3}}$[3] |

[1] Wang and Ji [2022]'s Theorem 3.1. It uses a stronger assumption, equivalent to Assumption 3 with $\alpha = 0$.
[2] For a clearer comparison with other bounds, we keep "$N$" in both numerators and denominators in the bounds of FL-AP.
[3] For FL-GraB, $\tilde{L} = L + L_{2,\infty}(1+\alpha) + L_\infty$.

Our unified framework covers regularized-participation FL with AP (FL-AP), with RR (FL-RR) and with GraB (FL-GraB). They correspond to AP, RR and PairGraB in SGD, respectively. In particular, we propose FL-GraB (see Appendix C) by replacing the true gradients in GraBs with the pseudo-gradients (that is, $\mathbf{p}_q^n$ in Algorithm 2) of the local objective functions to generate the "good" permutation as the training order of clients. The key results are in Table 4.

**Analysis of FL-GraB.** See Table 4. The main difference (in convergence rates) lies in the last term. The upper bound of FL-GraB $\tilde{\mathcal{O}}((\frac{1}{NQ})^{\frac{2}{3}})$ dominants those of the other algorithms in terms of the number of epochs $Q$ and the number of clients $N$. This conclusion is aligned with that in SGD. Notably, the changes (from $\bar{\phi}_q$ to $\bar{\varphi}_q$, and from true gradients to pseudo-gradients) make the analysis of FL-GraB more complex than that of PairGraB in SGD.

## 5 Experiments

In this section, we use the following simulated experiments to validate the theory. Refer to Lu et al. [2022a] and Cooper et al. [2023] for the experiments of SGD on real data sets; refer to Appendix I for the experiments of FL on real data sets. The details of the experiments can be found in the code available at https://github.com/liyipeng00/ordering.

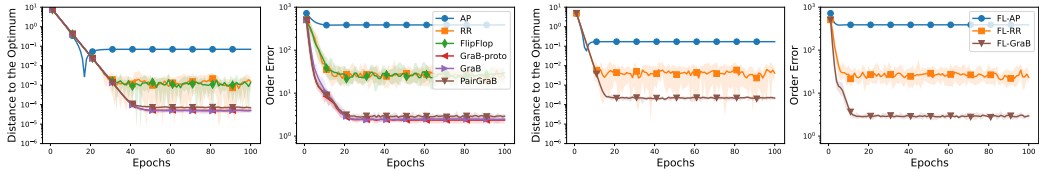

Figure 1: The experiments on simulated data. Shaded areas show the min-max values across 10 different random seeds. The left two figures are for SGD; the right two figures are for FL. For both SGD and FL, $\gamma$ is set to be the same for the algorithms; $N = 1000$. For FL, $K = 5$ and $S = 2$.

We use the one-dimensional functions $f_n(\mathbf{x}) = ((0.5 + a_n)\,\mathbb{1}_{\mathbf{x}<0} + a_n\mathbb{1}_{\mathbf{x}\geq 0})\,\mathbf{x}^2 + b_n\mathbf{x}$ as the local objective functions. We model $a_n \sim \mathcal{N}(0.5, 1)$ and $b_n \sim \mathcal{N}(0, 1)$ ($\mathcal{N}$ is the normal distribution). Here, $a_n$ and $b_n$ control the heterogeneity of the local objective functions. The observations on Figure 1 validate our theoretical results:

- The distance to the optimum (that is, $\|\mathbf{x} - \mathbf{x}^*\|$) and the order errors have the same trend, which validates that the order errors can measure the convergence rate.

- The performances of FlipFlop and RR are close. This does not contradict the conclusion in Rajput et al. [2022], which shows FlipFlop is better than RR on quadratic functions, given that the functions used in our simulation are strongly convex but not quadratic.

- The performances of PairGraB and GraB are close, and better than RR in both SGD and FL.

## 6   Conclusion

We study example ordering in permutation-based SGD and client ordering in regularized-participation FL. For SGD, we propose a more general assumption (Assumption 2) to bound the order error. Using it, we develop a unified framework for permutation-based SGD with arbitrary permutations of examples, including AP, IP (including RR, FlipFlop) and DP (including GraBs). Furthermore, we develop a unified framework for regularized-participation FL with arbitrary permutations of clients, including FL-AP, FL-RR and FL-GraB.

Limitations and possible future directions: First, explore new algorithm for SGD (no new algorithms are proposed for SGD in this work). Second, extend the framework to more practical scenarios for FL (our theory is for FL with regularized participation). Third, study example ordering in local updates for FL (we use GD as the local solver). Forth, explore the combination of permutation-based SGD and other algorithms (see Appendix L for an example combining permutation-based SGD with online learning).

## Acknowledgments and Disclosure of Funding

This work was supported in part by the National Natural Science Foundation of China under Grants 62371059 and 62571297, in part by the Fundamental Research Funds for the Central Universities under Grant 2242022k60006, and in part by the Shenzhen Science and Technology Program under Grant JCYJ20240813112301003.

We thank the anonymous Reviewers of ICML 2025 and NeurIPS 2025 for their insightful suggestions.

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

# Appendix

# A  Related Works

## A.1  Related Works

**Convergence analyses of permutation-based SGD.** Up to now, there have been a wealth of works analyzing the convergence of IG, SO and RR (the most classic permutation-based SGD algorithms): Nagaraj et al. [2019], Ahn et al. [2020], Mishchenko et al. [2020], Nguyen et al. [2021], Yun et al. [2021], Yu and Li [2023], Cai and Diakonikolas [2024], Liu and Zhou [2024] and Koloskova et al. [2024] analyzed their upper bounds and Safran and Shamir [2020], Safran and Shamir [2021], Rajput et al. [2020], Yun et al. [2022], Cha et al. [2023] and Kim et al. [2025b] analyzed their lower bounds. There are also some works analyzing their variants [Mishchenko et al., 2022; Malinovsky et al., 2023c; Liu and Zhou, 2024; Cai et al., 2024; Kim et al., 2025a], especially in the FL settings [Mishchenko et al., 2022; Yun et al., 2022; Horváth et al., 2022; Sadiev et al., 2023, 2024; Malinovsky et al., 2023b; Li and Lyu, 2023, 2025].

Several works [Rajput et al., 2022; Lu et al., 2022b; Mohtashami et al., 2022] started to explore the other permutation-based SGD algorithms beyond the simple IG, SO and RR. Rajput et al. [2022] proposed the FlipFlop variants (of IG, SO and RR), and proved that these FlipFlop variants are better than their corresponding original algorithms on the quadratic functions. In this paper, we only studied the FlipFlop variant of RR, introduced in Appendix F.3. Recently, Chae et al. [2024] extended the FlipFlop algorithm to stochastic extragradient methods for convex-concave objective functions. Lu et al. [2022b] and Mohtashami et al. [2022] proposed the Greedy Ordering algorithms to select the examples greedily to minimize the order error. However, the Greedy Ordering algorithms lack theoretical justification and suffer from non-trivial memory and computation overhead [Lu et al., 2022a]. Lu et al. [2022a] found that the goal to minimize the order error is aligned with the goal of herding [Welling, 2009], and proposed GraB based on the theory in Harvey and Samadi [2014] and Alweiss et al. [2021]. PairGraB first appeared in the public code of Lu et al. [2022a], and was later formally introduced in Cooper et al. [2023]. GraB has also been applied in distributed optimization (not FL) [Cooper et al., 2023] and multi-objective optimization [Yang and Kwok, 2025].

The most relevant works are the unified analyses of permutation-based SGD [Lu et al., 2022b; Mohtashami et al., 2022; Koloskova et al., 2024]. They all rely on Assumption 1 (they may consider an interval of arbitrary length, not necessarily an epoch); this assumption has been widely adopted in the subsequent works [Islamov et al., 2024; Li and Huang, 2024] for other settings beyond this paper. See Section 2 for the comparison with Lu et al. [2022b] and Koloskova et al. [2024].

**Convergence analyses of FL with regularized client participation.** The convergence of vanilla client participation pattern (the selection of clients across different rounds are independent) has been analyzed in the early works [Li et al., 2020; Karimireddy et al., 2020; Yang et al., 2021]. The convergence of regularized client participation pattern [Eichner et al., 2019] was initially analyzed in Wang and Ji [2022], Cho et al. [2023], Malinovsky et al. [2023a] and Demidovich et al. [2025], where Wang and Ji [2022] and Cho et al. [2023] considered regularized-participation FL with AP (FL-AP) and Malinovsky et al. [2023a] considered regularized-participation FL with RR (FL-RR). This work aims to develop a unified framework that includes these cases. See Tables 2 and 4 for the comparison with Wang and Ji [2022], Cho et al. [2023] and Malinovsky et al. [2023a]. Importantly, this work focuses on client ordering in FL with regularized participation, which is different from the studies of FL with arbitrary participation [Wang and Ji, 2022, 2024; Sun et al., 2025] and client sampling [Cho et al., 2022; Horváth et al., 2022].

Assumption 2 closely resembles the assumptions used in Gorbunov et al. [2020, 2021] in different contexts.

## A.2  Reformulating Existing Convergence Rates in Our Setting

In the main body, to facilitate comparison, we have reformulated the existing convergence rates to fit our setting. Here, we provide the details of how these rates are reformulated.

**The rates from Lu et al. [2022b].** We reformulate the results of Lu et al. [2022b] as the convergence rates for AP and RR reported in Table 1.

The correspondence between Lu et al. [2022b]'s notation and ours is as follows: $T$ in Lu et al. [2022b] corresponds to $NQ$ in this paper; $L$ corresponds to $L$; $\Delta$ corresponds to $F_0$; $A$ corresponds to $\varsigma$; $B$ corresponds to $\alpha$; $n$ corresponds to $N$; $\alpha$ corresponds to $\gamma$.

AP. Substituting the results of Lu et al. [2022b]'s Proposition 1 into Lu et al. [2022b]'s Theorem 1, and using the correspondence, we obtain

$$T = \tilde{\mathcal{O}}\left(\frac{L\Delta C}{\epsilon^3}\right) \stackrel{\text{Prop. 1}}{\Longrightarrow} T = \tilde{\mathcal{O}}\left(\frac{L\Delta nA}{\epsilon^3}\right),$$

where $B$ is set to $B = 0$ for comparison (implying that $\Phi = 0$). This complexity (number of iterations to achieve accuracy $\epsilon$) is equivalent to the convergence rate of $\tilde{\mathcal{O}}\left(\left(\frac{L\Delta nA}{T}\right)^{\frac{2}{3}}\right)$. Based on the notation correspondence, we obtain the rate of $\tilde{\mathcal{O}}\left(\left(\frac{LF_0 N\varsigma}{NQ}\right)^{\frac{2}{3}}\right)$, which matches the error term (the last term) in the convergence rate of AP in Table 1. The constraint of the step size for AP is not clearly given in Lu et al. [2022b]. So, for the optimization term (determined by the constraint of the step size), we use the one recovered by Koloskova et al. [2024, Table 1].

RR. Substituting the results of Lu et al. [2022b]'s Proposition 3 into Lu et al. [2022b]'s Theorem 1, following the steps as those in AP, using the constraint $\alpha \lesssim \frac{1}{Ln}$ (that is, $\gamma \lesssim \frac{1}{LN}$ in our setting) with our Lemma 1, and using the notation correspondence, we can obtain the rate of $\tilde{\mathcal{O}}\left(\frac{LF_0}{Q} + \left(\frac{LF_0\sqrt{N}\varsigma}{NQ}\right)^{\frac{2}{3}}\right)$.

**The rates from Koloskova et al. [2024].** We reformulate the results of Koloskova et al. [2024] as the convergence rates for AP and RR reported in Table 1, and the rate for classic SGD in Table 8.

The correspondence between Koloskova et al. [2024]'s notation and ours is as follows: $L$ in Koloskova et al. [2024] corresponds to $L$ in this paper; $F_0$ corresponds to $F_0$; $T$ denotes the number of iterations and corresponds to $NQ$; $n$ corresponds to $N$; $\sigma_{\text{SGD}}$ corresponds to $\varsigma$.

Using the notation correspondence, it is convenient to translate Koloskova et al. [2024]'s bounds into ours. In fact, Koloskova et al. [2024] gave the same rate for IG, SO and RR (their Examples 3.2, 3.3 and 3.4). We note that this rate also applies to AP.

**The rates from Lu et al. [2022a].** We reformulate the results of Lu et al. [2022a] as the convergence rate for GraB.

In the proof of Theorem 3 in Lu et al. [2022a], we can find the inequality

$$\frac{1}{K}\sum_{k=1}^{K}\|\nabla f(\boldsymbol{w}_k)\|^2 \lesssim \frac{f(\boldsymbol{w}_1) - f^*}{\alpha nK} + \frac{\alpha^2 n^2 \varsigma^2 L_{2,\infty}^2}{K} + \alpha^2 A^2 \varsigma^2 L_{2,\infty}^2.$$

The correspondence between Lu et al. [2022a]'s notation and ours is as follows: $f(\boldsymbol{w}_1) - f^*$ in Lu et al. [2022a] corresponds to $F_0$ in this paper; $\alpha$ corresponds to $\gamma$; $n$ corresponds to $N$; $K$ corresponds to $Q$; $L$, $L_\infty$ and $L_{2,\infty}$ correspond to $L$, $L_\infty$ and $L_{2,\infty}$ respectively; $\varsigma$ corresponds to $\varsigma$; $A$ corresponds to $C$; $\frac{1}{K}\sum_{k=1}^{K}\|\nabla f(\boldsymbol{w}_k)\|^2$ corresponds to $\frac{1}{Q}\sum_{q=0}^{Q-1}\|\nabla f(\mathbf{x}_q)\|^2$. Using the notation correspondence, we obtain

$$\frac{1}{Q}\sum_{q=0}^{Q-1}\|\nabla f(\mathbf{x}_q)\|^2 = \tilde{\mathcal{O}}\left(\frac{F_0}{\gamma NQ} + \gamma^2 L_{2,\infty}^2 N^2 \varsigma^2 \frac{1}{Q} + \gamma^2 L_{2,\infty}^2 \varsigma^2\right).$$

Then, using the constraint of the step size in Lu et al. [2022a]'s Theorem 3, and tuning the step size with Lemma 1, we can obtain the same rate as this paper.

**The rates from Cooper et al. [2023].** We reformulate the results of Cooper et al. [2023] as the convergence rate for PairGraB. CD-GraB [Cooper et al., 2023] is an extension of PairGraB. For Theorem 2, after setting $\sigma = 0$ and $m = 1$, and using the notation correspondence (that is, $F_1$ in Lu et al. [2022a] corresponds to $F_0$ in this paper, $L$ to $L$, $L_\infty$ to $L_\infty$, $L_{2,\infty}$ to $L_{2,\infty}$, $T$ to $Q$, $\tilde{A}$ to $\tilde{C}$ and $n$ to $N$), we can obtain the same rate as this paper.

**The rates from Cho et al. [2023].** We reformulate the results of Cho et al. [2023] as the convergence rates for FL-AP reported in Table 2.

After comparing Cho et al. [2023]'s Algorithm 1 (their LocalSGD case) with ours, we find that their setting is more general than ours, so we need to adapt their results to match ours: Setting $N = \frac{M}{K}$

and $\sigma = 0$ in their Theorem 2 gives

$$\mathbb{E}\left[F(\mathbf{w}^{(K,0)}) - F^*\right] = \tilde{\mathcal{O}}\left(\frac{\overline{K}^2(F(\mathbf{w}^{(0,0)}) - F^*)}{MT^2} + \frac{\kappa^2\overline{K}^2\alpha^2}{\mu T^2} + \frac{\kappa^2\nu^2}{\mu N^2 T^2}\right).$$

In addition, comparing their model update rule given in Section 3 (Problem Formulation) and Equation (37), we notice that a factor of $\frac{1}{N}$ is missing in Equation (37). The model rule of Equation (37) is used in their Equation (54), causing the missing of the factor of $N^2$ in the term caused by local updates in their bounds. As a result, we obtain

$$\mathbb{E}\left[F(\mathbf{w}^{(K,0)}) - F^*\right] = \tilde{\mathcal{O}}\left(\frac{\overline{K}^2(F(\mathbf{w}^{(0,0)}) - F^*)}{MT^2} + \frac{\kappa^2\overline{K}^2\alpha^2}{\mu T^2} + \frac{\kappa^2{\color{red}N^2}\nu^2}{\mu N^2 T^2}\right),$$

where we highlight the missing factor $N^2$ in red.

After the adapting, the correspondence between Cho et al. [2023]'s notation and ours is as follows: $\overline{K}$ in Cho et al. [2023] corresponds to $\frac{N}{S}$ in the main body of this paper; $F(\mathbf{w}^{(K,0)}) - F^*$ corresponds to $F_Q$; $F(\mathbf{w}^{(0,0)}) - F^*$ corresponds to $F_0$; $T$ corresponds to $\frac{N}{S}Q$; $N$ corresponds to $S$; $M$ corresponds to $N$; both $\alpha$ and $\nu$ correspond to $\varsigma$. Using the notation correspondence, we obtain

$$\mathbb{E}\left[F_Q\right] = \tilde{\mathcal{O}}\left(\frac{F_0}{NQ^2} + \frac{L^2S^2\varsigma^2}{\mu^3 N^2 Q^2} + \frac{L^2\varsigma^2}{\mu^3 Q^2}\right).$$

**The rates from Malinovsky et al. [2023a].** We reformulate the results of Malinovsky et al. [2023a] as the convergence rates for FL-RR reported in Table 2.

Malinovsky et al. [2023a] uses RR as the local solver while this paper uses GD, so we need to adapt their results to fit our setting: Setting $\sigma_\star = \tilde{\sigma}_\star$ in their Theorem 6.1 gives

$$\mathbb{E}\left\|x_T - x_\star\right\|^2 \lesssim (1 - \gamma\mu)^{NRT}\left\|x_0 - x_\star\right\|^2 + \frac{\gamma^2}{\mu}\left(\frac{LMN^2}{C^2}\tilde{\sigma}_\star^2 + LN^2\tilde{\sigma}_\star^2 + LN\tilde{\sigma}_\star^2\right)$$

$$\lesssim (1 - \gamma\mu)^{NRT}\left\|x_0 - x_\star\right\|^2 + \gamma^2\frac{1}{\mu}LMN^2\frac{1}{C^2}\sigma_\star^2 + \gamma^2\frac{1}{\mu}LN^2\tilde{\sigma}_\star^2$$

$$\lesssim \exp\left(-\gamma\mu NRT\right)\left\|x_0 - x_\star\right\|^2 + \gamma^2\frac{1}{\mu}LMN^2\frac{1}{C^2}\sigma_\star^2 + \gamma^2\frac{1}{\mu}LN^2\tilde{\sigma}_\star^2.$$

Then, using $\frac{\mu}{2}\left\|x - x_*\right\|^2 \leq f(x) - f(x_\star) \leq \frac{L}{2}\left\|x - x_*\right\|^2$, we obtain

$$f(x_T) - f(x_\star) \lesssim \frac{L}{\mu}(f(x_0) - f_\star)\exp\left(-\gamma\mu NRT\right) + \gamma^2\frac{1}{\mu}LMN^2\frac{1}{C^2}\sigma_\star^2 + \gamma^2\frac{1}{\mu}LN^2\tilde{\sigma}_\star^2.$$

After the adapting, the correspondence between Malinovsky et al. [2023a]'s notation and ours is as follows: $L$ in Malinovsky et al. [2023a] corresponds to $L$ in this paper; $\mu$ corresponds to $\mu$; $f(x_T) - f(x_\star)$ corresponds to $F_T$; $f(x_0) - f(x_\star)$ corresponds to $F_0$; $\gamma$ corresponds to $\gamma$; $C$ corresponds to $S$; $M$ corresponds to $N$; $N$ corresponds to $K$; $R$ corresponds to $\frac{N}{S}$; $T$ corresponds to $Q$; $\tilde{\sigma}_\star^2$ corresponds to $\varsigma$. Using the notation correspondence, we obtain

$$F_Q \leq \frac{L}{\mu}F_0\exp\left(-\gamma\mu KN\frac{1}{S}Q\right) + \gamma^2\frac{L^2K^2N\varsigma^2}{\mu S^2} + \gamma^2\frac{L^2K^2\varsigma^2}{\mu}.$$

In addition, we need the constraint of the step size $\gamma \leq \frac{1}{L}$. Then, tuning the step size with Koloskova et al. [2020, Lemma 15], we obtain the reported bound in Table 2.

**The rates from Wang and Ji [2022].** We reformulate the results of Wang and Ji [2022] as the convergence rate for FL-AP reported in Table 2 and the rate for FL with independent participation in Table 9.

Wang and Ji [2022]'s framework covers multiple participation patterns (including the regularized participation and independent participation), and uses SGD as the local solver.

For regularized participation, we set $\tilde{\delta}(P) = 0$ (see Wang and Ji [2022]'s Proposition 4.2) and $\sigma = 0$ in Wang and Ji [2022]'s Theorem 3.1, which gives

$$\min_t \mathbb{E}\left[\|\nabla f(\mathbf{x}_t)\|^2\right] = \mathcal{O}\left(\frac{\mathcal{F}}{\gamma\eta IT} + \gamma^2 L^2 I^2 \tilde{\nu}^2 + \gamma^2 L^2 I^2 P^2 \tilde{\beta}^2\right).$$

The correspondence between Wang and Ji [2022]'s notation and ours is as follows: $\mathcal{F}$ in Wang and Ji [2022] corresponds to $F_0$ in this paper; $\gamma$ corresponds to $\gamma$; $\eta$ corresponds to $\eta$; $I$ corresponds to $K$; $P$ corresponds to $\frac{N}{S}$; $T$ corresponds to $N\frac{1}{S}Q$; both $\tilde{\beta}$ and $\tilde{\nu}$ correspond to $\varsigma$ (see Wang and Ji [2022]'s Section 4.1, Discussion on $\tilde{\beta}^2$ and $\tilde{\nu}$: Decomposition of Divergence); $\min_t \mathbb{E}\left[\|\nabla f(\mathbf{x}_t)\|^2\right]$ corresponds to $\min_t \mathbb{E}\left[\|\nabla f(\mathbf{y}_t)\|^2\right]$. Using the notation correspondence, we obtain the bound

$$\min_t \mathbb{E}\left[\|\nabla f(\mathbf{y}_t)\|^2\right] = \mathcal{O}\left(\frac{F_0}{\gamma\eta KN\frac{1}{S}Q} + \gamma^2 L^2 K^2 \varsigma^2 + \gamma^2 L^2 K^2 N^2 \frac{1}{S^2}\varsigma^2\right),$$

where $\mathbf{y}_t$ denotes the model parameter vector in iteration $t$ (see Appendix K). Then, setting $\eta = 1$ and tuning the step size with Lemma 1, we obtain the reported bound in Table 2.

For independent participation, we set $\tilde{\delta}(P) = \Theta(\frac{N^2 d^2}{P})$ (see Wang and Ji [2022]'s Proposition 4.7) and $\sigma = 0$ in Wang and Ji [2022]'s Theorem 3.1, which gives

$$\min_t \mathbb{E}\left[\|\nabla f(\mathbf{x}_t)\|^2\right] = \tilde{\mathcal{O}}\left(\frac{\mathcal{F}}{\gamma\eta IT} + \gamma^2 L^2 I^2 \tilde{\nu}^2 + \gamma^2 L^2 I^2 P^2 \tilde{\beta}^2 + \frac{N^2 d^2}{P}\right).$$

Using the notation correspondence, we obtain the bound

$$\min_t \mathbb{E}\left[\|\nabla f(\mathbf{y}_t)\|^2\right] = \tilde{\mathcal{O}}\left(\frac{F_0}{\gamma\eta KN\frac{1}{S}Q} + \gamma^2 L^2 K^2 \varsigma^2 + \gamma^2 L^2 K^2 P^2 \varsigma^2 + \frac{N^2 \varsigma^2}{P}\right).$$

For comparison, we set $\eta = 1$ and $P = \Theta(\frac{1}{\gamma LK}S)$. This gives

$$\min_t \mathbb{E}\left[\|\nabla f(\mathbf{y}_t)\|^2\right] = \tilde{\mathcal{O}}\left(\frac{F_0}{\gamma KN\frac{1}{S}Q} + \gamma^2 L^2 K^2 \varsigma^2 + \varsigma^2 + \gamma LK\frac{1}{S}N^2\varsigma^2\right).$$

Notably, under the same setting ($\eta = 1$ and $P = \Theta(\frac{1}{\gamma LK}S)$), there is a non-vanishing term in Wang and Ji [2022] when $\gamma \to 0$. Then, following the same steps as Proposition 14, tuning the step size with Lemma 1, we obtain

$$\min_t \mathbb{E}\left[\|\nabla f(\mathbf{y}_t)\|^2\right] = \tilde{\mathcal{O}}\left(\frac{LF_0}{T} + \left(\frac{LF_0\varsigma}{T}\right)^{\frac{2}{3}} + \left(\frac{LF_0 N^2 \varsigma^2}{ST}\right)^{\frac{1}{2}} + \varsigma^2\right).$$

**The rates from Yang et al. [2021].** We reformulate the results of Yang et al. [2021] as the convergence rate for FL with independent participation in Table 9.

Yang et al. [2021] uses SGD as the local solver while this paper uses GD, so we adapt their results to fit out setting: Setting $\sigma_L = 0$ in Theorem 2 (Strategy 1) gives

$$\min_{t \in [T]} \mathbb{E}[\|\nabla f(\mathbf{x}_t)\|^2] \lesssim \frac{f_0 - f_*}{\eta\eta_L KT} + \frac{LK\eta\eta_L}{n}\sigma_G^2 + K^2\eta_L^2 L^2 \sigma_G^2 + \frac{K^3\eta\eta_L^3 L^3}{n}\sigma_G^2.$$

Then, using $\eta_L\eta KL \lesssim 1$, we obtain

$$\min_{t \in [T]} \mathbb{E}[\|\nabla f(\mathbf{x}_t)\|^2] \lesssim \frac{f_0 - f_*}{\eta\eta_L KT} + \frac{LK\eta\eta_L}{n}\sigma_G^2 + K^2\eta_L^2 L^2 \sigma_G^2 + \frac{K^2\eta_L^2 L^2}{n}\sigma_G^2$$

$$\lesssim \frac{f_0 - f_*}{\eta\eta_L KT} + \frac{LK\eta\eta_L}{n}\sigma_G^2 + K^2\eta_L^2 L^2 \sigma_G^2,$$

where in the last inequality, the forth term is subsumed into the third term on the right hand side.

The correspondence between Yang et al. [2021]'s notation and ours is as follows: $f_0 - f_*$ in Yang et al. [2021] corresponds to $F_0$ in this paper; $\eta$ corresponds to $\eta$; $\eta_L$ corresponds to $\gamma$; $K$ corresponds

to $K$; $T$ corresponds to $N\frac{1}{S}Q$; $L$ corresponds to $L$; $n$ corresponds to $S$; $\sigma_G$ corresponds to $\varsigma$, $\min_{t \in [T]} \mathbb{E}[\|\nabla f(\mathbf{x}_t)\|^2]$ corresponds to $\min_t \mathbb{E} \|\nabla f(\mathbf{y}_t)\|^2$. Using the notation correspondence, we obtain

$$\min_t \mathbb{E} \|\nabla f(\mathbf{x}_t)\|^2 = \mathcal{O}\left( \frac{F_0}{\gamma \eta K N \frac{1}{S} Q} + \gamma^2 L^2 K^2 \varsigma^2 + \frac{\gamma \eta L K}{S} \varsigma^2 \right).$$

Setting $\eta = 1$ and tuning the step size with Lemma 1 ($\gamma \eta L K \lesssim 1$ and $\gamma L K \lesssim 1$), we obtain

$$\min_t \mathbb{E} \|\nabla f(\mathbf{x}_t)\|^2 = \mathcal{O}\left( \frac{LF_0}{T} + \left( \frac{LF_0\varsigma}{T} \right)^{\frac{2}{3}} + \left( \frac{LF_0\varsigma^2}{ST} \right)^{\frac{1}{2}} \right).$$

## B  Notations

Table 5: Summary of key notations.

| Notation | Description |
|---|---|
| $Q$ | Number of epochs. |
| $N$ | Number of local objective functions. |
| $K$ | Number of local steps in FL. |
| $S$ | Number of participating clients in each round in FL. |
| $\mu$ | Strong convexity constant or PL condition constant. |
| $L_{p,p'}$ | Smoothness constants (see Definition 3). |
| $A, B, D$ | Constants in Assumption 2 and Theorems 1 and 2. |
| $\tilde{B}, \tilde{D}$ | Constants in Theorems 1 and 2. |
| $d$ | Dimension of the model parameter vector. |
| $\alpha, \varsigma$ | Constants in Assumption 3. |
| $G_q$ | $G_q := \sqrt{\alpha^2 \|\nabla f(\mathbf{x}_q)\|^2 + \varsigma^2}$; see Equation (5). |
| $\gamma$ | Step size. |
| $\eta$ | Global step size (in FL). |
| $\bar{\phi}$ | Order Error in SGD. |
| $\bar{\varphi}$ | Order Error in FL. |
| $\pi$ | A permutation of $\{0, 1, \dots, N-1\}$. It serves as the order of examples or clients. |
| $\pi(n)$ | The $(n+1)$-th element of permutation $\pi$. |
| $f$ | Global objective function. |
| $f_n$ | Local objective function. It represents examples in SGD and clients in FL. |
| $F_0$ | $F_0 = f(\mathbf{x}_0) - f_*$. |
| $\mathbf{x}$ | Model parameter vector. |
| $\mathbf{x}_q^n$ | Parameter vector after $n$ steps in epoch $q$ (in SGD). |
| $\mathbf{x}_{q,k}^n$ | Parameter vector after $k$ local updates in client $n$ in epoch $q$ (in FL). |
| $\mathbf{p}_q^n$ | Pseudo-gradient of client $n$ in epoch $q$ in FL. |

Key notations are summarized in Table 5.

**Norm.** We use $\|\cdot\|_p$ to denote the Lebesgue $p$-norm; unless otherwise stated, we use $\|\cdot\|$ to denote the Lebesgue 2-norm.

**Set.** We let $[n] := \{1, 2, \dots, n\}$ for $n \in \mathbb{N}^+$ and $\{x_i\}_{i \in \mathcal{S}} := \{x_i \mid i \in \mathcal{S}\}$ for any set $\mathcal{S}$. We let $|\mathcal{S}|$ be the size of any set $\mathcal{S}$.

**Big O notations.** We use $\lesssim$ to denote "less than" up to some numerical constants and polylogarithmic factors, and $\gtrsim$ and $\asymp$ are defined likewise. We also use the big O notations, $\tilde{O}, \mathcal{O}, \Omega$, where $\mathcal{O}$, $\Omega$ hide numerical constants, $\tilde{\mathcal{O}}$ hides numerical constants and polylogarithmic factors.

**Notations in proofs.** For convenience, we will use "$T_n$" to denote the $n$-th term on the right hand side in some equation in the following proofs. We will use $\pm$ to mean "add $(+)$" and then "subtract $(-)$" the term: $a \pm b$ means $a - b + b$.

For Assumption 3, we let

$$G_q := \sqrt{\alpha^2 \left\| \nabla f(\mathbf{x}_q) \right\|^2 + \varsigma^2}, \tag{5}$$

which gives

$$\left\| \nabla f_n(\mathbf{x}_q) - \nabla f(\mathbf{x}_q) \right\|^2 \leq \alpha^2 \left\| \nabla f(\mathbf{x}_q) \right\|^2 + \varsigma^2 = G_q^2.$$

Importantly, $\pi$ is a permutation of $\{0, 1, \ldots, N-1\}$, and it serves as the training orders of data examples in SGD or training orders of clients in FL. Next, we need to define an operation on $\pi$ as done in Lu et al. [2022a, Appendix B] and Cooper et al. [2023, Appendix C.4]:

$$\pi^{-1}(i) := j \text{ such that } \pi(j) = i, \quad i, j \in \{0, 1, \ldots, N-1\}.$$

It represents that the index of $i$ in the permutation $\pi$ is $j$, where $i, j \in \{0, 1, \ldots, N-1\}$. This operation will be very useful in Appendices F.5, F.6, F.7, F.8, H.3. According to the definition, it follows that

$$\pi^{-1}\left(\pi(j)\right) = j.$$

It can be proved as follows: Assume that $\pi^{-1}\left(\pi(j)\right) = k \neq j$. Then, according to the definition, we obtain $\pi(j) = \pi(k)$, which implies that $j = k$. This contradicts our assumption. Thus, we have $\pi^{-1}\left(\pi(j)\right) = k = j$.

## C   Algorithms

In this section, we provide more details about GraBs.

### C.1   Preliminaries of GraBs

---

**Algorithm 3:** Balancing [Alweiss et al., 2021]

1 **Function** Balance($\{\mathbf{z}_n\}_{n=0}^{N-1}$)
2      Initialize running sum $\mathbf{s}$, hyperparameter $c$
3      Initialize $\{\epsilon_n\}$ for assigned signs
4      **for** $n = 0, \ldots, N-1$ **do**
5          Compute $\tilde{p} \leftarrow \frac{1}{2} - \frac{\langle \mathbf{s}, \mathbf{z}_n \rangle}{2c}$
6          Assign signs:
             $\epsilon_n \leftarrow +1$ with probability $\tilde{p}$;
             $\epsilon_n \leftarrow -1$ with probability $1 - \tilde{p}$
7          Update $\mathbf{s} \leftarrow \mathbf{s} + \epsilon_n \cdot \mathbf{z}_n$
8      **return** the assigned signs $\{\epsilon_n\}$

---

**Algorithm 4:** Reordering [Harvey and Samadi, 2014]

1 **Function** Reorder($\pi, \{\epsilon_n\}_{n=0}^{N-1}$)
2      Initialize two lists $L_{\text{positive}} \leftarrow []$, $L_{\text{negative}} \leftarrow []$
3      **for** $n = 0, \ldots, N-1$ **do**
4          **if** $\epsilon_n = +1$ **then**
5              Append $\pi(n)$ to $L_{\text{positive}}$
6          **else**
7              Append $\pi(n)$ to $L_{\text{negative}}$
8      $\pi' = \text{concatenate}(L_{\text{positive}}, \text{reverse}(L_{\text{negative}}))$
9      **return** the new order $\pi'$

---

**Algorithm 5:** Basic Balancing and Reordering

1 **Function** BasicBR($\pi, \{\mathbf{z}_n\}_{n=0}^{N-1}, \mathbf{m}$)[a]
2      Centering: $\{\mathbf{c}_n := \mathbf{z}_n - \mathbf{m}\}_{n=0}^{N-1}$
3      $\{\epsilon_n\}_{n=0}^{N-1} \leftarrow$ Balance($\{\mathbf{c}_n\}_{n=0}^{N-1}$)
4      $\pi' \leftarrow$ Reorder($\pi, \{\epsilon_n\}_{n=0}^{N-1}$)
5      **return** $\pi'$

---

[a]The mean vector $\mathbf{m}$ is used to center the input vectors $\{\mathbf{z}_n\}_{n=0}^{N-1}$ (Line 2). In most cases, it is the average of the input vectors $\frac{1}{N} \sum_{n=0}^{N-1} \mathbf{z}_n$, except in the original GraB algorithm, where it is replaced by an estimate of the actual average.

---

**Algorithm 6:** Pair Balancing and Reordering

1 **Function** PairBR($\pi, \{\mathbf{z}_n\}_{n=0}^{N-1}, \mathbf{m}$)
2      Centering: $\{\mathbf{c}_n := \mathbf{z}_n - \mathbf{m}\}_{n=0}^{N-1}$[a]
3      Compute $\{\mathbf{d}_l := \mathbf{c}_{2l} - \mathbf{c}_{2l+1}\}_{l=0}^{\frac{N}{2}-1}$
4      $\{\tilde{\epsilon}_l\}_{l=0}^{\frac{N}{2}-1} \leftarrow$ Balance($\{\mathbf{d}_l\}_{l=0}^{\frac{N}{2}-1}$)
5      Compute $\{\epsilon_n\}_{n=0}^{N-1}$ such that
         $\epsilon_{2l} = \tilde{\epsilon}_l$ and $\epsilon_{2l+1} = -\tilde{\epsilon}_l$ for
         $l = 0, \ldots, \frac{N}{2} - 1$
6      $\pi' \leftarrow$ Reorder($\pi, \{\epsilon_n\}_{n=0}^{N-1}$)
7      **return** $\pi'$

---

[a]The step of centering is not required in practical implementations

---

In Section 2.3, we have introduced the key ideas of GraBs. Now, we introduce the concrete GraB algorithms. We start from GraB-proto and PairGraB-proto, where the former is a simplified version of the original GraB algorithm [Lu et al., 2022a], and the latter is a simplified version of PairGraB algorithm [Lu et al., 2022a; Cooper et al., 2023].

- GraB-proto. Use `BasicBR` (Algorithm 5) as the `Permute` function in Algorithm 1, with the inputs of $\pi_q$, $\{\nabla f_{\pi_q(n)}(\mathbf{x}_q)\}_{n=0}^{N-1}$ and $\nabla f(\mathbf{x}_q)$, for each epoch $q$.

- PairGraB-proto. Use `PairBR` (Algorithm 6) as the `Permute` function in Algorithm 1, with the inputs of $\pi_q$, $\{\nabla f_{\pi_q(n)}(\mathbf{x}_q)\}_{n=0}^{N-1}$ and $\nabla f(\mathbf{x}_q)$, for each epoch $q$.

The main difference is that GraB-proto uses the basic balancing and reordering algorithm (`BasicBR`) while PairGraB-proto uses the pair balancing and reordering algorithm (`PairBR`). The advantage of `PairBR` is that it is free of *centering* the input vectors in the practical implementation. As shown in Algorithm 6 (Lines 3–4), it balances the difference of two centered vectors, which is equivalent to balancing the difference of the two original vectors as the mean vectors are canceled out:

$$\mathbf{d}_l = (\mathbf{z}_{2l} - \mathbf{m}) - (\mathbf{z}_{2l+1} - \mathbf{m}) = \mathbf{z}_{2l} - \mathbf{z}_{2l+1}.$$

This advantage makes it seamlessly compatible with online algorithms such as SGD. Notably, compared with the original GraB and PairGraB algorithms, whose implementation details are deferred to Appendix C.2, GraB-proto and PairGraB-proto are impractical in computation and storage, however, they are simple, and sufficient to support our theory.

Next, we briefly introduce the original GraB and PairGraB algorithms.

- GraB. Use `BasicBR` (Algorithm 5) as the `Permute` function in Algorithm 1, with the inputs of $\pi_q$, $\{\nabla f_{\pi_q(n)}(\mathbf{x}_q^n)\}_{n=0}^{N-1}$ and $\frac{1}{N}\sum_{n=0}^{N-1} \nabla f_{\pi_{q-1}(n)}(\mathbf{x}_{q-1}^n)$, for each epoch $q$.

- PairGraB. Use `PairBR` (Algorithm 6) as the `Permute` function in Algorithm 1, with the inputs of $\pi_q$, $\{\nabla f_{\pi_q(n)}(\mathbf{x}_q^n)\}_{n=0}^{N-1}$ and $\frac{1}{N}\sum_{n=0}^{N-1} \nabla f_{\pi_q(n)}(\mathbf{x}_q^n)$, for each epoch $q$.

They replace $\nabla f_{\pi_q(n)}(\mathbf{x}_q)$ in their prototype versions with the easily accessible $\nabla f_{\pi_q(n)}(\mathbf{x}_q^n)$, reducing the unnecessary computational cost. In addition, for GraB, to overcome the challenge of centering the gradients in the `BasicBR` algorithm, GraB uses the average of the stale gradients as the estimate of the actual average of the fresh gradients, to "center" the (fresh) gradients. This trick is not required for PairGraB. See the implementation details in Algorithms 9 and 11.

In FL, we propose regularized-participation FL with GraB, which uses the pseudo-gradients $\{\mathbf{p}_q^n\}_{n=0}^{N-1}$ to generate the permutations.

- FL-GraB. Use `PairBR` (Algorithm 6) as the `Permute` function in Algorithm 2, with the inputs of $\pi_q$, $\{\mathbf{p}_q^n\}_{n=0}^{N-1}$ and $\frac{1}{N}\sum_{n=0}^{N-1} \mathbf{p}_q^n$, for each epoch $q$.

The main differences of GraB algorithms are summarized in Table 6.

Table 6: The main differences of GraB Algorithms.

| Algorithm | Permute | Inputs of `Permute` (in epoch $q$) |
|---|---|---|
| GraB-proto (Prop. 1) | `BasicBR` | $\pi_q, \{\nabla f_{\pi_q(n)}(\mathbf{x}_q)\}_{n=0}^{N-1}, \frac{1}{N}\sum_{n=0}^{N-1} \nabla f_{\pi_q(n)}(\mathbf{x}_q)$ |
| PairGraB-proto (Prop. 5) | `PairBR` | $\pi_q, \{\nabla f_{\pi_q(n)}(\mathbf{x}_q)\}_{n=0}^{N-1}, \frac{1}{N}\sum_{n=0}^{N-1} \nabla f_{\pi_q(n)}(\mathbf{x}_q)$ |
| GraB (Prop. 6) | `BasicBR` | $\pi_q, \{\nabla f_{\pi_q(n)}(\mathbf{x}_q^n)\}_{n=0}^{N-1}, \frac{1}{N}\sum_{n=0}^{N-1} \nabla f_{\pi_{q-1}(n)}(\mathbf{x}_{q-1}^n)$ |
| PairGraB (Prop. 7) | `PairBR` | $\pi_q, \{\nabla f_{\pi_q(n)}(\mathbf{x}_q^n)\}_{n=0}^{N-1}, \frac{1}{N}\sum_{n=0}^{N-1} \nabla f_{\pi_q(n)}(\mathbf{x}_q^n)$ |
| FL-GraB (Prop. 10) | `PairBR` | $\pi_q, \{\mathbf{p}_q^n\}_{n=0}^{N-1}, \frac{1}{N}\sum_{n=0}^{N-1} \mathbf{p}_q^n$ |

## C.2 Implementations of GraBs

The practical implementations of GraB are provided in Algorithms 9 and 10. The implementation of PairGraB is provided in Algorithm 11. The implementation of FL-GraB is provided in Algorithm 12.

As done in Lu et al. [2022a] and Cooper et al. [2023], we use Algorithm 7 for the theories in this paper, while we use Algorithm 8 for the experiments in Appendix I.

Notably, Algorithm 9 (the original algorithm in Lu et al. [2022a, Algorithm 4]) is logically equivalent to Algorithm 10. Compared with Algorithm 9, which updates the new order at the end of each step (Lines 11–14), Algorithm 10 generates the new order at the end of each epoch (Line 12). In fact, in Algorithm 10, we can reorder the examples for multiple times with the same signs (see Line 12), which may be useful in practice [Wei, 2023]. Similar variants can also be formulated for Algorithms 11 and 12.

| **Algorithm 7:** Assign signs. [Alweiss et al., 2021] | **Algorithm 8:** Assign signs without normalization. [Lu et al., 2022a, Algorithm 5] |
|---|---|
| 1 **Function** AssignSign($\mathbf{s}$, $\mathbf{z}$, $c$)[a] 
 2    Compute $\tilde{p} \leftarrow \frac{1}{2} - \frac{\langle \mathbf{s}, \mathbf{z} \rangle}{2c}$ 
 3    Assign signs: 
      $\epsilon \leftarrow +1$ with probability $\tilde{p}$; 
      $\epsilon \leftarrow -1$ with probability $1 - \tilde{p}$ 
 4    **return** $\epsilon$ | 1 **Function** AssignSign($\mathbf{s}$, $\mathbf{z}$) 
 2    **if** $\|\mathbf{s} + \mathbf{z}\| < \|\mathbf{s} - \mathbf{z}\|$ **then** 
 3      $\epsilon \leftarrow +1$ 
 4    **else** 
 5      $\epsilon \leftarrow -1$ 
 6    **return** $\epsilon$ |

[a] $c$ is a hyperparameter. See Lu et al. [2022a, Theorem 4].

---

**Algorithm 9:** GraB [Lu et al., 2022a, Algorithm 4]

**Input:** $\pi_0$, $\mathbf{x}_0$; **Output:** $\{\mathbf{x}_q\}$
1 Initialize $\mathbf{s} \leftarrow \mathbf{0}$, $\mathbf{m} \leftarrow \mathbf{0}$, $\mathbf{m}_{\text{stale}} \leftarrow \mathbf{0}$
2 **for** $q = 0, 1, \ldots, Q-1$ **do**
3    $\mathbf{s} \leftarrow \mathbf{0}$; $\mathbf{m}_{\text{stale}} \leftarrow \mathbf{m}$; $\mathbf{m} \leftarrow \mathbf{0}$; $l \leftarrow 0$, $r \leftarrow N-1$
4    **for** $n = 0, 1, \ldots, N-1$ **do**
5      Compute the gradient $\nabla f_{\pi_q(n)}(\mathbf{x}_q^n)$
6      Update the parameter: $\mathbf{x}_q^{n+1} \leftarrow \mathbf{x}_q^n - \gamma \nabla f_{\pi_q(n)}(\mathbf{x}_q^n)$
7      Update the mean: $\mathbf{m} \leftarrow \mathbf{m} + \frac{1}{N} \nabla f_{\pi_q(n)}(\mathbf{x}_q^n)$
8      Center the gradient $\mathbf{c} \leftarrow \nabla f_{\pi_q(n)}(\mathbf{x}_q^n) - \mathbf{m}_{\text{stale}}$
9      Assign the sign: $\epsilon \leftarrow$ AssignSign($\mathbf{s}$, $\mathbf{c}$)
10      Update the sign sum: $\mathbf{s} \leftarrow \mathbf{s} + \epsilon \cdot \nabla f_{\pi_q(n)}(\mathbf{x}_q^n)$
11      **if** $\epsilon = +1$ **then**
12        $\pi_{q+1}(l) \leftarrow \pi_q(n)$; $l \leftarrow l+1$.
13      **else**
14        $\pi_{q+1}(r) \leftarrow \pi_q(n)$; $r \leftarrow r-1$.
15    Update the parameter: $\mathbf{x}_{q+1} \leftarrow \mathbf{x}_q^N$

**Algorithm 10:** GraB

**Input:** $\pi_0, \mathbf{x}_0$; **Output:** $\{\mathbf{x}_q\}$

1  Initialize $\mathbf{s} \leftarrow \mathbf{0}, \mathbf{m} \leftarrow \mathbf{0}, \mathbf{m}_{\text{stale}} \leftarrow \mathbf{0}$

2  **for** $q = 0, 1, \ldots, Q - 1$ **do**

3  $\quad$ $\mathbf{s} \leftarrow \mathbf{0}; \mathbf{m}_{\text{stale}} \leftarrow \mathbf{m}; \mathbf{m} \leftarrow \mathbf{0}; l \leftarrow 0, r \leftarrow N - 1$

4  $\quad$ **for** $n = 0, 1, \ldots, N - 1$ **do**

5  $\quad\quad$ Compute the gradient $\nabla f_{\pi_q(n)}(\mathbf{x}_q^n)$

6  $\quad\quad$ Update the parameter: $\mathbf{x}_q^{n+1} \leftarrow \mathbf{x}_q^n - \gamma \nabla f_{\pi_q(n)}(\mathbf{x}_q^n)$

7  $\quad\quad$ Update the mean: $\mathbf{m} \leftarrow \mathbf{m} + \frac{1}{N} \nabla f_{\pi_q(n)}(\mathbf{x}_q^n)$

8  $\quad\quad$ Center the gradient $\mathbf{c} \leftarrow \nabla f_{\pi_q(n)}(\mathbf{x}_q^n) - \mathbf{m}_{\text{stale}}$

9  $\quad\quad$ Assign the sign: $\epsilon_n \leftarrow \texttt{AssignSign}(\mathbf{s}, \mathbf{c})$

10  $\quad\quad$ Update the sign sum: $\mathbf{s} \leftarrow \mathbf{s} + \epsilon_n \cdot \nabla f_{\pi_q(n)}(\mathbf{x}_q^n)$

11  $\quad$ Update the parameter: $\mathbf{x}_{q+1} \leftarrow \mathbf{x}_q^N$

12  $\quad$ $\pi_{q+1} \leftarrow \texttt{Reorder}(\pi_q, \{\epsilon_n\}_{n=0}^{N-1})^a$

---

[a] We can reorder the examples for multiple times with the same signs in this step.

**Algorithm 11:** PairGraB

**Input:** $\pi_0, \mathbf{x}_0$; **Output:** $\{\mathbf{x}_q\}$

1  **for** $q = 0, 1, \ldots, Q - 1$ **do**

2  $\quad$ $\mathbf{s} \leftarrow \mathbf{0}; \mathbf{d} \leftarrow \mathbf{0}, l \leftarrow 0, r \leftarrow N - 1$

3  $\quad$ **for** $n = 0, 1, \ldots, N - 1$ **do**

4  $\quad\quad$ Compute the gradient $\nabla f_{\pi_q(n)}(\mathbf{x}_q^n)$

5  $\quad\quad$ Update the parameter: $\mathbf{x}_q^{n+1} \leftarrow \mathbf{x}_q^n - \gamma \nabla f_{\pi_q(n)}(\mathbf{x}_q^n)$

6  $\quad\quad$ **if** $(n + 1) \mod 2 = 0$ **then**

7  $\quad\quad\quad$ Compute the difference: $\mathbf{d} \leftarrow \nabla f_{\pi_q(n-1)} - \nabla f_{\pi_q(n)}$

8  $\quad\quad\quad$ Assign the sign: $\epsilon \leftarrow \texttt{AssignSign}(\mathbf{s}, \mathbf{d})$

9  $\quad\quad\quad$ Update the sign sum: $\mathbf{s} \leftarrow \mathbf{s} + \epsilon \cdot \mathbf{d}$

10  $\quad\quad\quad$ **if** $\epsilon = +1$ **then**

11  $\quad\quad\quad\quad$ $\pi_{q+1}(l) \leftarrow \pi_q(n); l \leftarrow l + 1$

12  $\quad\quad\quad\quad$ $\pi_{q+1}(r) \leftarrow \pi_q(n - 1); r \leftarrow r - 1$

13  $\quad\quad\quad$ **else**

14  $\quad\quad\quad\quad$ $\pi_{q+1}(l) \leftarrow \pi_q(n - 1); l \leftarrow l + 1$

15  $\quad\quad\quad\quad$ $\pi_{q+1}(r) \leftarrow \pi_q(n); r \leftarrow r - 1$

16  $\quad$ Update the parameter: $\mathbf{x}_{q+1} \leftarrow \mathbf{x}_q^N$

**Algorithm 12:** FL-GraB (Server-side)

---

**Input:** $\pi_0, \mathbf{x}_0$; **Output:** $\{\mathbf{x}_q\}$

1 **for** $q = 0, 1, \ldots, Q - 1$ **do**
2     $\mathbf{s} \leftarrow \mathbf{0}; \mathbf{d} \leftarrow \mathbf{0}, l \leftarrow 0, r \leftarrow N - 1$
3     **for** $n = 0, 1, \ldots, N - 1$ **do**
4         Get the pseudo-gradient $\mathbf{p}_q^n = \sum_{k=0}^{K-1} \nabla f_{\pi_q(n)}(\mathbf{x}_{q,k}^n)$
        /* Update the parameter */
5         **if** $(n + 1) \mod S = 0$ **then**
6             $\mathbf{w} \leftarrow \mathbf{w} - \sum_{s=0}^{S-1} \mathbf{p}_q^{n-S}$
7         **if** $(n + 1) \mod 2 = 0$ **then**
            /* Balance */
8             Compute the difference: $\mathbf{d} \leftarrow \nabla f_{\pi_q(n-1)} - \nabla f_{\pi_q(n)}$
9             Assign the signs: $\epsilon \leftarrow \mathtt{AssignSign}(\mathbf{s}, \mathbf{d})$
10            Update the sign sum: $\mathbf{s} \leftarrow \mathbf{s} + \epsilon \cdot \mathbf{d}$
            /* Update the new order */
11             **if** $\epsilon = +1$ **then**
12                 $\pi_{q+1}(l) \leftarrow \pi_q(n); l \leftarrow l + 1$
13                 $\pi_{q+1}(r) \leftarrow \pi_q(n - 1); r \leftarrow r - 1$
14             **else**
15                 $\pi_{q+1}(l) \leftarrow \pi_q(n - 1); l \leftarrow l + 1$
16                 $\pi_{q+1}(r) \leftarrow \pi_q(n); r \leftarrow r - 1$
        /* Update the parameter */
17     $\mathbf{x}_{q+1} \leftarrow \mathbf{x}_q - \eta(\mathbf{x}_q - \mathbf{w})$

---

# D   Helper Lemmas

**Lemma 1.** *For any parameters $r_0 > 0$, $T > 0$, $c > 0$ and $\gamma \leq \frac{1}{d}$, there exists constant step sizes $\gamma = \min\left\{\frac{1}{d}, \left(\frac{cr_0}{T}\right)^{\frac{1}{3}}\right\} \leq \frac{1}{d}$ such that*

$$\Psi_T := \frac{r_0}{\gamma T} + c\gamma^2 \leq \frac{dr_0}{T} + 2\frac{c^{\frac{1}{3}}r_0^{\frac{2}{3}}}{T^{\frac{2}{3}}} = \mathcal{O}\left(\frac{dr_0}{T} + \frac{c^{\frac{1}{3}}r_0^{\frac{2}{3}}}{T^{\frac{2}{3}}}\right).$$

*Proof.* If $\frac{1}{d} \leq \left(\frac{r_0}{cT}\right)^{\frac{1}{3}}$, choosing $\gamma = \frac{1}{d}$ gives

$$\Psi_T = \frac{dr_0}{T} + \frac{c}{d^2} \leq \frac{dr_0}{T} + \frac{c^{\frac{1}{3}}r_0^{\frac{2}{3}}}{T^{\frac{2}{3}}}.$$

If $\left(\frac{r_0}{cT}\right)^{\frac{1}{3}} \leq \frac{1}{d}$, choosing $\gamma = \left(\frac{r_0}{cT}\right)^{\frac{1}{3}}$ gives

$$\Psi_T = \frac{dr_0}{T} + \frac{c}{d^2} \leq \frac{c^{\frac{1}{3}}r_0^{\frac{2}{3}}}{T^{\frac{2}{3}}} + \frac{c^{\frac{1}{3}}r_0^{\frac{2}{3}}}{T^{\frac{2}{3}}} \leq 2\frac{c^{\frac{1}{3}}r_0^{\frac{2}{3}}}{T^{\frac{2}{3}}}.$$

Thus,

$$\Psi_T \leq \frac{dr_0}{T} + 2\frac{c^{\frac{1}{3}}r_0^{\frac{2}{3}}}{T^{\frac{2}{3}}} = \mathcal{O}\left(\frac{dr_0}{T} + \frac{c^{\frac{1}{3}}r_0^{\frac{2}{3}}}{T^{\frac{2}{3}}}\right).$$

$\square$

**Lemma 2.** *For any parameters $r_0 > 0$, $T > 0$, $c > 0$ and $\gamma \leq \frac{1}{d}$, there exists constant step sizes $\gamma = \min\left\{\frac{1}{d}, \left(\frac{cr_0}{T}\right)^{\frac{1}{3}}\right\} \leq \frac{1}{d}$ such that*

$$\Psi_T := \frac{r_0}{\gamma T} + c\gamma \leq \frac{dr_0}{T} + 2\frac{c^{\frac{1}{2}}r_0^{\frac{1}{2}}}{T^{\frac{1}{2}}} = \mathcal{O}\left(\frac{dr_0}{T} + \frac{c^{\frac{1}{2}}r_0^{\frac{1}{2}}}{T^{\frac{1}{2}}}\right).$$

*Proof.* If $\frac{1}{d} \leq \left(\frac{r_0}{cT}\right)^{\frac{1}{2}}$, choosing $\gamma = \frac{1}{d}$ gives

$$\Psi_T = \frac{dr_0}{T} + \frac{c}{d} \leq \frac{dr_0}{T} + \frac{c^{\frac{1}{2}} r_0^{\frac{1}{2}}}{T^{\frac{1}{2}}}.$$

If $\left(\frac{r_0}{cT}\right)^{\frac{1}{2}} \leq \frac{1}{d}$, choosing $\gamma = \left(\frac{r_0}{cT}\right)^{\frac{1}{2}}$ gives

$$\Psi_T = \frac{dr_0}{T} + \frac{c}{d^2} \leq \frac{c^{\frac{1}{2}} r_0^{\frac{1}{2}}}{T^{\frac{1}{2}}} + \frac{c^{\frac{1}{2}} r_0^{\frac{1}{2}}}{T^{\frac{1}{2}}} \leq 2 \frac{c^{\frac{1}{2}} r_0^{\frac{1}{2}}}{T^{\frac{1}{2}}}.$$

Thus,

$$\Psi_T \leq \frac{dr_0}{T} + 2 \frac{c^{\frac{1}{2}} r_0^{\frac{1}{2}}}{T^{\frac{1}{2}}} = \mathcal{O}\left(\frac{dr_0}{T} + \frac{c^{\frac{1}{2}} r_0^{\frac{1}{2}}}{T^{\frac{1}{2}}}\right).$$

$\square$

**Lemma 3.** *Consider $N$ vectors $\{\mathbf{z}_n\}_{n=0}^{N-1}$ and a permutation $\pi$ of $\{0, 1, \ldots, N-1\}$. Assign the signs $\{\epsilon_n\}_{n=0}^{N-1}$ ($\epsilon_n \in \{-1, +1\}$) by the balancing algorithms (such as Algorithm 3) to the permuted vectors under the permutation $\pi$ (that is, $\{\mathbf{z}_{\pi(n)}\}_{n=0}^{N-1}$). Let $\pi'$ be the new permutation produced by Algorithm 4 with the input of the old permutation $\pi$ and the assigned signs $\{\epsilon_n\}_{n=0}^{N-1}$. Then,*

$$\max_{n \in [N]} \left\|\sum_{i=0}^{n-1} \mathbf{z}_{\pi'(i)}\right\|_\infty \leq \frac{1}{2} \max_{n \in [N]} \left\|\sum_{i=0}^{n-1} \mathbf{z}_{\pi(i)}\right\|_\infty + \frac{1}{2} \max_{n \in [N]} \left\|\sum_{i=0}^{n-1} \epsilon_i \cdot \mathbf{z}_{\pi(i)}\right\|_\infty + \left\|\sum_{i=0}^{N-1} \mathbf{z}_i\right\|_\infty.$$

*Furthermore, suppose that the signs $\{\epsilon_n\}_{n=0}^{N-1}$ are assigned by Algorithm 3. If $\|\mathbf{z}_n\|_2 \leq a$ for all $n \in \{0, 1, \ldots, N-1\}$ and $\left\|\sum_{i=0}^{N-1} \mathbf{z}_i\right\|_\infty \leq b$, then, with probability at least $1 - \delta$,*

$$\max_{n \in [N]} \left\|\sum_{i=0}^{n-1} \mathbf{z}_{\pi'(i)}\right\|_\infty \leq \frac{1}{2} \max_{n \in [N]} \left\|\sum_{i=0}^{n-1} \mathbf{z}_{\pi(i)}\right\|_\infty + \frac{1}{2} Ca + b,$$

*where $C = 30 \log(\frac{dN}{\delta}) = \mathcal{O}\left(\log\left(\frac{dN}{\delta}\right)\right) = \tilde{\mathcal{O}}(1)$ is from Alweiss et al. [2021, Theorem 1.1].*

*Proof.* This is Lemma 5 in Lu et al. [2022a] and we reproduce it for completeness.

Let $M^+ = \{i \in \{0, 1, \ldots, N-1\} \mid \epsilon_i = +1\}$ and $M^- = \{i \in \{0, 1, \ldots, N-1\} \mid \epsilon_i = -1\}$. Then, for any $n \in \{1, 2 \ldots, N\}$,

$$\sum_{i=0}^{n-1} \mathbf{z}_{\pi(i)} + \sum_{i=0}^{n-1} \epsilon_i \cdot \mathbf{z}_{\pi(i)} = 2 \cdot \sum_{i \in M^+ \cap \{0, 1, \ldots, n-1\}} \mathbf{z}_{\pi(i)}, \tag{6}$$

$$\sum_{i=0}^{n-1} \mathbf{z}_{\pi(i)} - \sum_{i=0}^{n-1} \epsilon_i \cdot \mathbf{z}_{\pi(i)} = 2 \cdot \sum_{i \in M^- \cap \{0, 1, \ldots, n-1\}} \mathbf{z}_{\pi(i)}. \tag{7}$$

By using triangular inequality, for any $n \in \{1, 2 \ldots, N\}$, we have

$$\left\|\sum_{i \in M^+ \cap \{0, 1, \ldots, n-1\}} \mathbf{z}_{\pi(i)}\right\|_\infty \leq \frac{1}{2} \left\|\sum_{i=0}^{n-1} \mathbf{z}_{\pi(i)}\right\|_\infty + \frac{1}{2} \left\|\sum_{i=0}^{n-1} \epsilon_i \cdot \mathbf{z}_{\pi(i)}\right\|_\infty,$$

$$\left\|\sum_{i \in M^- \cap \{0, 1, \ldots, n-1\}} \mathbf{z}_{\pi(i)}\right\|_\infty \leq \frac{1}{2} \left\|\sum_{i=0}^{n-1} \mathbf{z}_{\pi(i)}\right\|_\infty + \frac{1}{2} \left\|\sum_{i=0}^{n-1} \epsilon_i \cdot \mathbf{z}_{\pi(i)}\right\|_\infty.$$

Next, we consider the upper bound of $\left\|\sum_{i=0}^{n'-1} \mathbf{z}_{\pi'(i)}\right\|_\infty$ for all $n' \in \{1, 2, \ldots, N\}$. Recall that Algorithm 4 puts the vectors with positive assigned signs in the front of the new permutation and the vectors with negative assigned signs in the back of the new permutation.

If $n' \leq |M^+|$ ($|M^+|$ denotes the size of $M^+$), we obtain

$$\left\| \sum_{i=0}^{n'-1} \mathbf{z}_{\pi'(i)} \right\|_{\infty} \leq \max_{n \in [N]} \left\| \sum_{i \in M^+ \cap \{0,1,\ldots,n-1\}} \mathbf{z}_{\pi(i)} \right\|_{\infty}$$

$$\leq \frac{1}{2} \max_{n \in [N]} \left\| \sum_{i=0}^{n-1} \mathbf{z}_{\pi(i)} \right\|_{\infty} + \frac{1}{2} \max_{n \in [N]} \left\| \sum_{i=0}^{n-1} \epsilon_i \cdot \mathbf{z}_{\pi(i)} \right\|_{\infty}.$$

If $n' > |M^+|$ ($|M^-|$ denotes the size of $M^+$), we obtain

$$\left\| \sum_{i=0}^{n'-1} \mathbf{z}_{\pi'(i)} \right\|_{\infty} = \left\| \sum_{i=0}^{N-1} \mathbf{z}_{\pi'(i)} - \sum_{i=n'}^{N-1} \mathbf{z}_{\pi'(i)} \right\|_{\infty}$$

$$\leq \left\| \sum_{i=0}^{N-1} \mathbf{z}_{\pi'(i)} \right\|_{\infty} + \left\| \sum_{i=n'}^{N-1} \mathbf{z}_{\pi'(i)} \right\|_{\infty}$$

$$\leq \left\| \sum_{i=0}^{N-1} \mathbf{z}_{\pi'(i)} \right\|_{\infty} + \max_{n \in [N]} \left\| \sum_{i \in M^- \cap \{0,1,\ldots,n-1\}} \mathbf{z}_{\pi(i)} \right\|_{\infty}$$

$$\leq \left\| \sum_{i=0}^{N-1} \mathbf{z}_i \right\|_{\infty} + \frac{1}{2} \max_{n \in [N]} \left\| \sum_{i=0}^{n-1} \mathbf{z}_{\pi(i)} \right\|_{\infty} + \frac{1}{2} \max_{n \in [N]} \left\| \sum_{i=0}^{n-1} \epsilon_i \cdot \mathbf{z}_{\pi(i)} \right\|_{\infty}.$$

Thus we combine the two cases and obtain the relation

$$\max_{n \in [N]} \left\| \sum_{i=0}^{n-1} \mathbf{z}_{\pi'(i)} \right\|_{\infty} \leq \frac{1}{2} \max_{n \in [N]} \left\| \sum_{i=0}^{n-1} \mathbf{z}_{\pi(i)} \right\|_{\infty} + \frac{1}{2} \max_{n \in [N]} \left\| \sum_{i=0}^{n-1} \epsilon_i \cdot \mathbf{z}_{\pi(i)} \right\|_{\infty} + \left\| \sum_{i=0}^{N-1} \mathbf{z}_i \right\|_{\infty}.$$

Using Alweiss et al. [2021]'s Theorem 1.1, for all $n \in [N]$, we have

$$\left\| \sum_{i=0}^{n-1} \epsilon_i \cdot \mathbf{z}_{\pi(i)} \right\|_{\infty} = \left\| \sum_{i=0}^{n-1} \epsilon_i \cdot \frac{\mathbf{z}_{\pi(i)}}{\max_{j \in \{0,1,\ldots,N-1\}} \left\| \mathbf{z}_{\pi(j)} \right\|_2} \right\|_{\infty} \cdot \max_{j \in \{0,1,\ldots,N-1\}} \left\| \mathbf{z}_{\pi(j)} \right\|_2 \leq Ca.$$

Then, using $\left\| \sum_{i=0}^{N-1} \mathbf{z}_i \right\|_{\infty} \leq b$, we obtain the claimed bound. $\qquad\square$

**Lemma 4.** *Let $\pi$, $\{\mathbf{z}_{\pi(n)}\}_{n=0}^{N-1}$ and $\frac{1}{N} \sum_{n=0}^{N-1} \mathbf{z}_{\pi(n)}$ be the inputs of Algorithm 6, and $\pi'$ be the corresponding output. Suppose that $N \mod 2 = 0$. If $\|\mathbf{z}_n\|_2 \leq a$ for all $n \in \{0,1,\ldots,N-1\}$ and $\left\| \sum_{i=0}^{N-1} \mathbf{z}_i \right\|_{\infty} \leq b$, then, with probability at least $1 - \delta$,*

$$\max_{n \in [N]} \left\| \sum_{i=0}^{n-1} \mathbf{z}_{\pi'(i)} \right\|_{\infty} \leq \frac{1}{2} \max_{n \in [N]} \left\| \sum_{i=0}^{n-1} \mathbf{z}_{\pi(i)} \right\|_{\infty} + Ca + b,$$

*where $C = 30 \log(\frac{dN}{2\delta}) = \mathcal{O}\left(\log\left(\frac{dN}{\delta}\right)\right) = \tilde{\mathcal{O}}(1)$ is from Alweiss et al. [2021, Theorem 1.1].*

*Proof.* This is Lemma 1 in Cooper et al. [2023] and we reproduce it for completeness.

We use $\tilde{\epsilon}_j$ to denote the assigned sign of $\mathbf{d}_j = \mathbf{z}_{\pi(2j)} - \mathbf{z}_{\pi(2j+1)}$ for all $j \in \{0,1,\ldots,\frac{N}{2}-1\}$; we use $\epsilon_i$ to denote the assigned sign of $\mathbf{z}_{\pi(i)}$ for all $i \in \{0,1,\ldots,N-1\}$. Since $\{\mathbf{d}_j\}_{j=0}^{\frac{N}{2}-1}$ is the input of Algorithm 3, according to Alweiss et al. [2021]'s Theorem 1.1, for all $l \in \{1,2,\ldots,\frac{N}{2}\}$,

$$\left\| \sum_{j=0}^{l-1} \tilde{\epsilon}_j \mathbf{d}_j \right\|_{\infty} = \left\| \sum_{j=0}^{l-1} \tilde{\epsilon}_j \frac{\mathbf{d}_j}{\max_{j \in \{0,1,\ldots,l-1\}} \|\mathbf{d}_j\|_2} \right\|_{\infty} \cdot \max_{j \in \{0,1,\ldots,l-1\}} \|\mathbf{d}_j\|_2$$

$$\leq C \max_{j \in \{0,1,\ldots,l-1\}} \|\mathbf{d}_j\|_2 \leq 2Ca,$$

where the last inequality is because for any $j \in \{0, 1, \ldots \frac{N}{2} - 1\}$,

$$\|\mathbf{d}_j\|_2 = \|\mathbf{z}_{\pi(2j)} - \mathbf{z}_{\pi(2j+1)}\|_2 \le \|\mathbf{z}_{\pi(2j)}\|_2 + \|\mathbf{z}_{\pi(2j+1)}\|_2 \le 2a.$$

We define $x_l$ and $y_l$ for $l \in \{1, 2, \ldots, \frac{N}{2}\}$,

$$x_l = \sum_{j=0}^{l-1} \left(\mathbf{z}_{\pi(2j)} + \mathbf{z}_{\pi(2j+1)}\right) = \sum_{i=0}^{2l-1} \mathbf{z}_{\pi(i)},$$

$$y_l = \sum_{j=0}^{l-1} \left(\epsilon_{2j}\mathbf{z}_{\pi(2j)} + \epsilon_{2j+1}\mathbf{z}_{\pi(2j+1)}\right) = \sum_{j=0}^{l-1} \left(\tilde{\epsilon}_j\mathbf{z}_{\pi(2j)} - \tilde{\epsilon}_j\mathbf{z}_{\pi(2j+1)}\right) = \sum_{j=0}^{l-1} \tilde{\epsilon}_j\mathbf{d}_j.$$

Let $M^+ = \{i \in \{0, 1, \ldots, N-1\} \mid \epsilon_i = +1\}$ and $M^- = \{i \in \{0, 1, \ldots, N-1\} \mid \epsilon_i = -1\}$. Then, for all $l \in \{1, 2, \ldots, \frac{N}{2}\}$, it follows that

$$\sum_{i \in M^+ \cap \{0,1,\ldots,2l-1\}} \mathbf{z}_{\pi(i)} = \frac{1}{2} \sum_{j=0}^{l-1} \left((1+\tilde{\epsilon}_j)\mathbf{z}_{\pi(2j)} + (1-\tilde{\epsilon}_j)\mathbf{z}_{\pi(2j+1)}\right) = \frac{1}{2}x_l + \frac{1}{2}y_l,$$

$$\sum_{i \in M^- \cap \{0,1,\ldots,2l-1\}} \mathbf{z}_{\pi(i)} = \frac{1}{2} \sum_{j=0}^{l-1} \left((1-\tilde{\epsilon}_j)\mathbf{z}_{\pi(2j)} + (1+\tilde{\epsilon}_j)\mathbf{z}_{\pi(2j+1)}\right) = \frac{1}{2}x_l - \frac{1}{2}y_l.$$

By using the triangle inequality, for all $l \in \{1, 2, \ldots, \frac{N}{2}\}$, we obtain

$$\left\| \sum_{i \in M^+ \cap \{0,1,\ldots,2l-1\}} \mathbf{z}_{\pi(i)} \right\|_\infty \le \frac{1}{2} \|x_l\|_\infty + \frac{1}{2} \|y_l\|_\infty$$

$$= \frac{1}{2} \left\| \sum_{i=0}^{2l-1} \mathbf{z}_{\pi(i)} \right\|_\infty + \frac{1}{2} \left\| \sum_{j=0}^{l-1} \tilde{\epsilon}_j \cdot \mathbf{d}_j \right\|_\infty$$

$$\le \frac{1}{2} \left\| \sum_{i=0}^{2l-1} \mathbf{z}_{\pi(i)} \right\|_\infty + Ca,$$

$$\left\| \sum_{i \in M^- \cap \{0,1,\ldots,2l-1\}} \mathbf{z}_{\pi(i)} \right\|_\infty \le \frac{1}{2} \left\| \sum_{i=0}^{2l-1} \mathbf{z}_{\pi(i)} \right\|_\infty + Ca.$$

Next, we consider the upper bound of $\left\| \sum_{i=0}^{l'-1} \mathbf{z}_{\pi'(i)} \right\|_\infty$ for all $l' \in \{1, 2, \ldots, N\}$. Recall that Algorithm 4 puts the vectors with positive assigned signs in the front of the new permutation and the vectors with negative assigned signs in the back of the new permutation.

If $l' \in \{1, 2, \ldots, \frac{N}{2}\}$, we obtain

$$\left\| \sum_{i=0}^{l'-1} \mathbf{z}_{\pi'(i)} \right\|_\infty = \left\| \sum_{i \in M^+ \cap \{0,1,\ldots,2l'-1\}} \mathbf{z}_{\pi(i)} \right\|_\infty$$

$$\le \frac{1}{2} \left\| \sum_{i=0}^{2l'-1} \mathbf{z}_{\pi(i)} \right\|_\infty + Ca.$$

Note that if $l' \in \{1, 2, \ldots, \frac{N}{2}\}$, then $2l' \in \{2, 4, \ldots, N\}$. Thus, we obtain

$$\left\| \sum_{i=0}^{l'-1} \mathbf{z}_{\pi'(i)} \right\|_\infty \le \frac{1}{2} \max_{n \in [N]} \left\| \sum_{i=0}^{n-1} \mathbf{z}_{\pi(i)} \right\|_\infty + Ca.$$

If $l' \in \{\frac{N}{2}+1, \frac{N}{2}+2, \ldots, N\}$, we obtain

$$\left\| \sum_{i=0}^{l'-1} \mathbf{z}_{\pi'(i)} \right\|_\infty = \left\| \sum_{i=0}^{N-1} \mathbf{z}_{\pi'(i)} - \sum_{i=l'}^{N-1} \mathbf{z}_{\pi'(i)} \right\|_\infty$$

$$\leq \left\| \sum_{i=0}^{N-1} \mathbf{z}_{\pi'(i)} \right\|_\infty + \left\| \sum_{i=l'}^{N-1} \mathbf{z}_{\pi'(i)} \right\|_\infty$$

$$= \left\| \sum_{i=0}^{N-1} \mathbf{z}_{\pi'(i)} \right\|_\infty + \left\| \sum_{i \in M^- \cap \{0,1,\ldots,2(N-l')-1\}} \mathbf{z}_{\pi(i)} \right\|_\infty$$

$$\leq \left\| \sum_{i=0}^{N-1} \mathbf{z}_i \right\|_\infty + \frac{1}{2} \left\| \sum_{i=0}^{2(N-l')-1} \mathbf{z}_{\pi(i)} \right\|_\infty + \frac{1}{2} \left\| \sum_{j=0}^{(N-l')-1} \tilde{\epsilon}_j \cdot \mathbf{d}_j \right\|_\infty .$$

Note that if $l' \in \{\frac{N}{2}+1, \frac{N}{2}+2, \ldots, N\}$, then $(N-l') \in \{0,1,\ldots,\frac{N}{2}-1\}$ and $2(N-l') \in \{0,2,\ldots,N-2\}$. Thus,

$$\left\| \sum_{i=0}^{l'-1} \mathbf{z}_{\pi'(i)} \right\|_\infty \leq \left\| \sum_{i=0}^{N-1} \mathbf{z}_i \right\|_\infty + \frac{1}{2} \max_{n \in [N]} \left\| \sum_{i=0}^{n-1} \mathbf{z}_{\pi(i)} \right\|_\infty + Ca .$$

Thus, combining the two cases and using $\left\| \sum_{i=0}^{N-1} \mathbf{z}_i \right\|_\infty \leq b$, we obtain

$$\max_{n \in [N]} \left\| \sum_{i=0}^{n-1} \mathbf{z}_{\pi'(i)} \right\|_\infty \leq \frac{1}{2} \max_{n \in [N]} \left\| \sum_{i=0}^{n-1} \mathbf{z}_{\pi(i)} \right\|_\infty + Ca + b ,$$

which is the claimed bound. $\qquad \square$

**Lemma 5.** *Let* $\pi$, $\{\mathbf{z}_{\pi(n)}\}_{n=0}^{N-1}$ *and* $\frac{1}{N}\sum_{n=0}^{N-1} \mathbf{z}_{\pi(n)}$ *be the inputs of Algorithm 6, and* $\pi'$ *be the corresponding output. Suppose that* $N \mod S = 0$ *and* $S \mod 2 = 0$. *If* $\|\mathbf{z}_n\|_2 \leq a$ *for all* $n \in \{0,1,\ldots,N-1\}$ *and* $\left\| \sum_{i=0}^{N-1} \mathbf{z}_i \right\|_\infty \leq b$, *then, with probability at least* $1-\delta$,

$$\max_{m \in \{S, 2S, \ldots, N\}} \left\| \sum_{i=0}^{m-1} \mathbf{z}_{\pi'(i)} \right\|_\infty \leq \frac{1}{2} \max_{m \in \{S, 2S, \ldots, N\}} \left\| \sum_{i=0}^{m-1} \mathbf{z}_{\pi(i)} \right\|_\infty + Ca + b ,$$

*where* $C = 30 \log(\frac{dN}{2\delta}) = \mathcal{O}\left(\log\left(\frac{dN}{\delta}\right)\right) = \tilde{\mathcal{O}}(1)$ *is from Alweiss et al. [2021, Theorem 1.1].*

*Proof.* This lemma is first introduced in this paper and tailored to FL-GraB.

We use $\tilde{\epsilon}_j$ to denote the assigned sign of $\mathbf{d}_j = \mathbf{z}_{\pi(2j)} - \mathbf{z}_{\pi(2j+1)}$ for all $j \in \{0,1,\ldots\frac{N}{2}-1\}$; we use $\epsilon_i$ to denote the assigned sign of $\mathbf{z}_{\pi(i)}$ for all $i \in \{0,1,\ldots,N-1\}$. Since $\{\mathbf{d}_j\}_{j=0}^{\frac{N}{2}-1}$ is the input of Algorithm 3, according to Alweiss et al. [2021]'s Theorem 1.1, for all $l \in \{1, 2, \ldots, \frac{N}{2}\}$,

$$\left\| \sum_{j=0}^{l-1} \tilde{\epsilon}_j \mathbf{d}_j \right\|_\infty = \left\| \sum_{j=0}^{l-1} \tilde{\epsilon}_j \frac{\mathbf{d}_j}{\max_{j \in \{0,1,\ldots,l-1\}} \|\mathbf{d}_j\|_2} \right\|_\infty \cdot \max_{j \in \{0,1,\ldots,l-1\}} \|\mathbf{d}_j\|_2$$

$$\leq C \max_{j \in \{0,1,\ldots,l-1\}} \|\mathbf{d}_j\|_2 \leq 2Ca .$$

where the last inequality is because for any $j \in \{0,1,\ldots\frac{N}{2}-1\}$,

$$\|\mathbf{d}_j\|_2 = \left\| \mathbf{z}_{\pi(2j)} - \mathbf{z}_{\pi(2j+1)} \right\|_2 \leq \left\| \mathbf{z}_{\pi(2j)} \right\|_2 + \left\| \mathbf{z}_{\pi(2j+1)} \right\|_2 \leq 2a .$$

We define $x_l$ and $y_l$ for $l \in \{1, 2, \ldots, \frac{N}{2}\}$,

$$x_l = \sum_{j=0}^{l-1} \left( \mathbf{z}_{\pi(2j)} + \mathbf{z}_{\pi(2j+1)} \right) = \sum_{i=0}^{2l-1} \mathbf{z}_{\pi(i)},$$

$$y_l = \sum_{j=0}^{l-1} \left( \epsilon_{2j} \mathbf{z}_{\pi(2j)} + \epsilon_{2j+1} \mathbf{z}_{\pi(2j+1)} \right) = \sum_{j=0}^{l-1} \left( \tilde{\epsilon}_j \mathbf{z}_{\pi(2j)} - \tilde{\epsilon}_j \mathbf{z}_{\pi(2j+1)} \right) = \sum_{j=0}^{l-1} \tilde{\epsilon}_j \mathbf{d}_j .$$

Let $M^+ = \{i \in \{0, 1, \ldots, N-1\} \mid \epsilon_i = +1\}$ and $M^- = \{i \in \{0, 1, \ldots, N-1\} \mid \epsilon_i = -1\}$. Then, for all $l \in \{1, 2, \ldots, \frac{N}{2}\}$, it follows that

$$\sum_{i \in M^+ \cap \{0,1,\ldots,2l-1\}} \mathbf{z}_{\pi(i)} = \frac{1}{2} \sum_{j=0}^{l-1} \left( (1 + \tilde{\epsilon}_j) \mathbf{z}_{\pi(2j)} + (1 - \tilde{\epsilon}_j) \mathbf{z}_{\pi(2j+1)} \right) = \frac{1}{2} x_l + \frac{1}{2} y_l ,$$

$$\sum_{i \in M^- \cap \{0,1,\ldots,2l-1\}} \mathbf{z}_{\pi(i)} = \frac{1}{2} \sum_{j=0}^{l-1} \left( (1 - \tilde{\epsilon}_j) \mathbf{z}_{\pi(2j)} + (1 + \tilde{\epsilon}_j) \mathbf{z}_{\pi(2j+1)} \right) = \frac{1}{2} x_l - \frac{1}{2} y_l .$$

By using the triangle inequality, for all $l \in \{1, 2, \ldots, \frac{N}{2}\}$, we obtain

$$\left\| \sum_{i \in M^+ \cap \{0,1,\ldots,2l-1\}} \mathbf{z}_{\pi(i)} \right\|_\infty \leq \frac{1}{2} \|x_l\|_\infty + \frac{1}{2} \|y_l\|_\infty$$

$$= \frac{1}{2} \left\| \sum_{i=0}^{2l-1} \mathbf{z}_{\pi(i)} \right\|_\infty + \frac{1}{2} \left\| \sum_{j=0}^{l-1} \tilde{\epsilon}_j \cdot \mathbf{d}_j \right\|_\infty$$

$$\leq \frac{1}{2} \left\| \sum_{i=0}^{2l-1} \mathbf{z}_{\pi(i)} \right\|_\infty + Ca ,$$

$$\left\| \sum_{i \in M^- \cap \{0,1,\ldots,2l-1\}} \mathbf{z}_{\pi(i)} \right\|_\infty \leq \frac{1}{2} \left\| \sum_{i=0}^{2l-1} \mathbf{z}_{\pi(i)} \right\|_\infty + Ca .$$

Next, we consider the upper bound of $\left\| \sum_{i=0}^{l'-1} \mathbf{z}_{\pi'(i)} \right\|_\infty$ for all $l' \in \{\frac{1}{2}S, S, \frac{3}{2}S, \ldots, \frac{N}{S} \cdot S\}$.

If $l' \leq \frac{N}{S} \cdot \frac{1}{2}S$, or equivalently, $l' \in \{\frac{1}{2}S, S, \frac{3}{2}S, \ldots, \frac{N}{S} \cdot \frac{1}{2}S\} \subseteq \{1, 2, \ldots, \frac{N}{2}\}$, then we obtain

$$\left\| \sum_{i=0}^{l'-1} \mathbf{z}_{\pi'(i)} \right\|_\infty = \left\| \sum_{i \in M^+ \cap \{0,1,\ldots,2l'-1\}} \mathbf{z}_{\pi(i)} \right\|_\infty \leq \frac{1}{2} \left\| \sum_{i=0}^{2l'-1} \mathbf{z}_{\pi(i)} \right\|_\infty + Ca .$$

Then, note that if $l' \in \{\frac{1}{2}S, S, \frac{3}{2}S, \ldots, \frac{N}{S} \cdot \frac{1}{2}S\}$, which implies that $2l' \in \{S, 2S, 3S, \ldots, N\}$, then

$$\left\| \sum_{i=0}^{l'-1} \mathbf{z}_{\pi'(i)} \right\|_\infty \leq \frac{1}{2} \max_{m \in \{S, 2S, 3S, \ldots, N\}} \left\| \sum_{i=0}^{m-1} \mathbf{z}_{\pi(i)} \right\|_\infty + Ca .$$

If $l' > \frac{N}{S} \cdot \frac{1}{2} S$, or equivalently, $l' \in \{\left(\frac{N}{S} + 1\right) \frac{S}{2}, \left(\frac{N}{S} + 2\right) \frac{S}{2}, \ldots, N\}$, then we obtain

$$
\begin{aligned}
\left\| \sum_{i=0}^{l'-1} \mathbf{z}_{\pi'(i)} \right\|_\infty &= \left\| \sum_{i=0}^{N-1} \mathbf{z}_{\pi'(i)} - \sum_{i=l'}^{N-1} \mathbf{z}_{\pi'(i)} \right\|_\infty \\
&\leq \left\| \sum_{i=0}^{N-1} \mathbf{z}_{\pi'(i)} \right\|_\infty + \left\| \sum_{i=l'}^{N-1} \mathbf{z}_{\pi'(i)} \right\|_\infty \\
&= \left\| \sum_{i=0}^{N-1} \mathbf{z}_{\pi'(i)} \right\|_\infty + \left\| \sum_{i \in M^- \cap \{0,1,\ldots,2(N-l')-1\}} \mathbf{z}_{\pi(i)} \right\|_\infty \\
&\leq \left\| \sum_{i=0}^{N-1} \mathbf{z}_{\pi'(i)} \right\|_\infty + \frac{1}{2} \left\| \sum_{i=0}^{2(N-l')-1} \mathbf{z}_{\pi(i)} \right\|_\infty + \frac{1}{2} \left\| \sum_{j=0}^{(N-l')-1} \tilde{\epsilon}_j \mathbf{d}_j \right\|_\infty .
\end{aligned}
$$

Note that if $l' \in \{\left(\frac{N}{S} + 1\right) \frac{S}{2}, \left(\frac{N}{S} + 2\right) \frac{S}{2}, \ldots, N\}$, then $(N - l') \in \{0, \frac{S}{2}, S, \ldots, \left(\frac{N}{S} - 1\right) \frac{S}{2}\}$ and $2(N - l') \in \{0, S, 2S, \ldots, N - S\}$. Thus, we obtain

$$
\begin{aligned}
\left\| \sum_{i=0}^{l'-1} \mathbf{z}_{\pi'(i)} \right\|_\infty &\leq \left\| \sum_{i=0}^{N-1} \mathbf{z}_{\pi(i)} \right\|_\infty + \frac{1}{2} \max_{m \in \{S, 2S, 3S, \ldots, N-S\}} \left\| \sum_{i=0}^{m-1} \mathbf{z}_{\pi(i)} \right\|_\infty + Ca \\
&\leq \left\| \sum_{i=0}^{N-1} \mathbf{z}_{\pi(i)} \right\|_\infty + \frac{1}{2} \max_{m \in \{S, 2S, 3S, \ldots, N\}} \left\| \sum_{i=0}^{m-1} \mathbf{z}_{\pi(i)} \right\|_\infty + Ca .
\end{aligned}
$$

The bounds for these two cases hold for all $l' \in \{\frac{1}{2}S, S, \frac{3}{2}S, \ldots, \frac{N}{S} \cdot S\}$, which means that

$$
\max_{l' \in \{\frac{1}{2}S, S, \frac{3}{2}S, \ldots, \frac{N}{S} \cdot S\}} \left\| \sum_{i=0}^{l'-1} \mathbf{z}_{\pi'(i)} \right\|_\infty \leq \frac{1}{2} \max_{m \in \{S, 2S, 3S, \ldots, N\}} \left\| \sum_{i=0}^{m-1} \mathbf{z}_{\pi(i)} \right\|_\infty + \left\| \sum_{i=0}^{N-1} \mathbf{z}_{\pi(i)} \right\|_\infty + Ca .
$$

Since $\{S, 2S, 3S, \ldots, \frac{N}{S} \cdot S\} \subseteq \{\frac{1}{2}S, S, \frac{3}{2}S, \ldots, \frac{N}{S} \cdot S\}$, then

$$
\max_{m \in \{S, 2S, 3S, \ldots, N\}} \left\| \sum_{i=0}^{m-1} \mathbf{z}_{\pi'(i)} \right\|_\infty \leq \frac{1}{2} \max_{m \in \{S, 2S, 3S, \ldots, N\}} \left\| \sum_{i=0}^{m-1} \mathbf{z}_{\pi(i)} \right\|_\infty + \left\| \sum_{i=0}^{N-1} \mathbf{z}_{\pi(i)} \right\|_\infty + Ca .
$$

Using $\left\| \sum_{i=0}^{N-1} \mathbf{z}_i \right\|_\infty \leq b$, we obtain the claimed bound. $\qquad\square$

# E   Theorem 1

## E.1   Proof of Theorem 1

We define the maximum parameter deviation (drift) in any epoch $q$, $\Delta_q$ as

$$
\Delta_q = \max_{n \in [N]} \left\| \mathbf{x}_q^n - \mathbf{x}_q^0 \right\|_p .
$$

**Lemma 6.** *If $\gamma L_p N \leq \frac{1}{32}$, the maximum parameter drift is bounded:*

$$
\begin{aligned}
\Delta_q &\leq \frac{32}{31} \gamma \bar{\phi}_q + \frac{32}{31} \gamma N \left\| \nabla f(\mathbf{x}_q) \right\| , \\
(\Delta_q)^2 &\leq 3\gamma^2 \left( \bar{\phi}_q \right)^2 + 3\gamma^2 N^2 \left\| \nabla f(\mathbf{x}_q) \right\|^2 .
\end{aligned}
$$

*Proof.* For any $n \in [N]$, it follows that

$$\left\| \mathbf{x}_q^n - \mathbf{x}_q^0 \right\|_p$$

$$= \gamma \left\| \sum_{i=0}^{n-1} \nabla f_{\pi_q(i)}(\mathbf{x}_q^i) \right\|_p$$

$$= \gamma \left\| \sum_{i=0}^{n-1} \left( \nabla f_{\pi(i)}(\mathbf{x}_q^i) - \nabla f_{\pi(i)}(\mathbf{x}_q^0) + \nabla f_{\pi_q(i)}(\mathbf{x}_q^0) - \nabla f(\mathbf{x}_q^0) + \nabla f(\mathbf{x}_q^0) \right) \right\|_p$$

$$\leq \gamma \left\| \sum_{i=0}^{n-1} \left( \nabla f_{\pi_q(i)}(\mathbf{x}_q^i) - \nabla f_{\pi_q(i)}(\mathbf{x}_q^0) \right) \right\|_p + \gamma \left\| \sum_{i=0}^{n-1} \left( \nabla f_{\pi_q(i)}(\mathbf{x}_q^0) - \nabla f(\mathbf{x}_q^0) \right) \right\|_p + \gamma \left\| \sum_{i=0}^{n-1} \nabla f(\mathbf{x}_q^0) \right\|_p$$

$$\leq \gamma L_p \sum_{i=0}^{n-1} \left\| \mathbf{x}_q^i - \mathbf{x}_q^0 \right\|_p + \gamma \phi^n + \gamma n \left\| \nabla f(\mathbf{x}_q^0) \right\|_p$$

$$\leq \gamma L_p N \Delta_q + \gamma \bar{\phi} + \gamma N \left\| \nabla f(\mathbf{x}_q^0) \right\|_p.$$

Note that this bound holds for any $n \in [N]$. This means

$$\Delta_q \leq \gamma L_p N \Delta_q + \gamma \bar{\phi}_q + \gamma N \left\| \nabla f(\mathbf{x}_q) \right\|_p.$$

Then, using $\gamma L_p N \leq \frac{1}{32}$, we have

$$\Delta_q \leq \frac{32}{31} \gamma \bar{\phi}_q + \frac{32}{31} \gamma N \left\| \nabla f(\mathbf{x}_q) \right\|_p,$$

$$(\Delta_q)^2 \leq 3\gamma^2 \left( \bar{\phi}_q \right)^2 + 3\gamma^2 N^2 \left\| \nabla f(\mathbf{x}_q) \right\|_p^2.$$

At last, using $\|\mathbf{x}\|_p \leq \|\mathbf{x}\|$ for $\mathbf{x} \in \mathbb{R}^d$ and $p \geq 2$, we obtain the claim of this lemma. $\qquad \square$

*Proof of Theorem 1.* For permutation-based SGD, the cumulative updates over any epoch $q$ are

$$\mathbf{x}_{q+1} - \mathbf{x}_q = -\gamma \sum_{n=0}^{N-1} \nabla f_{\pi_q(n)}(\mathbf{x}_q^n). \tag{8}$$

Since the global objective function $f$ is $L$-smooth, it follows that

$$f(\mathbf{x}_{q+1}) \leq f(\mathbf{x}_q) + \langle \nabla f(\mathbf{x}_q), \mathbf{x}_{q+1} - \mathbf{x}_q \rangle + \frac{1}{2} L \left\| \mathbf{x}_{q+1} - \mathbf{x}_q \right\|^2. \tag{9}$$

Using Equation (8), we obtain

$$\langle \nabla f(\mathbf{x}_q), \mathbf{x}_{q+1} - \mathbf{x}_q \rangle$$

$$= -\gamma N \left\langle \nabla f(\mathbf{x}_q), \frac{1}{N} \sum_{n=0}^{N-1} \nabla f_{\pi(n)}(\mathbf{x}_q^n) \right\rangle$$

$$= -\frac{1}{2} \gamma N \left\| \nabla f(\mathbf{x}_q) \right\|^2 - \frac{1}{2} \gamma N \left\| \frac{1}{N} \sum_{n=0}^{N-1} \nabla f_{\pi_q(n)}(\mathbf{x}_q^n) \right\|^2 + \frac{1}{2} \gamma N \left\| \frac{1}{N} \sum_{n=0}^{N-1} \nabla f_{\pi_q(n)}(\mathbf{x}_q^n) - \nabla f(\mathbf{x}_q) \right\|^2,$$

where the second equality uses $2\langle \mathbf{x}, \mathbf{y} \rangle = \|\mathbf{x}\|^2 + \|\mathbf{y}\|^2 - \|\mathbf{x} - \mathbf{y}\|^2$. Using Equation (8), we obtain

$$\frac{1}{2} L \left\| \mathbf{x}_{q+1} - \mathbf{x}_q \right\|^2 = \frac{1}{2} \gamma^2 L N^2 \left\| \frac{1}{N} \sum_{n=0}^{N-1} \nabla f_{\pi_q(n)}(\mathbf{x}_q^n) \right\|^2.$$

Plugging back the preceding two inequalities into Equation (9), we obtain

$$f(\mathbf{x}_{q+1}) \leq f(\mathbf{x}_q) + \langle \nabla f(\mathbf{x}_q), \mathbf{x}_{q+1} - \mathbf{x}_q \rangle + \frac{1}{2} L \left\| \mathbf{x}_{q+1} - \mathbf{x}_q \right\|^2$$

$$\leq f(\mathbf{x}_q) - \frac{1}{2} \gamma N \left\| \nabla f(\mathbf{x}_q) \right\|^2 - \frac{1}{2} \gamma N (1 - \gamma L N) \mathbb{E} \left\| \frac{1}{N} \sum_{n=0}^{N-1} \nabla f_{\pi_q(n)}(\mathbf{x}_q^n) \right\|^2$$

$$+ \frac{1}{2} \gamma N \left\| \frac{1}{N} \sum_{n=0}^{N-1} \nabla f_{\pi_q(n)}(\mathbf{x}_q^n) - \nabla f(\mathbf{x}_q) \right\|^2.$$

Since $\gamma L N \leq 1$, we obtain

$$f(\mathbf{x}_{q+1}) \leq f(\mathbf{x}_q) - \frac{1}{2}\gamma N \left\| \nabla f(\mathbf{x}_q) \right\|^2 + \frac{1}{2}\gamma N \left\| \frac{1}{N} \sum_{n=0}^{N-1} \nabla f_{\pi_q(n)}(\mathbf{x}_q^n) - \nabla f(\mathbf{x}_q) \right\|^2. \qquad (10)$$

Since each local objective function $f_n$ is $L_{2,p}$-smooth, we have

$$\mathrm{T}_3 \text{ in } (10) = \frac{1}{2}\gamma N \left\| \frac{1}{N} \sum_{n=0}^{N-1} \nabla f_{\pi_q(n)}(\mathbf{x}_q^n) - \nabla f(\mathbf{x}_q) \right\|^2$$

$$= \frac{1}{2}\gamma N \left\| \frac{1}{N} \sum_{n=0}^{N-1} \left( \nabla f_{\pi_q(n)}(\mathbf{x}_q^n) - \nabla f_{\pi_q(n)}(\mathbf{x}_q) \right) \right\|^2$$

$$\leq \frac{1}{2}\gamma L_{2,p}^2 \sum_{n=0}^{N-1} \left\| \mathbf{x}_q^n - \mathbf{x}_q \right\|_p^2.$$

Plugging the preceding inequality back into Equation (10), we obtain

$$f(\mathbf{x}_{q+1}) \leq f(\mathbf{x}_q) - \frac{1}{2}\gamma N \left\| \nabla f(\mathbf{x}_q) \right\|^2 + \frac{1}{2}\gamma L_{2,p}^2 \sum_{n=0}^{N-1} \left\| \mathbf{x}_q^n - \mathbf{x}_q \right\|_p^2$$

$$\leq f(\mathbf{x}_q) - \frac{1}{2}\gamma N \left\| \nabla f(\mathbf{x}_q) \right\|^2 + \frac{1}{2}\gamma L_{2,p}^2 N \left( \Delta_q \right)^2$$

$$\leq f(\mathbf{x}_q) - \frac{1}{2}\gamma N \left\| \nabla f(\mathbf{x}_q) \right\|^2 + \frac{1}{2}\gamma L_{2,p}^2 N \left( 3\gamma^2 \left( \bar{\phi}_q \right)^2 + 3\gamma^2 N^2 \left\| \nabla f(\mathbf{x}_q) \right\|^2 \right)$$

$$\leq f(\mathbf{x}_q) - \frac{1}{2}\gamma N \left( 1 - 3\gamma^2 L_{2,p}^2 N^2 \right) \left\| \nabla f(\mathbf{x}_q) \right\|^2 + \frac{3}{2}\gamma^3 L_{2,p}^2 N \left( \bar{\phi}_q \right)^2$$

$$\leq f(\mathbf{x}_q) - \frac{255}{512}\gamma N \left\| \nabla f(\mathbf{x}_q) \right\|^2 + 2\gamma^3 L_{2,p}^2 N \left( \bar{\phi}_q \right)^2, \qquad (11)$$

where the third inequality uses Lemma 6 and the last inequality uses $\gamma L_{2,p} N \leq \frac{1}{32}$. Then,

$$f(\mathbf{x}_{q+1}) - f(\mathbf{x}_q) \leq -\frac{255}{512}\gamma N \left\| \nabla f(\mathbf{x}_q) \right\|^2 + 2\gamma^3 L_{2,p}^2 N \left( \bar{\phi}_q \right)^2$$

Average both sides over $q \in \{0, 1, \ldots, Q-1\}$, we obtain

$$\frac{f(\mathbf{x}_Q) - f(\mathbf{x}_0)}{\gamma N Q} \leq -\frac{255}{512} \frac{1}{Q} \sum_{q=0}^{Q-1} \left\| \nabla f(\mathbf{x}_q) \right\|^2 + 2\gamma^2 L_{2,p}^2 \frac{1}{Q} \sum_{q=0}^{Q-1} \left( \bar{\phi}_q \right)^2. \qquad (12)$$

Since

$$\left( \bar{\phi}_q \right)^2 \leq A_1 \left( \bar{\phi}_{q-1} \right)^2 + A_2 \left( \bar{\phi}_{q-2} \right)^2 + \cdots + A_\nu \left( \bar{\phi}_{q-\nu} \right)^2$$
$$+ B_0 \left\| \nabla f(\mathbf{x}_q) \right\|^2 + B_1 \left\| \nabla f(\mathbf{x}_{q-1}) \right\|^2 + \cdots + B_\nu \left\| \nabla f(\mathbf{x}_{q-\nu}) \right\|^2 + D,$$

we obtain

$$\frac{1}{Q} \sum_{q=0}^{Q-1} \left( \bar{\phi}_q \right)^2 \leq \frac{1}{\left( 1 - \sum_{i=1}^{\nu} A_i \right)} \frac{1}{Q} \sum_{i=0}^{\nu-1} \left( \bar{\phi}_i \right)^2$$

$$+ \frac{\left( \sum_{i=0}^{\nu} B_i \right)}{\left( 1 - \sum_{i=1}^{\nu} A_i \right)} \frac{1}{Q} \sum_{q=0}^{Q-1} \left\| \nabla f(\mathbf{x}_q) \right\|^2 + \frac{1}{\left( 1 - \sum_{i=1}^{\nu} A_i \right)} D. \qquad (13)$$

For $\left( \bar{\phi}_i \right)^2$ $(0 \leq i \leq \nu - 1)$, we have

$$\left( \bar{\phi}_i \right)^2 \leq \tilde{B} \left\| \nabla f(\mathbf{x}_i) \right\|^2 + \tilde{D} \implies \sum_{i=0}^{\nu-1} \left( \bar{\phi}_i \right)^2 \leq \tilde{B} \sum_{q=0}^{Q-1} \left\| \nabla f(\mathbf{x}_q) \right\|^2 + \nu \tilde{D}. \qquad (14)$$

Plugging Equation (14) into Equation (13), and the resulting inequality into Equation (12), and then using the condition

$$\gamma \leq \min\left\{\frac{1}{4} \cdot \frac{\sqrt{1 - \sum_{i=1}^{\nu} A_i}}{L_{2,p}\sqrt{\sum_{i=0}^{\nu} B_i}}, \frac{1}{4} \cdot \frac{\sqrt{1 - \sum_{i=1}^{\nu} A_i}}{L_{2,p}\sqrt{\tilde{B}}}\right\},$$

we obtain

$$\frac{1}{Q}\sum_{q=0}^{Q-1}\|\nabla f(\mathbf{x}_q)\|^2 \leq 5 \cdot \frac{f(\mathbf{x}_0) - f(\mathbf{x}_Q)}{\gamma N Q} + c \cdot \gamma^2 L_{2,p}^2 \frac{1}{Q}\nu\tilde{D} + c \cdot \gamma^2 L_{2,p}^2 D,$$

where $c = \frac{10}{(1 - \sum_{i=1}^{\nu} A_i)}$. Using $f(\mathbf{x}_0) - f(\mathbf{x}_Q) \leq f(\mathbf{x}_0) - f_* = F_0$, we obtain the claimed bound.

At last, we summarize the constraints on the step size $\gamma$:

$$\gamma \leq \min\left\{\frac{1}{4} \cdot \frac{\sqrt{1 - \sum_{i=1}^{\nu} A_i}}{L_{2,p}\sqrt{\sum_{i=0}^{\nu} B_i}}, \frac{1}{4} \cdot \frac{\sqrt{1 - \sum_{i=1}^{\nu} A_i}}{L_{2,p}\sqrt{\tilde{B}}}\right\},$$

$$\gamma L N \leq 1,$$

$$\gamma L_{2,p} N \leq \frac{1}{32},$$

$$\gamma L_p N \leq \frac{1}{32},$$

where the last one is from Lemma 6. For simplicity, we use a tighter constraint

$$\gamma \leq \min\left\{\frac{1}{LN}, \frac{1}{32L_{2,p}N}, \frac{\sqrt{1 - \sum_{i=1}^{\nu} A_i}}{4L_{2,p}\sqrt{\sum_{i=0}^{\nu} B_i}}, \frac{\sqrt{1 - \sum_{i=1}^{\nu} A_i}}{4L_{2,p}\sqrt{\tilde{B}}}, \frac{1}{32L_p N}\right\}.$$

$\square$

## F Special Cases in SGD

As shown in Theorem 1, the constraint of $\gamma$ relies on the choices of all $A_i$ and $B_i$ (for $i \in \{1, 2, \ldots, \nu\}$). For clarity, we summarize the constraints of the existing permutation-based SGD algorithms in Table 7. It can be seen that the choices of $A_i$ and $B_i$ do not impose stronger constraints than the existing works [Lu et al., 2022b,a; Cooper et al., 2023; Koloskova et al., 2024].

Table 7: Specific choices of $A_i$, $B_i$ and $D$ for different algorithms. The coefficients not explicitly specified equal 0. The numerical constants and polylogarithmic factors of $B_i$ and $D$ are omitted.

| Algorithm | $B_0$ | $A_1$ | $B_1$ | $A_2$ | $B_2$ | $D$ | $\gamma$ |
|---|---|---|---|---|---|---|---|
| AP | $N^2\alpha^2$ | 0 | 0 | 0 | 0 | $N^2\varsigma^2$ | $\gamma \lesssim \frac{1}{LN(1+\alpha)}$ |
| RR/FlipFlop | $N^2\alpha^2$ | 0 | 0 | 0 | 0 | $N\varsigma^2$ | $\gamma \lesssim \frac{1}{LN(1+\alpha/\sqrt{N})}$ |
| GraB-proto | 0 | $\frac{3}{4}^{(1)}$ | $N^2 + \alpha^2$ | 0 | 0 | $\varsigma^2$ | $\gamma \lesssim \min\{\frac{1}{LN}, \frac{1}{L_{2,\infty}N(1+\alpha)}, \frac{1}{L_\infty N}\}$ |
| GraB | 0 | $\frac{3}{5}^{(1)}$ | $N^2 + \alpha^2$ | $\frac{1}{50}^{(1)}$ $N^2$ | | $\varsigma^2$ | $\gamma \lesssim \min\{\frac{1}{LN}, \frac{1}{L_{2,\infty}N(1+\alpha)}, \frac{1}{L_\infty N}\}$ |
| PairGraB | 0 | $\frac{4}{5}^{(1)}$ | $N^2 + \alpha^2$ | 0 | 0 | $\varsigma^2$ | $\gamma \lesssim \min\{\frac{1}{LN}, \frac{1}{L_{2,\infty}N(1+\alpha)}, \frac{1}{L_\infty N}\}$ |

$^1$ $A_i$ may take other values as long as $\sum_{i=1}^{\nu} A_i < 1$ for GraBs.

### F.1 Arbitrary Permutation (AP)

**Proposition 2** (AP). *Suppose that Assumption 3 holds. Then, we can obtain that, for $q \geq 0$,*

$$\left(\bar{\phi}_q\right)^2 \leq N^2\|\nabla f(\mathbf{x}_q)\|^2 + N^2\varsigma^2.$$

*Applying Theorem 1 and tuning the step size, we obtain*

$$\frac{1}{Q}\sum_{q=0}^{Q-1}\|\nabla f(\mathbf{x}_q)\|^2 = \mathcal{O}\left(\frac{LF_0}{Q} + \left(\frac{LF_0\varsigma}{Q}\right)^{\frac{2}{3}}\right).$$

*Proof.* For any $q$, it follows that

$$\left(\bar{\phi}_q\right)^2 = \max_{n \in [N]} \left\| \sum_{i=0}^{n-1} \left(\nabla f_{\pi_q(i)}(\mathbf{x}_q) - \nabla f(\mathbf{x}_q)\right) \right\|^2$$

$$\leq \max_{n \in [N]} \left\{ n \sum_{i=0}^{n-1} \left\| \nabla f_{\pi_q(i)}(\mathbf{x}_q) - \nabla f(\mathbf{x}_q) \right\|^2 \right\}$$

$$\leq \max_{n \in [N]} \left\{ n^2 G_q^2 \right\}$$

$$\leq N^2 \alpha^2 \left\| \nabla f(\mathbf{x}_q) \right\|^2 + N^2 \varsigma^2.$$

In this example, for Theorem 1, $p = 2$, $\nu = 0$, $B_0 = N^2\alpha^2$, $D = N^2\varsigma^2$ and $c = 10$. These lead to

$$\frac{1}{Q} \sum_{q=0}^{Q-1} \left\| \nabla f(\mathbf{x}_q) \right\|^2 = \mathcal{O}\left( \frac{F_0}{\gamma N Q} + \gamma^2 L^2 N^2 \varsigma^2 \right).$$

Next, we summarize the constraints:

$$\gamma \leq \min \left\{ \frac{1}{LN}, \frac{1}{32 L_{2,p} N}, \frac{\sqrt{1 - \sum_{i=1}^{\nu} A_i}}{4 L_{2,p} \sqrt{\sum_{i=0}^{\nu} B_i}}, \frac{\sqrt{1 - \sum_{i=1}^{\nu} A_i}}{4 L_{2,p} \sqrt{\tilde{B}}}, \frac{1}{32 L_p N} \right\}.$$

It is from Theorem 1. For simplicity, we can use a tighter constraint

$$\gamma \leq \frac{1}{32 L N (1 + \alpha)}.$$

After we use the effective step size $\tilde{\gamma} := \gamma N$, the constraint becomes

$$\tilde{\gamma} \leq \frac{1}{32 L (1 + \alpha)},$$

and the upper bound becomes

$$\frac{1}{Q} \sum_{q=0}^{Q-1} \left\| \nabla f(\mathbf{x}_q) \right\|^2 = \mathcal{O}\left( \frac{F_0}{\tilde{\gamma} Q} + \tilde{\gamma}^2 L^2 \varsigma^2 \right).$$

Applying Lemma 1, we obtain

$$\frac{1}{Q} \sum_{q=0}^{Q-1} \left\| \nabla f(\mathbf{x}_q) \right\|^2 = \mathcal{O}\left( \frac{L F_0 (1 + \alpha)}{Q} + \left( \frac{L F_0 N \varsigma}{N Q} \right)^{\frac{2}{3}} \right).$$

$\square$

### F.2  Random Reshuffling (RR)

**Proposition 3** (RR)**.** *Suppose that Assumption 3 holds. Then, we obtain that, for $q \geq 0$, with probability at least $1 - \delta$,*

$$\left(\bar{\phi}_q\right)^2 \leq 4 \log^2 \left( \frac{8}{\delta} \right) \left( N \alpha^2 \left\| \nabla f(\mathbf{x}_q) \right\|^2 + N \varsigma^2 \right).$$

*Applying Theorem 1 and tuning the step size, we obtain that, with probability at least $1 - Q\delta$,*

$$\frac{1}{Q} \sum_{q=0}^{Q-1} \left\| \nabla f(\mathbf{x}_q) \right\|^2 = \tilde{\mathcal{O}}\left( \frac{L F_0 \left(1 + \frac{\alpha}{\sqrt{N}}\right)}{Q} + \left( \frac{L F_0 \sqrt{N} \varsigma}{N Q} \right)^{\frac{2}{3}} \right).$$

*Proof.* Since the permutations $\{\pi_q\}$ are independent across different epochs, for any $q$, when conditional on $\mathbf{x}_q$, we obtain that, with probability at least $1 - \delta$,

$$\left(\bar{\phi}_q\right)^2 = \max_{n \in [N]} \left\| \sum_{i=0}^{n-1} \left(\nabla f_{\pi_q(i)}(\mathbf{x}_q) - \nabla f(\mathbf{x}_q)\right) \right\|^2 \leq N C \alpha^2 \left\| \nabla f(\mathbf{x}_q) \right\|^2 + N C \varsigma^2, \quad (15)$$

where $C = 4\log^2\left(\frac{8}{\delta}\right)$ and the last inequality uses Yu and Li [2023]'s Proposition 2.3.

In this example, for Theorem 1, $p = 2$, $\nu = 0$, $B_0 = NC\alpha^2$, $D = NC\varsigma^2$ and $c = 10$. These lead to

$$\frac{1}{Q}\sum_{q=0}^{Q-1}\|\nabla f(\mathbf{x}_q)\|^2 = \mathcal{O}\left(\frac{F_0}{\gamma NQ} + \gamma^2 L^2 NC\varsigma^2\right),$$

Equation (15) is used for each epoch (that is, for $Q$ times), so, by the union bound, the preceding bound holds with probability at least $1 - Q\delta$. Next, we summarize the constraints:

$$\gamma \leq \min\left\{\frac{1}{LN}, \frac{1}{32L_{2,p}N}, \frac{\sqrt{1 - \sum_{i=1}^{\nu}A_i}}{4L_{2,p}\sqrt{\sum_{i=0}^{\nu}B_i}}, \frac{\sqrt{1 - \sum_{i=1}^{\nu}A_i}}{4L_{2,p}\sqrt{\tilde{B}}}, \frac{1}{32L_pN}\right\}.$$

It is from Theorem 1. For simplicity, we can use a tighter constraint

$$\gamma \leq \frac{1}{32L\left(N + \sqrt{NC}\alpha\right)}.$$

After we use the effective step size $\tilde{\gamma} := \gamma N$, the constrain becomes

$$\tilde{\gamma} \leq \frac{1}{32L\left(1 + \sqrt{\frac{C}{N}}\alpha\right)},$$

and the upper bound becomes

$$\frac{1}{Q}\sum_{q=0}^{Q-1}\|\nabla f(\mathbf{x}_q)\|^2 = \tilde{\mathcal{O}}\left(\frac{F_0}{\tilde{\gamma}Q} + \tilde{\gamma}^2 L^2 \frac{1}{N}\varsigma^2\right).$$

Applying Lemma 1, we obtain

$$\frac{1}{Q}\sum_{q=0}^{Q-1}\|\nabla f(\mathbf{x}_q)\|^2 = \tilde{\mathcal{O}}\left(\frac{LF_0\left(1 + \frac{\alpha}{\sqrt{N}}\right)}{Q} + \left(\frac{LF_0\sqrt{N}\varsigma}{NQ}\right)^{\frac{2}{3}}\right).$$

$\square$

### F.3 FlipFlop

**Introduction of FlipFlop.** The permutation in each even epoch ($q = 0, 2, \ldots$) is generated randomly and independently; the permutation in each odd epoch ($q = 1, 3, \ldots$) is the reversed version of the previous epoch's permutation. Take $N = 3$ as an example: If $\pi_0 = (0, 1, 2)$, then $\pi_1 = (2, 1, 0)$.

**FlipFlop belongs to IP.** We regard each even epoch and its succeeding odd epoch as a meta epoch. Accordingly, we regard the permutation in such a meta epoch as the meta permutation (denoted as $\sigma$). For instance, if the original permutations are $\pi_0 = (0, 1, 2)$ and $\pi_1 = (2, 1, 0)$, then the meta permutation is $\sigma_0 = (\pi_0, \pi_1) = (\pi_0, \text{reverse}(\pi_0)) = (0, 1, 2, 2, 1, 0)$. Now, we see that the meta permutations are independent. Thus, it should be seen as a variant of IP.

**Analysis of FlipFlop.** Since the length of each meta permutation is $2N$ and the meta permutations are independent, we define the order error of each meta epoch $m$ ($m = 0, 1, \ldots, \frac{Q}{2} - 1$) as $\bar{\psi}_m := \max_{n\in[2N]}\left\{\|\sum_{i=0}^{n-1}(\nabla f_{\sigma_m(i)}(\mathbf{x}_{2m}) - \nabla f(\mathbf{x}_{2m}))\|\right\}$ to distinguish it from the original order error $\bar{\phi}_q$. As shown in Proposition 4, FlipFlop achieves the same convergence rate as RR. This is aligned with the conclusion of Chae et al. [2024], and the observation of Lu et al. [2022a].

**Proposition 4** (FlipFlop)**.** *For FlipFlop, we assume that $Q \mod 2 = 0$. Suppose that Assumption 3 holds. Then, we can obtain that, for $m = 0, 1, \ldots, \frac{Q}{2} - 1$, with probability at least $1 - \delta$,*

$$\left(\bar{\psi}_m\right)^2 \leq 4\log^2\left(\frac{8}{\delta}\right)\left(N\|\nabla f(\mathbf{x}_{2m})\|^2 + N\varsigma^2\right).$$

*Applying Theorem 1 and tuning the step size, we obtain that, with probability at least $1 - {(Q\delta)}/{2}$,*

$$\frac{1}{Q/2}\sum_{m=0}^{Q/2-1}\|\nabla f(\mathbf{x}_{2m})\|^2 = \tilde{\mathcal{O}}\left(\frac{LF_0\left(1 + \frac{\alpha}{\sqrt{N}}\right)}{Q} + \left(\frac{LF_0\sqrt{N}\varsigma}{NQ}\right)^{\frac{2}{3}}\right).$$

*Proof.* FlipFlop is one variant of IP. For clarity, we define the order error of each meta epoch $m$ as

$$\bar{\psi}_m := \max_{n \in [2N]} \left\{ \psi_m^n := \left\| \sum_{i=0}^{n-1} (\nabla f_{\sigma_m(i)}(\mathbf{x}_{2m}) - \nabla f(\mathbf{x}_{2m})) \right\| \right\},$$

where $\sigma_m$ is the meta permutation $\sigma_m = (\pi_{2m}, \pi_{2m+1}) = (\pi_{2m}, \text{reverse}(\pi_{2m}))$. Since the meta permutations are independent, we next deal with each $\bar{\psi}_m$ separately, and thus drop the subscripts $m$. For each $1 \le n \le N$,

$$(\psi^n)^2 = \left\| \sum_{i=0}^{n-1} \left( \nabla f_{\sigma(i)}(\mathbf{x}) - \nabla f(\mathbf{x}) \right) \right\|^2$$

$$= \left\| \sum_{i=0}^{n-1} \left( \nabla f_{\pi(i)}(\mathbf{x}) - \nabla f(x) \right) \right\|^2$$

$$\le NC\alpha^2 \left\| \nabla f(\mathbf{x}) \right\|^2 + NC\varsigma^2,$$

where $C = 4 \log^2 \left( \frac{8}{\delta} \right)$ and the last inequality uses Yu and Li [2023]'s Proposition 2.3. Similar to Chae et al. [2024, Lemma E.5], we obtain that, for each $N + 1 \le n \le 2N$,

$$(\psi^n)^2 = \left\| \sum_{i=0}^{n-1} \left( \nabla f_{\sigma(i)}(\mathbf{x}) - \nabla f(\mathbf{x}) \right) \right\|^2$$

$$= \left\| \sum_{i=0}^{2N-1} \left( \nabla f_{\sigma(i)}(\mathbf{x}) - \nabla f(\mathbf{x}) \right) - \sum_{i=0}^{n-1} \left( \nabla f_{\sigma(i)}(\mathbf{x}) - \nabla f(\mathbf{x}) \right) \right\|^2$$

$$= \left\| \sum_{i=n}^{2N-1} \left( \nabla f_{\sigma(i)}(\mathbf{x}) - \nabla f(\mathbf{x}) \right) \right\|^2$$

$$= \left\| \sum_{i=0}^{(2N-n)-1} \left( \nabla f_{\pi(i)}(\mathbf{x}) - \nabla f(\mathbf{x}) \right) \right\|^2$$

$$\le NC\alpha^2 \left\| \nabla f(\mathbf{x}) \right\|^2 + NC\varsigma^2.$$

This implies that $\bar{\psi}_m \le NC\varsigma^2 + NC\alpha^2 \left\| \nabla f(\mathbf{x}_{2m}) \right\|^2$ for each meta epoch $m$. Following the same steps as those of RR in Appendix F.2, we obtain

$$\frac{1}{Q/2} \sum_{m=0}^{Q/2-1} \left\| \nabla f(\mathbf{x}_{2m}) \right\|^2 = \tilde{\mathcal{O}} \left( \frac{LF_0 \left( 1 + \frac{\alpha}{\sqrt{N}} \right)}{Q} + \left( \frac{LF_0 \sqrt{N}\varsigma}{NQ} \right)^{\frac{2}{3}} \right),$$

which shows the sames rate as that of RR.

$\square$

### F.4  One Permutation (OP)

OP, the simplest DP algorithm, is one persistent topic due to its simple implementation. However, as shown in the prior works [Mishchenko et al., 2020; Koloskova et al., 2024], for non-convex objective functions, its convergence rate is no better than AP. In this section, we show that our framework can still help analyze the convergence of OP.

As done in the prior works, we can use the bound of AP as the bound of OP. However, *this general bound (for AP) cannot catch the key characteristic of OP* that the initial permutation is reused for the subsequent epochs. To further explore the potential of OP, we have the following analysis.

In OP, the key characteristic is that *the initial permutation is reused* for the subsequent epochs. To fully use this characteristic, we try to establish the relation between $\bar{\phi}_q$ and $\bar{\phi}_0$. Specifically, for all $q \ge 1$ and $n \in [N]$ (here, $p \ge 2$),

$$\phi_q^n \le 2LN \left\| \mathbf{x}_q - \mathbf{x}_0 \right\|_p + \left\| \sum_{i=0}^{n-1} \left( \nabla f_{\pi_q(i)}(\mathbf{x}_0) - \nabla f(\mathbf{x}_0) \right) \right\|_p \le 2LN \left\| \mathbf{x}_q - \mathbf{x}_0 \right\|_p + \bar{\phi}_0,$$

where the last inequality is due to $\pi_q = \pi_0$ for all $q \geq 1$. Under the standard settings, to bound the term $\|\mathbf{x}_q - \mathbf{x}_0\|_p$, the step size $\gamma$ needs to be made very small (that is, $\gamma \lesssim \frac{1}{LNQ}$), causing a very slow convergence. Therefore, to use the dependence property of OP, we need additional assumptions to bound $\|\mathbf{x}_q - \mathbf{x}_0\|_p$ [Lu et al., 2022b]. Further research is left for future work.

### F.5 GraB-proto

GraB-proto: Use `BasicBR` (Algorithm 5) as the `Permute` function in Algorithm 1, with the inputs of $\pi_q$, $\{\nabla f_{\pi_q(n)}(\mathbf{x}_q)\}_{n=0}^{N-1}$ and $\nabla f(\mathbf{x}_q)$, for each epoch $q$.

Thus, the key idea of our proof is as follows:

$$\bar{\phi}_{q+1} \to \max_{n \in [N]} \left\| \sum_{i=0}^{n-1} \left( \nabla f_{\pi_{q+1}(i)}(\mathbf{x}_q) - \nabla f(\mathbf{x}_q) \right) \right\|_\infty \overset{\text{Lemma 3}}{\to} \bar{\phi}_q.$$

*Proof of Proposition 1.* We need to find the relation between $\bar{\phi}_q$ and $\bar{\phi}_{q-1}$ for $q \geq 1$. For any $n \in [N]$,

$$
\begin{aligned}
\phi_{q+1}^n &= \left\| \sum_{i=0}^{n-1} \left( \nabla f_{\pi_{q+1}(i)}(\mathbf{x}_{q+1}) - \nabla f(\mathbf{x}_{q+1}) \right) \right\|_\infty \\
&\leq \left\| \sum_{i=0}^{n-1} \left( \nabla f_{\pi_{q+1}(i)}(\mathbf{x}_{q+1}) - \nabla f_{\pi_{q+1}(i)}(\mathbf{x}_q) \right) \right\|_\infty + \left\| \sum_{i=0}^{n-1} \left( \nabla f(\mathbf{x}_{q+1}) - \nabla f(\mathbf{x}_q) \right) \right\|_\infty \\
&\quad + \left\| \sum_{i=0}^{n-1} \left( \nabla f_{\pi_{q+1}(i)}(\mathbf{x}_q) - \nabla f(\mathbf{x}_q) \right) \right\|_\infty \\
&\leq \sum_{i=0}^{n-1} \left\| \nabla f_{\pi_{q+1}(i)}(\mathbf{x}_{q+1}) - \nabla f_{\pi_{q+1}(i)}(\mathbf{x}_q) \right\|_\infty + \sum_{i=0}^{n-1} \left\| \nabla f(\mathbf{x}_{q+1}) - \nabla f(\mathbf{x}_q) \right\|_\infty \\
&\quad + \left\| \sum_{i=0}^{n-1} \left( \nabla f_{\pi_{q+1}(i)}(\mathbf{x}_q) - \nabla f(\mathbf{x}_q) \right) \right\|_\infty \\
&\leq 2 L_\infty n \left\| \mathbf{x}_{q+1} - \mathbf{x}_q \right\|_\infty + \left\| \sum_{i=0}^{n-1} \left( \nabla f_{\pi_{q+1}(i)}(\mathbf{x}_q) - \nabla f(\mathbf{x}_q) \right) \right\|_\infty.
\end{aligned}
$$

Since the above inequality holds for all $n \in [N]$, we have

$$\bar{\phi}_{q+1} \leq 2 L_\infty N \left\| \mathbf{x}_{q+1} - \mathbf{x}_q \right\|_\infty + \max_{n \in [N]} \left\| \sum_{i=0}^{n-1} \left( \nabla f_{\pi_{q+1}(i)}(\mathbf{x}_q) - \nabla f(\mathbf{x}_q) \right) \right\|_\infty.$$

Note that $\nabla f_{\pi_q(i)}(\mathbf{x}_q) - \nabla f(\mathbf{x}_q)$ and $\nabla f_{\pi_{q+1}(i)}(\mathbf{x}_q) - \nabla f(\mathbf{x}_q)$ correspond to $\mathbf{z}_{\pi(i)}$ and $\mathbf{z}_{\pi'(i)}$ in Lemma 3, respectively. In GraB-proto, since

$$\|\nabla f_i(\mathbf{x}_q) - f(\mathbf{x}_q)\| \leq G_q, \quad \forall i \in \{0, 1, \ldots, N-1\},$$
$$\left\| \sum_{i=0}^{N-1} \left( \nabla f_i(\mathbf{x}_q) - f(\mathbf{x}_q) \right) \right\|_\infty = 0,$$

we apply Lemma 3 with $a = \varsigma$ and $b = 0$, and obtain

$$
\begin{aligned}
\max_{n \in [N]} \left\| \sum_{i=0}^{N-1} \left( \nabla f_{\pi_{q+1}(i)}(\mathbf{x}_q) - \nabla f(\mathbf{x}_q) \right) \right\|_\infty &\leq \frac{1}{2} \max_{n \in [N]} \left\| \sum_{i=0}^{n-1} \left( \nabla f_{\pi_q(i)}(\mathbf{x}_q) - \nabla f(\mathbf{x}_q) \right) \right\|_\infty + \frac{1}{2} C G_q \\
&= \frac{1}{2} \bar{\phi}_q + \frac{1}{2} C G_q.
\end{aligned}
$$

Using Lemma 6 that $\Delta_q \leq \frac{32}{31}\gamma\bar{\phi}_q + \frac{32}{31}\gamma N \|\nabla f(\mathbf{x}_q)\|$, we obtain

$$\bar{\phi}_{q+1} \leq 2L_\infty N \|\mathbf{x}_{q+1} - \mathbf{x}_q\|_\infty + \max_{n\in[N]} \left\|\sum_{i=0}^{n-1} \left(\nabla f_{\pi_{q+1}(i)}(\mathbf{x}_q) - \nabla f(\mathbf{x}_q)\right)\right\|_\infty$$

$$\leq 2L_\infty N \left(\frac{32}{31}\gamma\bar{\phi}_q + \frac{32}{31}\gamma N \|\nabla f(\mathbf{x}_q)\|\right) + \left(\frac{1}{2}\bar{\phi}_q + \frac{1}{2}CG_q\right)$$

$$\leq \frac{35}{62}\bar{\phi}_q + \frac{2}{31}N \|\nabla f(\mathbf{x}_q)\| + \frac{1}{2}CG_q,$$

where the last inequality uses $\gamma L_\infty N \leq \frac{1}{32}$. Next, we obtain

$$\left(\bar{\phi}_{q+1}\right)^2 \leq \frac{3}{4}\left(\bar{\phi}_q\right)^2 + \left(\frac{1}{50}N^2 + C^2\alpha^2\right) \|\nabla f(\mathbf{x}_q)\|^2 + C^2\varsigma^2.$$

As a result, the relation between $\bar{\phi}_q$ and $\bar{\phi}_{q-1}$ is

$$\left(\bar{\phi}_q\right)^2 \leq \frac{3}{4}\left(\bar{\phi}_{q-1}\right)^2 + \left(\frac{1}{50}N^2 + C^2\alpha^2\right) \|\nabla f(\mathbf{x}_{q-1})\|^2 + C^2\varsigma^2, \tag{16}$$

for $q \geq 1$. In addition, we need to obtain the bound of $\left(\bar{\phi}_0\right)^2$:

$$\left(\bar{\phi}_0\right)^2 = \max_{n\in[N]} \left\|\sum_{i=0}^{n-1} \left(\nabla f_{\pi_0(i)}(\mathbf{x}_0) - \nabla f(\mathbf{x}_0)\right)\right\|^2 \leq N^2\alpha^2 \|\nabla f(\mathbf{x}_0)\|^2 + N^2\varsigma^2.$$

In this example, for Theorem 1, $p = \infty$, $\nu = 1$, $A_1 = \frac{3}{4}$, $B_0 = 0$, $B_1 = \frac{1}{50}N^2 + C^2\alpha^2$, $D = C^2\varsigma^2$, $\tilde{B} = N^2\alpha^2$ and $\tilde{D} = N^2\varsigma^2$, and $c = 40$. These lead to

$$\frac{1}{Q}\sum_{q=0}^{Q-1} \|\nabla f(\mathbf{x}_q)\|^2 = \mathcal{O}\left(\frac{F_0}{\gamma NQ} + \gamma^2 L_{2,\infty}^2 N^2 \frac{1}{Q}\varsigma^2 + \gamma^2 L_{2,\infty}^2 C^2\varsigma^2\right).$$

where $F_0 = f(\mathbf{x}_0) - f_*$. Lemma 3 is used for each epoch (that is, for $Q$ times), so, by the union bound, the preceding bound holds with probability at least $1 - Q\delta$.

Next, we summarize the constraints on the step size:

$$\gamma \leq \min\left\{\frac{1}{LN}, \frac{1}{32L_{2,p}N}, \frac{\sqrt{1-\sum_{i=1}^{\nu}A_i}}{4L_{2,p}\sqrt{\sum_{i=0}^{\nu}B_i}}, \frac{\sqrt{1-\sum_{i=1}^{\nu}A_i}}{4L_{2,p}\sqrt{\tilde{B}}}, \frac{1}{32L_pN}\right\},$$

$$\gamma \leq \frac{1}{32L_\infty N},$$

where the first one is from Theorem 1 and the other is from the derivation of the relation. For simplicity, we can use a tighter constraint

$$\gamma \leq \min\left\{\frac{1}{LN}, \frac{1}{32L_{2,\infty}N(1+\alpha)}, \frac{1}{32L_\infty N}\right\}.$$

After we use the effective step size $\tilde{\gamma} := \gamma N$, the constraint will be

$$\tilde{\gamma} \leq \min\left\{\frac{1}{L}, \frac{1}{32L_{2,\infty}(1+\alpha)}, \frac{1}{32L_\infty}\right\},$$

and the upper bound will be

$$\frac{1}{Q}\sum_{q=0}^{Q-1} \|\nabla f(\mathbf{x}_q)\|^2 = \mathcal{O}\left(\frac{F_0}{\tilde{\gamma}Q} + \tilde{\gamma}^2 L_{2,\infty}^2 \frac{1}{Q}\varsigma^2 + \tilde{\gamma}^2 L_{2,\infty}^2 \frac{1}{N^2}C^2\varsigma^2\right).$$

Applying Lemma 1, we obtain

$$\frac{1}{Q}\sum_{q=0}^{Q-1} \|\nabla f(\mathbf{x}_q)\|^2 = \mathcal{O}\left(\frac{(L + L_{2,\infty}(1+\alpha) + L_\infty)F_0}{Q} + \frac{(L_{2,\infty}F_0\varsigma)^{\frac{2}{3}}}{Q} + \left(\frac{L_{2,\infty}F_0C\varsigma}{NQ}\right)^{\frac{2}{3}}\right).$$

$\square$

### F.6 PairGraB-proto

**Proposition 5** (PairGraB-proto). *Suppose that Assumption 3 holds. If each $f_n$ is $L_\infty$-smooth and $\gamma \leq \frac{1}{32L_\infty N}$, we obtain that, for $q = 0$, $\left(\bar{\phi}_0\right)^2 \leq N^2\alpha^2 \|\nabla f(\mathbf{x}_0)\|^2 + N^2\varsigma^2$, and for $q \geq 1$, with probability at least $1 - \delta$,*

$$\left(\bar{\phi}_q\right)^2 \leq \frac{3}{4}\left(\bar{\phi}_{q-1}\right)^2 + \left(\frac{1}{50}N^2 + 4C^2\alpha^2\right)\|\nabla f(\mathbf{x}_{q-1})\|^2 + 4C^2\varsigma^2,$$

*where $C = \mathcal{O}\left(\log\left(\frac{dN}{\delta}\right)\right) = \tilde{\mathcal{O}}(1)$. Applying Theorem 1 and tuning the step size, we obtain that, with probability at least $1 - Q\delta$,*

$$\frac{1}{Q}\sum_{q=0}^{Q-1}\|\nabla f(\mathbf{x}_q)\|^2 = \mathcal{O}\left(\frac{\tilde{L}F_0 + (L_{2,\infty}F_0\varsigma)^{\frac{2}{3}}}{Q} + \left(\frac{L_{2,\infty}F_0C\varsigma}{NQ}\right)^{\frac{2}{3}}\right),$$

*where $\tilde{L} = L + L_{2,\infty}(1 + \alpha) + L_\infty$.*

PairGraB-proto. Use `PairBR` (Algorithm 6) as the `Permute` function in Algorithm 1, with the inputs of $\pi_q$, $\{\nabla f_{\pi_q(n)}(\mathbf{x}_q)\}_{n=0}^{N-1}$ and $\nabla f(\mathbf{x}_q)$, for each epoch $q$.

Thus, the key idea of our proof is as follows:

$$\bar{\phi}_{q+1} \to \max_{n \in [N]}\left\|\sum_{i=0}^{n-1}\left(\nabla f_{\pi_{q+1}(i)}(\mathbf{x}_q) - \nabla f(\mathbf{x}_q)\right)\right\|_\infty \overset{\text{Lemma 4}}{\to} \bar{\phi}_q.$$

*Proof.* The proof of Example 5 is almost identical to that of Example 1, except that Lemma 3 is replaced by Lemma 4. This difference only causes that some numerical constants are changed accordingly.

$\square$

### F.7 GraB

**Proposition 6** (GraB). *Suppose that Assumption 3 holds. If each $f_n$ is $L_{2,\infty}$-smooth and $L_\infty$-smooth and $\gamma \leq \min\{\frac{1}{128L_{2,\infty}C}, \frac{1}{128L_\infty N}\}$, we obtain that, for $q = 0, 1$, $\left(\bar{\phi}_q\right)^2 \leq N^2\alpha^2 \|\nabla f(\mathbf{x}_q)\|^2 + N^2\varsigma^2$, and for $q \geq 2$, with probability at least $1 - \delta$,*

$$\left(\bar{\phi}_q\right)^2 \leq \frac{3}{5}\left(\bar{\phi}_{q-1}\right)^2 + \frac{1}{50}\left(\bar{\phi}_{q-2}\right)^2$$
$$+ \left(\frac{1}{50}N^2 + 2C^2\alpha^2\right)\|\nabla f(\mathbf{x}_{q-1})\|^2 + \frac{1}{50}N^2\|\nabla f(\mathbf{x}_{q-2})\|^2 + 2C^2\varsigma^2,$$

*where $C = \mathcal{O}\left(\log\left(\frac{dN}{\delta}\right)\right) = \tilde{\mathcal{O}}(1)$. Applying Theorem 1 and tuning the step size, we obtain that, with probability at least $1 - Q\delta$,*

$$\frac{1}{Q}\sum_{q=0}^{Q-1}\|\nabla f(\mathbf{x}_q)\|^2 = \mathcal{O}\left(\frac{\tilde{L}F_0 + (L_{2,\infty}F_0\varsigma)^{\frac{2}{3}}}{Q} + \left(\frac{L_{2,\infty}F_0C\varsigma}{NQ}\right)^{\frac{2}{3}}\right),$$

*where $\tilde{L} = L + L_{2,\infty}\left(1 + \frac{C}{N} + \alpha\right) + L_\infty$.*

GraB. Use `BasicBR` (Algorithm 5) as the `Permute` function in Algorithm 1, with the inputs of $\pi_q$, $\{\nabla f_{\pi_q(n)}(\mathbf{x}_q^n)\}_{n=0}^{N-1}$ and $\frac{1}{N}\sum_{n=0}^{N-1}\nabla f_{\pi_{q-1}(n)}(\mathbf{x}_{q-1}^n)$, for each epoch $q$.

Thus, the key idea of our proof is as follows:

$$\bar{\phi}_{q+1} \to \max_{n \in [N]}\left\|\sum_{i=0}^{n-1}\left(\nabla f_{\pi_{q+1}(i)}\left(\mathbf{x}_q^{\pi_q^{-1}(\pi_{q+1}(i))}\right) - \frac{1}{N}\sum_{l=0}^{N-1}\nabla f_{\pi_{q-1}(l)}(\mathbf{x}_{q-1}^l)\right)\right\|_\infty$$

$$\overset{\text{Lemma 3}}{\to} \max_{n \in [N]}\left\|\sum_{i=0}^{n-1}\left(\nabla f_{\pi_q(i)}\left(\mathbf{x}_q^i\right) - \frac{1}{N}\sum_{l=0}^{N-1}\nabla f_{\pi_{q-1}(l)}(\mathbf{x}_{q-1}^l)\right)\right\|_\infty \to \bar{\phi}_q.$$

*Proof.* We need to find the relation between $\bar{\phi}_{q+1}$ and $\bar{\phi}_q$.

$$\phi_{q+1}^n = \left\| \sum_{i=0}^{n-1} \left( \nabla f_{\pi_{q+1}(i)}(\mathbf{x}_{q+1}) - \nabla f(\mathbf{x}_{q+1}) \right) \right\|_\infty$$

$$\leq \left\| \sum_{i=0}^{n-1} \left( \nabla f_{\pi_{q+1}(i)}(\mathbf{x}_{q+1}) - \nabla f_{\pi_{q+1}(i)} \left( \mathbf{x}_q^{\pi_q^{-1}(\pi_{q+1}(i))} \right) \right) \right\|_\infty$$

$$+ \left\| \sum_{i=0}^{n-1} \left( \frac{1}{N} \sum_{l=0}^{N-1} \nabla f_{\pi_{q+1}(l)}(\mathbf{x}_{q+1}) - \frac{1}{N} \sum_{l=0}^{N-1} \nabla f_{\pi_{q-1}(l)}(\mathbf{x}_{q-1}^l) \right) \right\|_\infty$$

$$+ \left\| \sum_{i=0}^{n-1} \left( \nabla f_{\pi_{q+1}(i)} \left( \mathbf{x}_q^{\pi_q^{-1}(\pi_{q+1}(i))} \right) - \frac{1}{N} \sum_{l=0}^{N-1} \nabla f_{\pi_{q-1}(l)}(\mathbf{x}_{q-1}^l) \right) \right\|_\infty. \qquad (17)$$

Then,

$$T_1 \text{ in } (17) = \left\| \sum_{i=0}^{n-1} \left( \nabla f_{\pi_{q+1}(i)}(\mathbf{x}_{q+1}) - \nabla f_{\pi_{q+1}(i)} \left( \mathbf{x}_q^{\pi_q^{-1}(\pi_{q+1}(i))} \right) \right) \right\|_\infty$$

$$\leq \sum_{i=0}^{n-1} \left\| \nabla f_{\pi_{q+1}(i)}(\mathbf{x}_{q+1}) - \nabla f_{\pi_{q+1}(i)} \left( \mathbf{x}_q^{\pi_q^{-1}(\pi_{q+1}(i))} \right) \right\|_\infty$$

$$\leq L_\infty \sum_{i=0}^{n-1} \left\| \mathbf{x}_{q+1} - \mathbf{x}_q^{\pi_q^{-1}(\pi_{q+1}(i))} \right\|_\infty$$

$$\leq L_\infty \sum_{i=0}^{n-1} \left( \|\mathbf{x}_{q+1} - \mathbf{x}_q\|_\infty + \left\| \mathbf{x}_q - \mathbf{x}_q^{\pi_q^{-1}(\pi_{q+1}(i))} \right\|_\infty \right)$$

$$\leq 2L_\infty N \Delta_q,$$

$$T_2 \text{ in } (17) = \left\| \sum_{i=0}^{n-1} \left( \frac{1}{N} \sum_{l=0}^{N-1} \nabla f_{\pi_{q+1}(l)}(\mathbf{x}_{q+1}) - \frac{1}{N} \sum_{l=0}^{N-1} \nabla f_{\pi_{q-1}(l)} \left( \mathbf{x}_{q-1}^l \right) \right) \right\|_\infty$$

$$= \left\| \sum_{i=0}^{n-1} \left( \frac{1}{N} \sum_{l=0}^{N-1} \nabla f_l(\mathbf{x}_{q+1}) - \frac{1}{N} \sum_{l=0}^{N-1} \nabla f_l \left( \mathbf{x}_{q-1}^{\pi_{q-1}^{-1}(l)} \right) \right) \right\|_\infty$$

$$\leq \sum_{i=0}^{n-1} \frac{1}{N} \sum_{l=0}^{N-1} \left\| \nabla f_l(\mathbf{x}_{q+1}) - \nabla f_l \left( \mathbf{x}_{q-1}^{\pi_{q-1}^{-1}(l)} \right) \right\|_\infty$$

$$\leq L_\infty \sum_{i=0}^{n-1} \frac{1}{N} \sum_{l=0}^{N-1} \left\| \mathbf{x}_{q+1} - \mathbf{x}_{q-1}^{\pi_{q-1}^{-1}(l)} \right\|_\infty$$

$$\leq L_\infty \sum_{i=0}^{n-1} \frac{1}{N} \sum_{l=0}^{N-1} \left( \|\mathbf{x}_{q+1} - \mathbf{x}_q\|_\infty + \|\mathbf{x}_q - \mathbf{x}_{q-1}\|_\infty + \left\| \mathbf{x}_{q-1} - \mathbf{x}_{q-1}^{\pi_{q-1}^{-1}(l)} \right\|_\infty \right)$$

$$\leq L_\infty N \Delta_q + 2L_\infty N \Delta_{q-1}.$$

Since the preceding inequalities hold for all $n \in [N]$, we have

$$\bar{\phi}_{q+1} \leq 3L_\infty N \Delta_q + 2L_\infty N \Delta_{q-1}$$

$$+ \max_{n \in [N]} \left\| \sum_{i=0}^{n-1} \left( \nabla f_{\pi_{q+1}(i)} \left( \mathbf{x}_q^{\pi_q^{-1}(\pi_{q+1}(i))} \right) - \frac{1}{N} \sum_{l=0}^{N-1} \nabla f_{\pi_{q-1}(l)}(\mathbf{x}_{q-1}^l) \right) \right\|_\infty. \qquad (18)$$

Note that $\nabla f_{\pi_q(i)} \left( \mathbf{x}_q^i \right) - \frac{1}{N} \sum_{l=0}^{N-1} \nabla f_{\pi_{q-1}(l)}(\mathbf{x}_{q-1}^l)$ and $\nabla f_{\pi_{q+1}(i)} \left( \mathbf{x}_q^{\pi_q^{-1}(\pi_{q+1}(i))} \right) - \frac{1}{N} \sum_{l=0}^{N-1} \nabla f_{\pi_{q-1}(l)}(\mathbf{x}_{q-1}^l)$ correspond to $\mathbf{z}_{\pi(i)}$ and $\mathbf{z}_{\pi'(i)}$ in Lemma 3, respectively. We next

obtain the upper bounds of

$$\left\|\mathbf{z}_{\pi(i)}\right\|_2, \left\|\sum_{i=0}^{N-1} \mathbf{z}_{\pi(i)}\right\|_\infty \text{ and } \max_{n\in[N]} \left\|\sum_{i=0}^{n-1} \mathbf{z}_{\pi(i)}\right\|_\infty,$$

and then apply Lemma 3 to the last term on the right hand side in Equation (18).

$$\left\|\mathbf{z}_{\pi(i)}\right\|_2$$
$$= \left\|\nabla f_{\pi_q(i)}(\mathbf{x}_q^i) - \frac{1}{N}\sum_{l=0}^{N-1} \nabla f_{\pi_{q-1}(l)}\left(\mathbf{x}_{q-1}^l\right)\right\|_2$$
$$= \left\|\left(\nabla f_{\pi_q(i)}(\mathbf{x}_q^i) - \frac{1}{N}\sum_{l=0}^{N-1} \nabla f_{\pi_{q-1}(l)}\left(\mathbf{x}_{q-1}^l\right)\right) \pm \left(\nabla f_{\pi_q(i)}(\mathbf{x}_q) - \frac{1}{N}\sum_{l=0}^{N-1} \nabla f_{\pi_q(l)}(\mathbf{x}_q)\right)\right\|_2$$
$$\leq \left\|\nabla f_{\pi_q(i)}(\mathbf{x}_q^i) - \nabla f_{\pi_q(i)}(\mathbf{x}_q)\right\|_2 + \left\|\frac{1}{N}\sum_{l=0}^{N-1} \nabla f_{\pi_{q-1}(l)}\left(\mathbf{x}_{q-1}^l\right) - \frac{1}{N}\sum_{l=0}^{N-1} \nabla f_{\pi_q(l)}(\mathbf{x}_q)\right\|_2 + G_q$$
$$\leq \left\|\nabla f_{\pi_q(i)}(\mathbf{x}_q^i) - \nabla f_{\pi_q(i)}(\mathbf{x}_q)\right\|_2 + \left\|\frac{1}{N}\sum_{l=0}^{N-1} \nabla f_l\left(\mathbf{x}_{q-1}^{\pi_{q-1}^{-1}(l)}\right) - \frac{1}{N}\sum_{l=0}^{N-1} \nabla f_l(\mathbf{x}_q)\right\|_2 + G_q$$
$$\leq L_{2,\infty}\left\|\mathbf{x}_q^i - \mathbf{x}_q\right\|_\infty + \frac{1}{N}\sum_{l=0}^{N-1} L_{2,\infty}\left\|\mathbf{x}_{q-1}^{\pi_{q-1}^{-1}(l)} - \mathbf{x}_q\right\|_\infty + G_q$$
$$\leq L_{2,\infty}\left\|\mathbf{x}_q^i - \mathbf{x}_q\right\|_\infty + \frac{1}{N}\sum_{l=0}^{N-1} L_{2,\infty}\left(\left\|\mathbf{x}_{q-1}^{\pi_{q-1}^{-1}(l)} - \mathbf{x}_{q-1}\right\|_\infty + \left\|\mathbf{x}_{q-1} - \mathbf{x}_q\right\|_\infty\right) + G_q$$
$$\leq L_{2,\infty}\Delta_q + 2L_{2,\infty}\Delta_{q-1} + G_q.$$

The preceding inequality holds for any $i \in \{0, 1, \ldots, N-1\}$.

$$\left\|\sum_{i=0}^{N-1} \mathbf{z}_{\pi(i)}\right\|_\infty = \left\|\sum_{i=0}^{N-1}\left(\nabla f_{\pi_q(i)}(\mathbf{x}_q^i) - \frac{1}{N}\sum_{l=0}^{N-1} \nabla f_{\pi_{q-1}(l)}(\mathbf{x}_{q-1}^l)\right)\right\|_\infty$$
$$= \left\|\sum_{i=0}^{N-1} \nabla f_{\pi_q(i)}(\mathbf{x}_q^i) - \sum_{i=0}^{N-1} \nabla f_{\pi_{q-1}(i)}(\mathbf{x}_{q-1}^i)\right\|_\infty$$
$$= \left\|\sum_{i=0}^{N-1} \nabla f_i\left(\mathbf{x}_q^{\pi_q^{-1}(i)}\right) - \sum_{i=0}^{N-1} \nabla f_i\left(\mathbf{x}_{q-1}^{\pi_{q-1}^{-1}(i)}\right)\right\|_\infty$$
$$\leq \sum_{i=0}^{N-1} \left\|\nabla f_i\left(\mathbf{x}_q^{\pi_q^{-1}(i)}\right) - \nabla f_i\left(\mathbf{x}_{q-1}^{\pi_{q-1}^{-1}(i)}\right)\right\|_\infty$$
$$\leq L_\infty \sum_{i=0}^{N-1} \left\|\mathbf{x}_q^{\pi_q^{-1}(i)} - \mathbf{x}_{q-1}^{\pi_{q-1}^{-1}(i)}\right\|_\infty$$
$$\leq L_\infty \sum_{i=0}^{N-1} \left(\left\|\mathbf{x}_q^{\pi_q^{-1}(i)} - \mathbf{x}_q\right\|_\infty + \left\|\mathbf{x}_q - \mathbf{x}_{q-1}\right\|_\infty + \left\|\mathbf{x}_{q-1} - \mathbf{x}_{q-1}^{\pi_{q-1}^{-1}(i)}\right\|_\infty\right)$$
$$\leq L_\infty N\Delta_q + 2L_\infty N\Delta_{q-1}.$$

For any $n \in [N]$, we have

$$\left\| \sum_{i=0}^{n-1} \mathbf{z}_{\pi(i)} \right\|_\infty$$

$$= \left\| \sum_{i=0}^{n-1} \left( \nabla f_{\pi_q(i)} \left(\mathbf{x}_q^i\right) - \frac{1}{N} \sum_{l=0}^{N-1} \nabla f_{\pi_{q-1}(l)}(\mathbf{x}_{q-1}^l) \right) \right\|_\infty$$

$$= \left\| \sum_{i=0}^{n-1} \left( \left( \nabla f_{\pi_q(i)} \left(\mathbf{x}_q^i\right) - \frac{1}{N} \sum_{l=0}^{N-1} \nabla f_{\pi_{q-1}(l)}(\mathbf{x}_{q-1}^l) \right) \pm \left( \nabla f_{\pi_q(i)} \left(\mathbf{x}_q\right) - \frac{1}{N} \sum_{l=0}^{N-1} \nabla f_{\pi_q(l)}(\mathbf{x}_q) \right) \right) \right\|_\infty$$

$$\leq \left\| \sum_{i=0}^{n-1} \left( \nabla f_{\pi_q(i)}(\mathbf{x}_q^i) - \nabla f_{\pi_q(i)}(\mathbf{x}_q) \right) \right\|_\infty + \left\| \sum_{i=0}^{n-1} \left( \frac{1}{N} \sum_{l=0}^{N-1} \nabla f_{\pi_{q-1}(l)}(\mathbf{x}_{q-1}^l) - \frac{1}{N} \sum_{l=0}^{N-1} \nabla f_{\pi_q(l)}(\mathbf{x}_q) \right) \right\|_\infty + \bar{\phi}_q$$

$$\leq \left\| \sum_{i=0}^{n-1} \left( \nabla f_{\pi_q(i)}(\mathbf{x}_q^i) - \nabla f_{\pi_q(i)}(\mathbf{x}_q) \right) \right\|_\infty + \left\| \sum_{i=0}^{n-1} \left( \frac{1}{N} \sum_{l=0}^{N-1} \nabla f_l \left( \mathbf{x}_{q-1}^{\pi_{q-1}^{-1}(l)} \right) - \frac{1}{N} \sum_{l=0}^{N-1} \nabla f_l(\mathbf{x}_q) \right) \right\|_\infty + \bar{\phi}_q$$

$$\leq \sum_{i=0}^{n-1} \left\| \nabla f_{\pi_q(i)}(\mathbf{x}_q^i) - \nabla f_{\pi_q(i)}(\mathbf{x}_q) \right\|_\infty + \sum_{i=0}^{n-1} \frac{1}{N} \sum_{l=0}^{N-1} \left\| \nabla f_l \left( \mathbf{x}_{q-1}^{\pi_{q-1}^{-1}(l)} \right) - \nabla f_l(\mathbf{x}_q) \right\|_\infty + \bar{\phi}_q$$

$$\leq L_\infty \sum_{i=0}^{n-1} \left\| \mathbf{x}_q^i - \mathbf{x}_q \right\|_\infty + L_\infty \sum_{i=0}^{n-1} \frac{1}{N} \sum_{l=0}^{N-1} \left\| \mathbf{x}_{q-1}^{\pi_{q-1}^{-1}(l)} - \mathbf{x}_q \right\|_\infty + \bar{\phi}_q$$

$$\leq L_\infty \sum_{i=0}^{n-1} \left\| \mathbf{x}_q^i - \mathbf{x}_q \right\|_\infty + L_\infty \sum_{i=0}^{n-1} \frac{1}{N} \sum_{l=0}^{N-1} \left( \left\| \mathbf{x}_{q-1}^{\pi_{q-1}^{-1}(l)} - \mathbf{x}_{q-1} \right\|_\infty + \left\| \mathbf{x}_{q-1} - \mathbf{x}_q \right\|_\infty \right) + \bar{\phi}_q$$

$$\leq L_\infty N \Delta_q + 2 L_\infty N \Delta_{q-1} + \bar{\phi}_q.$$

Since it holds for all $n \in [N]$, we have

$$\max_{n \in [N]} \left\| \sum_{i=0}^{n-1} \mathbf{z}_{\pi(i)} \right\|_\infty \leq L_\infty N \Delta_q + 2 L_\infty N \Delta_{q-1} + \bar{\phi}_q.$$

Now, applying Lemma 3 to the last term on the right hand side in Equation (18), we obtain

$$\bar{\phi}_{q+1} \leq (3 L_\infty N \Delta_q + 2 L_\infty N \Delta_{q-1}) + \frac{1}{2} \left( L_\infty N \Delta_q + 2 L_\infty N \Delta_{q-1} + \bar{\phi}_q \right)$$

$$+ (L_\infty N \Delta_q + 2 L_\infty N \Delta_{q-1}) + \frac{1}{2} C \left( L_{2,\infty} \Delta_q + 2 L_{2,\infty} \Delta_{q-1} + G_q \right)$$

$$\leq \left( \frac{9}{2} L_\infty N + \frac{1}{2} C L_{2,\infty} \right) \Delta_q + (5 L_\infty N + C L_{2,\infty}) \Delta_{q-1} + \frac{1}{2} \bar{\phi}_q + \frac{1}{2} C G_q$$

$$\leq \left( \frac{9}{2} L_\infty N + \frac{1}{2} C L_{2,\infty} \right) \left( \frac{32}{31} \gamma \bar{\phi}_q + \frac{32}{31} \gamma N \left\| \nabla f(\mathbf{x}_q) \right\| \right)$$

$$+ (5 L_\infty N + C L_{2,\infty}) \left( \frac{32}{31} \gamma \bar{\phi}_{q-1} + \frac{32}{31} \gamma N \left\| \nabla f(\mathbf{x}_{q-1}) \right\| \right) + \frac{1}{2} \bar{\phi}_q + \frac{1}{2} C G_q,$$

where the last inequality uses Lemma 6. If $\gamma L_\infty N \leq \frac{1}{128}$ and $\gamma L_{2,\infty} C \leq \frac{1}{128}$, then $\left( \frac{9}{2} \gamma L_\infty N + \frac{1}{2} \gamma C L_{2,\infty} \right) \cdot \frac{32}{31} \leq \frac{5}{124}$ and $(5 \gamma L_\infty N + C \gamma L_{2,\infty}) \cdot \frac{32}{31} \leq \frac{6}{124}$; we obtain

$$\bar{\phi}_{q+1} \leq \frac{67}{124} \bar{\phi}_q + \frac{6}{124} \bar{\phi}_{q-1} + \frac{5}{124} N \left\| \nabla f(\mathbf{x}_q) \right\| + \frac{6}{124} N \left\| \nabla f(\mathbf{x}_{q-1}) \right\| + \frac{1}{2} C G_q.$$

Then, we obtain

$$\left( \bar{\phi}_{q+1} \right)^2 \leq \frac{3}{5} \left( \bar{\phi}_q \right)^2 + \frac{1}{50} \left( \bar{\phi}_{q-1} \right)^2$$

$$+ \left( \frac{1}{50} N^2 + 2 C^2 \alpha^2 \right) \left\| \nabla f(\mathbf{x}_q) \right\|^2 + \frac{1}{50} N^2 \left\| \nabla f(\mathbf{x}_{q-1}) \right\|^2 + 2 C^2 \varsigma^2.$$

So the relation between $\bar{\phi}_q$ and $\bar{\phi}_{q-1}$ is

$$\left(\bar{\phi}_q\right)^2 \le \frac{3}{5}\left(\bar{\phi}_{q-1}\right)^2 + \frac{1}{50}\left(\bar{\phi}_{q-2}\right)^2$$
$$+ \left(\frac{1}{50}N^2 + 2C^2\alpha^2\right)\|\nabla f(\mathbf{x}_{q-1})\|^2 + \frac{1}{50}N^2\|\nabla f(\mathbf{x}_{q-2})\|^2 + 2C^2\varsigma^2,$$

for $q \ge 2$. We have $\left(\bar{\phi}_0\right)^2 \le N^2\alpha^2\|\nabla f(\mathbf{x}_0)\|^2 + N^2\varsigma^2$ and $\left(\bar{\phi}_1\right)^2 \le N^2\alpha^2\|\nabla f(\mathbf{x}_1)\|^2 + N^2\varsigma^2$.

In this example, for Theorem 1, $p = \infty$, $\nu = 2$, $A_1 = \frac{3}{5}$, $A_2 = \frac{1}{50}$, $B_0 = 0$, $B_1 = \left(\frac{1}{50}N^2 + 2C^2\alpha^2\right)$, $B_2 = \frac{1}{50}N^2$, $D = 2C^2\varsigma^2$, $\tilde{B} = N^2\alpha^2$, $\tilde{D} = N^2\varsigma^2$ and $c = 25$. These lead to

$$\frac{1}{Q}\sum_{q=0}^{Q-1}\|\nabla f(\mathbf{x}_q)\|^2 = \mathcal{O}\left(\frac{F_0}{\gamma N Q} + \gamma^2 L_{2,\infty}^2 N^2\frac{1}{Q}\varsigma^2 + \gamma^2 L_{2,\infty}^2 C^2\varsigma^2\right).$$

where $F_0 = f(\mathbf{x}_0) - f_*$. Lemma 3 is used for each epoch (that is, for $Q$ times), so, by the union bound, the preceding bound holds with probability at least $1 - Q\delta$.

Next, we summarize the constraints on the step size:

$$\gamma \le \min\left\{\frac{1}{LN}, \frac{1}{32L_{2,p}N}, \frac{\sqrt{1 - \sum_{i=1}^{\nu} A_i}}{4L_{2,p}\sqrt{\sum_{i=0}^{\nu} B_i}}, \frac{\sqrt{1 - \sum_{i=1}^{\nu} A_i}}{4L_{2,p}\sqrt{\tilde{B}}}, \frac{1}{32L_p N}\right\},$$

$$\gamma \le \frac{1}{128L_\infty N},$$

$$\gamma \le \frac{1}{128L_{2,\infty}C},$$

where the first one is from Theorem 1 and the others is from the derivation of the relation. For simplicity, we can use a tighter constraint

$$\gamma \le \min\left\{\frac{1}{LN}, \frac{1}{128L_{2,\infty}(N + C + N\alpha)}, \frac{1}{128L_\infty N}\right\}.$$

After we use the effective step size $\tilde{\gamma} := \gamma N$, the constraint will be

$$\tilde{\gamma} \le \min\left\{\frac{1}{L}, \frac{1}{128L_{2,\infty}\left(1 + \frac{C}{N} + \alpha\right)}, \frac{1}{128L_\infty}\right\},$$

and the upper bound will be

$$\frac{1}{Q}\sum_{q=0}^{Q-1}\|\nabla f(\mathbf{x}_q)\|^2 = \mathcal{O}\left(\frac{F_0}{\tilde{\gamma}Q} + \tilde{\gamma}^2 L_{2,\infty}^2\frac{1}{Q}\varsigma^2 + \tilde{\gamma}^2 L_{2,\infty}^2\frac{1}{N^2}C^2\varsigma^2\right).$$

Applying Lemma 1, we obtain

$$\frac{1}{Q}\sum_{q=0}^{Q-1}\|\nabla f(\mathbf{x}_q)\|^2$$
$$= \mathcal{O}\left(\frac{\left(L + L_{2,\infty}\left(1 + \frac{C}{N} + \alpha\right) + L_\infty\right)F_0}{Q} + \frac{(L_{2,\infty}F_0\varsigma)^{\frac{2}{3}}}{Q} + \left(\frac{L_{2,\infty}F_0 C\varsigma}{NQ}\right)^{\frac{2}{3}}\right).$$

$\square$

### F.8 PairGraB

**Proposition 7** (PairGraB). *Suppose that Assumption 3 holds and that $N \mod 2 = 0$. If each $f_n$ is $L_{2,\infty}$-smooth and $L_\infty$-smooth and $\gamma \le \min\{\frac{1}{64L_{2,\infty}C}, \frac{1}{64L_\infty N}\}$, we obtain that, for $q = 0$, $\left(\bar{\phi}_0\right)^2 \le N^2\alpha^2\|\nabla f(\mathbf{x}_0)\|^2 + N^2\varsigma^2$, and for $q \ge 1$, with probability at least $1 - \delta$,*

$$\left(\bar{\phi}_q\right)^2 \le \frac{4}{5}\left(\bar{\phi}_{q-1}\right)^2 + \left(\frac{3}{50}N^2 + 4C^2\alpha^2\right)\|\nabla f(\mathbf{x}_{q-1})\|^2 + 4C^2\varsigma^2,$$

where $C = \mathcal{O}\left(\log\left(\frac{dN}{\delta}\right)\right) = \tilde{\mathcal{O}}(1)$. *Applying Theorem 1 and tuning the step size, we obtain that, with probability at least $1 - Q\delta$,*

$$\frac{1}{Q}\sum_{q=0}^{Q-1}\|\nabla f(\mathbf{x}_q)\|^2 = \mathcal{O}\left(\frac{\tilde{L}F_0 + (L_{2,\infty}F_0\varsigma)^{\frac{2}{3}}}{Q} + \left(\frac{L_{2,\infty}F_0 C\varsigma}{NQ}\right)^{\frac{2}{3}}\right),$$

*where $\tilde{L} = L + L_{2,\infty}\left(1 + \frac{C}{N} + \alpha\right) + L_\infty$.*

PairGraB. Use `PairBR` (Algorithm 6) as the `Permute` function in Algorithm 1, with the inputs of $\pi_q$, $\{\nabla f_{\pi_q(n)}(\mathbf{x}_q^n)\}_{n=0}^{N-1}$ and $\frac{1}{N}\sum_{n=0}^{N-1}\nabla f_{\pi_q(n)}(\mathbf{x}_q^n)$, for each epoch $q$.

Thus, the key idea of our proof is as follows:

$$\bar{\phi}_{q+1} \to \max_{n\in[N]}\left\|\sum_{i=0}^{n-1}\left(\nabla f_{\pi_{q+1}(i)}\left(\mathbf{x}_q^{\pi_q^{-1}(\pi_{q+1}(i))}\right) - \frac{1}{N}\sum_{l=0}^{N-1}\nabla f_{\pi_q(l)}(\mathbf{x}_q^l)\right)\right\|_\infty$$

$$\overset{\text{Lemma 4}}{\to}\left\|\sum_{i=0}^{n-1}\left(\nabla f_{\pi_q(i)}\left(\mathbf{x}_q^i\right) - \frac{1}{N}\sum_{l=0}^{N-1}\nabla f_{\pi_q(l)}(\mathbf{x}_q^l)\right)\right\|_\infty \to \bar{\phi}_q.$$

*Proof.* We need to find the relation between $\bar{\phi}_{q+1}$ and $\bar{\phi}_q$.

$$\phi_{q+1}^n = \left\|\sum_{i=0}^{n-1}\left(\nabla f_{\pi_{q+1}(i)}(\mathbf{x}_{q+1}) - \nabla f(\mathbf{x}_{q+1})\right)\right\|_\infty$$

$$\leq \left\|\sum_{i=0}^{n-1}\left(\nabla f_{\pi_{q+1}(i)}(\mathbf{x}_{q+1}) - \nabla f_{\pi_{q+1}(i)}\left(\mathbf{x}_q^{\pi_q^{-1}(\pi_{q+1}(i))}\right)\right)\right\|_\infty$$

$$+ \left\|\sum_{i=0}^{n-1}\left(\frac{1}{N}\sum_{l=0}^{N-1}\nabla f_{\pi_q(l)}(\mathbf{x}_{q+1}) - \frac{1}{N}\sum_{l=0}^{N-1}\nabla f_{\pi_q(l)}(\mathbf{x}_q^l)\right)\right\|_\infty$$

$$+ \left\|\sum_{i=0}^{n-1}\left(\nabla f_{\pi_{q+1}(i)}\left(\mathbf{x}_q^{\pi_q^{-1}(\pi_{q+1}(i))}\right) - \frac{1}{N}\sum_{l=0}^{N-1}\nabla f_{\pi_q(l)}(\mathbf{x}_q^l)\right)\right\|_\infty. \tag{19}$$

Then,

$$\text{T}_1 \text{ in (19)} = \left\|\sum_{i=0}^{n-1}\left(\nabla f_{\pi_{q+1}(i)}(\mathbf{x}_{q+1}) - \nabla f_{\pi_{q+1}(i)}\left(\mathbf{x}_q^{\pi_q^{-1}(\pi_{q+1}(i))}\right)\right)\right\|_\infty$$

$$\leq \sum_{i=0}^{n-1}\left\|\nabla f_{\pi_{q+1}(i)}(\mathbf{x}_{q+1}) - \nabla f_{\pi_{q+1}(i)}\left(\mathbf{x}_q^{\pi_q^{-1}(\pi_{q+1}(i))}\right)\right\|_\infty$$

$$\leq L_\infty\sum_{i=0}^{n-1}\left\|\mathbf{x}_{q+1} - \mathbf{x}_q^{\pi_q^{-1}(\pi_{q+1}(i))}\right\|_\infty$$

$$\leq L_\infty\sum_{i=0}^{n-1}\left(\|\mathbf{x}_{q+1} - \mathbf{x}_q\|_\infty + \left\|\mathbf{x}_q - \mathbf{x}_q^{\pi_q^{-1}(\pi_{q+1}(i))}\right\|_\infty\right)$$

$$\leq 2L_\infty N\Delta_q,$$

$$\text{T}_2 \text{ in (19)} = \left\| \sum_{i=0}^{n-1} \left( \frac{1}{N} \sum_{l=0}^{N-1} \nabla f_{\pi_q(l)}(\mathbf{x}_{q+1}) - \frac{1}{N} \sum_{l=0}^{N-1} \nabla f_{\pi_q(l)}(\mathbf{x}_q^l) \right) \right\|_\infty$$

$$\leq \sum_{i=0}^{n-1} \frac{1}{N} \sum_{l=0}^{N-1} \left\| \nabla f_{\pi_q(l)}(\mathbf{x}_{q+1}) - \nabla f_{\pi_q(l)}(\mathbf{x}_q^l) \right\|_\infty$$

$$\leq L_\infty \sum_{i=0}^{n-1} \frac{1}{N} \sum_{l=0}^{N-1} \left\| \mathbf{x}_{q+1} - \mathbf{x}_q^l \right\|_\infty$$

$$\leq L_\infty \sum_{i=0}^{n-1} \frac{1}{N} \sum_{l=0}^{N-1} \left( \left\| \mathbf{x}_{q+1} - \mathbf{x}_q \right\|_\infty + \left\| \mathbf{x}_q - \mathbf{x}_q^l \right\|_\infty \right)$$

$$\leq 2 L_\infty N \Delta_q \,.$$

Since the preceding inequalities hold for all $n \in [N]$, we have

$$\bar{\phi}_{q+1} \leq 4 L_\infty N \Delta_q + \max_{n \in [N]} \left\| \sum_{i=0}^{n-1} \left( \nabla f_{\pi_{q+1}(i)} \left( \mathbf{x}_q^{\pi_q^{-1}(\pi_{q+1}(i))} \right) - \frac{1}{N} \sum_{l=0}^{N-1} \nabla f_{\pi_q(l)}(\mathbf{x}_q^l) \right) \right\|_\infty . \tag{20}$$

Note that $\nabla f_{\pi_q(i)}\left(\mathbf{x}_q^i\right) - \frac{1}{N} \sum_{l=0}^{N-1} \nabla f_{\pi_q(l)}(\mathbf{x}_q^l)$ and $\nabla f_{\pi_{q+1}(i)}\left(\mathbf{x}_q^{\pi_q^{-1}(\pi_{q+1}(i))}\right) - \frac{1}{N} \sum_{l=0}^{N-1} \nabla f_{\pi_q(l)}(\mathbf{x}_q^l)$ correspond to $\mathbf{z}_{\pi(i)}$ and $\mathbf{z}_{\pi'(i)}$ in Lemma 4, respectively. We next derive the upper bounds of

$$\left\| \mathbf{z}_{\pi(i)} \right\|_2 , \left\| \sum_{i=0}^{N-1} \mathbf{z}_{\pi(i)} \right\|_\infty \text{ and } \max_{n \in [N]} \left\| \sum_{i=0}^{n-1} \mathbf{z}_{\pi(i)} \right\|_\infty ,$$

and then apply Lemma 4 to the last term on the right hand side in Equation (20).

$$\left\| \mathbf{z}_{\pi_q(i)} \right\|_2 = \left\| \nabla f_{\pi_q(i)}(\mathbf{x}_q^i) - \frac{1}{N} \sum_{l=0}^{N-1} \nabla f_{\pi_q(l)}(\mathbf{x}_q^l) \right\|_2$$

$$= \left\| \left( \nabla f_{\pi_q(i)}(\mathbf{x}_q^i) - \frac{1}{N} \sum_{l=0}^{N-1} \nabla f_{\pi_q(l)}(\mathbf{x}_q^l) \right) \pm \left( \nabla f_{\pi_q(i)}(\mathbf{x}_q) - \frac{1}{N} \sum_{l=0}^{N-1} \nabla f_{\pi_q(l)}(\mathbf{x}_q) \right) \right\|_2$$

$$\leq \left\| \nabla f_{\pi_q(i)}(\mathbf{x}_q^i) - \nabla f_{\pi_q(i)}(\mathbf{x}_q) \right\|_2 + \left\| \frac{1}{N} \sum_{l=0}^{N-1} \nabla f_{\pi_q(l)}(\mathbf{x}_q^l) - \frac{1}{N} \sum_{l=0}^{N-1} \nabla f_{\pi_q(l)}(\mathbf{x}_q) \right\|_2$$

$$+ \left\| \nabla f_{\pi_q(i)}(\mathbf{x}_q) - \frac{1}{N} \sum_{l=0}^{N-1} \nabla f_{\pi_q(l)}(\mathbf{x}_q) \right\|_2$$

$$\leq L_{2,\infty} \left\| \mathbf{x}_q^i - \mathbf{x}_q \right\|_\infty + \frac{1}{N} \sum_{l=0}^{N-1} L_{2,\infty} \left\| \mathbf{x}_q^l - \mathbf{x}_q \right\|_\infty + G_q$$

$$\leq 2 L_{2,\infty} \Delta_q + G_q,$$

$$\left\| \sum_{i=0}^{N-1} \mathbf{z}_{\pi_q(i)} \right\|_\infty = \left\| \sum_{i=0}^{N-1} \left( \nabla f_{\pi_q(i)}(\mathbf{x}_q^i) - \frac{1}{N} \sum_{l=0}^{N-1} \nabla f_{\pi_q(l)}(\mathbf{x}_q^l) \right) \right\|_\infty = 0 \,.$$

For any $n \in [N]$, we obtain

$$\left\| \sum_{i=0}^{n-1} \mathbf{z}_{\pi(i)} \right\|_\infty$$

$$= \left\| \sum_{i=0}^{n-1} \left( \nabla f_{\pi_q(i)}\left(\mathbf{x}_q^i\right) - \frac{1}{N} \sum_{l=0}^{N-1} \nabla f_{\pi_q(l)}(\mathbf{x}_q^l) \right) \right\|_\infty$$

$$= \left\| \sum_{i=0}^{n-1} \left( \left( \nabla f_{\pi_q(i)}\left(\mathbf{x}_q^i\right) - \frac{1}{N} \sum_{l=0}^{N-1} \nabla f_{\pi_q(l)}(\mathbf{x}_q^l) \right) \pm \left( \nabla f_{\pi_q(i)}\left(\mathbf{x}_q\right) - \frac{1}{N} \sum_{l=0}^{N-1} \nabla f_{\pi_q(l)}(\mathbf{x}_q) \right) \right) \right\|_\infty$$

$$\leq \left\| \sum_{i=0}^{n-1} \left( \nabla f_{\pi_q(i)}(\mathbf{x}_q^i) - \nabla f_{\pi_q(i)}(\mathbf{x}_q) \right) \right\|_\infty + \left\| \sum_{i=0}^{n-1} \left( \frac{1}{N} \sum_{l=0}^{N-1} \nabla f_{\pi_q(l)}(\mathbf{x}_q^l) - \frac{1}{N} \sum_{l=0}^{N-1} \nabla f_{\pi_q(l)}(\mathbf{x}_q) \right) \right\|_\infty + \bar{\phi}_q$$

$$\leq \sum_{i=0}^{n-1} \left\| \nabla f_{\pi_q(i)}(\mathbf{x}_q^i) - \nabla f_{\pi_q(i)}(\mathbf{x}_q) \right\|_\infty + \sum_{i=0}^{n-1} \frac{1}{N} \sum_{l=0}^{N-1} \left\| \nabla f_{\pi_q(l)}(\mathbf{x}_q^l) - \nabla f_{\pi_q(l)}(\mathbf{x}_q) \right\|_\infty + \bar{\phi}_q$$

$$\leq L_\infty \sum_{i=0}^{n-1} \left\| \mathbf{x}_q^i - \mathbf{x}_q \right\|_\infty + L_\infty \sum_{i=0}^{n-1} \frac{1}{N} \sum_{l=0}^{N-1} \left\| \mathbf{x}_q^l - \mathbf{x}_q \right\|_\infty + \bar{\phi}_q$$

$$\leq 2 L_\infty N \Delta_q + \bar{\phi}_q \,.$$

Since it holds for all $n \in [N]$, we obtain

$$\max_{n \in [N]} \left\| \sum_{i=0}^{n-1} \mathbf{z}_{\pi(i)} \right\|_\infty \leq 2 L_\infty N \Delta_q + \bar{\phi}_q \,.$$

Now, applying Lemma 4 to the last term on the right hand side in Equation (20), we obtain

$$\bar{\phi}_{q+1} \leq 4 L_\infty N \Delta_q + \frac{1}{2}\left(2L_\infty N \Delta_q + \bar{\phi}_q\right) + C\left(2L_{2,\infty}\Delta_q + G_q\right)$$

$$\leq \left(5L_\infty N + 2L_{2,\infty}C\right)\Delta_q + \frac{1}{2}\bar{\phi}_q + CG_q$$

$$\leq \left(5L_\infty N + 2L_{2,\infty}C\right)\left(\frac{32}{31}\gamma\bar{\phi}_q + \frac{32}{31}\gamma N \left\|\nabla f(\mathbf{x}_q)\right\|\right) + \frac{1}{2}\bar{\phi}_q + CG_q,$$

where the last inequality uses Lemma 6. If $\gamma L_\infty N \leq \frac{1}{64}$ and $\gamma L_{2,\infty}C \leq \frac{1}{64}$, we obtain

$$\bar{\phi}_{q+1} \leq \frac{38}{62}\bar{\phi}_q + \frac{7}{62}N\left\|\nabla f(\mathbf{x}_q)\right\|_\infty + CG_q.$$

Then, we obtain

$$\left(\bar{\phi}_{q+1}\right)^2 \leq \frac{4}{5}\left(\bar{\phi}_q\right)^2 + \left(\frac{3}{50}N^2 + 4C^2\alpha^2\right)\left\|\nabla f(\mathbf{x}_q)\right\|^2 + 4C^2\varsigma^2.$$

So the relation between $\bar{\phi}_q$ and $\bar{\phi}_{q-1}$ is

$$\left(\bar{\phi}_q\right)^2 \leq \frac{4}{5}\left(\bar{\phi}_{q-1}\right)^2 + \left(\frac{3}{50}N^2 + 4C^2\alpha^2\right)\left\|\nabla f(\mathbf{x}_{q-1})\right\|^2 + 4C^2\varsigma^2.$$

for $q \geq 1$. In addition, we have $\left(\bar{\phi}_0\right)^2 \leq N^2\alpha^2\left\|\nabla f(\mathbf{x}_0)\right\|^2 + N^2\varsigma^2$.

In this example, for Theorem 1, $p = \infty$, $\nu = 1$, $A_1 = \frac{4}{5}$, $B_0 = 0$, $B_1 = \frac{3}{50}N^2 + 4C^2\alpha^2$, $D = 4C^2\varsigma^2$, $\tilde{B} = N^2\alpha^2$, $\tilde{D} = N^2\varsigma^2$ and $c = 50$. These lead to

$$\frac{1}{Q}\sum_{q=0}^{Q-1}\left\|\nabla f(\mathbf{x}_q)\right\|^2 = \mathcal{O}\left(\frac{F_0}{\gamma N Q} + \gamma^2 L_{2,\infty}^2 N^2 \frac{1}{Q}\varsigma^2 + \gamma^2 L_{2,\infty}^2 C^2 \varsigma^2\right).$$

where $F_0 = f(\mathbf{x}_0) - f_*$. Lemma 4 is used for each epoch (that is, for $Q$ times), so, by the union bound, the preceding bound holds with probability at least $1 - Q\delta$.

Next, we summarize the constraints on the step size:

$$\gamma \le \min\left\{\frac{1}{LN}, \frac{1}{32L_{2,p}N}, \frac{\sqrt{1-\sum_{i=1}^{\nu}A_i}}{4L_{2,p}\sqrt{\sum_{i=0}^{\nu}B_i}}, \frac{\sqrt{1-\sum_{i=1}^{\nu}A_i}}{4L_{2,p}\sqrt{\tilde{B}}}, \frac{1}{32L_pN}\right\},$$

$$\gamma \le \frac{1}{64L_{\infty}N},$$

$$\gamma \le \frac{1}{64L_{2,\infty}C},$$

where the first one is from Theorem 1 and the others are from the derivation of the relation. For simplicity, we can use a tighter constraint

$$\gamma \le \min\left\{\frac{1}{LN}, \frac{1}{64L_{2,\infty}(N+C+N\alpha)}, \frac{1}{64L_{\infty}N}\right\}.$$

After we use the effective step size $\tilde{\gamma} := \gamma N$, the constraint will be

$$\tilde{\gamma} \le \min\left\{\frac{1}{L}, \frac{1}{64L_{2,\infty}\left(1+\frac{C}{N}+\alpha\right)}, \frac{1}{64L_{\infty}}\right\},$$

and the upper bound will be

$$\frac{1}{Q}\sum_{q=0}^{Q-1}\|\nabla f(\mathbf{x}_q)\|^2 = \mathcal{O}\left(\frac{F_0}{\tilde{\gamma}Q} + \tilde{\gamma}^2 L_{2,\infty}^2 \frac{1}{Q}\varsigma^2 + \tilde{\gamma}^2 L_{2,\infty}^2 \frac{1}{N^2}C^2\varsigma^2\right).$$

Applying Lemma 1, we obtain

$$\frac{1}{Q}\sum_{q=0}^{Q-1}\|\nabla f(\mathbf{x}_q)\|^2$$

$$= \mathcal{O}\left(\frac{\left(L+L_{2,\infty}\left(1+\frac{C}{N}+\alpha\right)+L_{\infty}\right)F_0}{Q} + \frac{(L_{2,\infty}F_0\varsigma)^{\frac{2}{3}}}{Q} + \left(\frac{L_{2,\infty}F_0C\varsigma}{NQ}\right)^{\frac{2}{3}}\right).$$

$\square$

### F.9  Refinement of the High Probability Bounds from $Q\delta$ to $\delta$

To maintain consistency with Lu et al. [2022a] and Cooper et al. [2023], we use a failure probability of $Q\delta$ rather than $\delta$ in the main body. This can lead to looser bounds as $Q$ increases. This section shows that the framework can also provide bounds that hold with probability at least $1-\delta$.

Considering that the bound for AP is deterministic (not a probabilistic one), we next only discuss IP and DP:

**IP.** Taking RR as an example. Starting from Equation (15), we obtain that, with probability at least $1-\delta'$,

$$(\bar{\phi}_q)^2 \le 4\log^2\left(\frac{8}{\delta'}\right)U,$$

where $U = N\alpha^2\|\nabla f(x_q)\|^2 + N\varsigma^2$ for brevity. That is,

$$\mathbb{P}\left((\bar{\phi}_q)^2 \ge 4\log^2\left(\frac{8}{\delta'}\right)U\right) \le \delta'$$

for any $q \in \{0, 1, \ldots, Q-1\}$. Then, applying the union bound for $q \in \{0, 1, \ldots, Q-1\}$, we obtain

$$\mathbb{P}\left(\exists q, (\bar{\phi}_q)^2 \ge 4\log^2\left(\frac{8}{\delta'}\right)U\right) \le Q\delta'.$$

Then, setting $\delta = \frac{\delta'}{Q}$, we obtain

$$\mathbb{P}\left(\exists q, (\bar{\phi}_q)^2 \ge 4\log^2\left(\frac{8Q}{\delta}\right)U\right) \le \delta.$$

That is, with probability at least $1 - \delta$, $(\bar{\phi}_q)^2 \leq 4 \log^2 \left( \frac{8Q}{\delta} \right) U$ for all $q \in \{0, 1, \ldots, Q-1\}$. Then, using these bounds, we obtain the desired bound that holds with probability at least $1 - \delta$.

**DP.** Taking GraB-proto as an example. Starting from Equation (16), we obtain that, with probability at least $1 - \delta'$,

$$(\bar{\phi}_q)^2 \leq \log^2 \left( \frac{dN}{\delta'} \right) U + V,$$

where $U = 900(\alpha^2 \|\nabla f(x_{q-1})\|^2 + \varsigma^2)$ and $V = \frac{3}{4}(\bar{\phi}_{q-1})^2 + \frac{1}{50} N^2 \|\nabla f(x_{q-1})\|^2$. That is,

$$\mathbb{P} \left( (\bar{\phi}_q)^2 \geq \log^2 \left( \frac{dN}{\delta'} \right) U + V \right) \leq \delta'$$

for any $q \in \{1, 2, \ldots, Q-1\}$. Then, applying the union bound for $q \in \{1, 2, \ldots, Q-1\}$, we obtain

$$\mathbb{P} \left( \exists q, (\bar{\phi}_q)^2 \geq \log^2 \left( \frac{dN}{\delta'} \right) U + V \right) \leq (Q-1)\delta'.$$

Then, setting $\delta = \frac{\delta'}{Q-1}$, we obtain

$$\mathbb{P} \left( \exists q, (\bar{\phi}_q)^2 \geq \log^2 \left( \frac{dN(Q-1)}{\delta} \right) U + V \right) \leq (Q-1)\delta.$$

That is, with probability at least $1 - \delta$, $(\bar{\phi}_q)^2 \leq \log^2 \left( \frac{dN(Q-1)}{\delta} \right) U + V$ for all $q \in \{1, 2, \ldots, Q-1\}$. Then, using the bound of $(\bar{\phi}_0)^2$ (it is deterministic) and these bounds, we obtain the desired bound that holds with probability at least $1 - \delta$.

# G    Theorem 2

## G.1    Order Error in FL

*Theoretical understanding of Definition 2.* We can prove that, for small finite step sizes, the cumulative updates in one epoch are

$$\mathbf{x}_{q+1} - \mathbf{x}_q$$

$$= -\gamma \frac{1}{S} \sum_{n=0}^{N-1} \sum_{k=0}^{K-1} \nabla f_{\pi_q(n)} \left( \mathbf{x}_{q,k}^n \right)$$

$$= -\gamma \frac{1}{S} \sum_{n=0}^{N-1} \sum_{k=0}^{K-1} \nabla f_{\pi(n)} \left( \mathbf{x}_q \right)$$

$$+ \gamma^2 \frac{1}{S} \sum_{n=0}^{N-1} \sum_{k=0}^{K-1} \nabla \nabla f_{\pi(n)}(\mathbf{x}_q) \sum_{j=0}^{k-1} \nabla f_{\pi(n)} \left( \mathbf{x}_q \right)$$

$$+ \gamma^2 \frac{1}{S} \sum_{n=0}^{N-1} \sum_{k=0}^{K-1} \nabla \nabla f_{\pi(n)}(\mathbf{x}_q) \frac{1}{S} \sum_{i=0}^{v(n)-1} \sum_{j=0}^{K-1} \nabla f_{\pi(i)} \left( \mathbf{x}_q \right) + \mathcal{O} \left( \gamma^3 K^3 N^3 \frac{1}{S^3} \right). \tag{21}$$

Similar to the analysis in the main body, it can be seen that the error vectors are caused by the second and third terms on the right hand side in Equation (21). Note that when we consider $\nabla \nabla f_{\pi(n)}(\mathbf{x}_0^0) \approx L$, the second term can be also seen as a optimization vector (with the same direction as $\nabla f(\mathbf{x}_{q,0}^0)$). This is mainly because the local solver is the classic SGD in our setup, and it can be different when the local solver is the permutation-based SGD. As a result, we next focus on the third term. With a similar decomposition in the main body, our goal turns to suppress the error vector as follows

$$\text{Error vector} = \gamma^2 \frac{1}{S} \sum_{n=0}^{N-1} \sum_{k=0}^{K-1} \nabla \nabla f_{\pi(n)}(\mathbf{x}_q) \frac{1}{S} \sum_{i=0}^{v(n)-1} \sum_{j=0}^{K-1} \left( \nabla f_{\pi(i)} \left( \mathbf{x}_{q,0}^0 \right) - \nabla f_{\pi(i)} \left( \mathbf{x}_q \right) \right).$$

One straightforward way is to minimize the norm of error vector

$$\|\text{Error vector}\| \leq \gamma^2 L \left\| \frac{1}{S} \sum_{n=0}^{N-1} \sum_{k=0}^{K-1} \frac{1}{S} \sum_{i=0}^{v(n)-1} \sum_{j=0}^{K-1} \left( \nabla f_{\pi(i)}\left(\mathbf{x}_q\right) - \nabla f_{\pi(i)}\left(\mathbf{x}_q\right) \right) \right\|$$

$$\leq \gamma^2 L K^2 \frac{1}{S^2} \sum_{n=0}^{N-1} \left\| \sum_{i=0}^{v(n)-1} \left( \nabla f_{\pi(i)}\left(\mathbf{x}_q\right) - \nabla f_{\pi(i)}\left(\mathbf{x}_q\right) \right) \right\|$$

$$\leq \gamma^2 L K^2 N \frac{1}{S^2} \bar{\varphi}_q \, .$$

## G.2  Proof of Theorem 2

To avoid ambiguity, we define

$$\tilde{\mathbf{x}}_{q+1} := \mathbf{x}_{q,K}^{N-1} = \mathbf{x}_{q,0}^{N}.$$

Due to the amplified updates (see Lines 5 and 6) [Wang and Ji, 2022], we have

$$\tilde{\mathbf{x}}_{q+1} - \mathbf{x}_q = -\gamma \ \frac{1}{S} \sum_{n=0}^{N-1} \sum_{k=0}^{K-1} \nabla f_{\pi_q(n)}\left(\mathbf{x}_{q,k}^n\right),$$

$$\mathbf{x}_{q+1} - \mathbf{x}_q = -\gamma \boldsymbol{\eta} \frac{1}{S} \sum_{n=0}^{N-1} \sum_{k=0}^{K-1} \nabla f_{\pi_q(n)}\left(\mathbf{x}_{q,k}^n\right).$$

We define the maximum parameter deviation (drift) of FL in any epoch $q$, $\Delta_q$ as

$$\Delta_q = \max \left\{ \max_{\substack{n \in \{0, \ldots, N-1\} \\ k \in \{0, \ldots, K-1\}}} \left\| \mathbf{x}_{q,k}^n - \mathbf{x}_q \right\|_p, \left\| \tilde{\mathbf{x}}_{q+1} - \mathbf{x}_q \right\|_p \right\}.$$

Then, we obtain the relation

$$\left\| \mathbf{x}_{q+1} - \mathbf{x}_q \right\|_p = \eta \left\| \tilde{\mathbf{x}}_{q+1} - \mathbf{x}_q \right\|_p \leq \eta \Delta_q.$$

**Lemma 7.** *We first prove that if $\gamma L_p K N \frac{1}{S} \leq \frac{1}{32}$, the maximum parameter drift in FL is bounded:*

$$\Delta_q \leq \frac{32}{31} \gamma K \frac{1}{S} \bar{\varphi}_q + \frac{32}{31} \gamma K N \frac{1}{S} \left\| \nabla f(\mathbf{x}_q) \right\| + \frac{32}{31} \gamma K G_q,$$

$$(\Delta_q)^2 \leq 4\gamma^2 K^2 \frac{1}{S^2} \left( \bar{\varphi}_q \right)^2 + 4\gamma^2 K^2 N^2 \frac{1}{S^2} \left\| \nabla f(\mathbf{x}_q) \right\|^2 + 4\gamma^2 K^2 G_q^2.$$

*Proof.* Let $v(n) = \lfloor \frac{n}{S} \rfloor \cdot S$. Then,

$$\mathbf{x}_{q,k}^n - \mathbf{x}_q = \mathbf{x}_{q,k}^n - \mathbf{x}_{q,0}^n + \underbrace{\mathbf{x}_{q,0}^n - \mathbf{x}_{q,0}^{v(n)}}_{=0} + \mathbf{x}_{q,0}^{v(n)} - \mathbf{x}_q$$

$$= -\gamma \sum_{j=0}^{k-1} \nabla f_{\pi_q(n)}\left(\mathbf{x}_{q,j}^n\right) - \gamma \frac{1}{S} \sum_{i=0}^{v(n)-1} \sum_{j=0}^{K-1} \nabla f_{\pi_q(i)}\left(\mathbf{x}_{q,j}^i\right).$$

For any $q > 0$ and all $n \in \{0, 1, \ldots, N-1\}$ and $k \in \{0, 1, \ldots, K-1\}$, it follows that

$$\left\| \mathbf{x}_{q,k}^n - \mathbf{x}_q \right\|_p = \left\| \gamma \sum_{j=0}^{k-1} \nabla f_{\pi_q(n)}(\mathbf{x}_{q,j}^n) + \gamma \frac{1}{S} \sum_{i=0}^{v(n)-1} \sum_{j=0}^{K-1} \nabla f_{\pi_q(i)}(\mathbf{x}_{q,j}^i) \right\|_p$$

$$\leq \left\| \gamma \sum_{j=0}^{k-1} \nabla f_{\pi_q(n)}(\mathbf{x}_{q,j}^n) \right\|_p + \left\| \gamma \frac{1}{S} \sum_{i=0}^{v(n)-1} \sum_{j=0}^{K-1} \nabla f_{\pi_q(i)}(\mathbf{x}_{q,j}^i) \right\|_p. \quad (22)$$

Then, we bound the two terms on the right hand side in Equation (22) respectively.

$T_1$ in (22)

$$= \left\| \gamma \sum_{j=0}^{k-1} \nabla f_{\pi_q(n)}(\mathbf{x}_{q,j}^n) \right\|_p$$

$$\leq \gamma \left\| \sum_{j=0}^{k-1} \left( \nabla f_{\pi_q(n)}(\mathbf{x}_{q,j}^n) - \nabla f_{\pi_q(n)}(\mathbf{x}_q) \right) \right\|_p + \gamma \left\| \sum_{j=0}^{k-1} \left( \nabla f_{\pi_q(n)}(\mathbf{x}_q) - \nabla f(\mathbf{x}_q) \right) \right\|_\infty$$

$$+ \gamma \left\| \sum_{j=0}^{k-1} \left( \nabla f(\mathbf{x}_q) \right) \right\|_p$$

$$\leq \gamma \sum_{j=0}^{k-1} \left\| \nabla f_{\pi_q(n)}(\mathbf{x}_{q,j}^n) - \nabla f_{\pi_q(n)}(\mathbf{x}_q) \right\|_p + \gamma \sum_{j=0}^{k-1} \left\| \nabla f_{\pi_q(n)}(\mathbf{x}_q) - \nabla f(\mathbf{x}_q) \right\|_p + \gamma \sum_{j=0}^{k-1} \left\| \nabla f(\mathbf{x}_q) \right\|_p$$

$$\leq \gamma L_p \sum_{j=0}^{k-1} \left\| \mathbf{x}_{q,j}^n - \mathbf{x}_q \right\|_p + \gamma \sum_{j=0}^{k-1} G_q + \gamma \sum_{j=0}^{k-1} \left\| \nabla f(\mathbf{x}_q) \right\|_p$$

$$\leq \gamma L_p K \Delta_q + \gamma K G_q + \gamma K \left\| \nabla f(\mathbf{x}_q) \right\|_p ,$$

$T_2$ in (22)

$$= \left\| \gamma \frac{1}{S} \sum_{i=0}^{v(n)-1} \sum_{j=0}^{K-1} \nabla f_{\pi_q(i)}(\mathbf{x}_{q,j}^i) \right\|_p$$

$$= \gamma \frac{1}{S} \left\| \sum_{i=0}^{v(n)-1} \sum_{j=0}^{K-1} \left( \nabla f_{\pi_q(i)}(\mathbf{x}_{q,j}^i) - \nabla f_{\pi_q(i)}(\mathbf{x}_q) + \nabla f_{\pi_q(i)}(\mathbf{x}_q) - \nabla f(\mathbf{x}_q) + \nabla f(\mathbf{x}_q) \right) \right\|_p$$

$$\leq \gamma \frac{1}{S} \left\| \sum_{i=0}^{v(n)-1} \sum_{j=0}^{K-1} \left( \nabla f_{\pi_q(i)}(\mathbf{x}_{q,j}^i) - \nabla f_{\pi_q(i)}(\mathbf{x}_q) \right) \right\|_p + \gamma \frac{1}{S} \left\| \sum_{i=0}^{v(n)-1} \sum_{j=0}^{K-1} \left( \nabla f_{\pi_q(i)}(\mathbf{x}_q) - \nabla f(\mathbf{x}_q) \right) \right\|_p$$

$$+ \gamma \frac{1}{S} \left\| \sum_{i=0}^{v(n)-1} \sum_{j=0}^{K-1} \left( \nabla f(\mathbf{x}_q) \right) \right\|_p$$

$$\leq \gamma \frac{1}{S} \sum_{i=0}^{v(n)-1} \sum_{j=0}^{K-1} \left\| \nabla f_{\pi_q(i)}(\mathbf{x}_{q,j}^i) - \nabla f_{\pi_q(i)}(\mathbf{x}_q) \right\|_p + \gamma K \frac{1}{S} \varphi_q^{v(n)} + \gamma \frac{1}{S} \sum_{i=0}^{v(n)-1} \sum_{j=0}^{K-1} \left\| \nabla f(\mathbf{x}_q) \right\|_p$$

$$\leq \gamma L_p \frac{1}{S} \sum_{i=0}^{v(n)-1} \sum_{j=0}^{K-1} \left\| \mathbf{x}_{q,j}^i - \mathbf{x}_q \right\|_p + \gamma K \frac{1}{S} \varphi_q^{v(n)} + \gamma K \left( v(n) \right) \frac{1}{S} \left\| \nabla f(\mathbf{x}_q) \right\|_p$$

$$\leq \gamma L_p K \left( v(n) \right) \frac{1}{S} \Delta_q + \gamma K \frac{1}{S} \bar{\varphi}_q + \gamma K \left( v(n) \right) \frac{1}{S} \left\| \nabla f(\mathbf{x}_q) \right\|_p .$$

Next, we return to the upper bound of $\| \mathbf{x}_{q,k}^n - \mathbf{x}_q \|_p$ for any $n, k$ such that $nK + k \leq NK$. If $k = 0$, then $v(n) \leq N$ and the first term on the right hand in Equation (22) equals zero, so we obtain

$$\left\| \mathbf{x}_{q,k}^n - \mathbf{x}_q \right\|_p \leq \gamma L_p K N \frac{1}{S} \Delta_q + \gamma K \frac{1}{S} \bar{\varphi}_q + \gamma K N \frac{1}{S} \left\| \nabla f(\mathbf{x}_q) \right\|_p .$$

If $k > 0$, then $v(n) \leq N - S$, so we obtain

$$
\begin{aligned}
\left\|\mathbf{x}_{q,k}^n - \mathbf{x}_q\right\|_p &\leq \gamma L_p K \Delta_q + \gamma K G_q + \gamma K \|\nabla f(\mathbf{x}_q)\|_p \\
&\quad + \gamma L_p K \left(v(n)\right) \frac{1}{S} \Delta_q + \gamma K \frac{1}{S} \bar{\varphi}_q + \gamma K \left(v(n)\right) \frac{1}{S} \|\nabla f(\mathbf{x}_q)\|_p \\
&\leq \gamma L_p K \Delta_q + \gamma K G_q + \gamma K \|\nabla f(\mathbf{x}_q)\|_p \\
&\quad + \gamma L_p K \left(\frac{N}{S} - 1\right) \Delta_q + \gamma K \frac{1}{S} \bar{\varphi}_q + \gamma K \left(\frac{N}{S} - 1\right) \|\nabla f(\mathbf{x}_q)\|_p \\
&\leq \gamma L_p K N \frac{1}{S} \Delta_q + \gamma K \frac{1}{S} \bar{\varphi}_q + \gamma K N \frac{1}{S} \|\nabla f(\mathbf{x}_q)\|_p + \gamma K G_q.
\end{aligned}
$$

Therefore, for any $n, k$ such that $nK + k \leq NK$, we obtain

$$
\begin{aligned}
\Delta_q &= \max_{n,k} \left\|\mathbf{x}_{q,k}^n - \mathbf{x}_q\right\|_p \\
&\leq \gamma L_p K N \frac{1}{S} \Delta_q + \gamma K \frac{1}{S} \bar{\varphi}_q + \gamma K N \frac{1}{S} \|\nabla f(\mathbf{x}_q)\|_p + \gamma K G_q.
\end{aligned}
$$

Then, if $\gamma L_p K N \frac{1}{S} \leq \frac{1}{32}$, we obtain

$$
\Delta_q \leq \frac{32}{31} \gamma K \frac{1}{S} \bar{\varphi}_q + \frac{32}{31} \gamma K N \frac{1}{S} \|\nabla f(\mathbf{x}_q)\|_p + \frac{32}{31} \gamma K G_q.
$$

It also implies that

$$
(\Delta_q)^2 \leq 4\gamma^2 K^2 \frac{1}{S^2} (\bar{\varphi}_q)^2 + 4\gamma^2 K^2 N^2 \frac{1}{S^2} \|\nabla f(\mathbf{x}_q)\|_p^2 + 4\gamma^2 K^2 G_q^2.
$$

At last, using $\|\mathbf{x}\|_q \leq \|\mathbf{x}\|$ for $\mathbf{x} \in \mathbb{R}^d$ and $p \geq 2$, we obtain the claim of this lemma. $\qquad \square$

*Proof of Theorem 2.* For FL with regularized participation (Algorithm 2), the cumulative updates over any epoch $q$ are

$$
\mathbf{x}_{q+1} - \mathbf{x}_q = -\gamma \eta \frac{1}{S} \sum_{n=0}^{N-1} \sum_{k=0}^{K-1} \nabla f_{\pi_q(n)}\left(\mathbf{x}_{q,k}^n\right). \tag{23}
$$

Since the global objective function $f$ is $L$-smooth, it follows that

$$
f(\mathbf{x}_{q+1}) - f(\mathbf{x}_q) \leq \langle \nabla f(\mathbf{x}_q), \mathbf{x}_{q+1} - \mathbf{x}_q \rangle + \frac{1}{2} L \|\mathbf{x}_{q+1} - \mathbf{x}_q\|^2. \tag{24}
$$

Using Equation (23), we obtain

$$
\begin{aligned}
&\langle \nabla f(\mathbf{x}_q), \mathbf{x}_{q+1} - \mathbf{x}_q \rangle \\
&= -\gamma \eta \frac{1}{S} KN \left[ \left\langle \nabla f(\mathbf{x}_q), \frac{1}{N} \sum_{n=0}^{N-1} \frac{1}{K} \sum_{k=0}^{K-1} \nabla f_{\pi_q(n)}(\mathbf{x}_{q,k}^n) \right\rangle \right] \\
&= -\frac{1}{2} \gamma \eta \frac{1}{S} KN \|\nabla f(\mathbf{x}_q)\|^2 - \frac{1}{2} \gamma \eta \frac{1}{S} KN \left\| \frac{1}{N} \sum_{n=0}^{N-1} \frac{1}{K} \sum_{k=0}^{K-1} \nabla f_{\pi_q(n)}(\mathbf{x}_{q,k}^n) \right\|^2 \\
&\quad + \frac{1}{2} \gamma \eta \frac{1}{S} KN \left\| \frac{1}{N} \sum_{n=0}^{N-1} \frac{1}{K} \sum_{k=0}^{K-1} \nabla f_{\pi_q(n)}(\mathbf{x}_{q,k}^n) - \nabla f(\mathbf{x}_q) \right\|^2,
\end{aligned}
$$

where the second equality uses $2\langle \mathbf{x}, \mathbf{y} \rangle = \|\mathbf{x}\|^2 + \|\mathbf{y}\|^2 - \|\mathbf{x} - \mathbf{y}\|^2$. Using Equation (23), we obtain

$$
\begin{aligned}
\frac{1}{2} L \|\mathbf{x}_{q+1} - \mathbf{x}_q\|^2 &= \frac{1}{2} L \left\| \gamma \eta \frac{1}{S} \sum_{n=0}^{N-1} \sum_{k=0}^{K-1} \nabla f_{\pi_q(n)}(\mathbf{x}_{q,k}^n) \right\|^2 \\
&= \frac{1}{2} \gamma^2 \eta^2 L \frac{1}{S^2} K^2 N^2 \left\| \frac{1}{N} \sum_{n=0}^{N-1} \frac{1}{K} \sum_{k=0}^{K-1} \nabla f_{\pi_q(n)}(\mathbf{x}_{q,k}^n) \right\|^2.
\end{aligned}
$$

Plugging the preceding inequalities back into Equation (24), and using $\gamma\eta LKN\frac{1}{S} \leq 1$, we obtain

$$
f(\mathbf{x}_{q+1}) - f(\mathbf{x}_q)
$$

$$
\leq -\frac{1}{2}\gamma\eta\frac{1}{S}KN\|\nabla f(\mathbf{x}_q)\|^2 + \frac{1}{2}\gamma\eta\frac{1}{S}KN\left\|\frac{1}{N}\sum_{n=0}^{N-1}\frac{1}{K}\sum_{k=0}^{K-1}\nabla f_{\pi_q(n)}(\mathbf{x}_{q,k}^n) - \nabla f(\mathbf{x}_q)\right\|^2
$$

$$
\leq -\frac{1}{2}\gamma\eta\frac{1}{S}KN\|\nabla f(\mathbf{x}_q)\|^2 + \frac{1}{2}\gamma\eta L_{2,p}^2\frac{1}{S}\sum_{n=0}^{N-1}\sum_{k=0}^{K-1}\|\mathbf{x}_{q,k}^n - \mathbf{x}_q\|_p^2
$$

$$
\leq -\frac{1}{2}\gamma\eta\frac{1}{S}KN\|\nabla f(\mathbf{x}_q)\|^2 + \frac{1}{2}\gamma\eta L_{2,p}^2 KN\frac{1}{S}(\Delta_q)^2,
$$

where the second inequality is because $f_{\pi_q(n)}$ is $L_{2,p}$ smooth for all $n$. Applying Lemma 7, and the constraints $\gamma L_{2,p}KN\frac{1}{S} \leq \frac{1}{32}$ and $\gamma L_{2,p}K\alpha \leq \frac{1}{32}$, we obtain

$$
f(\mathbf{x}_{q+1}) - f(\mathbf{x}_q)
$$
$$
\leq -\frac{127}{256}\gamma\eta KN\frac{1}{S}\|\nabla f(\mathbf{x}_q)\|^2 + 2\gamma^3\eta L_{2,p}^2 K^3 N\frac{1}{S^3}(\bar{\varphi}_q)^2 + 2\gamma^3\eta L_{2,p}^2 K^3 N\frac{1}{S}\varsigma^2. \tag{25}
$$

Averaging over $q \in \{0, 1, \ldots, Q-1\}$, we obtain

$$
\frac{f(\mathbf{x}_Q) - f(\mathbf{x}_0)}{\gamma\eta KN\frac{1}{S}Q} \leq -\frac{127}{256}\frac{1}{Q}\sum_{q=0}^{Q-1}\|\nabla f(\mathbf{x}_q)\|^2 + 2\gamma^2 L_{2,p}^2 K^2\frac{1}{S^2}\frac{1}{Q}\sum_{q=0}^{Q-1}(\bar{\varphi}_q)^2 + 2\gamma^2 L_{2,p}^2 K^2\varsigma^2.
$$
$$
\tag{26}
$$

The following steps are similar to those in Theorem 1. Since

$$
(\bar{\varphi}_q)^2 \leq A_1(\bar{\varphi}_{q-1})^2 + A_2(\bar{\varphi}_{q-2})^2 + \cdots + A_\nu(\bar{\varphi}_{q-\nu})^2
$$
$$
+ B_0\|\nabla f(\mathbf{x}_q)\|^2 + B_1\|\nabla f(\mathbf{x}_{q-1})\|^2 + \cdots + B_\nu\|\nabla f(\mathbf{x}_{q-\nu})\|^2 + D,
$$

we obtain

$$
\frac{1}{Q}\sum_{q=0}^{Q-1}(\bar{\varphi}_q)^2 \leq \frac{1}{(1-\sum_{i=1}^\nu A_i)}\frac{1}{Q}\sum_{i=0}^{\nu-1}(\bar{\varphi}_i)^2
$$
$$
+ \frac{(\sum_{i=0}^\nu B_i)}{(1-\sum_{i=1}^\nu A_i)}\frac{1}{Q}\sum_{q=0}^{Q-1}\|\nabla f(\mathbf{x}_q)\|^2 + \frac{1}{(1-\sum_{i=1}^\nu A_i)}D. \tag{27}
$$

For $(\bar{\varphi}_i)^2$ $(0 \leq i \leq \nu - 1)$, we have

$$
(\bar{\varphi}_i)^2 \leq \tilde{B}\|\nabla f(\mathbf{x}_i)\|^2 + \tilde{D} \implies \sum_{i=0}^{\nu-1}(\bar{\varphi}_i)^2 \leq \tilde{B}\sum_{q=0}^{Q-1}\|\nabla f(\mathbf{x}_q)\|^2 + \nu\tilde{D}. \tag{28}
$$

Substituting Equation (28) into Equation (27), and the resulting inequality into Equation (26), and then using the condition

$$
\gamma \leq \min\left\{\frac{1}{4}\cdot\frac{\sqrt{1-\sum_{i=1}^\nu A_i}}{L_{2,p}K\frac{1}{S}\sqrt{\sum_{i=0}^\nu B_i}}, \frac{1}{4}\cdot\frac{\sqrt{1-\sum_{i=1}^\nu A_i}}{L_{2,p}K\frac{1}{S}M\alpha}\right\},
$$

we obtain

$$
\frac{1}{Q}\sum_{q=0}^{Q-1}\|\nabla f(\mathbf{x}_q)\|^2 \leq 5\cdot\frac{f(\mathbf{x}_0) - f(\mathbf{x}_Q)}{\gamma\eta KN\frac{1}{S}Q}
$$
$$
+ c\cdot\gamma^2 L_{2,p}^2 K^2\frac{1}{S^2}\frac{1}{Q}\nu\tilde{D} + 2\gamma^2 L_{2,p}^2 K^2\varsigma^2 + c\cdot\gamma^2 L_{2,p}^2 K^2\frac{1}{S^2}D,
$$

where $c = \frac{10}{(1-\sum_{i=1}^\nu A_i)}$. Using $f(\mathbf{x}_0) - f(\mathbf{x}_Q) \leq f(\mathbf{x}_0) - f_* = F_0$, we obtain the claimed bound.

At last, we summarize the constraints on the step sizes $\gamma$ and $\eta$:

$$\gamma \leq \min\left\{\frac{1}{4} \cdot \frac{\sqrt{1 - \sum_{i=1}^{\nu} A_i}}{L_{2,p} K \frac{1}{S} \sqrt{\sum_{i=0}^{\nu} B_i}}, \frac{1}{4} \cdot \frac{\sqrt{1 - \sum_{i=1}^{\nu} A_i}}{L_{2,p} K \frac{1}{S} \sqrt{\tilde{B}}}\right\},$$

$$\gamma \eta L K N \frac{1}{S} \leq 1,$$

$$\gamma L_{2,p} K N \frac{1}{S} \leq \frac{1}{32},$$

$$\gamma L_{2,p} K \alpha \leq \frac{1}{32},$$

$$\gamma L_p K N \frac{1}{S} \leq \frac{1}{32},$$

where the last one is from Lemma 7. For simplicity, we use a stricter constraint

$$\gamma \leq \min\left\{\frac{1}{\eta L K N \frac{1}{S}}, \frac{1}{32 L_{2,p} K N \frac{1}{S}(1+\alpha)}, \frac{\sqrt{1 - \sum_{i=1}^{\nu} A_i}}{4 L_{2,p} K \frac{1}{S} \sqrt{\sum_{i=0}^{\nu} B_i}}, \frac{\sqrt{1 - \sum_{i=1}^{\nu} A_i}}{4 L_{2,p} K \frac{1}{S} \sqrt{\tilde{B}}}, \frac{1}{32 L_p K N \frac{1}{S}}\right\}.$$

$\square$

### G.3  PL Condition

In this section, we translate our bounds in Theorem 2 with $\mu$-PL condition

$$\|\nabla f(\mathbf{x})\|^2 \geq 2\mu \left(f(\mathbf{x}) - f_*\right), \ \forall \mathbf{x} \in \mathbb{R}^d.$$

Using Theorem 3 and the relations derived in Appendix H, we can obtain the bounds in Table 2 for FL-AP, FL-RR and FL-GraB. Here, we tune the step size with Koloskova et al. [2020, Lemma 15].

**Theorem 3.** *Unless explicitly stated, we assume the same conditions as those in Theorem 3. Suppose that all the local objective functions $f_n$ satisfy the $\mu$-PL condition. If*

$$\gamma \leq \min\left\{\frac{1}{32 \eta L K N \frac{1}{S}}, \frac{1}{32 L_{2,p} K N \frac{1}{S}(1+\alpha)}, \frac{\sqrt{1 - \sum_{i=1}^{\nu} A_i c^i}}{4 L_{2,p} K \frac{1}{S} \sqrt{\tilde{B} + \sum_{i=0}^{\nu} B_i c^i}}, \frac{1}{32 L_p K N \frac{1}{S}}\right\},$$

*we obtain*

$$f(\mathbf{x}_Q) - f_* \leq \left(1 - \frac{1}{2}\gamma \eta \mu K N \frac{1}{S}\right)^Q \left(f(\mathbf{x}_0) - f_* + c_1 \cdot \frac{\tilde{D}}{LN^2}\right)$$

$$+ 4\gamma^2 \frac{1}{\mu} L_{2,p}^2 K^2 \varsigma^2 + c_2 \cdot \gamma^2 \frac{1}{\mu} L_{2,p}^2 K^2 \frac{1}{S^2} D.$$

*where $c = \frac{64}{63}$, $c_1 = \frac{2\sum_{i=1}^{\nu} c^i}{1 - \sum_{i=1}^{\nu} A_i c^i}$ and $c_2 = \frac{4}{1 - \sum_{i=1}^{\nu} A_i c^i}$.*

*Proof.* Starting from Equation (25), we obtain

$$f(\mathbf{x}_{q+1}) - f_* \leq f(\mathbf{x}_q) - f_* - \frac{127}{256}\gamma \eta K N \frac{1}{S} \|\nabla f(\mathbf{x}_q)\|^2$$

$$+ 2\gamma^3 \eta L_{2,p}^2 K^3 N \frac{1}{S^3} (\bar{\varphi}_q)^2 + 2\gamma^3 \eta L_{2,p}^2 K^3 N \frac{1}{S} \varsigma^2$$

$$\leq f(\mathbf{x}_q) - f_* - \frac{1}{4}\gamma \eta K N \frac{1}{S} \|\nabla f(\mathbf{x}_q)\|^2 - \frac{63}{256}\gamma \eta K N \frac{1}{S} \|\nabla f(\mathbf{x}_q)\|^2$$

$$+ 2\gamma^3 \eta L_{2,p}^2 K^3 N \frac{1}{S^3} (\bar{\varphi}_q)^2 + 2\gamma^3 \eta L_{2,p}^2 K^3 N \frac{1}{S} \varsigma^2$$

$$\leq \left(1 - \frac{1}{2}\gamma \eta \mu K N \frac{1}{S}\right) (f(\mathbf{x}_q) - f_*) - \frac{63}{256}\gamma \eta K N \frac{1}{S} \|\nabla f(\mathbf{x}_q)\|^2$$

$$+ 2\gamma^3 \eta L_{2,p}^2 K^3 N \frac{1}{S^3} (\bar{\varphi}_q)^2 + 2\gamma^3 \eta L_{2,p}^2 K^3 N \frac{1}{S} \varsigma^2,$$

where the last inequality is due to PL condition, $\|\nabla f(\mathbf{x})\|^2 \geq 2\mu\left(f(\mathbf{x}) - f_*\right)$. Letting $F_q := f(\mathbf{x}_q) - f_*$ and $\rho = 1 - \frac{1}{2}\gamma\eta\mu KN\frac{1}{S}$, and then applying the recursion repeatedly, we obtain

$$
F_Q \leq \rho^Q F_0 - \frac{63}{256}\gamma\eta KN\frac{1}{S}\sum_{q=0}^{Q-1}\rho^{Q-1-q}\|\nabla f(\mathbf{x}_q)\|^2
$$

$$
+ 2\gamma^3\eta L_{2,p}^2 K^3 N\frac{1}{S^3}\sum_{q=0}^{Q-1}\rho^{Q-1-q}\left(\bar{\varphi}_q\right)^2 + 2\gamma^3\eta L_{2,p}^2 K^3 N\frac{1}{S}\varsigma^2\sum_{q=0}^{Q-1}\rho^{Q-1-q}. \qquad (29)
$$

Since for $0 \leq i \leq \nu - 1$,

$$
\left(\bar{\varphi}_i\right)^2 \leq \tilde{B}\|\nabla f(\mathbf{x}_i)\|^2 + \tilde{D}
$$

and for $q \geq \nu$,

$$
\left(\bar{\varphi}_q\right)^2 \leq A_1\left(\bar{\varphi}_{q-1}\right)^2 + A_2\left(\bar{\varphi}_{q-2}\right)^2 + \cdots + A_\nu\left(\bar{\varphi}_{q-\nu}\right)^2
$$
$$
+ B_0\|\nabla f(\mathbf{x}_q)\|^2 + B_1\|\nabla f(\mathbf{x}_{q-1})\|_p^2 + \cdots + B_\nu\|\nabla f(\mathbf{x}_{q-\nu})\|_p^2 + D,
$$

after some lengthy calculations, we can obtain that

$$
\sum_{q=\nu}^{Q-1}\rho^{Q-1-q}\left(\bar{\phi}_q\right)^2 \leq \left(A_1\frac{1}{\rho} + A_2\frac{1}{\rho^2} + \cdots + A_\nu\frac{1}{\rho^\nu}\right)\sum_{q=0}^{Q-1}\rho^{Q-1-q}\left(\bar{\phi}_q\right)^2
$$

$$
+ \left(B_0 + B_1\frac{1}{\rho} + \cdots + B_\nu\frac{1}{\rho^\nu}\right)\sum_{q=0}^{Q-1}\rho^{Q-1-q}\|\nabla f(\mathbf{x}_q)\|^2
$$

$$
+ \frac{1}{1-\rho}D, \qquad (30)
$$

and

$$
\sum_{q=0}^{\nu-1}\rho^{Q-1-q}\left(\bar{\phi}_q\right)^2 \leq \tilde{B}\sum_{q=0}^{Q-1}\rho^{Q-1-q}\|\nabla f(\mathbf{x}_q)\|^2 + \rho^Q\tilde{D}\sum_{q=0}^{\nu-1}\rho^{-1-q}. \qquad (31)
$$

Combining Equation (30) and Equation (31) gives

$$
\sum_{q=0}^{Q-1}\rho^{Q-1-q}\left(\bar{\varphi}_q\right)^2 \leq \mathcal{A}\left(\tilde{B}+\mathcal{B}\right)\sum_{q=0}^{Q-1}\rho^{Q-1-q}\|\nabla f(\mathbf{x}_q)\|^2 + \frac{\mathcal{A}D}{1-\rho} + \mathcal{A}\tilde{D}\rho^Q\sum_{q=0}^{\nu-1}\rho^{-1-q}, \quad (32)
$$

where $\mathcal{A} = \dfrac{1}{1 - \left(A_1\frac{1}{\rho} + A_2\frac{1}{\rho^2} + \cdots + A_\nu\frac{1}{\rho^\nu}\right)}$ and $\mathcal{B} = B_0 + B_1\frac{1}{\rho} + \cdots + B_\nu\frac{1}{\rho^\nu}$.

Substituting $\sum_{q=0}^{Q-1}\rho^{Q-1-q}\left(\bar{\varphi}_q\right)^2$ in Equation (29) with Equation (32) yields

$$
F_Q \leq \rho^Q F_0 + 2\gamma^3\eta L_{2,p}^2 K^3 N\frac{1}{S^3}\mathcal{A}\tilde{D}\rho^Q\sum_{q=0}^{\nu-1}\rho^{-1-q}
$$

$$
+ 2\gamma^3\eta L_{2,p}^2 K^3 N\frac{1}{S}\frac{\varsigma^2}{1-\rho} + 2\gamma^3\eta L_{2,p}^2 K^3 N\frac{1}{S^3}\frac{\mathcal{A}D}{1-\rho}
$$

$$
- \gamma\eta KN\frac{1}{S}\left(\frac{63}{256} - 2\gamma^2 L_{2,p}^2 K^2\frac{1}{S^2}\mathcal{A}\left(\tilde{B}+\mathcal{B}\right)\right)\sum_{q=0}^{Q-1}\rho^{Q-1-q}\|\nabla f(\mathbf{x}_q)\|^2.
$$

Since $\rho = 1 - \frac{1}{2}\gamma\eta\mu KN\frac{1}{S}$ and $\gamma\eta LKN\frac{1}{S} \leq \frac{1}{32}$, we obtain

$$
\frac{1}{1-\rho} = \frac{2}{\gamma\eta\mu KN\frac{1}{S}} \quad \text{and} \quad \frac{1}{\rho} \leq \frac{64}{63}.
$$

For simplicity, we let $c = \frac{64}{63}$. Then,

$$\mathcal{A} = \frac{1}{1 - \left(A_1 \frac{1}{\rho} + A_2 \frac{1}{\rho^2} + \cdots + A_\nu \frac{1}{\rho^\nu}\right)} \leq \frac{1}{1 - (A_1 c + A_2 c^2 + \cdots + A_\nu c^\nu)} = \frac{1}{1 - \sum_{i=1}^{\nu} A_i c^i},$$

$$\mathcal{B} = B_0 + B_1 \frac{1}{\rho} + \cdots + B_\nu \frac{1}{\rho^\nu} \leq B_0 + B_1 c + \cdots + B_\nu c^\nu = \sum_{i=0}^{\nu} B_i c^i,$$

and $$\sum_{q=0}^{\nu-1} \rho^{-1-q} = \frac{1}{\rho} + \frac{1}{\rho^2} + \cdots + \frac{1}{\rho^\nu} \leq c + c^2 + \cdots + c^\nu = \sum_{i=1}^{\nu} c^i.$$

Next, after we use the condition

$$\gamma \leq \frac{\sqrt{1 - \sum_{i=1}^{\nu} A_i c^i}}{4 L_{2,p} K \frac{1}{S} \sqrt{\tilde{B} + \sum_{i=0}^{\nu} B_i c^i}},$$

and the upper bounds of $\mathcal{A}$, $\mathcal{B}$ and $\sum_{q=0}^{\nu-1} \rho^{-1-q}$, we obtain

$$F_Q \leq \rho^Q F_0 + 2\gamma^3 \eta L_{2,p}^2 K^3 N \frac{1}{S^3} \mathcal{A} \tilde{D} \rho^Q \sum_{q=0}^{\nu-1} \rho^{-1-q}$$

$$+ 2\gamma^3 \eta L_{2,p}^2 K^3 N \frac{1}{S} \frac{\varsigma^2}{1-\rho} + 2\gamma^3 \eta L_{2,p}^2 K^3 N \frac{1}{S^3} \frac{\mathcal{A} D}{1-\rho}$$

$$\leq \rho^Q F_0 + \frac{2 \sum_{i=1}^{\nu} c^i}{1 - \sum_{i=1}^{\nu} A_i c^i} \cdot \rho^Q \frac{\tilde{D}}{LN^2}$$

$$+ 4\gamma^2 \frac{1}{\mu} L_{2,p}^2 K^2 \varsigma^2 + \frac{4}{1 - \sum_{i=1}^{\nu} A_i c^i} \cdot \gamma^2 \frac{1}{\mu} L_{2,p}^2 K^2 \frac{1}{S^2} D.$$

At last, we summarize the constraints on the step sizes $\gamma$ and $\eta$:

$$\gamma \leq \frac{\sqrt{1 - \sum_{i=1}^{\nu} A_i c^i}}{4 L_{2,p} K \frac{1}{S} \sqrt{\tilde{B} + \sum_{i=0}^{\nu} B_i c^i}},$$

$$\gamma \eta L K N \frac{1}{S} \leq \frac{1}{32},$$

$$\gamma L_{2,p} K N \frac{1}{S} \leq \frac{1}{32},$$

$$\gamma L_{2,p} K \alpha \leq \frac{1}{32},$$

$$\gamma L_p K N \frac{1}{S} \leq \frac{1}{32},$$

where the last one is from Lemma 7. For simplicity, we use a stricter constraint

$$\gamma \leq \min \left\{ \frac{1}{32 \eta L K N \frac{1}{S}}, \frac{1}{32 L_{2,p} K N \frac{1}{S}(1+\alpha)}, \frac{\sqrt{1 - \sum_{i=1}^{\nu} A_i c^i}}{4 L_{2,p} K \frac{1}{S} \sqrt{\tilde{B} + \sum_{i=0}^{\nu} B_i c^i}}, \frac{1}{32 L_p K N \frac{1}{S}} \right\}.$$

$\square$

## H  Special Cases in FL

In this section, we provide proofs of the examples of FL.

## H.1 FL-AP

**Proposition 8** (FL-AP). *Suppose that Assumption 3 holds. Then, we obtain that, for $q \geq 0$,*

$$(\bar{\varphi}_q)^2 \leq N^2 \alpha^2 \|\nabla f(\mathbf{x}_q)\|^2 + N^2 \varsigma^2.$$

*Applying Theorem 2, we obtain*

$$\frac{1}{Q} \sum_{q=0}^{Q-1} \|\nabla f(\mathbf{x}_q)\|^2 = \mathcal{O}\left(\frac{F_0}{\gamma \eta K N \frac{1}{S} Q} + \gamma^2 L^2 K^2 \varsigma^2 + \gamma^2 L^2 K^2 N^2 \frac{1}{S^2} \varsigma^2\right).$$

*If we set $\eta = 1$ and tune the step size, the upper bound becomes*
$\mathcal{O}\left(\frac{LF_0(1+\alpha)}{Q} + \left(\frac{LF_0 S\varsigma}{NQ}\right)^{\frac{2}{3}} + \left(\frac{LF_0 N\varsigma}{NQ}\right)^{\frac{2}{3}}\right).$

*Proof.* For any epoch $q$,

$$(\bar{\varphi}_q)^2 = \max_{n \in [N]} \left\| \sum_{i=0}^{v(n)-1} \left(\nabla f_{\pi_q(i)}(\mathbf{x}_q) - \nabla f(\mathbf{x}_q)\right) \right\|^2 \leq N^2 G_q^2 \leq N^2 \alpha^2 \|\nabla f(\mathbf{x}_q)\|^2 + N^2 \varsigma^2.$$

In this example, for Theorem 2, $p = 2$, $\nu = 0$, $B_0 = N^2 \alpha^2$, $D = N^2 \varsigma^2$ and $c = 10$. These lead to

$$\frac{1}{Q} \sum_{q=0}^{Q-1} \|\nabla f(\mathbf{x}_q)\|^2 = \mathcal{O}\left(\frac{F_0}{\gamma \eta K N \frac{1}{S} Q} + \gamma^2 L^2 K^2 \varsigma^2 + \gamma^2 L^2 K^2 N^2 \frac{1}{S^2} \varsigma^2\right).$$

Next, we summarize the constraints:

$$\gamma \leq \min\left\{ \frac{1}{\eta L K N \frac{1}{S}}, \frac{1}{32 L_{2,p} K N \frac{1}{S}(1+\alpha)}, \frac{\sqrt{1 - \sum_{i=1}^{\nu} A_i}}{4 L_{2,p} K \frac{1}{S} \sqrt{\sum_{i=0}^{\nu} B_i}}, \frac{\sqrt{1 - \sum_{i=1}^{\nu} A_i}}{4 L_{2,p} K \frac{1}{S} \sqrt{\tilde{B}}}, \frac{1}{32 L_p K N \frac{1}{S}} \right\}.$$

It is from Theorem 2. For simplicity, we can use a tighter constraint

$$\gamma \leq \frac{1}{32 L K N \frac{1}{S}(1 + \eta + \alpha)}.$$

After we use the effective step size $\tilde{\gamma} := \gamma \eta K N \frac{1}{S}$, the constraint becomes

$$\tilde{\gamma} \leq \frac{1}{32 L \left(1 + \frac{1}{\eta} + \frac{\alpha}{\eta}\right)},$$

and the upper bound becomes

$$\frac{1}{Q} \sum_{q=0}^{Q-1} \|\nabla f(\mathbf{x}_q)\|^2 = \mathcal{O}\left(\frac{F_0}{\tilde{\gamma} Q} + \tilde{\gamma}^2 L^2 \frac{1}{\eta^2 N^2 \frac{1}{S^2}} \varsigma^2 + \tilde{\gamma}^2 L^2 \frac{1}{\eta^2 N^2} N^2 \varsigma^2\right).$$

Applying Lemma 1, we obtain

$$\frac{1}{Q} \sum_{q=0}^{Q-1} \|\nabla f(\mathbf{x}_q)\|^2 = \mathcal{O}\left(\frac{LF_0(1 + \eta + \alpha)}{\eta Q} + \left(\frac{LF_0 S\varsigma}{\eta NQ}\right)^{\frac{2}{3}} + \left(\frac{LF_0 N\varsigma}{\eta NQ}\right)^{\frac{2}{3}}\right).$$

For comparison with other algorithms, we set $\eta = 1$, and get

$$\frac{1}{Q} \sum_{q=0}^{Q-1} \|\nabla f(\mathbf{x}_q)\|^2 = \mathcal{O}\left(\frac{LF_0(1+\alpha)}{Q} + \left(\frac{LF_0 S\varsigma}{NQ}\right)^{\frac{2}{3}} + \left(\frac{LF_0 N\varsigma}{NQ}\right)^{\frac{2}{3}}\right).$$

$\square$

## H.2 FL-RR

**Proposition 9** (FL-RR). *Suppose that Assumption 3 holds. Then, we obtrain, for $q \geq 0$, with probability at least $1 - \delta$,*

$$\left(\bar{\varphi}_q\right)^2 \leq 4 \log^2 \left(8/\delta\right) \left( N\alpha^2 \left\| \nabla f(\mathbf{x}_q) \right\|^2 + N\varsigma^2 \right).$$

*Applying Theorem 2, we obtain that, with probability at least $1 - Q\delta$,*

$$\frac{1}{Q} \sum_{q=0}^{Q-1} \left\| \nabla f(\mathbf{x}_q) \right\|^2 = \tilde{\mathcal{O}} \left( \frac{F_0}{\gamma \eta K N^{\frac{1}{S}} Q} + \gamma^2 L^2 K^2 \varsigma^2 + \gamma^2 L^2 K^2 N \frac{1}{S^2} \varsigma^2 \right).$$

*If we set $\eta = 1$ and tune the step size, the upper bound becomes $\tilde{\mathcal{O}} \left( \frac{LF_0(1+\alpha)}{Q} + \left( \frac{LF_0 S\varsigma}{NQ} \right)^{\frac{2}{3}} + \left( \frac{LF_0 \sqrt{N}\varsigma}{NQ} \right)^{\frac{2}{3}} \right).$*

*Proof.* Since the permutations $\{\pi_q\}$ are independent across different epochs, for any $q$, when conditional on $\mathbf{x}_q$, we obtain that, with probability at least $1 - \delta$,

$$\left(\bar{\phi}_q\right)^2 = \max_{n \in [N]} \left\| \sum_{i=0}^{v(n)-1} \left( \nabla f_{\pi_q(i)}(\mathbf{x}_q) - \nabla f(\mathbf{x}_q) \right) \right\|^2 \leq NC\varsigma^2 + NC\alpha^2 \left\| \nabla f(\mathbf{x}_q) \right\|^2, \quad (33)$$

where $C = 4 \log^2 \left( \frac{8}{\delta} \right)$ and the last inequality is due to Yu and Li [2023]'s Proposition 2.3.

In this example, for Theorem 2, $p = 2$, $\nu = 0$, $B_0 = NC\alpha^2$, $D = NC\varsigma^2$ and $c = 10$. These lead to

$$\frac{1}{Q} \sum_{q=0}^{Q-1} \left\| \nabla f(\mathbf{x}_q) \right\|^2 = \mathcal{O} \left( \frac{F_0}{\gamma \eta K N^{\frac{1}{S}} Q} + \gamma^2 L^2 K^2 \varsigma^2 + \gamma^2 L^2 K^2 NC \frac{1}{S^2} \varsigma^2 \right),$$

where $F_0 = f(\mathbf{x}_0) - f_*$ and $L = L_{2,p} = L_p$ when $p = 2$. Equation (33) is used for each epoch (that is, for $Q$ times), so, by the union bound, the preceding bound holds with probability at least $1 - Q\delta$.

Next, we summarize the constraints:

$$\gamma \leq \min \left\{ \frac{1}{\eta L K N^{\frac{1}{S}}}, \frac{1}{32 L_{2,p} K N^{\frac{1}{S}} (1+\alpha)}, \frac{\sqrt{1 - \sum_{i=1}^{\nu} A_i}}{4 L_{2,p} K^{\frac{1}{S}} \sqrt{\sum_{i=0}^{\nu} B_i}}, \frac{\sqrt{1 - \sum_{i=1}^{\nu} A_i}}{4 L_{2,p} K^{\frac{1}{S}} \sqrt{\tilde{B}}}, \frac{1}{32 L_p K N^{\frac{1}{S}}} \right\}.$$

It is from Theorem 2. For simplicity, we can use a tighter constraint

$$\gamma \leq \frac{1}{32 L K N^{\frac{1}{S}} \left( 1 + \eta + \alpha + \sqrt{\frac{C}{N}} \alpha \right)}.$$

After we use the effective step size $\tilde{\gamma} := \gamma \eta K N^{\frac{1}{S}}$, the constrain becomes

$$\tilde{\gamma} \leq \frac{1}{32 L \left( 1 + \frac{1}{\eta} + \frac{\alpha}{\eta} + \frac{\sqrt{\frac{C}{N}} \alpha}{\eta} \right)},$$

and the upper bound becomes

$$\frac{1}{Q} \sum_{q=0}^{Q-1} \left\| \nabla f(\mathbf{x}_q) \right\|^2 = \tilde{\mathcal{O}} \left( \frac{F_0}{\tilde{\gamma} Q} + \tilde{\gamma}^2 L^2 \frac{1}{\eta^2 N^2 \frac{1}{S^2}} \varsigma^2 + \tilde{\gamma}^2 L^2 \frac{1}{\eta^2 N^2} N\varsigma^2 \right).$$

Applying Lemma 1, we obtain

$$\frac{1}{Q} \sum_{q=0}^{Q-1} \left\| \nabla f(\mathbf{x}_q) \right\|^2 = \tilde{\mathcal{O}} \left( \frac{LF_0 \left( 1 + \eta + \alpha \right)}{\eta Q} + \left( \frac{LF_0 S\varsigma}{\eta NQ} \right)^{\frac{2}{3}} + \left( \frac{LF_0 \sqrt{N}\varsigma}{\eta NQ} \right)^{\frac{2}{3}} \right).$$

For comparison with other algorithms, we set $\eta = 1$, and get

$$\frac{1}{Q} \sum_{q=0}^{Q-1} \left\| \nabla f(\mathbf{x}_q) \right\|^2 = \tilde{\mathcal{O}} \left( \frac{LF_0 \left( 1 + \alpha \right)}{Q} + \left( \frac{LF_0 S\varsigma}{NQ} \right)^{\frac{2}{3}} + \left( \frac{LF_0 \sqrt{N}\varsigma}{NQ} \right)^{\frac{2}{3}} \right).$$

$\square$

### H.3 FL-GraB

**Proposition 10** (FL-GraB). *Suppose that Assumption 3 holds and that $N \bmod S = 0$ and $S \bmod 2 = 0$. If each $f_n$ is $L_{2,\infty}$-smooth and $L_\infty$-smooth and $\gamma \leq \min\{\frac{1}{128L_{2,\infty}KC\frac{1}{S}}, \frac{1}{128(1+\eta)L_\infty KN\frac{1}{S}}\}$, we obtain that, for $q = 0$, $(\bar{\varphi}_0)^2 \leq N^2 \|\nabla f(\mathbf{x}_0)\|^2 + N^2 \varsigma^2$ and for $q \geq 1$, with probability at least $1 - \delta$,*

$$(\bar{\varphi}_q)^2 \leq \frac{3}{5}(\bar{\varphi}_{q-1})^2 + \left(\frac{1}{96}N^2 + 6C^2\alpha^2\right)\|\nabla f(\mathbf{x}_{q-1})\|^2 + \frac{1}{96}S^2\varsigma^2 + 6C^2\varsigma^2,$$

*where $C = \mathcal{O}\left(\log\left(\frac{dN}{\delta}\right)\right) = \tilde{\mathcal{O}}(1)$. Applying Theorem 2, we obtain that, with probability at least $1 - Q\delta$,*

$$\frac{1}{Q}\sum_{q=0}^{Q-1}\|\nabla f(\mathbf{x}_q)\|^2$$

$$= \mathcal{O}\left(\frac{F_0}{\gamma\eta KN\frac{1}{S}Q} + \gamma^2 L_{2,\infty}^2 K^2 N^2 \frac{1}{S^2}\frac{1}{Q}\varsigma^2 + \gamma^2 L_{2,\infty}^2 K^2 \varsigma^2 + \gamma^2 L_{2,\infty}^2 K^2 C^2 \frac{1}{S^2}\varsigma^2\right).$$

*After we set $\eta = 1$ and tune the step size, the upper bound becomes*
$\mathcal{O}\left(\frac{\tilde{L}F_0 + (L_{2,\infty}F_0\varsigma)^{\frac{2}{3}}}{Q} + \left(\frac{L_{2,\infty}F_0 S\varsigma}{NQ}\right)^{\frac{2}{3}} + \left(\frac{L_{2,\infty}F_0 C\varsigma}{NQ}\right)^{\frac{2}{3}}\right)$ *where $\tilde{L} = L + L_{2,\infty}\left(1 + \frac{C}{N} + \alpha\right) + L_\infty$.*

FL-GraB. Use `PairBR` (Algorithm 6) as the `Permute` function in Algorithm 2, with the inputs of $\pi_q$, $\{\mathbf{p}_q^n\}_{n=0}^{N-1}$ and $\frac{1}{N}\sum_{n=0}^{N-1}\mathbf{p}_q^n$, for each epoch $q$.

Thus, the key idea of our proof is as follows:

$$\bar{\varphi}_{q+1} \to \max_{m\in\{S,2S,\ldots,N\}}\left\|\sum_{i=0}^{m-1}\left(\sum_{j=0}^{K-1}\nabla f_{\pi_q(i)}\left(\mathbf{x}_{q,j}^i\right) - \frac{1}{N}\sum_{l=0}^{N-1}\sum_{j=0}^{K-1}\nabla f_{\pi_q(l)}\left(\mathbf{x}_{q,j}^l\right)\right)\right\|_\infty$$

$$\overset{\text{Lemma 5}}{\to} \max_{m\in\{S,2S,\ldots,N\}}\left\|\sum_{i=0}^{m-1}\left(\sum_{j=0}^{K-1}\nabla f_{\pi_{q+1}(i)}\left(\mathbf{x}_{q,j}^{\pi_q^{-1}(\pi_{q+1}(i))}\right) - \frac{1}{N}\sum_{l=0}^{N-1}\sum_{j=0}^{K-1}\nabla f_{\pi_q(l)}\left(\mathbf{x}_{q,j}^l\right)\right)\right\|_\infty$$

$$\to \bar{\varphi}_q.$$

*Proof.* We need to find the relation between $\bar{\varphi}_{q+1}$ and $\varphi_q$. For all $m \in \{S, 2S, \ldots, N\}$,

$$\varphi_{q+1}^m = \left\|\sum_{i=0}^{m-1}\left(\nabla f_{\pi_{q+1}(i)}\left(\mathbf{x}_{q+1}\right) - \nabla f\left(\mathbf{x}_{q+1}\right)\right)\right\|_\infty$$

$$= \frac{1}{K}\left\|\sum_{i=0}^{m-1}\sum_{j=0}^{K-1}\left(\nabla f_{\pi_{q+1}(i)}\left(\mathbf{x}_{q+1}\right) - \nabla f\left(\mathbf{x}_{q+1}\right)\right)\right\|_\infty$$

$$\leq \frac{1}{K}\left\|\sum_{i=0}^{m-1}\sum_{j=0}^{K-1}\nabla f_{\pi_{q+1}(i)}\left(\mathbf{x}_{q+1}\right) - \sum_{i=0}^{m-1}\sum_{j=0}^{K-1}\nabla f_{\pi_{q+1}(i)}\left(\mathbf{x}_{q,j}^{\pi_q^{-1}(\pi_{q+1}(i))}\right)\right\|_\infty$$

$$+ \frac{1}{K}\left\|\sum_{i=0}^{m-1}\sum_{j=0}^{K-1}\frac{1}{N}\sum_{l=0}^{N-1}\nabla f_{\pi_q(l)}\left(\mathbf{x}_{q+1}\right) - \sum_{i=0}^{m-1}\frac{1}{N}\sum_{l=0}^{N-1}\sum_{j=0}^{K-1}\nabla f_{\pi_q(l)}\left(\mathbf{x}_{q,j}^l\right)\right\|_\infty$$

$$+ \frac{1}{K}\left\|\sum_{i=0}^{m-1}\left(\sum_{j=0}^{K-1}\nabla f_{\pi_{q+1}(i)}\left(\mathbf{x}_{q,j}^{\pi_q^{-1}(\pi_{q+1}(i))}\right) - \frac{1}{N}\sum_{l=0}^{N-1}\sum_{j=0}^{K-1}\nabla f_{\pi_q(l)}\left(\mathbf{x}_{q,j}^l\right)\right)\right\|_\infty, \quad (34)$$

where the last inequality is due to $\nabla f(\mathbf{x}_{q+1}) = \frac{1}{N} \sum_{l=0}^{N-1} \nabla f_{\pi_q(l)}(\mathbf{x}_{q+1})$. Then,

$$
\text{T}_1 \text{ in (34)} = \frac{1}{K} \left\| \sum_{i=0}^{m-1} \sum_{j=0}^{K-1} \nabla f_{\pi_{q+1}(i)}(\mathbf{x}_{q+1}) - \sum_{i=0}^{m-1} \sum_{j=0}^{K-1} \nabla f_{\pi_{q+1}(i)} \left( \mathbf{x}_{q,j}^{\pi_q^{-1}(\pi_{q+1}(i))} \right) \right\|_{\infty}
$$

$$
\leq \frac{1}{K} \sum_{i=0}^{m-1} \sum_{j=0}^{K-1} \left\| \nabla f_{\pi_{q+1}(i)}(\mathbf{x}_{q+1}) - \nabla f_{\pi_{q+1}(i)} \left( \mathbf{x}_{q,j}^{\pi_q^{-1}(\pi_{q+1}(i))} \right) \right\|_{\infty}
$$

$$
\leq L_{\infty} \frac{1}{K} \sum_{i=0}^{m-1} \sum_{j=0}^{K-1} \left\| \mathbf{x}_{q+1} - \mathbf{x}_{q,j}^{\pi_q^{-1}(\pi_{q+1}(i))} \right\|_{\infty}
$$

$$
\leq L_{\infty} \frac{1}{K} \sum_{i=0}^{m-1} \sum_{j=0}^{K-1} \left( \left\| \mathbf{x}_{q+1} - \mathbf{x}_q \right\|_{\infty} + \left\| \mathbf{x}_q - \mathbf{x}_{q,j}^{\pi_q^{-1}(\pi_{q+1}(i))} \right\|_{\infty} \right)
$$

$$
\leq L_{\infty} \frac{1}{K} \sum_{i=0}^{m-1} \sum_{j=0}^{K-1} (\eta \Delta_q + \Delta_q)
$$

$$
\leq L_{\infty} N (\eta \Delta_q + \Delta_q),
$$

$$
\text{T}_2 \text{ in (34)} = \frac{1}{K} \left\| \sum_{i=0}^{m-1} \sum_{j=0}^{K-1} \frac{1}{N} \sum_{l=0}^{N-1} \nabla f_{\pi_q(l)}(\mathbf{x}_{q+1}) - \sum_{i=0}^{m-1} \frac{1}{N} \sum_{l=0}^{N-1} \sum_{j=0}^{K-1} \nabla f_{\pi_q(l)} \left( \mathbf{x}_{q,j}^l \right) \right\|_{\infty}
$$

$$
\leq \frac{1}{K} \sum_{i=0}^{m-1} \sum_{j=0}^{K-1} \frac{1}{N} \sum_{l=0}^{N-1} \left\| \nabla f_{\pi_q(l)}(\mathbf{x}_{q+1}) - \nabla f_{\pi_q(l)} \left( \mathbf{x}_{q,j}^l \right) \right\|_{\infty}
$$

$$
\leq L_{\infty} \frac{1}{K} \sum_{i=0}^{m-1} \sum_{j=0}^{K-1} \frac{1}{N} \sum_{l=0}^{N-1} \left\| \mathbf{x}_{q+1} - \mathbf{x}_{q,j}^l \right\|_{\infty}
$$

$$
\leq L_{\infty} \frac{1}{K} \sum_{i=0}^{m-1} \sum_{j=0}^{K-1} \frac{1}{N} \sum_{l=0}^{N-1} \left( \left\| \mathbf{x}_{q+1} - \mathbf{x}_q \right\|_{\infty} + \left\| \mathbf{x}_q - \mathbf{x}_{q,j}^l \right\|_{\infty} \right)
$$

$$
\leq L_{\infty} \frac{1}{K} \sum_{i=0}^{m-1} \sum_{j=0}^{K-1} \frac{1}{N} \sum_{l=0}^{N-1} (\eta \Delta_q + \Delta_q)
$$

$$
\leq L_{\infty} N (\eta \Delta_q + \Delta_q).
$$

Since it holds for any $m \in \{S, 2S, \ldots, N\}$, we have

$$
\bar{\varphi}_{q+1} \leq 2 L_{\infty} N (\eta \Delta_q + \Delta_q)
$$

$$
+ \frac{1}{K} \max_{m \in \{S, 2S, \ldots, N\}} \left\| \sum_{i=0}^{m-1} \left( \sum_{j=0}^{K-1} \nabla f_{\pi_{q+1}(i)} \left( \mathbf{x}_{q,j}^{\pi_q^{-1}(\pi_{q+1}(i))} \right) - \frac{1}{N} \sum_{l=0}^{N-1} \sum_{j=0}^{K-1} \nabla f_{\pi_q(l)} \left( \mathbf{x}_{q,j}^l \right) \right) \right\|_{\infty}.
$$

$$
\tag{35}
$$

Note that

$$
\sum_{j=0}^{K-1} \nabla f_{\pi_q(i)} \left( \mathbf{x}_{q,j}^i \right) - \frac{1}{N} \sum_{l=0}^{N-1} \sum_{j=0}^{K-1} \nabla f_{\pi_q(l)} \left( \mathbf{x}_{q,j}^l \right)
$$

and

$$
\sum_{j=0}^{K-1} \nabla f_{\pi_{q+1}(i)} \left( \mathbf{x}_{q,j}^{\pi_q^{-1}(\pi_{q+1}(i))} \right) - \frac{1}{N} \sum_{l=0}^{N-1} \sum_{j=0}^{K-1} \nabla f_{\pi_q(l)} \left( \mathbf{x}_{q,j}^l \right)
$$

correspond to $\mathbf{z}_{\pi(i)}$ and $\mathbf{z}_{\pi'(i)}$ in Lemma 5, respectively. We next obtain the upper bounds of

$$\left\|\mathbf{z}_{\pi(i)}\right\|_2, \left\|\sum_{i=0}^{N-1}\mathbf{z}_{\pi(i)}\right\|_\infty \text{ and } \max_{n\in[N]}\left\|\sum_{i=0}^{n-1}\mathbf{z}_{\pi(i)}\right\|_\infty,$$

and then apply Lemma 5 to the last term on the right hand side in Equation (35).

$$
\begin{aligned}
\left\|\mathbf{z}_{\pi(i)}\right\|_2 &= \left\|\sum_{j=0}^{K-1}\nabla f_{\pi_q(i)}\left(\mathbf{x}_{q,j}^i\right) - \frac{1}{N}\sum_{l=0}^{N-1}\sum_{j=0}^{K-1}\nabla f_{\pi_q(l)}\left(\mathbf{x}_{q,j}^l\right)\right\|_2 \\
&\leq \left\|\sum_{j=0}^{K-1}\nabla f_{\pi_q(i)}\left(\mathbf{x}_{q,j}^i\right) - \sum_{j=0}^{K-1}\nabla f_{\pi_q(i)}\left(\mathbf{x}_q\right)\right\|_2 \\
&\quad + \left\|\frac{1}{N}\sum_{l=0}^{N-1}\sum_{j=0}^{K-1}\nabla f_{\pi_q(l)}\left(\mathbf{x}_{q,j}^l\right) - \frac{1}{N}\sum_{l=0}^{N-1}\sum_{j=0}^{K-1}\nabla f_{\pi_q(l)}\left(\mathbf{x}_q\right)\right\|_2 \\
&\quad + \left\|\sum_{j=0}^{K-1}\nabla f_{\pi_q(i)}\left(\mathbf{x}_q\right) - \frac{1}{N}\sum_{l=0}^{N-1}\sum_{j=0}^{K-1}\nabla f_{\pi_q(l)}\left(\mathbf{x}_q\right)\right\|_2 \\
&\leq \sum_{j=0}^{K-1}\left\|\nabla f_{\pi_q(i)}\left(\mathbf{x}_{q,j}^i\right) - \nabla f_{\pi_q(i)}\left(\mathbf{x}_q\right)\right\|_2 \\
&\quad + \frac{1}{N}\sum_{l=0}^{N-1}\sum_{j=0}^{K-1}\left\|\nabla f_{\pi_q(l)}\left(\mathbf{x}_{q,j}^l\right) - \nabla f_{\pi_q(l)}\left(\mathbf{x}_q\right)\right\|_2 + KG_q \\
&\leq L_{2,\infty}\sum_{j=0}^{K-1}\left\|\mathbf{x}_{q,j}^i - \mathbf{x}_q\right\|_\infty + L_{2,\infty}\frac{1}{N}\sum_{l=0}^{N-1}\sum_{j=0}^{K-1}\left\|\mathbf{x}_{q,j}^l - \mathbf{x}_q\right\|_\infty + KG_q \\
&\leq L_{2,\infty}\sum_{j=0}^{K-1}\Delta_q + L_{2,\infty}\frac{1}{N}\sum_{l=0}^{N-1}\sum_{j=0}^{K-1}\Delta_q + KG_q \\
&\leq 2L_{2,\infty}K\Delta_q + KG_q,
\end{aligned}
$$

$$\left\|\sum_{i=0}^{N-1}\mathbf{z}_{\pi_q(i)}\right\|_\infty = \left\|\sum_{i=0}^{N-1}\left(\sum_{j=0}^{K-1}\nabla f_{\pi_q(i)}\left(\mathbf{x}_{q,j}^i\right) - \frac{1}{N}\sum_{l=0}^{N-1}\sum_{j=0}^{K-1}\nabla f_{\pi_q(l)}\left(\mathbf{x}_{q,j}^l\right)\right)\right\|_\infty = 0.$$

In addition, for any $m \in \{S, 2S, \ldots, N\}$, we have

$$
\begin{aligned}
\left\| \sum_{i=0}^{m-1} \mathbf{z}_{\pi(i)} \right\|_\infty &= \left\| \sum_{i=0}^{m-1} \left( \sum_{j=0}^{K-1} \nabla f_{\pi_q(i)} \left( \mathbf{x}_{q,j}^i \right) - \frac{1}{N} \sum_{l=0}^{N-1} \sum_{j=0}^{K-1} \nabla f_{\pi_q(l)} \left( \mathbf{x}_{q,j}^l \right) \right) \right\|_\infty \\
&\leq \left\| \sum_{i=0}^{m-1} \sum_{j=0}^{K-1} \nabla f_{\pi_q(i)} \left( \mathbf{x}_{q,j}^i \right) - \sum_{i=0}^{m-1} \sum_{j=0}^{K-1} \nabla f_{\pi_q(i)} \left( \mathbf{x}_q \right) \right\|_\infty \\
&\quad + \left\| \sum_{i=0}^{m-1} \frac{1}{N} \sum_{l=0}^{N-1} \sum_{j=0}^{K-1} \nabla f_{\pi_q(l)} \left( \mathbf{x}_{q,j}^l \right) - \sum_{i=0}^{m-1} \frac{1}{N} \sum_{l=0}^{N-1} \sum_{j=0}^{K-1} \nabla f_{\pi_q(l)} \left( \mathbf{x}_q \right) \right\|_\infty \\
&\quad + \left\| \sum_{i=0}^{m-1} \left( \sum_{j=0}^{K-1} \nabla f_{\pi_q(i)} \left( \mathbf{x}_q \right) - \frac{1}{N} \sum_{l=0}^{N-1} \sum_{j=0}^{K-1} \nabla f_{\pi_q(l)} \left( \mathbf{x}_q \right) \right) \right\|_\infty \\
&\leq \sum_{i=0}^{m-1} \sum_{j=0}^{K-1} \left\| \nabla f_{\pi_q(i)} \left( \mathbf{x}_{q,j}^i \right) - \nabla f_{\pi_q(i)} \left( \mathbf{x}_q \right) \right\|_\infty \\
&\quad + \sum_{i=0}^{m-1} \frac{1}{N} \sum_{l=0}^{N-1} \sum_{j=0}^{K-1} \left\| \nabla f_{\pi_q(l)} \left( \mathbf{x}_{q,j}^l \right) - \nabla f_{\pi_q(l)} \left( \mathbf{x}_q \right) \right\|_\infty + K \bar{\varphi}_q \\
&\leq L_\infty \sum_{i=0}^{m-1} \sum_{j=0}^{K-1} \left\| \mathbf{x}_{q,j}^i - \mathbf{x}_q \right\|_\infty + L_\infty \sum_{i=0}^{m-1} \frac{1}{N} \sum_{l=0}^{N-1} \sum_{j=0}^{K-1} \left\| \mathbf{x}_{q,j}^l - \mathbf{x}_q \right\|_\infty + K \bar{\varphi}_q \\
&\leq L_\infty \sum_{i=0}^{m-1} \sum_{j=0}^{K-1} \Delta_q + L_\infty \sum_{i=0}^{m-1} \frac{1}{N} \sum_{l=0}^{N-1} \sum_{j=0}^{K-1} \Delta_q + K \bar{\varphi}_q \\
&\leq 2 L_\infty K N \Delta_q + K \bar{\varphi}_q .
\end{aligned}
$$

Since it holds for all $m \in \{S, 2S, \ldots, N\}$, we have

$$
\max_{m \in \{S, 2S, \ldots, N\}} \left\| \sum_{i=0}^{m-1} \mathbf{z}_{\pi(i)} \right\|_\infty \leq 2 L_\infty K N \Delta_q + K \bar{\varphi}_q .
$$

Now, applying Lemma 5 to the last term on the right hand side in Equation (35), we obtain

$$
\begin{aligned}
\bar{\varphi}_{q+1} &\leq 2 L_\infty N \left( \eta \Delta_q + \Delta_q \right) + \frac{1}{2} \left( 2 L_\infty N \Delta_q + \bar{\varphi}_q \right) + C \left( 2 L_{2,\infty} \Delta_q + G_q \right) \\
&\leq \left( (3 + 2\eta) L_\infty N + 2 L_{2,\infty} C \right) \Delta_q + \frac{1}{2} \bar{\varphi}_q + C G_q .
\end{aligned}
$$

Applying Lemma 7, we obtain

$$
\begin{aligned}
\bar{\varphi}_{q+1} &\leq \left( (3 + 2\eta) L_\infty N + 2 L_{2,\infty} C \right) \Delta_q + \frac{1}{2} \bar{\varphi}_q + C G_q \\
&\leq \left( (3 + 2\eta) L_\infty N + 2 L_{2,\infty} C \right) \cdot \frac{32}{31} \gamma K \frac{1}{S} \left( \bar{\varphi}_q + N \| \nabla f(\mathbf{x}_q) \| + S G_q \right) + \frac{1}{2} \bar{\varphi}_q + C G_q \\
&\leq \frac{13}{24} \bar{\varphi}_q + \frac{1}{24} N \| \nabla f(\mathbf{x}_q) \| + \frac{1}{24} S G_q + C G_q .
\end{aligned}
$$

where the last inequality uses $\gamma(1 + \eta) L_\infty K N \frac{1}{S} \leq \frac{1}{128}$ and $\gamma L_{2,\infty} K C \frac{1}{S} \leq \frac{1}{128}$. Then, we obtain

$$
\left( \bar{\varphi}_{q+1} \right)^2 \leq \frac{3}{5} \left( \bar{\varphi}_q \right)^2 + \frac{1}{96} N^2 \| \nabla f(\mathbf{x}_q) \|^2 + 6 C^2 \alpha^2 \| \nabla f(\mathbf{x}_q) \|^2 + \frac{1}{96} S^2 \varsigma^2 + 6 C^2 \varsigma^2 .
$$

So the relation between $\bar{\varphi}_q$ and $\bar{\varphi}_{q-1}$ is

$$
\left( \bar{\varphi}_q \right)^2 \leq \frac{3}{5} \left( \bar{\varphi}_{q-1} \right)^2 + \frac{1}{96} N^2 \| \nabla f(\mathbf{x}_{q-1}) \|^2 + 6 C^2 \alpha^2 \| \nabla f(\mathbf{x}_{q-1}) \|^2 + \frac{1}{96} S^2 \varsigma^2 + 6 C^2 \varsigma^2 ,
$$

for $q \geq 1$. In addition, we have $(\bar{\varphi}_0)^2 \leq N^2 \|\nabla f(\mathbf{x}_0)\|^2 + N^2 \varsigma^2$.

In this example, for Theorem 2, $p = \infty$, $\nu = 1$, $A_1 = \frac{3}{5}$, $B_0 = 0$, $B_1 = \frac{1}{96}N^2 + 6C^2\alpha^2$, $D = 6C^2\varsigma^2$, $\tilde{B} = N^2\alpha^2$, $\tilde{D} = N^2\varsigma^2$ and $c = 25$. These lead to

$$
\frac{1}{Q} \sum_{q=0}^{Q-1} \|\nabla f(\mathbf{x}_q)\|^2
$$
$$
= \mathcal{O}\left( \frac{F_0}{\gamma\eta K N\frac{1}{S}Q} + \gamma^2 L_{2,\infty}^2 K^2 N^2 \frac{1}{S^2}\frac{1}{Q}\varsigma^2 + \gamma^2 L_{2,\infty}^2 K^2 \varsigma^2 + \gamma^2 L_{2,\infty}^2 K^2 C^2 \frac{1}{S^2}\varsigma^2 \right),
$$

where $F_0 = f(\mathbf{x}_0) - f_*$. Lemma 4 is used for each epoch (that is, for $Q$ times), so, by the union bound, the preceding bound holds with probability at least $1 - Q\delta$.

Next, we summarize the constraints on the step size:

$$
\gamma \leq \min\left\{ \frac{1}{\eta LK N\frac{1}{S}}, \frac{1}{32L_{2,p}K N\frac{1}{S}(1+\alpha)}, \frac{\sqrt{1 - \sum_{i=1}^{\nu} A_i}}{4L_{2,p}K\frac{1}{S}\sqrt{\sum_{i=0}^{\nu} B_i}}, \frac{\sqrt{1 - \sum_{i=1}^{\nu} A_i}}{4L_{2,p}K\frac{1}{S}\sqrt{\tilde{B}}}, \frac{1}{32L_p K N\frac{1}{S}} \right\},
$$
$$
\gamma \leq \frac{1}{128(1+\eta)L_\infty K N\frac{1}{S}},
$$
$$
\gamma \leq \frac{1}{128L_{2,\infty}KC\frac{1}{S}}.
$$

where the first one is from Theorem 1 and the others are from the derivation of the relation. For simplicity, we can use a tighter constraint

$$
\gamma \leq \min\left\{ \frac{1}{\eta LK N\frac{1}{S}}, \frac{1}{128L_{2,\infty}K(N + C + N\alpha)\frac{1}{S}}, \frac{1}{128(1+\eta)L_\infty K N\frac{1}{S}} \right\}.
$$

After we use the effective step size $\tilde{\gamma} := \gamma\eta K N\frac{1}{S}$, the constraint becomes

$$
\tilde{\gamma} \leq \min\left\{ \frac{1}{L}, \frac{\eta}{128L_{2,\infty}\left(1 + \frac{C}{N} + \alpha\right)}, \frac{\eta}{128(1+\eta)L_\infty} \right\},
$$

and the upper bound becomes

$$
\frac{1}{Q} \sum_{q=0}^{Q-1} \|\nabla f(\mathbf{x}_q)\|^2 = \mathcal{O}\left( \frac{F_0}{\tilde{\gamma}Q} + \tilde{\gamma}^2 L_{2,\infty}^2 \frac{1}{\eta^2}\frac{1}{Q}\varsigma^2 + \tilde{\gamma}^2 L_{2,\infty}^2 \frac{1}{\eta^2 N^2 \frac{1}{S^2}}\varsigma^2 + \tilde{\gamma}^2 L_{2,\infty}^2 \frac{1}{\eta^2 N^2}C^2\varsigma^2 \right).
$$

Applying Lemma 1, and setting $\eta = 1$, we obtain

$$
\frac{1}{Q} \sum_{q=0}^{Q-1} \|\nabla f(\mathbf{x}_q)\|^2
$$
$$
= \mathcal{O}\left( \frac{\left(L + L_{2,\infty}\left(1 + \frac{C}{N} + \alpha\right) + L_\infty\right) F_0}{Q} + \frac{(L_{2,\infty}F_0\varsigma)^{\frac{2}{3}}}{Q} + \left(\frac{L_{2,\infty}F_0 S\varsigma}{NQ}\right)^{\frac{2}{3}} + \left(\frac{L_{2,\infty}F_0 C\varsigma}{NQ}\right)^{\frac{2}{3}} \right).
$$

$\square$

# I  Experiments

In this section, we provide the experimental results of FL on real data sets. Refer to Lu et al. [2022a] and Cooper et al. [2023] for the experimental results of SGD on real data sets.

**Algorithms.** We consider the two algorithms in (regularized-participation) FL in the main body: FL-RR and FL-GraB.

**Datasets and models.** We consider the datasets CIFAR-10 [Krizhevsky et al., 2009], CIFAR-100 [Krizhevsky et al., 2009] and CINIC-10 [Darlow et al., 2018]. We use the convolutional neural network (CNN) from [Acar et al., 2021] and ResNet-10 [He et al., 2016].

**Hyperparameters.** We partition the data examples by the way in McMahan et al. [2017] and Zeng et al. [2023] among $N = 1000$ clients, ensuring that each client contains data examples from about one label. We use SGD as the local solver with the learning rate being constant, the momentum being 0 and weight decay being 0. We set the global step size to $\eta = 1$. We set the total number of training rounds to 20000 (that is, $Q = 200$ epochs). For other setups, following those in Wang and Ji [2022], we set the number of participating clients in each training round to $S = 10$, the number of local update steps to $K = 5$, the mini-batch size to 16.

**Two-stage grid search.** We use a two-stage grid search for tuning the step size. Specifically, we first perform a *coarse-grained* search over a broad range of step sizes to identify a best step size at a high level. After that, based on the best step size found, we perform a *fine-grained* search around it by testing neighboring step sizes to find a more precise value. For instance, in the first stage, we can use a grid of $\{10^{-2}, 10^{-1}, 10^0\}$ to find the coarse-grained best step size; in the second stage, if the coarse-grained best step size is $10^{-1}$, we use the grid of $\{10^{-1.5}, 10^{-1}, 10^{0.5}\}$ to find the fine-grained best step size. Notably, we tune the step size by the two-stage grid search for FL-RR, and reuse the best step size for FL-GraB. We get that the best step size is $10^{-1} = 0.1$ for CNN; in the same way, we get that the best step size is $10^{-0.5} \approx 0.316$ for ResNet-10.

**Computational resources.** We use one machine with one CPU and three GPUs. The CPU is Intel(R) Xeon(R) Gold 5218R CPU with 2.10GHz. The GPU is NVIDIA GeForce RTX 4090. It takes about 3 hours, 3 hours and 4 hours for each single run of training CNN on CIFAR-10, CIFAR-100 and CINIC-10, respectively. It takes about 4 hours, 4 hours, and 5 hours for each single run of training ResNet-10 on CIFAR-10, CIFAR-100 and CINIC-10, respectively.

**Experimental results.** The experimental results are in Figures 2 and 3. We see that FL-GraB outperforms FL-RR across all tasks, especially in the early stages. This is aligned with our theory that the convergence rate of FL-GraB is better than that of FL-RR.

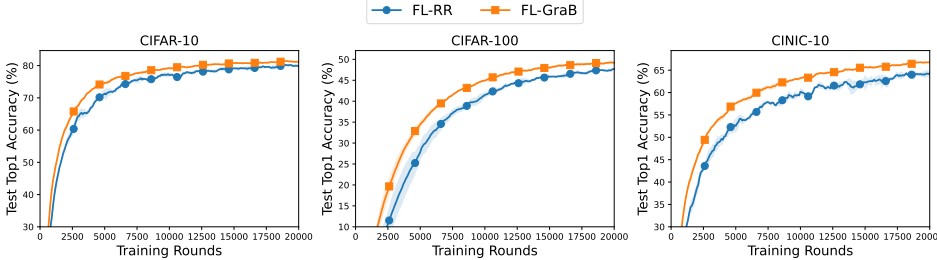

Figure 2: Test accuracy results for training CNN on CIFAR-10, CIFAR-100 and CINIC-10. As done in Wang and Ji [2022], we apply moving average on the recorded data points with a window length of 6; note that we record the results every 100 rounds (that is, one epoch). The shaded areas show the standard deviation across 5 random seeds.

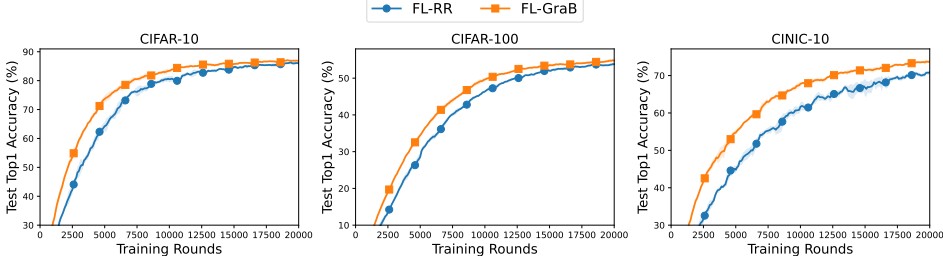

Figure 3: Test accuracy results for training ResNet-10 on CIFAR-10, CIFAR-100 and CINIC-10. As done in Wang and Ji [2022], we applied moving average on the recorded data points with a window length of 6; note that we record the results every 100 rounds (that is, one epoch). The shaded areas show the standard deviation across 5 random seeds.

# J Additional Extensions of SGD

In the main body, we considered periods of the same size as an epoch, which is sufficient for the convergence of permutation-based SGD. In fact, we can consider periods of arbitrary fixed sizes, by which we can extend the framework to include classic SGD. This mainly relies on the techniques developed in Lu et al. [2022b], Wang and Ji [2022] and Koloskova et al. [2024].

## J.1 Results

Let the size of the periods be $E$, which can be different from the epoch length $N$. To extend the results developed in the main body for this scenario, we make the following changes. First, before stating the adjusted results, we redefine some notations, that is, to reformulate those defined over "epochs" in terms of "periods". For instance, the order error $\bar{\phi}_q$ in any *period q* is defined as

$$\bar{\phi}_q := \max_{e \in [E]} \left\{ \phi_q^e := \left\| \sum_{i=0}^{e-1} \left( \nabla f_{\pi_q(i)}(\mathbf{x}_q) - \nabla f(\mathbf{x}_q) \right) \right\|_p \right\},$$

where $\pi_q$ denotes the order in period $q$ and $\mathbf{x}_q$ means the initial parameter vector in period $q$. See Definition 1 for comparison. Furthermore, we need to define

$$\psi_q^E := \max_{q \in \{0,1,\ldots,Q-1\}} \left\| \sum_{e=0}^{E-1} \nabla f_{\pi_q(e)}(\mathbf{x}_q^e) - \nabla f(\mathbf{x}_q) \right\|,$$

where $\mathbf{x}_q^n$ denotes the parameter vector after $e$ steps in period $q$.

Theorem 4 provides the extended framework for SGD.

**Theorem 4.** *Let the global objective function $f$ be $L$-smooth and each local objective functions $f_n$ be $L_{2,p}$-smooth and $L_p$-smooth ($p \geq 2$). Let $\nu \geq 0$ be a numerical constant. Suppose that there exist $\tilde{B}$ and $\tilde{D}$ such that for $0 \leq q \leq \nu - 1$,*

$$(\bar{\phi}_q)^2 \leq \tilde{B} \left\| \nabla f(\mathbf{x}_q) \right\|^2 + \tilde{D},$$

*and there exist $\{A_i\}$, $\{B_i\}$ and $D$ such that for $q \geq \nu$,*

$$(\bar{\phi}_q)^2 \leq \sum_{i=1}^{\nu} A_i (\bar{\phi}_{q-i})^2 + \sum_{i=0}^{\nu} B_i \left\| \nabla f(\mathbf{x}_{q-i}) \right\|^2 + D.$$

*If $\gamma \leq \min \left\{ \frac{1}{LE}, \frac{1}{32 L_{2,p} E}, \frac{\sqrt{1-\sum_{i=1}^{\nu} A_i}}{4 L_{2,p} \sqrt{\sum_{i=0}^{\nu} B_i}}, \frac{\sqrt{1-\sum_{i=1}^{\nu} A_i}}{4 L_{2,p} \sqrt{\tilde{B}}}, \frac{1}{32 L_p E} \right\}$, then*

$$\frac{1}{Q} \sum_{q=0}^{Q-1} \left\| \nabla f(\mathbf{x}_q) \right\|^2 \leq 10 \cdot \frac{F_0}{\gamma E Q} + c \cdot \gamma^2 L_{2,p}^2 \frac{1}{Q} \nu \tilde{D} + c \cdot \gamma^2 L_{2,p}^2 D + 10 \cdot \frac{1}{E^2} \left( \psi_q^E \right)^2,$$

*where $c = 30/(1-\sum_{i=1}^{\nu} A_i)$ is a numerical constant.*

When $E$ is a multiple of $N$ ($\psi_q^E = 0$), Theorem 4 is reduced to Theorem 1, including the permutation-based SGD algorithms. When $E = \Theta(\frac{1}{\gamma L})$ ($\psi_q^E$ is allowed to be nonzero), we can obtain the convergence rates of AP (Proposition 11) and classic SGD (Proposition 12). As shown in Table 8, theses rates match those of Koloskova et al. [2024].

**Assumption 4.** *There exists a nonnegative constant $\varsigma$ such that for any $n \in \{0, 1, \ldots, N-1\}$,*

$$\left\| \nabla f_n(\mathbf{x}) - \nabla f(\mathbf{x}) \right\|^2 \leq \varsigma^2, \ \forall \mathbf{x} \in \mathbb{R}^d.$$

**Proposition 11** (AP). *Suppose that Assumption 4 holds. Then, we can obtain that, for $q \geq 0$,*

$$\left( \bar{\phi}_q \right)^2 \leq N^2 \varsigma^2, \text{and } \psi_q^E \leq N^2 \varsigma^2.$$

*Applying Theorem 4, choosing $E = \Theta(\lfloor \frac{1}{\gamma L} \rfloor)$ and tuning the step size, for $E \leq \frac{T}{2}$, we obtain*

$$\min_{t \in \{0,1,\ldots T-1\}} \left\| \nabla f(\mathbf{y}_t) \right\|^2 = \mathcal{O} \left( \frac{L F_0}{T} + \min \left\{ \left( \frac{L F_0 N \varsigma}{T} \right)^{\frac{2}{3}}, \left( \frac{L F_0 N \varsigma^2}{T} \right)^{\frac{1}{2}} \right\} \right),$$

*where $\mathbf{y}_t$ denotes the model parameter vector in iteration $t$.*

**Proposition 12** (Classic SGD)**.** *Suppose that Assumption 4 holds. Then, we can obtain that, for $q \geq 0$, with probability at least $1 - \delta$,*

$$\left(\bar{\phi}_q\right)^2 \leq EC\varsigma^2, \text{ and } \psi_q^E \leq EC\varsigma^2.$$

*where $C = 9\log^2(\frac{8}{\delta})$. Applying Theorem 4, choosing $E = \Theta(\lfloor \frac{1}{\gamma L} \rfloor)$ and tuning the step size, for $E \leq \frac{T}{2}$, we obtain that, with probability at least $1 - T\delta$,*

$$\min_{t \in \{0,1,\dots T-1\}} \|\nabla f(\mathbf{y}_t)\|^2 = \mathcal{O}\left(\frac{LF_0}{T} + \left(\frac{LF_0\varsigma^2}{T}\right)^{\frac{1}{2}}\right),$$

*where $\mathbf{y}_t$ denotes the model parameter vector in iteration $t$.*

At last, we discuss Theorem 4 and the technique of setting $E = \Theta(\frac{1}{\gamma L})$:

- For classic SGD, Theorem 4 does not require the unbiased condition, which often used in the convergence analysis of classis SGD [Ghadimi and Lan, 2013]. This motivates a new approach for the convergence analysis of Markov chain SGD [Even, 2023; Beznosikov et al., 2023; Koloskova et al., 2024], where the unbiased condition is not satisfied.

- The standard approach (analyzing convergence over periods of a fixed size $E$) in permutation-based SGD causes a constraint of $\gamma LE \lesssim 1$. In the main body, $E = N$, causing the constraint on the step size $\gamma LN \lesssim 1$, which finally leads to a worse bound on the optimization term (See the "AP" row in Table 1). As noted in Koloskova et al. [2024], we can set $E = \Theta(\frac{1}{\gamma L})$ to circumvent the constraint of $\gamma LE \lesssim 1$, leading to a better bound on the optimization term for AP (Proposition 11).

- For AP, whose permutations are arbitrary (without any specific structure), analyzing convergence for arbitrary $E$ will not affect the statistical properties of these permutations. However, this does not apply to IP and DP. Take IP as an example. For IP, whose permutations are independently generated across epochs, analyzing convergence for $E = \Theta(\frac{1}{\gamma L})$ ($E$ does not necessarily equal $N$) will compromise the nice statistical properties of these permutations: The elements in different permutations may be divided into the same period and the elements in one permutation may be divided into different periods.

- For the cases $E = \Theta(\frac{1}{\gamma L})$, Theorem 4 and the previous works [Wang and Ji, 2022; Koloskova et al., 2024] use Assumption 4, which is stronger than Assumption 3 used in the cases $E$ is a multiple of $N$ (in the main body).

- Compared with Koloskova et al. [2024], one additional constraint $E \leq \frac{T}{2}$ is required for ours (Propositions 11 and 12) and Wang and Ji [2022].

Table 8: Comparison of convergence rates (of AP and classic SGD) with Koloskova et al. [2024]. Numerical constants and polylogarithmic factors are hided.

| Algorithm | Koloskova et al. [2024] | This work |
|---|---|---|
| AP | $\frac{LF_0}{T} + \min\left\{\left(\frac{LF_0 N\varsigma}{T}\right)^{\frac{2}{3}}, \left(\frac{LF_0 N\varsigma^2}{T}\right)^{\frac{1}{2}}\right\}$ [1] | $\frac{LF_0}{T} + \min\left\{\left(\frac{LF_0 N\varsigma}{T}\right)^{\frac{2}{3}}, \left(\frac{LF_0 N\varsigma^2}{T}\right)^{\frac{1}{2}}\right\}$ [2] |
| Classic SGD | $\frac{LF_0}{T} + \left(\frac{LF_0\varsigma^2}{T}\right)^{\frac{1}{2}}$ [1] | $\frac{LF_0}{T} + \left(\frac{LF_0\varsigma^2}{T}\right)^{\frac{1}{2}}$ [2] |

[1] These bounds are for $\frac{1}{T}\sum_{t=0}^{T-1}\mathbb{E}\|\nabla f(\mathbf{y}_t)\|^2$ and do not require the additional constraint $E \leq \frac{T}{2}$. Therefore, if these factors are taken into consideration, the bounds of Koloskova et al. [2024] are better than ours.
[2] These bounds are for $\min_{t \in \{0,1,\dots,T-1\}}\|\nabla f(\mathbf{y}_t)\|^2$ and require the additional constraint $E \leq \frac{T}{2}$.

## J.2 Proofs

This section provides the proofs of Theorem 4 and the helper lemmas for Theorem 4.

**Lemma 8.** *Suppose that there are $n$ vectors $\mathbf{x}_0, \mathbf{x}_1 \dots, \mathbf{x}_{n-1} \in \mathbb{R}^d$ such that $\frac{1}{n}\sum_{i=0}^{n-1}\mathbf{x}_i = \mathbf{0}$ and $\|\mathbf{x}_i\|^2 \leq \varsigma^2$ for any $i \in \{0, 1 \dots, n-1\}$. Suppose that the indices $\pi$ are sampled uniformly at*

*random from $\{0, 1, \ldots, n-1\}$ with replacement in an i.i.d. manner. Then, for any integer $m \geq 1$, the following inequality holds with probability at least $1 - \delta$:*

$$\left\| \sum_{i=0}^{m-1} \mathbf{x}_{\pi(i)} \right\|^2 \leq 9m\varsigma^2 \log^2 \left( \frac{8}{\delta} \right).$$

*Proof.* The proof is modified from Yu and Li [2023, Proposition 2.3]. For any $i \in \{0, 1 \ldots, n-1\}$, the matrix

$$\boldsymbol{X}_i = \begin{bmatrix} \mathbf{0}_{d \times d} & \mathbf{x}_i \\ \mathbf{x}_i^{\mathrm{T}} & 0 \end{bmatrix} \in \mathbb{R}^{(d+1) \times (d+1)}$$

can be constructed. It can be obtained that

$$\|\boldsymbol{X}_i\|_2 = \|\mathbf{x}_i\| \leq \varsigma,$$

$$\left\| \sum_{i=0}^{n-1} \boldsymbol{X}_i \right\|_2 \leq n\varsigma^2.$$

Then, applying Yu and Li [2023, Lemma 2.1] (it also holds for sampling with replacement) with $\lambda = n\varsigma^2$, $b = \varsigma$, we obtain

$$\mathbb{P} \left( \left\| \sum_{i=0}^{m-1} \boldsymbol{X}_{\pi(i)} \right\|_2 \geq s \right) \leq 4\tilde{d} \exp \left( -\frac{s^2/2}{\lambda m/n + bs/3} \right) = 4\tilde{d} \exp \left( -\frac{s^2/2}{m\varsigma^2 + \varsigma s/3} \right).$$

Solving $4\tilde{d} \exp \left( -\frac{s^2/2}{m\varsigma^2 + \varsigma s/3} \right) \leq \delta$ yields $s \geq \frac{\varsigma}{3} \log(\frac{4\tilde{d}}{\delta}) + \sqrt{\frac{\varsigma^2}{9} \log^2(\frac{4\tilde{d}}{\delta}) + 2m\varsigma^2 \log(\frac{4\tilde{d}}{\delta})}$. Using $\log(\frac{4\tilde{d}}{\delta}) > 1$, we obtain that if $s \geq 3\sqrt{m}\varsigma \log(\frac{4\tilde{d}}{\delta})$, then $4\tilde{d} \exp \left( -\frac{s^2/2}{m\varsigma^2 + \varsigma s/3} \right) \leq \delta$. As a result, we obtain

$$\mathbb{P} \left( \left\| \sum_{i=0}^{m-1} \boldsymbol{X}_{\pi(i)} \right\|_2 \geq 3\sqrt{m}\varsigma \log \left( \frac{4\tilde{d}}{\delta} \right) \right) \leq \delta.$$

Using $\left\| \sum_{i=0}^{m-1} \boldsymbol{X}_{\pi(i)} \right\|_2 = \left\| \sum_{i=0}^{m-1} \mathbf{x}_{\pi(i)} \right\|_2$ and $\tilde{d} = 2$, we obtain the claimed result. $\qquad \square$

*Proof of Theorem 4.* For SGD, the cumulative updates over any epoch $q$ are

$$\mathbf{x}_{q+1} - \mathbf{x}_q = -\gamma \sum_{e=0}^{E-1} \nabla f_{\pi_q(e)}(\mathbf{x}_q^e). \tag{36}$$

Since the global objective function $f$ is $L$-smooth, it follows that

$$f(\mathbf{x}_{q+1}) \leq f(\mathbf{x}_q) + \langle \nabla f(\mathbf{x}_q), \mathbf{x}_{q+1} - \mathbf{x}_q \rangle + \frac{1}{2} L \|\mathbf{x}_{q+1} - \mathbf{x}_q\|^2. \tag{37}$$

Using Equation (36), we obtain

$\langle \nabla f(\mathbf{x}_q), \mathbf{x}_{q+1} - \mathbf{x}_q \rangle$

$= -\gamma E \left\langle \nabla f(\mathbf{x}_q), \frac{1}{E} \sum_{e=0}^{E-1} \nabla f_{\pi_q(e)}(\mathbf{x}_q^e) \right\rangle$

$= -\frac{1}{2}\gamma E \|\nabla f(\mathbf{x}_q)\|^2 - \frac{1}{2}\gamma E \left\| \frac{1}{E} \sum_{n=0}^{E-1} \nabla f_{\pi_q(e)}(\mathbf{x}_q^e) \right\|^2 + \frac{1}{2}\gamma E \left\| \frac{1}{E} \sum_{n=0}^{E-1} \nabla f_{\pi_q(e)}(\mathbf{x}_q^e) - \nabla f(\mathbf{x}_q) \right\|^2,$

where the second equality uses $2\langle \mathbf{x}, \mathbf{y} \rangle = \|\mathbf{x}\|^2 + \|\mathbf{y}\|^2 - \|\mathbf{x} - \mathbf{y}\|^2$. Using Equation (36), we obtain

$$\frac{1}{2} L \|\mathbf{x}_{q+1} - \mathbf{x}_q\|^2 = \frac{1}{2}\gamma^2 LE^2 \left\| \frac{1}{E} \sum_{e=0}^{E-1} \nabla f_{\pi_q(e)}(\mathbf{x}_q^e) \right\|^2.$$

Next, plugging the preceding two equalities back in to Equation (37), we obtain

$$f(\mathbf{x}_{q+1}) \leq f(\mathbf{x}_q) + \langle \nabla f(\mathbf{x}_q), \mathbf{x}_{q+1} - \mathbf{x}_q \rangle + \frac{1}{2} L \|\mathbf{x}_{q+1} - \mathbf{x}_q\|^2$$

$$\leq f(\mathbf{x}_q) - \frac{1}{2} \gamma E \|\nabla f(\mathbf{x}_q)\|^2 - \frac{1}{2} \gamma E (1 - \gamma L E) \mathbb{E} \left\| \frac{1}{E} \sum_{e=0}^{E-1} \nabla f_{\pi_q(e)}(\mathbf{x}_q^e) \right\|^2$$

$$+ \frac{1}{2} \gamma E \left\| \frac{1}{E} \sum_{e=0}^{E-1} \nabla f_{\pi_q(e)}(\mathbf{x}_q^e) - \nabla f(\mathbf{x}_q) \right\|^2 .$$

Since $\gamma L E \leq 1$, we obtain

$$f(\mathbf{x}_{q+1}) \leq f(\mathbf{x}_q) - \frac{1}{2} \gamma E \|\nabla f(\mathbf{x}_q)\|^2 + \frac{1}{2} \gamma E \left\| \frac{1}{E} \sum_{e=0}^{E-1} \nabla f_{\pi_q(e)}(\mathbf{x}_q^e) - \nabla f(\mathbf{x}_q) \right\|^2 . \qquad (38)$$

Since each local objective function $f_n$ is $L_{2,p}$-smooth, we obtain

$$\text{T}_3 \text{ in (38)} = \frac{1}{2} \gamma E \left\| \frac{1}{E} \sum_{e=0}^{E-1} \nabla f_{\pi_q(e)}(\mathbf{x}_q^e) - \nabla f(\mathbf{x}_q) \right\|^2$$

$$\leq \gamma E \left\| \frac{1}{E} \sum_{e=0}^{E-1} \nabla f_{\pi_q(e)}(\mathbf{x}_q^e) - \frac{1}{E} \sum_{e=0}^{E-1} \nabla f_{\pi_q(e)}(\mathbf{x}_q) \right\|^2$$

$$+ \gamma E \left\| \frac{1}{E} \sum_{e=0}^{E-1} \nabla f_{\pi_q(e)}(\mathbf{x}_q) - \nabla f(\mathbf{x}_q) \right\|^2$$

$$\leq \gamma L_{2,p}^2 \sum_{e=0}^{E-1} \|\mathbf{x}_q^e - \mathbf{x}_q\|_p^2 + \gamma \frac{1}{E} \left( \psi_q^E \right)^2$$

$$\leq \gamma L_{2,p}^2 E \left( \Delta_q \right)^2 + \gamma \frac{1}{E} \left( \psi_q^E \right)^2 .$$

*This is the key change of the proof of Theorem 4 compared to Theorem 1.* Plugging the preceding inequality back into Equation (38), we obtain

$$f(\mathbf{x}_{q+1}) \leq f(\mathbf{x}_q) - \frac{1}{2} \gamma E \|\nabla f(\mathbf{x}_q)\|^2 + \gamma L_{2,p}^2 E \left( \Delta_q \right)^2 + \gamma \frac{1}{E} \left( \psi_q^E \right)^2$$

$$\leq f(\mathbf{x}_q) - \frac{1}{2} \gamma E \left( 1 - 6\gamma^2 L_{2,p}^2 E^2 \right) \|\nabla f(\mathbf{x}_q)\|^2 + 3\gamma^3 L_{2,p}^2 E \left( \bar{\phi}_q \right)^2 + \gamma \frac{1}{E} \left( \psi_q^E \right)^2$$

$$\leq f(\mathbf{x}_q) - \frac{509}{1024} \gamma E \|\nabla f(\mathbf{x}_q)\|^2 + 3\gamma^3 L_{2,p}^2 E \left( \bar{\phi}_q \right)^2 + \gamma \frac{1}{E} \left( \psi_q^E \right)^2 ,$$

where the second inequality uses Lemma 9 and the last inequality uses $\gamma L_{2,p} E \leq \frac{1}{32}$. Notably, the resulting inequality is almost identical to Equation (11), except an additional term $\gamma \frac{1}{E} \left( \psi_q^E \right)^2$. The term $\left( \psi_q^E \right)^2$ will not be dealt with further in this theorem, so the remaining steps of this proof are almost the same as those in Theorem 1. For these reasons, we omit the remaining steps. $\qquad \square$

To prove Theorem 4, we require the following lemma. In particular, we define the maximum parameter deviation (drift) in any period $q$, $\Delta_q$ as

$$\Delta_q = \max_{e \in [E]} \|\mathbf{x}_q^e - \mathbf{x}_q^0\|_p .$$

**Lemma 9.** *If $\gamma L_p E \leq \frac{1}{32}$, the maximum parameter drift is bounded:*

$$\Delta_q \leq \frac{32}{31} \gamma \bar{\phi}_q + \frac{32}{31} \gamma E \|\nabla f(\mathbf{x}_q)\| ,$$

$$\left( \Delta_q \right)^2 \leq 3\gamma^2 \left( \bar{\phi}_q \right)^2 + 3\gamma^2 E^2 \|\nabla f(\mathbf{x}_q)\|^2 .$$

*Proof.* The proof is identical to that of Lemma 6, except that the epoch-based notations are replaced by the period-based notations. □

*Proof of Proposition 11.* For any $q \in \{0, 1 \ldots, Q-1\}$ and $e \in [E]$,

$$\left(\phi_q^e\right)^2 = \left\|\sum_{i=0}^{e-1} \left(\nabla f_{\pi_q(i)}(\mathbf{x}_q) - \nabla f(\mathbf{x}_q)\right)\right\|^2 \leq 4 \min\{E, N\} N \varsigma^2. \tag{39}$$

See the detailed analyses of Equation (39) in Koloskova et al. [2024, Appendix C, Proofs of the bounds in Table 2] and Lu et al. [2022b, Appendix A.5, Justifications for Assumption 2 under various example orderings].

In this proposition, for Theorem 4, $p = 2$, $\nu = 0$, $D = 4 \min\{E, N\} N \varsigma^2$, $c = 30$ and $\psi_q^E = 4 \min\{E, N\} N \varsigma^2$. These lead to

$$\frac{1}{Q} \sum_{q=0}^{Q-1} \|\nabla f(\mathbf{x}_q)\|^2 = \mathcal{O}\left(\frac{F_0}{\gamma EQ} + \gamma^2 L^2 \min\{N^2, NE\}\varsigma^2 + \frac{1}{E^2} \min\{N^2, NE\}\varsigma^2\right),$$

with the constraint $\gamma \leq \frac{1}{32LE}$.

Then, choosing $E = \lfloor \frac{1}{32\gamma L} \rfloor$, we obtain

$$\frac{1}{Q} \sum_{q=0}^{Q-1} \|\nabla f(\mathbf{x}_q)\|^2 = \mathcal{O}\left(\frac{F_0}{\gamma EQ} + \min\{\gamma^2 L^2 N^2 \varsigma^2, \gamma LN\varsigma^2\}\right).$$

Using $\gamma \lesssim \frac{1}{L}$, which is due to $E = \lfloor \frac{1}{32\gamma L} \rfloor \geq 1$, and applying Lemmas 1 and 2, we obtain

$$\frac{1}{Q} \sum_{q=0}^{Q-1} \|\nabla f(\mathbf{x}_q)\|^2 = \mathcal{O}\left(\frac{LF_0}{EQ} + \min\left\{\left(\frac{LF_0 N\varsigma}{EQ}\right)^{\frac{2}{3}}, \left(\frac{LF_0 N\varsigma^2}{EQ}\right)^{\frac{1}{2}}\right\}\right).$$

Given that $\gamma$ can be chosen different in different propositions, $EQ$ ($E$ is chosen according to $\gamma$) can be different in different propositions. To ensure a fair comparison between algorithms, we use the total number of iterations $T$ instead of $NQ$ in the bounds:

$$\min_{t \in \{0, 1, \ldots T-1\}} \|\nabla f(\mathbf{y}_t)\|^2 \leq \min_{t \in \{0, E, \ldots, (\lfloor \frac{T}{E} \rfloor - 1)\}} \|\nabla f(\mathbf{y}_t)\|^2$$

$$= \min_{q \in \{0, 1, \ldots, Q-1\}} \|\nabla f(\mathbf{x}_q)\|^2$$

$$\leq \frac{1}{Q} \sum_{q=0}^{Q-1} \|\nabla f(\mathbf{x}_q)\|^2$$

$$= \mathcal{O}\left(\frac{LF_0}{T} + \min\left\{\left(\frac{LF_0 N\varsigma}{T}\right)^{\frac{2}{3}}, \left(\frac{LF_0 N\varsigma^2}{T}\right)^{\frac{1}{2}}\right\}\right),$$

where $\mathbf{y}_t$ denotes the model parameter vector in iteration $t$ and the last equality uses $\frac{1}{QE} = \frac{1}{\lfloor \frac{T}{E} \rfloor E} \leq \frac{T}{2}$ when $E \leq \frac{T}{2}$ [Wang and Ji, 2022, Appendix C, Proofs]. □

*Proof of Proposition 12.* For any $q \in \{0, 1 \ldots, Q-1\}$ and $e \in [E]$,

$$\left(\phi_q^e\right)^2 = \left\|\sum_{i=0}^{e-1} \left(\nabla f_{\pi_q(i)}(\mathbf{x}_q) - \nabla f(\mathbf{x}_q)\right)\right\|^2 \leq EC\varsigma^2. \tag{40}$$

where $C = 9 \log^2(\frac{8}{\delta})$ and the last inequality uses Lemma 8.

In this proposition, for Theorem 4, $p = 2$, $\nu = 0$, $D = NC\varsigma^2$, $c = 30$ and $\psi_q^E = EC\varsigma^2$. These give

$$\frac{1}{Q} \sum_{q=0}^{Q-1} \|\nabla f(\mathbf{x}_q)\|^2 = \tilde{\mathcal{O}}\left(\frac{F_0}{\gamma EQ} + \gamma^2 L^2 E\varsigma^2 + \frac{1}{E}\varsigma^2\right),$$

with the constraint $\gamma \leq \frac{1}{32LE}$. Since Equation (40) is used for each period (that is, for $Q$ times), so, by the union bound, the preceding bound holds with probability at least $1 - Q\delta$.

Then, choosing $E = \lfloor \frac{1}{32\gamma L} \rfloor$, we obtain

$$\frac{1}{Q} \sum_{q=0}^{Q-1} \|\nabla f(\mathbf{x}_q)\|^2 = \tilde{\mathcal{O}}\left( \frac{F_0}{\gamma EQ} + \gamma L\varsigma^2 \right).$$

Using $\gamma \lesssim \frac{1}{L}$ (due to $E \geq 1$) and applying Lemma 1, we obtain

$$\frac{1}{Q} \sum_{q=0}^{Q-1} \|\nabla f(\mathbf{x}_q)\|^2 = \tilde{\mathcal{O}}\left( \frac{LF_0}{EQ} + \left( \frac{LF_0\varsigma^2}{EQ} \right)^{\frac{1}{2}} \right).$$

To ensure a fair comparison between algorithms, we use the total number of iterations $T$ instead of $NQ$ in the bounds:

$$\min_{t \in \{0,1,\ldots T-1\}} \|\nabla f(\mathbf{y}_t)\|^2 = \tilde{\mathcal{O}}\left( \frac{LF_0}{T} + \left( \frac{LF_0\varsigma^2}{T} \right)^{\frac{1}{2}} \right),$$

where $\mathbf{y}_t$ denotes the model parameter vector in iteration $t$. $\qquad \square$

# K    Additional Extensions of FL

In this section, we extend the framework to cover the independent client participation pattern in Karimireddy et al. [2020], Yang et al. [2021] and Wang and Ji [2022] with the same way studied in Appendix J.

## K.1    Results

Similar to Appendix J, we redefine some notations, that is, to reformulate those defined over "epochs" in terms of "periods". For instance, the order error $\bar{\phi}_q$ in any *period* $q$ is defined as

$$\bar{\varphi}_q := \max_{e \in [E]} \left\{ \varphi_q^{v(e)} := \left\| \sum_{i=0}^{v(e)-1} \left( \nabla f_{\pi_q(i)}(\mathbf{x}_q) - \nabla f(\mathbf{x}_q) \right) \right\|_p \right\},$$

where $\pi_q$ denotes the order in period $q$ and $\mathbf{x}_q$ means the initial parameter vector in period $q$. See Definition 2 for comparison. Furthermore, we define

$$\psi_q^E := \max_{q \in \{0,1,\ldots,Q-1\}} \left\| \sum_{e=0}^{E-1} \nabla f_{\pi_q(e)}(\mathbf{x}_q^e) - \nabla f(\mathbf{x}_q) \right\|,$$

where $\mathbf{x}_q^e$ denotes the parameter vector after $k$ local updates in client $n$ in epoch $q$.

Theorem 5 provides the extended framework for FL, which covers the independent participation pattern (Propositions 13 and 14).

**Theorem 5.** *Let the global objective function $f$ be $L$-smooth and each local objective functions $f_n$ be $L_{2,p}$-smooth and $L_p$-smooth ($p \geq 2$). Suppose that Assumption 3 holds. Suppose $E \mod S = 0$. Let $\nu \geq 0$ be a numerical constant. Suppose that there exist $\tilde{B}$ and $\tilde{D}$ such that for $0 \leq q \leq \nu - 1$,*

$$(\bar{\varphi}_q)^2 \leq \tilde{B} \|\nabla f(\mathbf{x}_q)\|^2 + \tilde{D},$$

*and there exist $\{A_i\}$, $\{B_i\}$ and $D$ such that for $q \geq \nu$,*

$$(\bar{\varphi}_q)^2 \leq \sum_{i=1}^{\nu} A_i(\bar{\varphi}_{q-i})^2 + \sum_{i=0}^{\nu} B_i \|\nabla f(\mathbf{x}_{q-i})\|^2 + D.$$

*If $\gamma \leq \min \left\{ \frac{1}{\eta LKE\frac{1}{S}}, \frac{1}{32L_{2,p}KE\frac{1}{S}}, \frac{\sqrt{1-\sum_{i=1}^{\nu} A_i}}{4L_{2,p}K\frac{1}{S}\sqrt{\sum_{i=0}^{\nu} B_i}}, \frac{\sqrt{1-\sum_{i=1}^{\nu} A_i}}{4L_{2,p}K\frac{1}{S}\sqrt{\tilde{B}}}, \frac{1}{32L_pKE\frac{1}{S}} \right\}$, then*

$$\frac{1}{Q} \sum_{q=0}^{Q-1} \|\nabla f(\mathbf{x}_q)\|^2 \leq \frac{5F_0}{\gamma\eta KE\frac{1}{S}Q} + c\gamma^2 L_{2,p}^2 K^2 \frac{1}{S^2} \frac{1}{Q} \nu\tilde{D} + 20\gamma^2 L_{2,p}^2 K^2\varsigma^2$$

$$+ c\gamma^2 L_{2,p}^2 K^2 \frac{1}{S^2} D + 5\frac{1}{E^2} \left( \psi_q^E \right)^2,$$

*where $c = \frac{10}{(1-\sum_{i=1}^{\nu} A_i)}$ is a numerical constant.*

When $E$ is a multiple of $N$ ($\psi_q^E = 0$), and Theorem 5 is reduced to Theorem 2, covering the examples in FL with regularized participation. When $E = \Theta(\frac{1}{\gamma L})$ ($\psi_q^E$ is allowed to be nonzero), we can obtain the convergence rates of FL with independent participation. Two strategies appeared in previous works are considered (see Propositions 13 and 14). The comparison of convergence rates with the previous works [Yang et al., 2021; Wang and Ji, 2022] are provided in Table 9. Note that the assumption that the clients are selected according to Equation (46) is also used in Wang and Ji [2022, Proposition 4.7]. See the statement in Wang and Ji [2022]: "In the following, we assume that $\mu_{t_0}$ is chosen such that $\mu_{t_0} = \mathbb{E}[q_t^n]$ for all $n \in \{1, \ldots, N\}$ and $t \in \{t_0, t_0 + 1, \ldots, t_0 + P - 1\}$".

**Proposition 13** (Strategy 1). *Let the clients be selected uniformly at random with replacement in each round, independently across rounds. Suppose that Assumption 4 holds. Then, we obtain that, for $q \geq 0$, with probability at least $1 - \delta$,*

$$(\bar{\varphi}_q)^2 \leq EC\varsigma^2, \text{ and } \psi_q^E \leq EC\varsigma^2,$$

*where $C = 9\log^2(\frac{8}{\delta})$. Applying Theorem 5, choosing $E = \Theta(\lfloor \frac{1}{\gamma KL} \rfloor S)$ and tuning the step size, for $\frac{E}{S} \leq \frac{T}{2}$, we obtain that, with probability at least $1 - T\delta$,*

$$\min_{t \in \{0,1,\ldots T-1\}} \|\nabla f(\mathbf{y}_t)\|^2 = \tilde{\mathcal{O}}\left( \frac{LF_0}{T} + \left(\frac{LF_0\varsigma}{T}\right)^{\frac{2}{3}} + \left(\frac{LF_0\varsigma^2}{ST}\right)^{\frac{1}{2}} \right).$$

*where $\mathbf{y}_t$ denotes the model parameter vector in round $t$.*

**Proposition 14** (Strategy 2). *Let the clients be selected in a manner that satisfies Equation (46) in each round, independently across rounds. Suppose that Assumption 4 holds. Then, we obtain that, for $q \geq 0$, with probability at least $1 - \delta$,*

$$(\bar{\varphi}_q)^2 \leq ESC\varsigma^2, \text{ and } \psi_q^E \leq ESC\varsigma^2,$$

*where $C = 9\log^2(\frac{8}{\delta})$. Applying Theorem 5, choosing $E = \Theta(\lfloor \frac{1}{\gamma KL} \rfloor S)$ and tuning the step size, for $\frac{E}{S} \leq \frac{T}{2}$, we obtain that, with probability at least $1 - T\delta$,*

$$\min_{t \in \{0,1,\ldots T-1\}} \|\nabla f(\mathbf{y}_t)\|^2 = \tilde{\mathcal{O}}\left( \frac{LF_0}{T} + \left(\frac{LF_0\varsigma}{T}\right)^{\frac{2}{3}} + \left(\frac{LF_0\varsigma^2}{T}\right)^{\frac{1}{2}} \right).$$

*where $\mathbf{y}_t$ denotes the model parameter vector in round $t$.*

Table 9: Comparison of convergence rates (of FL with independent participation) with previous works. Numerical constants and polylogarithmic factors are hided. We set $\eta = 1$ for comparison.

| **Algorithm** | Previous works | This work |
|---|---|---|
| Strategy 1 | $\frac{LF_0}{T} + \left(\frac{LF_0\varsigma}{T}\right)^{\frac{2}{3}} + \left(\frac{LF_0\varsigma^2}{ST}\right)^{\frac{1}{2}}$ [1] | $\frac{LF_0}{T} + \left(\frac{LF_0\varsigma}{T}\right)^{\frac{2}{3}} + \left(\frac{LF_0\varsigma^2}{ST}\right)^{\frac{1}{2}}$ [2] |
| Strategy 2 | $\frac{LF_0}{T} + \left(\frac{LF_0\varsigma}{T}\right)^{\frac{2}{3}} + \left(\frac{LF_0N^2\varsigma^2}{ST}\right)^{\frac{1}{2}} + \varsigma$ [2,3] | $\frac{LF_0}{T} + \left(\frac{LF_0\varsigma}{T}\right)^{\frac{2}{3}} + \left(\frac{LF_0\varsigma^2}{T}\right)^{\frac{1}{2}}$ [2] |

[1] This bound is from Yang et al. [2021, Theorem 2]. This bound is for $\frac{1}{T}\sum_{t=0}^{T-1} \mathbb{E}\|\nabla f(\mathbf{y}_t)\|^2$ and do not require the additional constraint $\frac{E}{S} \leq \frac{T}{2}$. Therefore, if these factors are taken into consideration, the bounds of Yang et al. [2021] are better than ours.

[2] These bounds is for $\min_{t \in \{0,1,\ldots,T-1\}} \|\nabla f(\mathbf{y}_t)\|^2$ and requires the additional constraint $\frac{E}{S} \leq \frac{T}{2}$.

[3] This bound is from Wang and Ji [2022].

## K.2 Proofs

This section provides the proofs of Theorem 5 and the helper lemmas for Theorem 5.

To avoid ambiguity, we define

$$\tilde{\mathbf{x}}_{q+1} := \mathbf{x}_{q,K}^{E-1} = \mathbf{x}_{q,0}^E.$$

Due to the amplified updates [Wang and Ji, 2022], we have

$$\tilde{\mathbf{x}}_{q+1} - \mathbf{x}_q = -\gamma \, \frac{1}{S} \sum_{e=0}^{E-1} \sum_{k=0}^{K-1} \nabla f_{\pi_q(e)}\left(\mathbf{x}_{q,k}^e\right),$$

$$\mathbf{x}_{q+1} - \mathbf{x}_q = -\gamma \boldsymbol{\eta} \frac{1}{S} \sum_{e=0}^{E-1} \sum_{k=0}^{K-1} \nabla f_{\pi_q(e)}\left(\mathbf{x}_{q,k}^e\right).$$

We define the maximum parameter deviation (drift) of FL in any period $q$, $\Delta_q$ as

$$\Delta_q = \max\left\{ \max_{\substack{e\in\{0,\dots,E-1\}\\ k\in\{0,\dots,K-1\}}} \left\|\mathbf{x}_{q,k}^e - \mathbf{x}_q\right\|_p, \left\|\tilde{\mathbf{x}}_{q+1} - \mathbf{x}_q\right\|_p \right\}.$$

Then, we obtain the relation

$$\left\|\mathbf{x}_{q+1} - \mathbf{x}_q\right\|_p = \eta\left\|\tilde{\mathbf{x}}_{q+1} - \mathbf{x}_q\right\|_p \le \eta\Delta_q.$$

**Lemma 10.** *If $\gamma L_p K N \frac{1}{S} \le \frac{1}{32}$ and $E$ is a multiple of $S$, the maximum parameter drift in FL is bounded:*

$$\Delta_q \le \frac{32}{31}\gamma K \frac{1}{S}\bar{\varphi}_q + \frac{32}{31}\gamma K E \frac{1}{S}\left\|\nabla f(\mathbf{x}_q)\right\| + \frac{32}{31}\gamma K \varsigma,$$

$$(\Delta_q)^2 \le 4\gamma^2 K^2 \frac{1}{S^2}\left(\bar{\varphi}_q\right)^2 + 4\gamma^2 K^2 E^2 \frac{1}{S^2}\left\|\nabla f(\mathbf{x}_q)\right\|^2 + 4\gamma^2 K^2 \varsigma^2.$$

*Proof.* The proof is identical to that of Lemma 10, except that the epoch-based notations are replaced by the period-based notations. $\qquad\square$

*Proof of Theorem 5.* For FL with regularized participation (Algorithm 2), the cumulative updates over any period $q$ are

$$\mathbf{x}_{q+1} - \mathbf{x}_q = -\gamma\eta\frac{1}{S} \sum_{n=0}^{N-1} \sum_{k=0}^{K-1} \nabla f_{\pi_q(n)}\left(\mathbf{x}_{q,k}^n\right). \tag{41}$$

Since the global objective function $f$ is $L$-smooth, it follows that

$$f(\mathbf{x}_{q+1}) - f(\mathbf{x}_q) \le \langle \nabla f(\mathbf{x}_q), \mathbf{x}_{q+1} - \mathbf{x}_q \rangle + \frac{1}{2}L\left\|\mathbf{x}_{q+1} - \mathbf{x}_q\right\|^2. \tag{42}$$

Using Equation (41), we obtain

$$\langle \nabla f(\mathbf{x}_q), \mathbf{x}_{q+1} - \mathbf{x}_q \rangle$$

$$= -\gamma\eta\frac{1}{S}KE\left[\left\langle \nabla f(\mathbf{x}_q), \frac{1}{E}\sum_{e=0}^{E-1}\frac{1}{K}\sum_{k=0}^{K-1}\nabla f_{\pi_q(e)}(\mathbf{x}_{q,k}^e)\right\rangle\right]$$

$$= -\frac{1}{2}\gamma\eta\frac{1}{S}KE\left\|\nabla f(\mathbf{x}_q)\right\|^2 - \frac{1}{2}\gamma\eta\frac{1}{S}KE\left\|\frac{1}{E}\sum_{e=0}^{E-1}\frac{1}{K}\sum_{k=0}^{K-1}\nabla f_{\pi_q(e)}(\mathbf{x}_{q,k}^e)\right\|^2$$

$$+ \frac{1}{2}\gamma\eta\frac{1}{S}KE\left\|\frac{1}{E}\sum_{e=0}^{E-1}\frac{1}{K}\sum_{k=0}^{K-1}\nabla f_{\pi_q(e)}(\mathbf{x}_{q,k}^e) - \nabla f(\mathbf{x}_q)\right\|^2,$$

where the second equality uses $2\langle \mathbf{x}, \mathbf{y}\rangle = \|\mathbf{x}\|^2 + \|\mathbf{y}\|^2 - \|\mathbf{x} - \mathbf{y}\|^2$. Using Equation (41), we obtain

$$\frac{1}{2}L\left\|\mathbf{x}_{q+1} - \mathbf{x}_q\right\|^2 = \frac{1}{2}L\left\|\gamma\eta\frac{1}{S}\sum_{e=0}^{E-1}\sum_{k=0}^{K-1}\nabla f_{\pi_q(e)}(\mathbf{x}_{q,k}^e)\right\|^2$$

$$= \frac{1}{2}\gamma^2\eta^2 L\frac{1}{S^2}K^2E^2\left\|\frac{1}{E}\sum_{n=0}^{E-1}\frac{1}{K}\sum_{k=0}^{K-1}\nabla f_{\pi_q(e)}(\mathbf{x}_{q,k}^e)\right\|^2.$$

Plugging the preceding two equations back into Equation (42), and using $\gamma\eta LKN\frac{1}{S} \le 1$, we obtain

$$f(\mathbf{x}_{q+1}) - f(\mathbf{x}_q)$$
$$\le -\frac{1}{2}\gamma\eta\frac{1}{S}KE\|\nabla f(\mathbf{x}_q)\|^2 + \frac{1}{2}\gamma\eta\frac{1}{S}KE\left\|\frac{1}{E}\sum_{e=0}^{E-1}\frac{1}{K}\sum_{k=0}^{K-1}\nabla f_{\pi_q(e)}(\mathbf{x}_{q,k}^e) - \nabla f(\mathbf{x}_q)\right\|^2. \quad (43)$$

Then,

$$\text{T}_2 \text{ in } (43) = \frac{1}{2}\gamma\eta\frac{1}{S}KE\left\|\frac{1}{E}\sum_{e=0}^{E-1}\frac{1}{K}\sum_{k=0}^{K-1}\nabla f_{\pi_q(e)}(\mathbf{x}_{q,k}^e) - \nabla f(\mathbf{x}_q)\right\|^2$$

$$\le \gamma\eta\frac{1}{S}KE\left\|\frac{1}{E}\sum_{e=0}^{E-1}\frac{1}{K}\sum_{k=0}^{K-1}\nabla f_{\pi_q(e)}(\mathbf{x}_{q,k}^e) - \frac{1}{E}\sum_{e=0}^{E-1}\frac{1}{K}\sum_{k=0}^{K-1}\nabla f_{\pi_q(e)}(\mathbf{x}_q)\right\|^2$$

$$+ \gamma\eta\frac{1}{S}KE\left\|\frac{1}{E}\sum_{e=0}^{E-1}\frac{1}{K}\sum_{k=0}^{K-1}\nabla f_{\pi_q(e)}(\mathbf{x}_q) - \nabla f(\mathbf{x}_q)\right\|^2$$

$$\le \gamma\eta L_{2,p}^2\frac{1}{S}\sum_{e=0}^{E-1}\sum_{k=0}^{K-1}\left\|\mathbf{x}_{q,k}^e - \mathbf{x}_q\right\|_p^2$$

$$+ \gamma\eta\frac{1}{S}KE\left\|\frac{1}{E}\sum_{e=0}^{E-1}\frac{1}{K}\sum_{k=0}^{K-1}\nabla f_{\pi_q(e)}(\mathbf{x}_q) - \nabla f(\mathbf{x}_q)\right\|^2$$

$$\le \gamma\eta L_{2,p}^2 KE\frac{1}{S}\left(\Delta_q\right)^2 + \gamma\eta K\frac{1}{ES}\left(\psi_q^E\right)^2,$$

where the second inequality is because $f_{\pi_q(e)}$ is $L_{2,p}$ smooth and the last inequality uses the definitions of $\Delta_q$ and $\psi_q^E$. Plugging the preceding inequality back into Equation (43), we obtain

$$f(\mathbf{x}_{q+1}) - f(\mathbf{x}_q) \le -\frac{1}{2}\gamma\eta\frac{1}{S}KE\|\nabla f(\mathbf{x}_q)\|^2 + \gamma\eta L_{2,p}^2 KE\frac{1}{S}\left(\Delta_q\right)^2 + \gamma\eta K\frac{1}{ES}\left(\psi_q^E\right)^2.$$

Using Lemma 7 and $\gamma L_{2,p}KN\frac{1}{S} \le \frac{1}{32}$, we obtain

$$f(\mathbf{x}_{q+1}) - f(\mathbf{x}_q) \le -\frac{127}{256}\gamma\eta KE\frac{1}{S}\|\nabla f(\mathbf{x}_q)\|^2 + 4\gamma^3\eta L_{2,p}^2 K^3 E\frac{1}{S^3}\left(\bar\varphi_q\right)^2$$

$$+ 4\gamma^3\eta L_{2,p}^2 K^3 E\frac{1}{S}\varsigma^2 + \gamma\eta K\frac{1}{ES}\left(\psi_q^E\right)^2.$$

Note that the preceding inequality is almost identical to Equation (25), except an additional term $\gamma\eta K\frac{1}{ES}\left(\psi_q^E\right)^2$. The term $\left(\psi_q^E\right)^2$ will not be dealt with further in this theorem, so the remaining steps of this proof are almost the same as those in Theorem 2. For these reasons, we omit the remaining steps. $\qquad\square$

*Proof of Proposition 13.* For any $q \in \{0, 1 \ldots, Q-1\}$ and $e \in [E]$,

$$\left(\varphi_q^{v(e)}\right)^2 = \left\|\sum_{i=0}^{v(e)-1}\left(\nabla f_{\pi_q(i)}(\mathbf{x}_q) - \nabla f(\mathbf{x}_q)\right)\right\|^2 \le EC\varsigma^2 \quad (44)$$

where $C = 9\ln^2(\frac{8}{\delta})$ and the first inequality uses Lemma 8.

In this proposition, for Theorem 5, $p = 2$, $\nu = 0$, $D = EC\varsigma^2$, $c = 10$ and $\psi_q^E = EC\varsigma^2$. These give

$$\frac{1}{Q}\sum_{q=0}^{Q-1}\|\nabla f(\mathbf{x}_q)\|^2 = \tilde{\mathcal{O}}\left(\frac{F_0}{\gamma\eta KE\frac{1}{S}Q} + \gamma^2 L^2 K^2\varsigma^2 + \gamma^2 L^2 K^2 E\frac{1}{S^2}\varsigma^2 + \frac{1}{E}\varsigma^2\right),$$

with the constraint $\gamma \le \frac{1}{32\eta LKE\frac{1}{S}}$. Equation (44) is used for each period (that is, for $Q$ times), so, by the union bound, the preceding bound holds with probability at least $1 - Q\delta$.

Then, choosing $E = \lfloor \frac{1}{32\gamma LK} \rfloor S$, we obtain

$$\frac{1}{Q} \sum_{q=0}^{Q-1} \|\nabla f(\mathbf{x}_q)\|^2 = \tilde{\mathcal{O}} \left( \frac{F_0}{\gamma \eta K E \frac{1}{S} Q} + \gamma^2 L^2 K^2 \varsigma^2 + \gamma L K \frac{1}{S} \varsigma^2 \right).$$

Using $\gamma \lesssim \frac{1}{LK}$ (due to $E = \lfloor \frac{1}{32\gamma LK} \rfloor S \geq S$), and applying Lemmas 1 and 2, we obtain

$$\frac{1}{Q} \sum_{q=0}^{Q-1} \|\nabla f(\mathbf{x}_q)\|^2 = \tilde{\mathcal{O}} \left( \frac{LF_0}{\eta E \frac{1}{S} Q} + \left( \frac{LF_0\varsigma}{\eta E \frac{1}{S} Q} \right)^{\frac{2}{3}} + \left( \frac{LF_0\varsigma^2}{\eta E Q} \right)^{\frac{1}{2}} \right).$$

Given that $\gamma$ can be chosen different in different propositions, $EQ$ ($E$ is chosen according to $\gamma$) can be different in different propositions. To ensure a fair comparison between algorithms, we use the total number of iterations $T$ instead of $NQ$ in the bounds:

$$\min_{t \in \{0,1,\ldots T-1\}} \|\nabla f(\mathbf{y}_t)\|^2 \leq \min_{t \in \{0, \frac{E}{S}, \ldots, (\lfloor \frac{TS}{E} \rfloor - 1)\}} \|\nabla f(\mathbf{y}_t)\|^2$$

$$= \min_{q \in \{0,1,\ldots,Q-1\}} \|\nabla f(\mathbf{x}_q)\|^2$$

$$\leq \frac{1}{Q} \sum_{q=0}^{Q-1} \|\nabla f(\mathbf{x}_q)\|^2$$

$$= \tilde{\mathcal{O}} \left( \frac{LF_0}{\eta T} + \left( \frac{LF_0\varsigma}{\eta T} \right)^{\frac{2}{3}} + \left( \frac{LF_0\varsigma^2}{\eta ST} \right)^{\frac{1}{2}} \right),$$

where $\mathbf{y}_t$ denotes the model parameter vector in round $t$ and the last equality uses $\frac{1}{Q\frac{E}{S}} = \frac{1}{\lfloor \frac{TS}{E} \rfloor \frac{E}{S}} \leq \frac{2}{T}$ when $\frac{E}{S} \leq \frac{T}{2}$ [Wang and Ji, 2022, Appendix C, Proofs]. We set $\eta = 1$ for comparison:

$$\min_{t \in \{0,1,\ldots T-1\}} \|\nabla f(\mathbf{y}_t)\|^2 = \tilde{\mathcal{O}} \left( \frac{LF_0}{T} + \left( \frac{LF_0\varsigma}{T} \right)^{\frac{2}{3}} + \left( \frac{LF_0\varsigma^2}{ST} \right)^{\frac{1}{2}} \right).$$

$\square$

*Proof of Proposition 14.* For any $q \in \{0,1\ldots,Q-1\}$ and $e \in [E]$,

$$\left( \varphi_q^{v(e)} \right)^2 = \left\| \sum_{i=0}^{v(e)-1} \left( \nabla f_{\pi_q(i)}(\mathbf{x}_q) - \nabla f(\mathbf{x}_q) \right) \right\|^2$$

$$= \left\| \sum_{i=0}^{v(e)-1} \left( \frac{1}{S} \sum_{s=0}^{S-1} \nabla f_{\pi_q(\lfloor \frac{i}{S} \rfloor S + s)}(\mathbf{x}_q) - \nabla f(\mathbf{x}_q) \right) \right\|^2$$

$$\leq ESC \max_{i \in \{0,1\ldots,E-1\}} \left\| \frac{1}{S} \sum_{s=0}^{S-1} \nabla f_{\pi_q(\lfloor \frac{i}{S} \rfloor S + s)}(\mathbf{x}_q) - \nabla f(\mathbf{x}_q) \right\|^2$$

$$\leq ESC \max_{i \in \{0,1\ldots,E-1\}} \frac{1}{S} \sum_{s=0}^{S-1} \left\| \nabla f_{\pi_q(\lfloor \frac{i}{S} \rfloor S + s)}(\mathbf{x}_q) - \nabla f(\mathbf{x}_q) \right\|^2$$

$$\leq ESC\varsigma^2 \tag{45}$$

where $C = 9 \ln^2(\frac{8}{\delta})$ and the first inequality uses Lemma 8 and the condition

$$\mathbb{E} \left[ \frac{1}{S} \sum_{s=0}^{S-1} \nabla f_{\pi_q(\lfloor \frac{i}{S} \rfloor S + s)}(\mathbf{x}_q) \right] = \nabla f(\mathbf{x}_q) \tag{46}$$

for any $i \in \{0,1\ldots,E-1\}$ and $q \in \{0,1,\ldots,Q-1\}$.

In this proposition, for Theorem 5, $p = 2$, $\nu = 0$, $D = ESC\varsigma^2$, $c = 10$ and $\psi_q^E = ESC\varsigma^2$. These lead to

$$\frac{1}{Q}\sum_{q=0}^{Q-1}\|\nabla f(\mathbf{x}_q)\|^2 = \tilde{\mathcal{O}}\left(\frac{F_0}{\gamma\eta KE\frac{1}{S}Q} + \gamma^2 L^2 K^2\varsigma^2 + \gamma^2 L^2 K^2 E\frac{1}{S}\varsigma^2 + \frac{1}{E}S\varsigma^2\right),$$

with the constraint $\gamma \leq \frac{1}{32\eta LKE\frac{1}{S}}$. Equation (44) is used for each period (that is, for $Q$ times), so, by the union bound, we can obtain that the preceding bound holds with probability at least $1 - Q\delta$. Then, choosing $E = \lfloor\frac{1}{32\gamma KL}\rfloor S$, we obtain

$$\frac{1}{Q}\sum_{q=0}^{Q-1}\|\nabla f(\mathbf{x}_q)\|^2 = \tilde{\mathcal{O}}\left(\frac{F_0}{\gamma\eta KE\frac{1}{S}Q} + \gamma^2 L^2 K^2\varsigma^2 + \gamma LK\varsigma^2\right).$$

Using $\gamma \lesssim \frac{1}{LK}$ (due to $E = \lfloor\frac{1}{32\gamma LK}\rfloor S \geq S$), and applying Lemmas 1 and 2, we obtain

$$\frac{1}{Q}\sum_{q=0}^{Q-1}\|\nabla f(\mathbf{x}_q)\|^2 = \tilde{\mathcal{O}}\left(\frac{LF_0}{\eta E\frac{1}{S}Q} + \left(\frac{LF_0\varsigma}{\eta E\frac{1}{S}Q}\right)^{\frac{2}{3}} + \left(\frac{LF_0\varsigma^2}{\eta E\frac{1}{S}Q}\right)^{\frac{1}{2}}\right).$$

To ensure a fair comparison between algorithms, we use the total number of iterations $T$ instead of $NQ$ in the bounds. Following the same steps as Proposition 13, we obtain

$$\min_{t\in\{0,1,\ldots T-1\}}\|\nabla f(\mathbf{y}_t)\|^2 = \tilde{\mathcal{O}}\left(\frac{LF_0}{\eta T} + \left(\frac{LF_0\varsigma}{\eta T}\right)^{\frac{2}{3}} + \left(\frac{LF_0\varsigma^2}{\eta T}\right)^{\frac{1}{2}}\right),$$

where $\mathbf{y}_t$ denotes the model parameter vector in round $t$. We set $\eta = 1$ for comparison:

$$\min_{t\in\{0,1,\ldots T-1\}}\|\nabla f(\mathbf{y}_t)\|^2 = \tilde{\mathcal{O}}\left(\frac{LF_0}{T} + \left(\frac{LF_0\varsigma}{T}\right)^{\frac{2}{3}} + \left(\frac{LF_0\varsigma^2}{T}\right)^{\frac{1}{2}}\right).$$

$\square$

## L  Permutation-based SGD and Online Learning

In modern large-scale language model training, it may be impossible to actually shuffle all the data examples in a very large dataset. In this section, we analyze a hybrid algorithm that combines permutation-based SGD with online learning [Orabona, 2019; Hazan et al., 2016].

**The hybrid algorithm, notations and assumptions.** The data examples are sampled from a very large dataset in a online fashion. Each time $K$ data examples are sampled (which can be stored in a small memory), these data examples are permuted and then fed into the model. For simplicity, we use projected online gradient descent (see Orabona [2019, Algorithm 2.1]) to train this model with the shuffled examples. The details are in Algorithm 13, which combines Algorithm 1 and Orabona [2019, Algorithm 2.1]. Intuitively, Algorithm 13 is a hierarchical algorithm where the top-level part is projected online gradient descent and the bottom-level part is permutation-based SGD.

In this section, we need to redefine some notations. We call each group of $K$ examples a block, indexed by $q$. We use $f_{\pi_q(k)}$ to denote the loss function of the $k$-th example in block $q$, where $\pi_q$ is the permutation in block $q$. We use $F_q = \frac{1}{K}\sum_{k=0}^{K-1}f_k^q$ to denote the loss function of the block $q$. We use $\mathbf{x}$ to denote the iterates: $\mathbf{x}_q^k$ denotes the iterate in example $k$ and block $q$ and $\mathbf{x}_q$ denotes the initial iterate in block $q$.

---

**Algorithm 13:** An algorithm that combines permutation-based SGD with online learning

**Input:** $\mathcal{V}$, $\pi_0$, $\mathbf{x}_0 \in \mathcal{V}$
**Output:** $\{\mathbf{x}_q\}$
1 **for** $q = 0, 1, \ldots, Q-1$ **do**
2 $\quad$ Sample $K$ examples in an online fashion
3 $\quad$ **for** $k = 0, 1, \ldots, K-1$ **do**
4 $\quad\quad$ $\mathbf{x}_q^{k+1} \leftarrow \mathbf{x}_q^k - \gamma\nabla f_{\pi_q(k)}^q(\mathbf{x}_q^k)$
5 $\quad$ $\mathbf{x}_{q+1} \leftarrow \Pi_{\mathcal{V}}(\mathbf{x}_q^K) = \arg\min_{\mathbf{y}\in\mathcal{V}}\|\mathbf{x}_q^K - \mathbf{y}\|^2$
6 $\quad$ $\pi_{q+1} \leftarrow \texttt{Permute}(\cdots)$

---

The assumptions are as follows:

**Assumption 5.** *All iterations* $\mathbf{x} \in \mathcal{V}$, *where* $\mathcal{V}$ *is a closed convex set with diameter* $\mathcal{D}$.

**Assumption 6.** *There exists a constant* $\sigma$ *such that for any* $q \in \{0, 1, \ldots, Q - 1\}$,

$$\|\nabla F_q(\mathbf{x})\|^2 \le \sigma^2, \ \forall \mathbf{x} \in \mathbb{R}^d,$$

*where* $F_q(\mathbf{x}) = \frac{1}{K} \sum_{k=0}^{K-1} f_k^q(\mathbf{x})$.

Assumptions 5 and 6 are used in Orabona [2019, Theorem 2.13].

**Definition 4.** *The order error in any block* $q$ *is defined as*

$$\bar{\phi}_q = \max_{k \in [K]} \left\| \sum_{i=0}^{k-1} (\nabla f_{\pi_q(k)}^q(\mathbf{x}_q) - \nabla F_q(\mathbf{x}_q)) \right\|.$$

**Assumption 7.** *There exists a constant* $\varsigma$ *such that for any* $q \in \{0, 1, \ldots, Q - 1\}$ *and* $k \in \{0, 1, \ldots, K - 1\}$,

$$\|\nabla f_k^q(\mathbf{x}) - \nabla F_q(\mathbf{x})\|^2 \le \varsigma^2, \ \forall \mathbf{x} \in \mathbb{R}^d.$$

**Regret analysis.** The average regret bound of Algorithm 13 is given in Theorem 6. Suppose that Assumption 7 holds, we can obtain the bounds of $\bar{\phi}_q$ for AP and IP (as in Appendix F), by which we can further obtain the average regret bounds for AP and IP. These bounds imply that a nice permutation (the value of $\bar{\phi}_q$ is small) will lead to a nice bound. Thus, shuffling a subset of the dataset can also help improve the performance. Notably, in online learning, the examples appear only once, which means that the information gained from the previous blocks may not apply to the subsequent blocks. As a result, the original DP algorithms cannot be applied for online learning. Given the complexity, we leave DP to future works.

**Theorem 6.** *Suppose that Assumptions 5 and 6 hold. The average regret bound of Algorithm 13 is*

$$\frac{1}{Q} \sum_{q=0}^{Q-1} (\nabla F_q(\mathbf{x}_q) - \nabla F_q(\mathbf{u})) \le \frac{\mathcal{D}^2}{2\gamma K Q} + 2\gamma^2 L \left(\bar{\phi}_q\right)^2 + 2\gamma K \sigma^2,$$

*where* $\mathbf{u} \in \mathcal{V}$ *is an arbitrary competitor.*

*Proof.* This analysis is based on Orabona [2019, Theorem 2.13].

$$\begin{aligned}
\|\mathbf{x}_{q+1} - \mathbf{u}\|^2 - \|\mathbf{x}_q - \mathbf{u}\|^2 &= \left\|\Pi_{\mathcal{V}}(\mathbf{x}_q^K) - \mathbf{u}\right\|^2 - \|\mathbf{x}_q - \mathbf{u}\|^2 \\
&\le \left\|\mathbf{x}_q^K - \mathbf{u}\right\|^2 - \|\mathbf{x}_q - \mathbf{u}\|^2 \\
&\le \left\|\mathbf{x}_q - \gamma \sum_{k=0}^{K-1} \nabla f_{\pi_q(k)}^q(\mathbf{x}_q^k) - \mathbf{u}\right\|^2 - \|\mathbf{x}_q - \mathbf{u}\|^2 \\
&\le -2\gamma \sum_{k=0}^{K-1} \langle \mathbf{x}_q - \mathbf{u}, \nabla f_{\pi_q(k)}^q(\mathbf{x}_q^k) \rangle + \gamma^2 \|\sum_{k=0}^{K-1} \nabla f_{\pi_q(k)}^q(\mathbf{x}_q^k)\|^2 \quad (47)
\end{aligned}$$

where $\mathbf{u} \in \mathcal{V}$ is an arbitrary competitor, and the first inequality uses Orabona [2019, Proposition 2.11]. Then, using Karimireddy et al. [2020, Lemma 5], we obtain

$$\begin{aligned}
\text{T}_1 \text{ in } (47) &= -2\gamma \sum_{k=0}^{K-1} \langle \mathbf{x}_q - \mathbf{u}, \nabla f_{\pi_q(k)}^q(\mathbf{x}_q^k) \rangle \\
&\le -2\gamma \sum_{k=0}^{K-1} \left( f_{\pi_q(k)}^q(\mathbf{x}_q) - f_{\pi_q(k)}^q(\mathbf{u}) - L \left\|\mathbf{x}_q^k - \mathbf{x}_q\right\|^2 \right) \\
&= -2\gamma K \left(F_q(\mathbf{x}_q) - F_q(\mathbf{u})\right) + 2\gamma L \sum_{k=0}^{K-1} \left\|\mathbf{x}_q^k - \mathbf{x}_q\right\|^2,
\end{aligned}$$

and

$$\text{T}_2 \text{ in (47)} = \gamma^2 \left\| \sum_{k=0}^{K-1} \nabla f^q_{\pi_q(k)}(\mathbf{x}^k_q) \right\|^2$$

$$\leq 2\gamma^2 \left\| \sum_{k=0}^{K-1} \nabla f^q_{\pi_q(k)}(\mathbf{x}^k_q) - \sum_{k=0}^{K-1} \nabla f^q_{\pi_q(k)}(\mathbf{x}_q) \right\|^2 + 2\gamma^2 \left\| \sum_{k=0}^{K-1} \nabla f^q_{\pi_q(k)}(\mathbf{x}_q) \right\|^2$$

$$= 2\gamma^2 L^2 K \sum_{k=0}^{K-1} \left\| \mathbf{x}^k_q - \mathbf{x}_q \right\|^2 + 2\gamma^2 K^2 \nabla F_q(\mathbf{x}_q)$$

$$\leq 2\gamma^2 L^2 K \sum_{k=0}^{K-1} \left\| \mathbf{x}^k_q - \mathbf{x}_q \right\|^2 + 2\gamma^2 K^2 \sigma^2.$$

Plugging the preceding two bounds into Equation (47), we obtain

$$\|\mathbf{x}_{q+1} - \mathbf{u}\|^2 - \|\mathbf{x}_q - \mathbf{u}\|^2$$
$$\leq -2\gamma K \left( F_q(\mathbf{x}_q) - F_q(\mathbf{u}) \right) + 2\gamma L(1 + \gamma LK) \sum_{k=0}^{K-1} \left\| \mathbf{x}^k_q - \mathbf{x}_q \right\|^2 + 2\gamma^2 K^2 \sigma^2.$$

Note that Lemma 6 applies to the local updates (Lines 3–4) in Algorithm 13, and then we obtain

$$\|\mathbf{x}_{q+1} - \mathbf{u}\|^2 - \|\mathbf{x}_q - \mathbf{u}\|^2$$
$$\leq -2\gamma K \left( F_q(\mathbf{x}_q) - F_q(\mathbf{u}) \right) + 2\gamma LK(1 + \gamma LK)\Delta^2_q + 2\gamma^2 K^2 \sigma^2$$
$$\leq -2\gamma K \left( F_q(\mathbf{x}_q) - F_q(\mathbf{u}) \right)$$
$$\quad + 2\gamma LK(1 + \gamma LK)\left( 3\gamma^2 \left( \bar{\phi}_q \right)^2 + 3\gamma^2 K^2 \|\nabla F_q(\mathbf{x}_q)\|^2 \right) + 2\gamma^2 K^2 \sigma^2$$
$$\leq -2\gamma K \left( F_q(\mathbf{x}_q) - F_q(\mathbf{u}) \right) + 4\gamma^3 LK \left( \bar{\phi}_q \right)^2 + 4\gamma^3 LK^3 \|\nabla F_q(\mathbf{x}_q)\|^2 + 2\gamma^2 K^2 \sigma^2$$
$$\leq -2\gamma K \left( F_q(\mathbf{x}_q) - F_q(\mathbf{u}) \right) + 4\gamma^3 LK \left( \bar{\phi}_q \right)^2 + 4\gamma^3 LK^3 \sigma^2 + 2\gamma^2 K^2 \sigma^2$$
$$\leq -2\gamma K \left( F_q(\mathbf{x}_q) - F_q(\mathbf{u}) \right) + 4\gamma^3 LK \left( \bar{\phi}_q \right)^2 + 3\gamma^2 K^2 \sigma^2,$$

where the first inequality uses the definition $\Delta_q := \max_{k \in [K]} \left\| \mathbf{x}^k_q - \mathbf{x}_q \right\|^2$, the second inequality uses Lemma 6, the third inequality uses $\gamma LK \leq \frac{1}{32}$, the forth inequality uses Assumption 6 and the fifth inequality uses $\gamma LK \leq \frac{1}{32}$. Then,

$$F_q(\mathbf{x}_q) - F_q(\mathbf{u}) \leq \frac{\|\mathbf{x}_q - \mathbf{u}\|^2}{2\gamma K} - \frac{\|\mathbf{x}_{q+1} - \mathbf{u}\|^2}{2\gamma K} + 2\gamma^2 L \left( \bar{\phi}_q \right)^2 + 2\gamma K \sigma^2.$$

Averaging over $q$ in the preceding inequality, we obtain

$$\frac{1}{Q} \sum_{q=0}^{Q-1} \left( F_q(\mathbf{x}_q) - F_q(\mathbf{u}) \right) \leq \frac{\|\mathbf{x}_0 - \mathbf{u}\|^2 - \|\mathbf{x}_Q - \mathbf{u}\|^2}{2\gamma KQ} + 2\gamma^2 L \left( \bar{\phi}_q \right)^2 + 2\gamma K \sigma^2$$

$$\leq \frac{\mathcal{D}^2}{2\gamma KQ} + 2\gamma^2 L \left( \bar{\phi}_q \right)^2 + 2\gamma K \sigma^2,$$

where we use Assumption 5. Applying Lemmas 1 and 2, we obtain the average regret bound

$$\frac{1}{Q} \sum_{q=0}^{Q-1} \left( F_q(\mathbf{x}_q) - F_q(\mathbf{u}) \right) = \mathcal{O} \left( \frac{L\mathcal{D}^2}{Q} + \left( \frac{\sqrt{L}\mathcal{D}^2 \bar{\phi}_q}{KQ} \right)^{\frac{2}{3}} + \left( \frac{\sigma^2 \mathcal{D}^2}{Q} \right)^{\frac{1}{2}} \right).$$

$\square$

