# OpenReview forum: "A Unified Analysis of Stochastic Gradient Descent with Arbitrary Data Permutations and Beyond"
_NeurIPS.cc/2025/Conference — NeurIPS 2025 poster_

### Official Review · Reviewer_625o · 2025-06-14

**Clarity:** 2
**Significance:** 2
**Originality:** 2
**Rating:** 4
**Confidence:** 3

**Summary:**

The paper proposed a new general framework for analyzing the stochastic gradient descent algorithm with various kinds of permutations in the smooth setting. A more general assumption was introduced to handle the potential dependency of the permutations across different rounds. The framework was further extended to the federated setting.

**Questions:**

> 1. In practical federated learning setting, the data is often heterogeneous across different clients. How would this affects the algorithm, would it be reflected by the convergence guarantee?

> 2. Would the same kind of technique be applicable to other optimization algorithms besides SGD? Is there a specific reason of picking SGD as the base optimizer in this case?

> 3. How does the algorithm scales with large systems. In fact, is there a reason of choosing dependent permutation over the other algorithms?

**Ethical Concerns:**

["NO or VERY MINOR ethics concerns only"]

**Final Justification:**

The rebuttal of the authors addresses most of my concerns.

**Limitations:**

Yes.

**Paper Formatting Concerns:**

Some tables contain vertical lines which seem a bit strange.

**Quality:**

2

**Strengths And Weaknesses:**

Strengths:
> 1. The author proposed a novel assumption which is more general that handles potential dependencies of permutations effectively.
> 2. The framework proposed was further developed and extended into the federated learning setting.
> 3. Convergence guarantees were provided together with numerical experiments further supporting its validity.

Weaknesses:
> 1. The newly proposed assumption brings additional difficulty into practical implementations, in Theorem 1, the step size $\gamma$ seems to be dependent on the parameters such as $A_i,B_i$. Given the complexity in practice, I am not sure whether the theoretical indication actually can be helpful at all.

> 2. Do the authors have an example, where all of the assumptions in Theorem 1 holds? The sets of assumptions seem quite restrictive to me. In order to apply the theorem, do we need to determine all of $A_i, B_i$ and $D$? If it is the case, how can we actually determine them in a practical setting?

> 3. I am curious about the original motivation of the authors. If dependent permutation is better, why bother developing a theory that includes the three cases. Why not focusing alone on the dependent case.

> 4. Given the complication that the theories involve, I would suggest the authors test the algorithm on a broader scope of machine learning and deep learning tasks.

---

> ### Author Rebuttal · Authors · 2025-07-31
>
> > The newly proposed assumption brings additional difficulty into practical implementations, in Theorem 1, the step size $\gamma$ seems to be dependent on the parameters such as $A_i$, $B_i$. Given the complexity in practice, I am not sure whether the theoretical indication actually can be helpful at all.
>
> > Do the authors have an example, where all of the assumptions in Theorem 1 holds? The sets of assumptions seem quite restrictive to me. In order to apply the theorem, do we need to determine all of $A_i, B_i$ and $D$? If it is the case, how can we actually determine them in a practical setting?
>
> 1. The assumptions. The main assumptions used in Theorem 1 is $L$-smoothness (Definition 3) and Assumption 2. The main assumption used in propositions (for specific algorithms) is $L$-smoothness and Assumption 3. (i) $L$-smoothness and Assumption 3 are the standard assumptions in non-convex optimization. (ii) Assumption 2 has been proved to be weaker than Assumption 3 [D2]. (iii) We also acknowledge that, although the assumptions we adopt are standard, they may not hold in modern deep learning models (e.g., CNNs), which we think is out of the scope of this paper.
>
> 2. The constraint of the step size $\gamma$. The constraint of $\gamma$ relies on the choices of $A_i$, $B_i$, which may seem to be stringent. In fact, as shown in Table D1, in the given algorithms, the choices of $A_i$ and $B_i$ does not impose stronger constraints than the existing works [D2,D3,D4,D5].
>
>    [Table D1: Specific choices of $A_i$, $B_i$, D under the standard assumptions ($L$-smoothness and Assumption 3). The numerical constants and polylogarithmic factors of $B_i$ and $D$ are omitted. We use $\lesssim$ to denote "less than" up to some numerical constants and polylogarithmic factors.]
>
>    | Algorithm   | $B_0$       | $A_1$         | $B_1$            | $A_2$          | $B_2$ | $D$              | $\gamma$                                                     |
>    | ----------- | ----------- | ------------- | ---------------- | -------------- | ----- | ---------------- | ------------------------------------------------------------ |
>    | AP          | $N^2\alpha^2$           | 0             | 0                | 0              | 0     | $N^2\varsigma^2$ | $\gamma \lesssim \frac{1}{L(N+N\alpha)}$                     |
>    | RR/FlipFlop | $N\alpha^2$ | 0             | 0                | 0              | 0     | $N\varsigma^2$   | $\gamma \lesssim \frac{1}{L(N+\sqrt{N}\alpha)}$              |
>    | GraB-proto  | 0           | $\frac{3}{4}$ | $N^2 + \alpha^2$ | 0              | 0     | $\varsigma^2$    | $\gamma \lesssim \min \\{\frac{1}{LN}, \frac{1}{L_{2,\infty}N(1+\alpha)}, \frac{1}{L_{\infty}N}\\}$ |
>    | GraB        | 0           | $\frac{3}{5}$ | $N^2 + \alpha^2$ | $\frac{1}{50}$ | $N^2$ | $\varsigma^2$    | $\gamma \lesssim \min \\{\frac{1}{LN}, \frac{1}{L_{2,\infty}N(1+\alpha)}, \frac{1}{L_{\infty}N}\\}$ |
>    | PairGraB    | 0           | $\frac{4}{5}$ | $N^2 + \alpha^2$ | 0              | 0     | $\varsigma^2$    | $\gamma \lesssim \min \\{\frac{1}{LN}, \frac{1}{L_{2,\infty}N(1+\alpha)}, \frac{1}{L_{\infty}N}\\}$ |
>
> 3. The implementations of the algorithms. (i) As shown in Table D1, we can derive the values of $A_i$, $B_i$, $D$ for the given algorithms theoretically. This is mainly to use Theorem 1 to obtain their convergence guarantees, which is a important metric to measure the convergence of algorithms. (ii) However, it is not necessary to compute their values in practice. For example, for RR and FlipFlop, we only need to shuffle the dataset, and tune the step size (and other hyperparameters) in the usual way; for GraBs, we need to run Algorithms 9, 10, 11 (in Appendix C.2) and reuse the hyperparameters from RR (or tune them as RR). In addition, no additional hyperparameters are introduced in GraBs.
>
> [D1] Lan, First-order and Stochastic Optimization Methods for Machine Learning, 2020. (540+ citations)
>
> [D2] Lu et al, A General Analysis of Example-Selection for Stochastic Gradient Descent, ICLR, 2022.
>
> [D3] Koloskova et al., On Convergence of Incremental Gradient for Non-Convex Smooth Functions, ICML, 2024.
>
> [D4] Lu et al., GraB Finding Provably Better Data Permutations than Random Reshuffling, NeurIPS, 2022.
>
> [D5] Cooper et al., Coordinating Distributed Example Orders for Provably Accelerated Training, NeurIPS, 2023.
>
> > I am curious about the original motivation of the authors. If dependent permutation is better, why bother developing a theory that includes the three cases. Why not focusing alone on the dependent case.
>
> + Developing a unified framework is a consistent topic in permutation-based SGD [D2,D3,D6,D7,D8].
> + Existing frameworks cannot include the DP categories. Meanwhile, the DP algorithms have been shown to perform better in multiple areas (e.g., distributed learning [D4], multi-objective opti-
>   mization [D9], and also federated learning in this paper).
> + The framework can help us understand the impacts of the permutations in permutation-based SGD, e.g., the definition of the order error $\bar \phi_q$.
> + The framework can also be used to analyze the convergence of future permutation-based SGD algorithms.
>
> [D6] A Unified Convergence Analysis for Shuffling-Type Gradient Methods, JMLR, 2021.
>
> [D7] Characterizing & Finding Good Data Orderings for Fast Convergence of Sequential Gradient Method, arXiv, 2022.
>
> [D8] A Unified Convergence Theorem for Stochastic Optimization Methods, NeurIPS, 2022.
>
> [D9] Joint Gradient Balancing for Data Ordering in Finite-Sum Multi-Objective Optimization, ICLR, 2025.
>
> > Given the complication that the theories involve, I would suggest the authors test the algorithm on a broader scope of machine learning and deep learning tasks.
>
> As suggested, we will add more tasks in the revision. For the current version, the main contributions of this paper are theoretical.
>
> + We validate the theoretical conclusions through a simulated experiments (Section 5).
> + For SGD, our theoretical conclusions are aligned with the existing experiments in [D4,D5].
> + For FL, we use the experiments of training CNN/ResNet on CIFAR-10/CIFAR-100/CINIC-10 to validate the effectiveness of the new proposed FL-GraB (Appendix I), which shows that FL-GraB outperforms FL-RR in cross-device FL settings.
>
> > In practical federated learning setting, the data is often heterogeneous across different clients. How would this affects the algorithm, would it be reflected by the convergence guarantee?
>
> Yes, our convergence guarantees can reflect the heterogeneity. The guarantees are given in Table D3. Here, we mainly study the heterogeneity, so the local solver of FL is set to be GD. The level of the heterogeneity is measured by $\varsigma$: Larger value of $\varsigma$ means higher heterogeneity. The conclusions: (i) High heterogeneity will hurt the convergence. (ii) The convergence of FL-GraB is the best (when $Q$ and $N$ are large).
>
> [Table D2: Convergence guarantees for FL.]
>
> | Algorithm | Guarantee                                                    |
> | --------- | ------------------------------------------------------------ |
> | FL-AP     | $\frac{L F_0}{Q} + \left(\frac{LF_0 S\varsigma }{ NQ}\right)^{\frac{2}{3}} + \left(\frac{LF_0 N \varsigma }{ NQ}\right)^{\frac{2}{3}}$ |
> | FL-RR     | $\frac{L F_0\left( 1+\alpha\right)}{Q} + \left(\frac{LF_0 S\varsigma }{ NQ}\right)^{\frac{2}{3}} + \left(\frac{LF_0 \sqrt{N} \varsigma }{ NQ}\right)^{\frac{2}{3}}$ |
> | FL-GraB   | $\frac{\tilde L  F_0+\left (L_{2,\infty}F_0 \varsigma \right)^{\frac{2}{3}}}{Q} + \left(\frac{L_{2,\infty}F_0 S\varsigma }{ NQ}\right)^{\frac{2}{3}} + \left(\frac{L_{2,\infty}F_0 \varsigma }{ NQ}\right)^{\frac{2}{3}}$ |
>
> > Would the same kind of technique be applicable to other optimization algorithms besides SGD? Is there a specific reason of picking SGD as the base optimizer in this case?
>
> For now, the framework only applies to SGD. The reasons of picking SGD as the base optimizer.
>
> 1. SGD is simple and effective.
> 2. The existing theory of SGD is more well-developed.
> 3. We think studying SGD is a good starting point. Studying the effects of the permutations on other optimizers can be a promising future direction.
>
> > How does the algorithm scales with large systems. In fact, is there a reason of choosing dependent permutation over the other algorithms?
>
> We acknowledge that scalability remains an open issue in this version. We think our theory can serve as a first step toward understanding how permutations may affect convergence, potentially inspiring more scalable variants in future works.
>
> > Some tables contain vertical lines which seem a bit strange.
>
> As suggested, we will remove vertical lines in the revision.

---

> > ### Comment · Reviewer_625o · 2025-08-03
> >
> > I thank the authors for their efforts in drafting the rebuttal. The majority of my concerns are resolved.

---

### Official Review · Reviewer_MycC · 2025-07-01

**Clarity:** 3
**Significance:** 3
**Originality:** 3
**Rating:** 4
**Confidence:** 3

**Summary:**

This paper presents a unified framework for analyzing permutation-based Stochastic Gradient Descent (SGD), including arbitrary, independent, and dependent permutations. By introducing a more general assumption (Assumption 2) that captures inter-epoch permutation dependencies, the authors generalize existing analyses and derive upper bounds for prior algorithms (e.g., GraB). The framework is further extended to Federated Learning (FL) with regularized client participation. The authors propose FL-GraB, a new algorithm that improves FL training performance under permutation constraints.

**Questions:**

1. Can the unified framework be extended to cover independent client sampling, as in Wang & Ji (NeurIPS 2022)?
2. Can you provide a comparison or discussion of how FL-GraB performs against prior client sampling strategies, such as SCAFFOLD (Karimireddy et al., 2020)
3. Can you clarify the implementation details of the AP and FL-AP algorithms used in the experiments?

**Ethical Concerns:**

["NO or VERY MINOR ethics concerns only"]

**Final Justification:**

Technically solid paper and the authors have addressed my concerns in the rebuttal.

**Limitations:**

Yes

**Paper Formatting Concerns:**

According to the formatting instructions (Section 4.2), footnotes should be placed at the bottom of the page on which they appear.

**Quality:**

3

**Strengths And Weaknesses:**

Strengths
1. The paper presents a new upper bound for dependent permutation-based SGD algorithms.
2. It offers a clean and comprehensive theoretical analysis, with results that are consistent with prior work.
3. The findings are interesting: FlipFlop and PairGraB are shown to achieve upper bounds comparable to those of RR and the original GraB, respectively.
4. The proposed unified framework is extended to federated learning FL. FL-GraB empirically outperforms FL-RR in the experiments provided.

Weaknesses
1. Only covers regularised participation in FL; the much more common independent sampling setting (e.g., Wang & Ji 2022) seems outside the framework.
2. The advantage of FL-GraB or dependent client sampling is not clear. There is no comparison with FL baselines such as SCAFFOLD (Karimireddy et al., 2020) under heterogeneous data distributions.
3. The implementation AP / FL-AP are not clearly described, and the corresponding results are not discussed.
4. Figure 1 lacks an x-axis label (likely "epochs").
5. "Initilize" in Algorithm 2 line 4 should be "Initialize".

References
1. Wang, S., & Ji, M. A unified analysis of federated learning with arbitrary client participation. In Proceedings of the Conference on Neural Information Processing Systems (NeurIPS), 2022.
2. Karimireddy, et al. SCAFFOLD: Stochastic controlled averaging for federated learning. In Proceedings of the International Conference on Machine Learning (ICML), 2020.

---

> ### Author Rebuttal · Authors · 2025-07-31
>
> > Only covers regularised participation in FL; the much more common independent sampling setting (e.g., Wang & Ji 2022) seems outside the framework.
>
> > Can the unified framework be extended to cover independent client sampling, as in Wang & Ji (NeurIPS 2022)?
>
> Before drawing the conclusion, let us revisit the definition of independent participation (their Proposition 4.7) in [C1]: If $\{u_r^n\}$ is independent across $r$, for any $n$, where $u_r^n$ is the participation weight of client $n$ ($u_r^n$ is a random variable and is the key component of [C1]'s framework), and $r$ is the index of the communication round. In the experiments of their Figure D.1, they gave a more concrete example (a simplified version): "For independent participation, the subset of clients are selected randomly according to a uniform distribution, independently across rounds". As we can see, this simplified independent participation pattern is the typical pattern studied in [C2,C3].
>
> Now, we are sure that our unified framework can be extended to cover the independent participation pattern in [C1] (at least the simplified version).
>
> 1. *Cover the simplified independent participation pattern*. In our framework, we divide all rounds into smaller chunks (with the same size) and analyze each of the chunks separately. We denote the size of these chunks as $P$. In the original framework, for simplicity, we let $P=\frac{N}{S}$, where $N$ is the total number of the clients and $S$ is the participating clients in each round. To incorporate independent participation: (i) We treat $P$ as an algorithm-specific parameter. (ii) For independent participation, we set $P = \Theta(\frac{1}{\gamma L})$ as done explicitly in [C4] and implicitly in [C1]'s Proposition 4.7. (ii) For the original algorithms in our paper, we set $P=\frac{N}{S}$.
> 2. *Extension for arbitrary participation weights*. In the first step, we show that our framework can incorporate the independent participation with uniform participation weights (the simplified version). Next, we consider the arbitrary participation weights. The main challenge is due to the participation weight $u_q^n$ (of client $n$ in epoch $q$) can be arbitrary. One simple approach to circumvent this challenge is to apply different weights (in the parameter aggregation stage) to different clients' updates [C5,C6]. For example, if the participation weight of client $n$ is $u_q^n$ in epoch $q$, we can use the weight $\frac{1}{u_q^n}$ (we need to assume that $u_q^n>0$) in the parameter aggregation stage. By this, we can transform the complex problem (with arbitrary participation weights) into the original problem (with uniform participation weights) in this paper. Furthermore, as pointed out by [C6], the original FedAvg with arbitrary participation in fact optimizes a different objective from the true objective, and this modified algorithm (with different weights) has been proved to optimize the true objective and shown better performances [C6].
>
> [C1] Wang and Ji, A Unified Analysis of Federated Learning with Arbitrary Client Participation, NeurIPS, 2022.
>
> [C2] Karimireddy et al., SCAFFOLD: Stochastic Controlled Averaging for Federated Learning, ICML, 2020.
>
> [C3] Yang et al., Achieving Linear Speedup with Partial Worker Participation in Non-IID Federated Learning, ICLR, 2021.
>
> [C4] Koloskova et al., On Convergence of Incremental Gradient for Non-Convex Smooth Functions, ICML, 2024.
>
> [C5] Wang et al., Tackling the Objective Inconsistency Problem in Heterogeneous Federated Optimization, NeurIPS, 2020.
>
> [C6] Wang and Ji, A Lightweight Method for Tackling Unknown Participation Statistics in Federated Averaging, ICLR, 2024.
>
> > The advantage of FL-GraB or dependent client sampling is not clear. There is no comparison with FL baselines such as SCAFFOLD (Karimireddy et al., 2020) under heterogeneous data distributions.
>
> > Can you provide a comparison or discussion of how FL-GraB performs against prior client sampling strategies, such as SCAFFOLD (Karimireddy et al., 2020)
>
> We compare SCAFFOLD [C2] and FL-GraB theoretically and empirically.
>
> 1. *Theoretical Aspect*. According to the convergence guarantees, SCAFFOLD is immune to data heterogeneity, while FL-GraB still suffers from heterogeneity.
>
> 2. *Empirical Findings*. In cross-device FL settings (using the same experimental setup as our Appendix I), FL-GraB consistently outperforms both FL-RR and SCAFFOLD. The results are shown in Table C1. The client ordering for SCAFFOLD is the same as FL-RR.
>
>    *Key observations*: (i) FL-GraB outperforms FL-RR and SCAFFOLD, especially at the early stage. (ii) SCAFFOLD underperforms FL-RR, which can be due to the high heterogeneity [C7,C8] and the staleness of control variates [C6] in cross-device FL.
>
> [Table C1: Test accuracies (averaged over the last 10 epochs and 5 random seeds) for training CNN on CIFAR-10/CIFAR-100/CINIC-10.]
>
> | CIFAR-10  | FL-RR | SCAFFOLD | FL-GraB      |
> | --------- | ----- | -------- | ------------ |
> | epoch 100 | 77.2  | 71.1     | 79.12 (+1.9) |
> | epoch 200 | 80.0  | 76.4     | 81.20 (+1.2) |
>
> | CIFAR-100 | FL-RR | SCAFFOLD | FL-GraB     |
> | --------- | ----- | -------- | ----------- |
> | epoch 100 | 41.1  | 30.5     | 44.8 (+3.7) |
> | epoch 200 | 47.5  | 41.1     | 49.2 (+1.7) |
>
> | CINIC-10  | FL-RR | SCAFFOLD | FL-GraB     |
> | --------- | ----- | -------- | ----------- |
> | epoch 100 | 59.7  | 53.6     | 63.1 (+3.4) |
> | epoch 200 | 64.2  | 58.6     | 66.7 (+2.5) |
>
> [C7] TCT Convexifying Federated Learning using Bootstrapped Neural Tangent Kernels, NeurIPS, 2022.
>
> [C8] Federated Learning on Non-IID Data Silos An Experimental Study, ICDE, 2022. (1300+ citations)
>
>
> > The implementation AP / FL-AP are not clearly described, and the corresponding results are not discussed.
>
> > Can you clarify the implementation details of the AP and FL-AP algorithms used in the experiments?
>
> The specific implementations of AP and FL-AP (AP allows for arbitrary permutations) are as follows:
>
> 1. The data examples are sorted in ascending order with respect to the value of $b_n$.
> 2. This sorted permutation is used repeatedly for all the epochs.
> 3. Additional implementation details are available in our submitted code (specifically in the `train_IG()` function)
>
>
> > Figure 1 lacks an x-axis label (likely "epochs").
>
> > "Initilize" in Algorithm 2 line 4 should be "Initialize".
>
> > According to the formatting instructions (Section 4.2), footnotes should be placed at the bottom of the page on which they appear.
>
> We thank you for catching these presentation issues:
>
> 1. We will add "epochs" as the x-axis label for Figure 1.
> 2. "Initilize" in Algorithm 2, line 4 will be corrected to "Initialize".
> 3. We will revise our footnote usage to comply with formatting guidelines.

---

> ### Author Response · Authors · 2025-08-03
> **Details for covering the simplified version of the independent participation**
>
> **Details for covering the simplified version of the independent participation**
>
> > Only covers regularized participation in FL; the much more common independent sampling setting (e.g., Wang & Ji 2022) seems outside the framework.
>
> > Can the unified framework be extended to cover independent client sampling, as in Wang & Ji (NeurIPS 2022)?
>
> As a follow-up to the previous comment where we provided a proof sketch for the simplified version of the independent participation, we now present the derived bound for completeness.
>
> For simplicity, we let $\alpha =0$ in Assumption 3, i.e., $\\|\nabla f\_n(x) - \nabla f(x)\\|^2 \leq \varsigma^2$ for all $n$.
>
> When $P$ is arbitrary, we can get
> $$
> \frac{1}{Q} \sum_{q=0}^{Q-1} \\|  \nabla f(x_q) \\|^2 \lesssim \frac{F_0}{\gamma \eta \frac{1}{S}KPQ } + \gamma^2 L_{2,p}^2 K^2 \varsigma^2 + \gamma^2 L_{2,p}^2 \frac{1}{S^2}K^2 \frac{1}{Q} \sum_{q=0}^{Q-1}(\bar\varphi_q)^2 + \frac{1}{P^2} \frac{1}{Q} \sum_{q=0}^{Q-1}(\psi_q^P)^2
> $$
> where $( \psi_q^P)^2 = \\|\sum_{u=0}^{P-1} \nabla f_{\pi_q(u)}(x_{q}) - \nabla f(x_q)\\|^2$, and $\bar\varphi_q =\max_{u\in [P]} \\| \sum_{i=0}^{v(u)-1} (\nabla f_{\pi_q(i)}(x_q) - \nabla f(x_q)) \\|_p^2$ ($v(u) = \lfloor \frac{u}{S}\rfloor \cdot S$). Here, we define $(\psi_q^P)^2$ for simplicity, because we use different $\ell_p$-norms for $\psi$ and $\varphi$. It should be noted that $P$ here does not necessarily correspond to a single epoch. Therefore, $q$ and $Q$ should not be interpreted as the index and the total number of epochs, respectively. Here, we denote the total number of communication round as $R$. Note that we use the total number of communication rounds $R$ here, while we use the total number of epochs in the original paper.
>
> + Regularized participation (FL-AP, FL-RR, FL-GraB): In this case, we let $P = N$ or $P = c\cdot N$ ($c$ is a numerical constant). In this case, $(\psi_q^P)^2 = 0$, and the bound is reduced to that studied in the original paper.
>
> + Independent participation: In this case, we let $P = \Theta(\lfloor \frac{1}{\gamma KL}\rfloor \cdot S)$ ($P$ is a multiple of $S$) and $p=2$. Then, $(\psi_q^P)^2$ can be subsumed into the $(\bar\varphi_q)^2$. Then,
>   $$
>   \min_{r=0,\ldots,R-1} \\|\nabla f(x_r)\\|^2 \leq \frac{1}{Q} \sum_{q=0}^{Q-1} \\|  \nabla f(x_q) \\|^2 \lesssim \frac{F_0}{\gamma \eta KR} + \gamma^2 L^2 K^2 \varsigma^2 + \gamma^2 L^2 \frac{1}{S^2} K^2 (\bar\varphi_q)^2
>   $$
>    (i) When the clients are chosen randomly within each round and independently across rounds, then $(\bar \varphi_q)^2 \lesssim P\varsigma^2$ (with high probability; see [1]). Using this, we get the bound $O\left( \frac{LF_0}{R} + \left( \frac{LF_0\varsigma}{ R}  \right)^{2/3} +  \left( \frac{LF_0\varsigma^2}{ S R}  \right)^{1/2}  \right)$, where we let $\eta =1$ for comparison. This bound matches that in [2].
>
>    (ii) When the clients are chosen arbitrarily within each round and independently across rounds, then $(\bar \varphi_q)^2 \lesssim PS\varsigma^2$. Using this, we get the bound $O\left( \frac{LF_0}{R} + \left( \frac{LF_0\varsigma}{R}  \right)^{2/3} +  \left( \frac{LF_0\varsigma^2}{ R}  \right)^{1/2}  \right)$, where we let $\eta =1$ for comparison. This bound is better than [3].
>
>   | Independent participation (across rounds) | Arbitrary (within each round)                                | Random (within each round)                                   |
>   | ----------------------------------------- | ------------------------------------------------------------ | ------------------------------------------------------------ |
>   | Previous works                            | $O\left( \frac{LF_0}{R} + \left( \frac{LF_0\varsigma}{ R}  \right)^{2/3} +  \left( \frac{LF_0\varsigma^2}{ S R}  \right)^{1/2}  \right)$ [2] | $O\left( \frac{LF_0}{R} + \left( \frac{LF_0\varsigma}{R}  \right)^{2/3} +  \left( \frac{LF_0N\varsigma^2}{ R}  \right)^{1/2}  \right)$ [3] |
>   | This work                                 | $O\left( \frac{LF_0}{R} + \left( \frac{LF_0\varsigma}{ R}  \right)^{2/3} +  \left( \frac{LF_0\varsigma^2}{ S R}  \right)^{1/2}  \right)$ | $O\left( \frac{LF_0}{R} + \left( \frac{LF_0\varsigma}{R}  \right)^{2/3} +  \left( \frac{LF_0\varsigma^2}{ R}  \right)^{1/2}  \right)$ |
>
> It is worth noting that our framework captures **the impact of client ordering** in both regularized and independent participation patterns, which is not addressed in [3].
>
> [1] Yu and Li, High Probability Guarantees for Random Reshuffling, NeurIPS 2023 Workshop Heavy Tails in Machine Learning, 2023.
>
> [2] Yang et al., Achieving Linear Speedup with Partial Worker Participation in Non-IID Federated Learning, ICLR, 2021.
>
> [3] Wang and Ji, A Unified Analysis of Federated Learning with Arbitrary Client Participation, NeurIPS, 2022.

---

> > ### Comment · Reviewer_MycC · 2025-08-07
> >
> > Thanks for your response. I have no further questions.

---

### Official Review · Reviewer_iP1W · 2025-07-02

**Clarity:** 4
**Significance:** 3
**Originality:** 3
**Rating:** 5
**Confidence:** 4

**Summary:**

This paper gives a unified framework for studying different permutation-based algorithms. The framework is general enough to include arbitrary parameter orders, random reshuffling, and some adaptive shuffling algorithms such as GraB. The main novelty is in Assumption 2-- It connects the gradient error (difference between shuffled gradients and the true gradients) over an epoch *between* epochs. It vaguely reminds me of some of the unified assumptions made in prior work in other contexts like federated learning [1] where the gradient error at epoch $k$ is related to the error at epoch $k-1$. Theorem 1 gives a convergence rate under this assumption in the nonconvex and smooth setting, and is specialized to various algorithms in Propositions 1-7. Not all of these results are restatements of existing results, as e.g. for FlipFlop and PairGraB. Finally, the results are directly applicable to the case when the different "data points" represent clients in a federated setting, and we can control how we sample them.

[1] Gorbunov, Eduard, Filip Hanzely, and Peter Richtárik. "Local SGD: Unified theory and new efficient methods." International Conference on Artificial Intelligence and Statistics. PMLR, 2021.

**Questions:**

1. Can your analysis extend to a sort-of hybrid algorithm where we choose a number (say $K$ that is smaller than $N$), sample $K$ differrent data points and then shuffle them? This might be more relevant to modern large-scale language model training.
2. Can your analysis be used with the shuffling variance technique of [3] to recover the $1/(NQ)$ dependence at least in the strongly convex setting?

[3] Mishchenko, Konstantin, Ahmed Khaled, and Peter Richtárik. "Random reshuffling: Simple analysis with vast improvements." Advances in Neural Information Processing Systems 33 (2020): 17309-17320.

**Ethical Concerns:**

["NO or VERY MINOR ethics concerns only"]

**Final Justification:**

The paper is good and the authors have addressed most of my concerns in the rebuttal.

**Limitations:**

N/A.

**Paper Formatting Concerns:**

No concerns.

**Quality:**

3

**Strengths And Weaknesses:**

1. (Strength) The framework is quite elegant and unifies many existing results. It provides a direct point-of-view as to *why* methods like GraB work. I actually like that no new methods are introduced here, it is relatively rare to see papers that do a good job at taking a field with a diverse number of methods and find a unifying framework to look at them.
2. (Strength) The paper provides a new analysis for PairGraB, a practical variant of GraB that was not analyzed prior.
3. (Weakness) The paper's analysis framework is to treat all of these methods as equivalent to doing one large step of gradient descent plus deviation error. Unfortunately, this means that all the convergence results obtained here outperform SGD only in the same regime that GD outperforms SGD, and underperform SGD in the regime when SGD is better than GD. This can be seen from e.g. the last column in Table 1, where the paper's leading term scales as $1/Q$ rather than $1/(NQ)$ as in the analysis of [2]. This is unfortunately common to a lot of the literature on RR.
4. (Weakness) The relevance of shuffling-based algorithms to modern training is a bit in doubt, since now we've looped back to the case where datasets are far too large to actually shuffle through and we instead rely on sampling. This isn't really this paper's fault though but is a more generic concern. I guess I would ask: how might we use some of the insights here to develop methods where shuffle only a subset of the dataset?

[2] Koloskova, Anastasia, et al. "A unified theory of decentralized SGD with changing topology and local updates." International conference on machine learning. PMLR, 2020.

---

> ### Author Rebuttal · Authors · 2025-07-31
>
> > ... the paper's leading term scales as $1/Q$ rather than $1/(NQ)$ as in the analysis of [2]...
>
> > Can your analysis be used with the shuffling variance technique of [3] to recover the $1/(NQ)$ dependence at least in the strongly convex setting?
>
> 1. The shuffling variance technique [B1] cannot be used to achieve the better bound in the strongly convex setting. The failure is primarily due to technical reasons arising from the incorporation of DP. We provide the details in Detail 1.
> 2. Another technique developed in [B2] can be used to achieve the better bound $1/(NQ)$ for AP in the non-convex setting; however, whether a tighter upper bound can be achieved for the other algorithms (i.e., IP and DP) remains an open problem. We provide the details in Detail 2. In this paper, given that this advanced technique [B2] (i) only applies to AP and (ii) can complicate the framework substantially, we did not use it in the current version.
>
> **Detail 1: Details for using the shuffling variance technique in the strongly convex setting.**
>
> 1. Since the shuffling variance technique only applies to the strongly convex setting [B1], we assume the strong convexity of all local functions and the existene of the minimizer $x_\ast$ of $f$.
>
> 2. In the strongly convex setting, we need a new definition for the order error $(\bar\phi_q)^2 = \max_n \\| \sum_{i=0}^{n-1} ( \nabla f_{\pi_q(i)}(x_\ast)-\nabla f(x_\ast) ) \\|^2$. A similar assumption is made as $(\bar \phi_q)^2 \lesssim (\bar\phi_{q-1}) + (\bar\phi_{q-2})+\cdots+(\bar\phi_{q-\nu})+\cdots+ D$ ($\nu$ is a numerical constant).
>
> 3. To deal with DP, we need to define a more general shuffling variance (associated with the epoch $q$), $\sigma_q^2 = \max_{n=1,\ldots,N-1}[\frac{1}{\gamma}\mathcal{D}\_{f\_{\pi_q(n)}} (x_\ast^n, x_\ast) ]$ ($\mathcal{D}$ denotes the Bregman divergence, $N$ is the total number of local functions). When $\sigma_q$ is independent of $q$, it reduces to the original definition in [B1]'s Definition 2. According to [B1]'s Proposition 1, we get that $\sigma_q^2 \leq \frac{1}{2}\gamma L (\bar\phi_q)^2$.
>
> 4. Let us focus on the [B1]'s Theorem 1. See [B1]'s Equation (23). It is safe to follow the same steps as [B1] before Equation (23). For $\sigma_q^2$, we get the following equation rather than its Equation (23):
>
>    $$
>    \begin{align}
>    \mathbb{E} \\|x_Q-x_\ast \\|^2 &\leq (1-\gamma \mu)^{NQ} \\|x_0-x_\ast\\|^2 + 2\gamma^2\sum_{n=0}^{N-1}(1-\gamma \mu)^n\sum_{q=0}^{Q-1}(1-\gamma\mu)^{N(Q-q-1)}\sigma_q^2 \\\\
>    &\leq (1-\gamma \mu)^{NQ} \\|x_0-x_\ast\\|^2 + \gamma^3L\sum_{n=0}^{N-1}(1-\gamma \mu)^n\underbrace{\sum_{q=0}^{Q-1}(1-\gamma\mu)^{N(Q-q-1)}(\bar\phi_q)^2}_{T}
>    \end{align}
>    $$
>
>     We see that when $\sigma_q$ is independent of $q$ (the case in [B1]), as done in [B1], we can get that the second term on the right hand side is upper bounded by $2\frac{\gamma}{\mu}\sigma^2$. However, $\sigma_q$ varies with $q$ and is bounded by $(\bar \phi_q)^2$, which relies on the order errors in the previous epochs (e.g., $(\bar\phi_{q-1})^2$).
>
> 5. Next, our goal is to use the assumption that $(\bar \phi_q)^2 \lesssim (\bar\phi_{q-1}) + (\bar\phi_{q-2})+\cdots+(\bar\phi_{q-\nu})+ D$  to bound $T$. As shown in Theorem 3 in this paper, it requires that $\frac{1}{(1-\gamma\mu)^N}\leq c$, where $c>1$ is a numerical constant. Then, using the fact $(1-x)^t\leq \frac{1}{1+tx}$ for $x\in [0,1]$ and $t\geq 0$, we get $1+\gamma \mu N\leq (1-\gamma \mu)^N \leq c$. Thus, if $\frac{1}{(1-\gamma\mu)^N}\leq c$, then $\gamma \leq \frac{c-1}{\mu N}=O(\frac{1}{N})$. This constraint on $\gamma$ will cause a worse bound (on the optimization term).
>
> **Detail 2: Details for the technique in [B2].**
>
> 1. *The limitation of the standard approach.* As noted, we divide all iterations into smaller chunks (with the same size) and analyze each of the chunks separately. We denote the size of these chunks as $P$. This standard approach in permutation-based SGD cause a constraint of $\gamma LP \lesssim 1$. In this paper, $P=N$, where $N$ is the total number of the local objective functions. This causes the constraint on the step size $\gamma \lesssim \frac{1}{LN}$, which finally causes a worse bound on the optimization term.
> 2. *The core idea of the new technique [B2].* As noted in [B2], we can set $P=\Theta(\frac{1}{\gamma L})$ to circumvent the constraint of $\gamma L P \lesssim 1$. This technique finally leads to a better bound $O(\frac{1}{NQ})$ for AP. For AP, whose permutations are arbitrary (without any specific structure), analyzing convergence for arbitrary $P$ will not affect the statistical properties of these permutations. However, this does not apply to IP and DP. Take IP as an example. For IP, whose permutations are independently generated across epochs, analyzing convergence for $P=\Theta(\frac{1}{\gamma L})$ ($P$ does not necessarily equal $N$) will compromise the nice statistical properties of these permutations: The elements in different permutations may be divided into the same chunk and the elements in one permutation may be divided into different chunks.
> 3. *Use this new technique [B2] in our framework.* One straightforward way to use this technique is to establish the framework for arbitrary $P$, and then use $P$ as a algorithm-specific parameter. For example, for AP, we let $P=\Theta(\frac{1}{\gamma L})$, and for IP and DP, we let $P=N$. Given (i) that using this technique will complicate the framework substantially, (i) that this technique only applies to AP and the main contribution of this paper is to incorporate DP, we did not use it in this version.
>
> [B1] Random Reshuffling Simple Analysis with Vast Improvements, NeurIPS, 2020.
>
> [B2] On Convergence of Incremental Gradient for Non-Convex Smooth Functions, ICML, 2024.
>
>
> > (Weakness) The relevance of shuffling-based algorithms to modern training is a bit in doubt, since now we've looped back to the case where datasets are far too large to actually shuffle through and we instead rely on sampling. This isn't really this paper's fault though but is a more generic concern. I guess I would ask: how might we use some of the insights here to develop methods where shuffle only a subset of the dataset?
>
> > Can your analysis extend to a sort-of hybrid algorithm where we choose a number (say $K$ that is smaller than $N$), sample $K$ different data points and then shuffle them? This might be more relevant to modern large-scale language model training.
>
> This forward-looking suggestion naturally leads us to consider the integration of permutation-based SGD and online learning [B3,B4]. As far as we know, there is no existing works that combines these two theoretically. So we next try to analyze such a hybrid algorithm that you mentioned.
>
> 1. Algorithm. First, we online sample the data examples from a very large dataset. Then, every time $K$ examples are sampled, we shuffle these $K$ examples and feed them to a model. For simplicity, we use projected online gradient descent (GD) (see [B3]'s Algorithm 2.1) to train this model with the shuffled examples. Intuitively, this algorithm is a hybrid algorithm where the top-level part is online GD and the bottom-level part is permutation-based SGD.
>
> 2. Notations. We call each $K$ examples as a block. We use $f_{\pi_q(k)}$ to denote the loss function of the $k$-th example in block $q$, where $\pi_q$ is the permutation in block $q$. We use $F_q = \frac{1}{K}\sum_{k=0}^{K-1}f_{\pi_q(k)}$ to denote the loss function of the block $q$. We use $x$ to denote the iterates, $x_{q}^k$ to denote the iterate in example $k$ and block $q$, $x_q$ denote the initial iterate in block $q$.
>
> 3. Assumptions. We use the assumptions: (i) Assumption B1: All iterations $x\in \mathcal{V}$, where $\mathcal{V}$ is a closed convex set with diameter $D$. (ii) Assumption B2: $\\|\nabla F_q(x)\\| \leq \sigma$ for all $q$. Assumptions B1 and B2 are standard in online GD [B3, B4]. We also use the order error $\bar \phi_q = \max_k \\|\sum_{i=0}^{k-1}(\nabla f_{\pi_q(k)} - \nabla F_q(x))\\|$, $L$-smoothness.
>
> 4. Regret analyses. This analysis is based on [B3]'s Theorem 2.13. Here, we provide the key steps.
>
>     $$
>    \begin{align}
>    \\| x_{q+1} - u\\|^2 - \\| x_q - u\\|^2 & \leq \\| x_q - \gamma \sum_{k=0}^{K-1} \nabla f_{\pi_q(k)}(x_q^k) -u\\|^2 - \\| x_q - u\\|^2\\\\
>    &\leq \underbrace{-2\gamma\sum_{k=0}^{K-1} \langle x_q-u,\nabla f_{\pi_q(k)}(x_q^k)\rangle}\_{T_1} + \underbrace{\gamma^2 \\|\gamma\sum_{k=0}^{K-1}\nabla f_{\pi_q(k)}(x_q^k)\\|}_{T_2} \\\\
>    \end{align}
>    $$
>
>     where $u\in \mathcal{V}$ is an arbitrary competitor. Then, $T_1$ can be bounded by [B5]'s Lemma 5 and our Lemma 5. $T_2$ can be bounded by Jensen's inequality and our Lemma 5.
>
> 5. Regret bound. The average regret bound of the algorithm is as follows:
>    $$
>    \frac{1}{Q}\sum_{q=0}^{Q-1} ( \nabla F_q(x_q) - \nabla F_q(u)) = O\left( \frac{\sigma D}{\sqrt{Q}} + \frac{L^{1/3}D^{4/3}(\bar\phi_q)^{2/3}}{(KQ)^{2/3}} + \frac{D^2}{LK^2Q}\right),
>    $$
>    where the last term is due to the constraint $\gamma L N \lesssim 1$ of our Lemma 5. This framework includes AP and IP readily. The original online GD corresponds to AP here. This bound means that a nice permutation (the value of $\bar\phi_q$ is small) will lead to a nice bound. Thus, shuffling a subset of the dataset can also help improve the performance.
>
> Given the complexity, we leave DP for future works. Furthermore, we think combining permutation-based SGD and online learning can be a promising future direction. We thank you again for these valuable suggestions, which have inspired meaningful extensions to our work. We will incorporate these insights into the revised manuscript.
>
> [B3] A Modern Introduction to Online Learning, arXiv, 2019. (500+ citations)
>
> [B4] Introduction to Online Convex Optimization, Foundations and Trends® in Optimization, 2016. (2400+ citations)
>
> [B5] SCAFFOLD: Stochastic Controlled Averaging for Federated Learning, ICML, 2020.

---

> > ### Comment · Reviewer_iP1W · 2025-08-04
> >
> > Thanks for your rebuttal and comments.
> >
> > 1. I'm not really convinced by the argument against Shuffling Variance here. You have shown that one approach doesn't work, this doesn't mean no other approach is possible.
> >
> > 2. Thanks for your outline in Detail 2. You don't have to include all of it in the paper but it would be instructive to add a comment.
> >
> > 3. Thanks for this outline as well, it is what I suspected should hold.
> >
> > Thank you, I maintain my positive score.

---

### Official Review · Reviewer_ediT · 2025-07-05

**Clarity:** 4
**Significance:** 3
**Originality:** 3
**Rating:** 5
**Confidence:** 3

**Summary:**

This work focuses on the analysis of permutation-based SGD. The authors propose a framework that enables a unified analysis for the three categories: arbitrary permutations, independent permutations, and dependent permutations. They address the dependency of permutations by introducing a more general assumption. They extend the analysis to Federated Learning scenarios where client permutations are considered. The framework establishes new convergence rate results and also results that are consistent with existing works.

**Questions:**

Corresponding to the weakness, is there any hidden factors that prevent one from choosing, e.g., $\delta = o(1/Q^2)$ so that the bounds will hold with higher probability with increasing $Q$?

**Ethical Concerns:**

["NO or VERY MINOR ethics concerns only"]

**Final Justification:**

The authors have addressed my concern. I'd keep the original score.

**Limitations:**

Yes

**Quality:**

3

**Strengths And Weaknesses:**

Strength:
1. The paper is well-structured and clearly written.
2. The provide extensive comparisons with previous work.
3. The authors propose a more general assumption that leads to a unified theorem for all three classes of SGD with data permutations. Mainly, the assumption accounts for dependency of the permutations between epochs.
4. The authors define a new order error for Federated Learning and extend the techniques for analysis of the convergence of FL with regularized participation.
5. The authors apply their main theorems to analyze multiple existing algorithms. They show that the results match the existing bounds. They also establish new results for methods that previously lack convergence guarantees (namely, FL-GraB), which indicates the effect of client ordering in Federated Learning.

Weakness:
The probabilistic bounds becomes less and less effective with increasing $Q$ due to $Q \delta$.

---

> ### Author Rebuttal · Authors · 2025-07-31
>
> > Weakness: The probabilistic bounds becomes less and less effective with increasing $Q$ due to $Q\delta$.
>
> > Corresponding to the weakness, is there any hidden factors that prevent one from choosing, e.g., $\delta=o(1/Q^2)$ so that the bounds will hold with higher probability with increasing $Q$?
>
> As noted, our current version uses a failure probability of $Q\delta$ rather than $\delta$, which could lead to looser bounds as $Q$ increases. We adopted this formulation primarily to maintain consistency with [A1,A2], which established initial bounds for GraBs.
>
> However, we would like to clarify that our framework can provide bounds that hold with probability $1-\delta$, addressing your specific concern. Given that the bound for Arbitrary Permutations (AP) is deterministic (not a probabilistic one), we next only discuss Independent Permutations (IP) and Dependent Permutations (DP):
>
> 1. IP. Taking Random Reshuffling (RR) as an example. Starting from Equation (12) in this paper, we get that, with probability at least $1-\delta$, $(\bar \phi_q )^2 \leq 4\log^2(\frac{8}{\delta})U $, where $U=N\alpha^2\\|\nabla f(x_q)\\|^2 + N \varsigma^2$ for brevity. That is, $\Pr( (\bar\phi_q)^2 \geq 4\log^2(\frac{8}{\delta})U) \leq \delta$ for any $q$. Then, applying the union bound for $q=0,\ldots,Q-1$, we get $\Pr( \exists q, (\bar\phi_q)^2 \geq 4\log^2(\frac{8}{\delta})U) \leq Q\delta$. Then, setting $\delta = \frac{\delta}{Q}$, we get $\Pr( \exists q, (\bar\phi_q)^2 \geq 4\log^2(\frac{8Q}{\delta})U) \leq \delta$. That is, with probability $1-\delta$, $(\bar\phi_q)^2 \leq 4\log^2(\frac{8Q}{\delta})U$ for all $q=0,\ldots,Q-1$. Then, using these bounds, we get the desired bound that holds with $1-\delta$.
> 2. DP. Taking GraB-proto as an example. Starting from the equation between Lines 764 and 765, we get that, with probability $1-\delta$, $(\bar\phi_q)^2 \leq \log^2(\frac{dN}{\delta})U + V$ where $U = 900(\alpha^2\\|\nabla f(x_{q-1})\\|^2 + \varsigma^2)$ and $V = \frac{3}{4} (\bar\phi_{q-1})^2 + \frac{1}{50}N^2\\|\nabla f(x_{q-1})\\|^2$. That is, $\Pr( (\bar\phi_q)^2 \geq \log^2(\frac{dN}{\delta})U+V)\leq\delta$ for any $q= 1,\ldots,Q-1$. Then, applying the union bound for $q=1,\ldots,Q-1$, we get $\Pr( \exists q, (\bar\phi_q)^2 \geq \log^2(\frac{dN}{\delta})U+V) \leq (Q-1)\delta$. Then, setting $\delta = \frac{\delta}{Q-1}$, we get $\Pr( \exists q, (\bar\phi_q)^2 \geq \log^2(\frac{dN(Q-1)}{\delta})U+V) \leq (Q-1)\delta$. That is, with probability $1-\delta$, $(\bar\phi_q)^2 \leq \log^2(\frac{dN(Q-1)}{\delta})U+V)$ for all $q=1,\ldots,Q-1$. Then, using the bound of $(\bar\phi_0)^2$ (it is deterministic) and these bounds, we get the desired bound that holds with $1-\delta$.
>
> [A1] Lu et al., GraB Finding Provably Better Data Permutations than Random Reshuffling, NeurIPS, 2022.
>
> [A2] Cooper et al., Coordinating Distributed Example Orders for Provably Accelerated Training, NeurIPS, 2023.
>
> [A3] Yu and Li, High Probability Guarantees for Random Reshuffling, NeurIPS 2023 Workshop Heavy Tails in Machine Learning, 2023.

---

> > ### Comment · Reviewer_ediT · 2025-08-08
> >
> > Thanks to the authors for thoroughly addressing my concern.

---

### Decision · Program_Chairs · 2025-09-17

**Decision:**

Accept (poster)

**Comment:**

This paper proposes an assumption that can incorporate permutation-based SGD with *past-dependent* permutations into a unifying framework. The authors prove convergence results on nonconvex smooth optimization problems under the framework, which can be applied to various specific example selection schemes. Particularly, the analysis yields new convergence results for FlipFlop and some GraB variants. The paper also applies this framework to federated learning with regularized client participation, and derives new bounds.
All reviewers were positive about the paper and appreciated the usefulness of the new assumption. The authors put extensive efforts into addressing reviewers’ concerns and all of them were satisfied. I also find that the paper successfully pushes the boundary of permutation-based SGD analysis. I hence recommend acceptance; I would like to ask the authors to incorporate the rebuttal and the discussion into the final version of the paper.

A couple minor comments:
- From my own reading, I think “when $Q$ is large” in line 132 uses $Q$ without stating it denotes the number of epochs.
- I think it would be a particularly good idea to include Table D1 in the paper.